# VOCSIM: A TRAINING-FREE BENCHMARK FOR ZERO-SHOT CONTENT IDENTITY IN SINGLE-SOURCE AUDIO

## ABSTRACT

The goal of general-purpose audio representations is to map acoustically variable instances of the same event to nearby points, i.e., to resolve content identity in a zero-shot setting. We introduce VocSim, a training-free benchmark that measures this capability directly on 125k single-source clips aggregated from 19 corpora spanning human speech, animal vocalizations, and environmental sounds. By restricting to single-source audio, VocSim isolates content representation from source separation confounds. We evaluate embeddings with two training-free measures: local Precision@k and a point-wise Global Separation Rate (GSR) that contrasts each item's nearest inter-class distance with its mean intra-class distance. To calibrate GSR, we report lift over an empirical random baseline obtained by label permutation. Across diverse models, a simple pipeline—frozen Whisper encoder features with time–frequency pooling and label-free PCA—yields strong zero-shot performance. Yet VocSim also surfaces a consistent generalization gap: on blind, low-resource speech, local retrieval (P@k) drops sharply and the GSR lift over baseline is small, indicating that global class structure is only marginally better than chance. As external validation, top embeddings predict zebra finch perceptual similarity (80.9% triplet accuracy), improve downstream bioacoustic classification, and achieve state-of-the-art performance on the HEAR benchmark. We release data, code, and a public leaderboard to standardize evaluation of zero-shot audio similarity and to catalyze representations that better generalize across sound sources and recording conditions.

## 1 INTRODUCTION

The ability to judge similarity between arbitrary sounds underpins fundamental behaviors in humans and animals, from distinguishing phonetic contrasts in infancy (Kuhl, 2004) to song imitation in birds (Tchernichovski et al., 2001; Doupe & Kuhl, 1999). Biological auditory systems achieve this with remarkable robustness by extracting a stable **content identity**, a core acoustic signature that defines a sound event (e.g., a specific phone, word, or bird call). This process requires generalizing across nuisance variations such as speed, speaker identity, loudness, and recording conditions. In contrast, machine-learned audio representations often require extensive task-specific supervision and can fail when faced with novel sound categories or acoustic distortions (Xie & Virtanen, 2019; Cramer et al., 2019).

A core challenge for building general audio intelligence is therefore to develop embeddings that, without any training on the target classes, intrinsically organize sounds by their content identity. This zero-shot similarity task is a more fundamental test of representation quality than supervised classification, as it probes the inherent geometry of the learned embedding space. While representations are also used for many other purposes (e.g., reconstruction, enhancement, or regression), this clustering of acoustically variable instances of the same event is a necessary property for recognition, retrieval, and similarity judgments. VocSim focuses explicitly on this aspect of representation quality. While self-supervised models such as Wav2Vec 2.0 (Baevski et al., 2020) and Whisper (Radford et al., 2022) have achieved great success, the evaluation paradigm has largely focused on their performance in downstream tasks. Evaluation suites like HEAR (Turian et al., 2022) and SUPERB (Yang et al., 2021), while invaluable for benchmarking the performance of audio models, primarily measure *task-specific adaptability* by fine-tuning a model on a suite of downstream tasks. This leaves a critical gap in our understanding of whether these models have learned truly general principles of sound in their raw, untuned state.

To fill this methodological gap, we introduce **VocSim** (Figure 1), a benchmark engineered as a diagnostic instrument for zero-shot content identity. Its design ensures a rigorous evaluation by **stress-testing generalization**, aggregating 125,382 clips from 19 corpora diversifying acoustic factors, and by **isolating content representation**, focusing exclusively on single-source audio to avoid the confound of source separation.

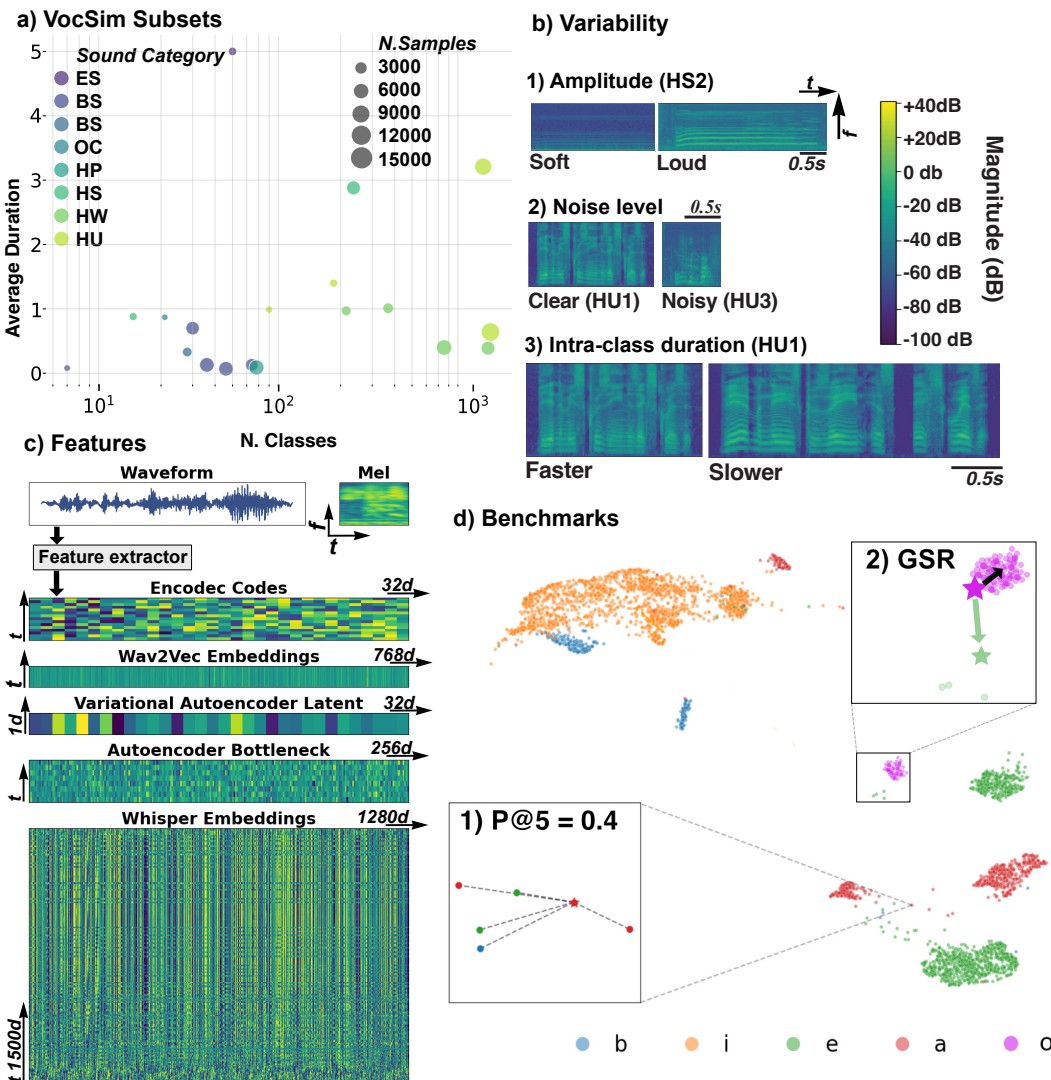

Figure 1: **Overview of the VocSim benchmark.** (a) **Dataset Composition:** VocSim aggregates 19 corpora, with dot size indicating sample count. (b) **Acoustic Variability:** Spectrograms show engineered variation to stress-test generalization. (c) **Feature Representations:** A diverse range of embeddings are evaluated. (d) **Zero-Shot Benchmarks:** We use two training-free metrics. **(1) Precision@k (P@k)** measures local neighborhood purity (e.g., 2 of 5 neighbors are correct, so P@5=0.4). **(2) Global Separation Rate (GSR)** provides a point-wise measure of class separation by comparing each point's average intra-class distance (Avg_ID, black arrow) to its nearest inter-class distance (NID, green arrow). The UMAP projection visualizes the embedding space, where different colors represent distinct classes (here, syllable types from BS3), illustrating how well embeddings form separable clusters.

Our extensive benchmarking of handcrafted features, self-supervised autoencoders, and large foundation models reveal several insights. We show that features from the Whisper encoder, refined with simple time-frequency pooling and PCA, form a powerful and effective representation for this task. Crucially, VocSim's design uncovers and quantifies a critical generalization gap. To do so, we introduce the Global Separation Rate (GSR), a metric that measures an embedding's discriminative

power relative to a random baseline. On a blind test set, we find that the **lift-over-random** of GSR collapses, revealing that even the best models organize novel classes only marginally better than chance, despite relatively high absolute GSR. The utility of our benchmark is validated by showing that the best-performing embeddings align with avian perceptual judgments (Zandberg et al., 2024), a key measure of biological plausibility given the scarcity of comparable human perceptual datasets, achieve superior performance in downstream bioacoustic classification (Goffinet et al., 2021), and establish new state-of-the-art results on the HEAR benchmark. By releasing VocSim, we provide a shared platform for developing audio representations that approach the flexibility and nuanced sensitivity of biological hearing. A full glossary of abbreviations for datasets, features, and metrics is provided in Appendix C.

**Contributions.**

- VocSim: a training-free benchmark for zero-shot content identity on 125k single-source clips spanning speech, animal vocalizations, and environmental sounds.
- A point-wise Global Separation Rate (GSR) with permutation-based calibration. We compute lift-over-random as a geometry-aware check that GSR is above chance for a given subset, but we rank models within each subset by their raw GSR (and P@k), not by lift.
- A simple, strong zero-shot baseline: frozen Whisper encoder with time–F pooling and label-free, per-subset PCA; Spearman distance is robust across domains. (F = frequency for spectrograms; feature/channel for hidden states.)
- A quantified OOD gap: on blind low-resource speech, P@k drops sharply and GSR lift is small, indicating class structure only marginally better than chance.
- External validation: alignment with avian perceptual similarity and improved fixed-feature bioacoustic classification. We release data, code, and a public leaderboard.

## 2 RELATED WORK

**Classical Audio Similarity.** Early work on audio matching paired handcrafted spectral features (MFCCs, spectral centroid) with dynamic-time warping (DTW) or statistical models (Davis & Mermelstein, 1980; "Múller, 2007; Aucouturier & Pachet, 2004). While effective for constrained tasks like speaker verification and limited-domain retrieval (Tzanetakis & Cook, 2002; Law et al., 2009), these methods lack robustness to variation in open-set similarity tasks.

**Self-Supervised and Foundation Audio Representations.** Inspired by successes in NLP and vision, the field has shifted to powerful learned audio encoders. The landscape includes raw-audio SSL models such as Wav2Vec 2.0 (Baevski et al., 2020), HuBERT (Hsu et al., 2021), and WavLM (Chen et al., 2022); spectrogram-based architectures including CNNs (PANNs (Kong et al., 2020)) and transformers trained with masked modeling or tagging (AST (Gong et al., 2021), PaSST (Koutini et al., 2022), BEATs (Chen et al., 2023), EAT (Chen et al., 2024)); and masked autoencoders such as AudioMAE (Huang et al., 2023). Large-scale foundation or multimodal models, Whisper (Radford et al., 2022) and CLAP (Elizalde et al., 2022), and neural codecs like EnCodec (Défossez et al., 2022) have set strong baselines across tasks, drawing on self-supervised objectives such as CPC (van den Oord et al., 2019), SimCLR (Chen et al., 2020), BYOL (Grill et al., 2020), DINO (Caron et al., 2021), and Barlow Twins (Zbontar et al., 2021). While these encoders excel under fine-tuning or linear probing, their intrinsic, training-free zero-shot similarity properties in single-source audio remain underexplored.

**The landscape of audio evaluation.** Most public evaluations target either (i) **task-specific adaptability**, where HEAR (Turian et al., 2022) and SUPERB (Yang et al., 2021) report fine-tuned or linearly probed performance on supervised tasks, or (ii) **complex scene analysis**, where AudioSet (Gemmeke et al., 2017) and DCASE (Mesaros et al., 2018) benchmark polyphonic event recognition. Similarly, benchmarks such as BirdSet (Rauch et al., 2025) and BIRB (Hamer et al., 2023) address the distinct challenge of bioacoustic scene analysis under domain shift. Large audio–text corpora such as LAION-Audio-630K (Wu et al., 2024) and WavCaps (Mei et al., 2024) are invaluable for *pre-training*, but they are not structured as evaluation suites.

What remains under-measured is the *intrinsic, training-free geometry* of frozen embeddings: do they already express content identity without any task-specific heads or labels? VocSim fills this gap by standardizing a strictly training-free evaluation on *single-source* audio and reporting complementary local and global measures (P@k and permutation-calibrated GSR), with held-out low-resource speech

to stress-test OOD generalization. A side-by-side comparison with existing resources is provided in Appendix D.3, Table 5.

**Zero-Shot Retrieval in Other Modalities.** Probing the intrinsic geometry of embeddings is an established paradigm in vision and NLP. Zero-shot retrieval benchmarks like CUB-200-2011 for images (Wah et al., 2011) and MMTEB for text (Enevoldsen et al., 2025) have proven instrumental for assessing generalization without fine-tuning. Inspired by this progress, VocSim brings this same foundational rigor to the audio domain, providing a necessary diagnostic for measuring the basis of general audio intelligence.

## 3 THE VOCSIM DATASET

VocSim is a large-scale benchmark of 125,382 audio clips designed to evaluate zero-shot content identity. Its primary goal is to test generalization by aggregating 19 distinct corpora (Table 1). Cross-source aggregation is a principled, standard strategy—as in GLUE (Wang et al., 2019) for NLP and SUPERB (Yang et al., 2021) for audio—when the objective is to probe robust generalization across diverse conditions. Constructing a single, de novo corpus with comparable breadth of vocal production systems, recording conditions, and acoustic characteristics would be prohibitively difficult; aggregation is therefore the practical and appropriate path to stress-test generalization.

Table 1: Characteristics of the 19 aggregated subsets comprising the VocSim benchmark. Subset IDs correspond to sound categories (e.g., BS: Bird Syllables, HP: Human Phones, HW: Human Words). Subsets marked with 'X' under 'Avail.' are reserved as blind test sets.

| ID | # Samples | Classes | Avg. Sam./Class (range) | Avg. Dur. (s) (min-max) ↓ | Avail. | DRI (dB) [*] |
|---|---|---|---|---|---|---|
| BS3 | 9988 | 46 | 217.1 (20-1374) | 0.07 (0.03-0.20) | ✓ | 20 |
| BS1 | 473 | 6 | 78.8 (78-79) | 0.08 (0.03-0.17) | ✓ | 20 |
| HP | 10687 | 68 | 157.2 (8-176) | 0.09 (0.03-0.49) | ✓ | 18 |
| BS2 | 10001 | 36 | 277.8 (6-1209) | 0.13 (0.03-0.26) | ✓ | 15 |
| BS4 | 7035 | 64 | 109.9 (9-129) | 0.13 (0.03-0.83) | ✓ | 31 |
| BC | 3321 | 28 | 118.6 (6-605) | 0.33 (0.03-2.99) | ✓ | 15 |
| HW2 | 8827 | 1324 | 6.7 (6-7) | 0.39 (0.08-1.00) | ✓ | 19 |
| HW1 | 11532 | 754 | 15.3 (6-100) | 0.40 (0.07-1.10) | ✓ | 20 |
| HU2 | 17041 | 1366 | 12.5 (6-130) | 0.64 (0.03-1.49) | ✓ | 22 |
| BS5 | 8244 | 30 | 274.8 (42-333) | 0.70 (0.03-4.99) | ✓ | 26 |
| OC1 | 441 | 21 | 21.0 (9-32) | 0.87 (0.28-5.32) | ✓ | 23 |
| HS1 | 1670 | 14 | 119.3 (6-713) | 0.88 (0.04-2.98) | ✓ | 26 |
| HW4 | 3497 | 215 | 16.3 (6-46) | 0.97 (0.46-2.00) | X | 15 |
| HU4 | 1001 | 80 | 12.5 (6-44) | 0.99 (0.47-1.98) | X | 15 |
| HW3 | 4540 | 368 | 12.3 (6-24) | 1.01 (0.32-2.00) | X | 20 |
| HU3 | 1703 | 183 | 9.3 (6-24) | 1.40 (0.40-3.00) | X | 20 |
| HS2 | 8918 | 236 | 37.8 (9-93) | 2.88 (0.04-6.00) | ✓ | 28 |
| HU1 | 14463 | 1245 | 11.6 (6-50) | 3.21 (1.24-6.10) | ✓ | 34 |
| ES1 | 2000 | 50 | 40.0 (40-40) | 5.00 (5.00-5.00) | ✓ | 35 |

[*] The Dynamic Range Index (DRI) was estimated by comparing the power in high-energy (80th percentile) versus low-energy (20th percentile) frames. As clips are pre-segmented vocalizations, this serves as a heuristic for the dynamic range between the signal and ambient noise within the recording.

Our design starts from the property VocSim measures: zero-shot content identity in single-source audio. This motivates three principles. First, **isolation of content**: we restrict to single-source recordings to decouple representation quality from source separation. Second, **acoustic-to-label** fidelity: within each subset, a label denotes the acoustic unit defined by the source corpus, so members of a class are intended to share a characteristic spectro–temporal signature. Concretely, phones (HP) are segment-level phonemes; words (HW) are word tokens; utterances (HU) are contiguous segments; call types (OC/BC) group calls of that type; and song "syllable types" are defined per individual bird (reflecting that labels like A/B/C have no cross-individual meaning). Third, **standardization**: we provide a uniform 16 kHz mono format to ensure broad model compatibility and remove spatial cues.

We then select 19 widely used, permissively licensed corpora that satisfy these constraints and jointly span three production domains, human speech, non-human vocalizations, and environmental sounds, while covering key nuisance factors (speaker/individual identity, channel, background, duration). We

deliberately exclude music and strongly polyphonic soundscapes (e.g., AudioSet (Gemmeke et al., 2017), MagnaTagATune (Law et al., 2009)) because our scope is single-source content identity under challenging but non-overlapping conditions.

Within this scope, class definitions span a controlled range of acoustic variability. Highly stereotyped units (e.g., segment-level phonemes; per-individual bird syllables) form tight acoustic classes, whereas words and utterances intentionally introduce greater intra-class variation due to speaker, prosody, and coarticulation; these serve as robustness stress tests rather than perfectly compact categories. We preserve source semantics without cross-dataset relabeling or collapsing (e.g., bird syllable labels remain unique within an individual), avoiding artificial classes that would conflate acoustically unrelated items.

VocSim's composition covers the three target domains with concrete sources: *Human speech and vocalizations* include phones, words, and utterances from LibriSpeech (Panayotov et al., 2015), VCTK (Yamagishi et al., 2019), and AMI (Carletta et al., 2005), plus non-verbal human sounds and vocal imitations (Cartwright & Pardo, 2015; Kim & Pardo, 2018). *Animal vocalizations* comprise six songbird corpora (zebra finch (Tomka et al., 2023; Elie & Theunissen, 2020; 2015; Clemens, 2021), Bengalese finch (Nicholson et al., 2022), canary (Giraudon et al., 2021; Cohen et al., 2022)) and 21 giant otter call types ("Mumm & Knórnschild, 2014). *Environmental sounds* are represented by ESC-50 (Piczak, 2015). To provide a stringent out-of-distribution test, we reserve a held-out, non-public blind set consisting of field recordings from low-resource languages (Shipibo–Conibo and Chintang (Stoll & Bickel, 2013)), evaluated via a secure, server-side protocol.

To systematically stress-test generalization, the aggregation introduces variability along four axes: (1) **duration** (sub-100 ms to 5 s), (2) **class granularity** (few-shot to many-shot classes), (3) **recording realism** (studio to noisy field), and (4) **intra-class variability** (speaker/individual identity, loudness, channel). We apply simple, uniform quality controls: a minimum class size of six items for stable local metrics; filtering of extreme duration outliers; and targeted down-weighting of extreme Zipfian tokens in large public speech corpora (see Algorithm 1). To sanity-check that observed structure reflects genuine signal rather than artifacts, we compute empirical random baselines by permuting labels within each subset and run controlled label-flip stress tests; both confirm that results lie well above chance and degrade smoothly under synthetic annotation noise (Appendix H.1, G).

As with any cross-source benchmark, some structural correlations remain intrinsic to the source corpora. We treat this variability as part of the benchmark's realism: a general-purpose embedding should identify content despite such nuisance factors. All models are evaluated under identical conditions, so relative performance provides a fair comparison of their ability to form coherent representations from naturalistic audio. For transparency about pre-training exposure, we also audit known overlaps between VocSim sources and model pre-training corpora and provide a summary in Appendix D.2.

## 4 BENCHMARK AND METHODS

Our evaluation methodology is designed to probe the intrinsic geometric structure of audio embeddings in a zero-shot setting. It involves extracting features from a wide range of models, deriving standardized fixed-length vectors, computing pairwise distances, and applying two complementary, training-free metrics that assess both local and global properties of the embedding space.

### 4.1 FEATURE REPRESENTATIONS

We benchmark an extensive array of audio feature extractors, ranging from traditional spectral features to large foundation models. These extractors produce either fixed-length vectors or, more commonly, variable-length sequences of frame-level embeddings. Our evaluation distinguishes between three categories:

- **Spectral Baseline:** 128-band log-Mel spectrograms (Davis & Mermelstein, 1980). To create a simple baseline, spectrograms are padded with zeros to the length of the longest clip in their subset and flattened.

- **Subset-Specific Unsupervised Baselines:** A Variational Autoencoder (VAE) and a standard Autoencoder (AE). These small models are trained unsupervised on a **per-subset basis**, including on the unlabeled audio of the blind test sets. Their performance serves to quantify the benefit of large-scale pre-training. As seen in our results, a small, domain-specific VAE can struggle more

on challenging out-of-distribution data than a simple spectral representation, underscoring the generalization power imparted by pre-training on massive, diverse datasets.

- **Zero-Shot Foundation Models:** These are large-scale models pre-trained on diverse external datasets, evaluated here in a strictly zero-shot fashion without any training or fine-tuning on VocSim data. They include self-supervised speech models (Wav2Vec 2.0 (Baevski et al., 2020), WavLM (Chen et al., 2022)), a masked autoencoder (AudioMAE (Huang et al., 2023)), a recent spectrogram-based transformer (EAT (Chen et al., 2024)), the BEATs model (Chen et al., 2023), another powerful transformer pretrained on AudioSet, and general-purpose systems like OpenAI's Whisper Encoder (Radford et al., 2022), its animal-tuned variant WhisperSeg (Gu et al., 2023), the text-supervised CLAP model (Elizalde et al., 2022), and neural audio codecs like EnCodec (Défossez et al., 2022). For EAT, we use the utterance-level [CLS] token embedding from the final layer.

Together, these encoders span the main families of modern audio representations: raw-waveform SSL speech models (Wav2Vec 2.0, WavLM), spectrogram transformers trained on tagging corpora (EAT, BEATs), masked autoencoding (AudioMAE), and large foundation or multimodal systems (Whisper, WhisperSeg, CLAP, EnCodec). For models that output sequences ($[T \times D]$), we form fixed-length vectors with simple time–F pooling: mean over time, mean over F, or their concatenation (F = frequency for spectrograms; feature/channel for hidden states). Means are computed over all output positions as produced by the encoder, no masking, padding exclusion, or VAD. Optional PCA compression is applied where stated; otherwise, we use the raw pooled vectors.

## 4.2 ZERO-SHOT SIMILARITY METRICS

Operating directly on the pairwise distance matrix, we evaluate embeddings using metrics that probe the geometric structure of the space without requiring label-based training. For each feature representation, we compute three dissimilarity metrics: **Cosine**, **Euclidean (L2)**, and **Spearman's rank correlation dissimilarity** $(1 - \rho)$ (see Appendix E).

**Precision@k (P@k).** A standard local metric that measures neighborhood coherence. For a set of $N$ clips, let $C(i)$ be the class of clip $i$ and $N_k(i)$ be the set of its $k$ nearest neighbors. P@k is the average fraction of neighbors that share the same class as their query item:

$$\text{P@k} = \frac{1}{N \cdot k} \sum_{i=1}^{N} \sum_{j \in N_k(i)} \mathbf{1}(C(j) = C(i))$$

where $\mathbf{1}(\cdot)$ is the indicator function. Unlike retrieval metrics such as mAP or Recall@k, designed for tasks with few positives, P@k is better suited to VocSim's goal of assessing dense class coherence, where every member of a class is a potential positive match.

**Global Separation Rate (GSR).** We designed the **Global Separation Rate (GSR)** as a robust global metric for class separability. For each point, it compares the distance to its nearest inter-class neighbor (**NID**) with its average intra-class distance (**Avg_ID**) using the formula (NID − Avg_ID)/(NID+Avg_ID+$\epsilon$), where $\epsilon$ is a small constant for numerical stability. The final GSR is the average of these point-wise scores, normalized to a percentage. Because GSR compares a minimum (NID) to an average (Avg_ID), its expected absolute value depends on the embedding geometry; we therefore calibrate it via a permutation baseline (Appendix G) and report **lift-over-random** as a diagnostic that a given configuration is meaningfully above chance. For model comparison within a subset, however, we still use the *raw* GSR (together with P@k); we do not compare absolute GSR values across different subsets.

**Zero-Shot Setting.** We evaluate all encoders strictly in a zero-shot regime: models are used as frozen feature extractors with no training or fine-tuning on VocSim tasks or labels. To substantiate this, we performed a systematic overlap audit between VocSim's public subsets and the documented pre-training corpora of all evaluated models (Appendix D.2). The audit shows that (i) the blind test sets and all ten non-human vocalization subsets exhibit no evidence of overlap with any model's pre-training data, while (ii) several public subsets plausibly overlap with specific models (e.g., LibriSpeech with Wav2Vec 2.0; ESC-50 with AudioMAE). Consequently, public-set results should be interpreted as transfer from pre-training objectives (e.g., ASR, tagging) to a novel acoustic-similarity task with new label schemes, whereas the blind low-resource speech sets provide the primary evidence of true out-of-distribution generalization.

# 5 RESULTS

Our extensive evaluation reveals a simple and strong approach to zero-shot audio similarity and precisely quantifies a persistent generalization gap. Full results for all configurations are in Appendix P. Unless otherwise noted, all averages reported in the main text and tables are macro (unweighted) across subsets; micro (sample-weighted) aggregates are explicitly labeled as such.

**Methodological Robustness.** We validated our key methodological choices through a series of ablations, detailed in Appendix H, to ensure that our conclusions reflect intrinsic properties of the embeddings rather than artifacts of the pipeline. A layer-wise sweep over all 32 Whisper encoder layers shows that performance is remarkably stable across depth, justifying our standard use of the final layer. Our choice of simple temporal pooling is supported by a comparison with a more complex, sequence-aware DTW re-ranking baseline (Appendix H.3), which yields no average performance gain. Label-noise experiments (Appendix H.1) further show that global metrics such as GSR degrade much more gracefully than local metrics like P@1 under synthetic label corruption. Finally, we compare our main findings with alternative clustering-based evaluations (NMI, Purity, ARI) and a metric correlation analysis (Appendix G.1), which demonstrate that P@k and GSR are consistent with other measures of class structure while capturing complementary aspects of local and global geometry.

**Whisper Encoder with Unsupervised Adaptation Performs Strongly.** A simple pipeline using a zero-shot Whisper encoder performs strongly on this task (Table 2). The best configuration, which we denote 'EWMTF D100' (Whisper large-v3 encoder with time–F pooling and label-free PCA to 100 Dimensions, using Spearman distance; F = feature/channel; see glossary in Appendix C), yields high zero-shot performance. On public data, it achieves a mean P@1 of 66.8% and a GSR of 41.7%. The confidence intervals, representing the average per-subset stability, confirm the robustness of these metrics. However, a clear generalization gap emerges on the blind test sets, where local retrieval coherence (P@k) drops sharply. The strong performance of Spearman distance across many configurations suggests its rank-based nature is robust to the feature scaling and high-dimensional geometry of these embedding spaces.

Notably, models that rank at or near the top of SUPERB (e.g., WavLM Large/Base+ Chen et al. (2022), HuBERT Large Hsu et al. (2021), and wav2vec 2.0 Large Baevski et al. (2020)) do not consistently top-rank on VocSim under our strictly frozen, training-free protocol. This divergence reinforces that SUPERB measures supervised adaptability, whereas VocSim probes intrinsic zero-shot geometry.

Table 2: Zero-shot performance on VocSim. Values are mean $\pm$ average margin of error across subsets. The margin of error is derived from per-subset 95% bootstrap confidence intervals, calculated over 300 resamples. Distances: S = Spearman $(1-\rho)$; E = Euclidean. "EWMTF D100" = Whisper encoder with time–F pooling and label-free, per-subset PCA to 100D (F = frequency for spectrograms; feature/channel for hidden states). Averages are macro (unweighted) across subsets; micro (sample-weighted) averages are labeled explicitly where used.

| Method | Dist | VocSim Public Sets (Avg.) | | | Blind Test Sets (Avg.) | | |
|---|---|---|---|---|---|---|---|
| | | P@1 ($\uparrow$) | P@5 ($\uparrow$) | GSR ($\uparrow$) | P@1 ($\uparrow$) | P@5 ($\uparrow$) | GSR ($\uparrow$) |
| *Whisper Encoder (Top Configurations)* | | | | | | | |
| **EWMTF D100 (PCA)** | **S** | **66.8**; $\pm$;**0.8** | **57.4**; $\pm$;**0.7** | **41.7**; $\pm$;**0.2** | **11.5**; $\pm$;**1.2** | **7.7**; $\pm$;**1.1** | **39.4**; $\pm$;**0.3** |
| EWMTF (Raw Pooled) | S | 61.5; $\pm$;0.8 | 53.0; $\pm$;0.8 | 40.2; $\pm$;0.3 | **11.5**; $\pm$;1.3 | **7.7**; $\pm$;0.9 | **39.4**; $\pm$;0.3 |
| *Other Foundation Models* | | | | | | | |
| CLAP D100 (PCA) | S | 63.7; $\pm$;0.7 | 55.6; $\pm$;0.6 | 38.1; $\pm$;0.2 | 8.1; $\pm$;1.4 | 5.2; $\pm$;1.0 | 36.2; $\pm$;0.2 |
| CLAP (Raw Pooled) | S | 61.7; $\pm$;0.5 | 54.4; $\pm$;0.5 | 33.8; $\pm$;0.3 | 8.1; $\pm$;1.3 | 5.2; $\pm$;1.0 | 36.2; $\pm$;0.4 |
| WavLM D100 (PCA) | E | 64.1; $\pm$;0.7 | 54.4; $\pm$;0.7 | 37.0; $\pm$;0.3 | 4.6; $\pm$;1.2 | 3.1; $\pm$;0.8 | 35.8; $\pm$;0.3 |
| WavLM (Raw Pooled) | E | 62.8; $\pm$;0.6 | 51.4; $\pm$;0.7 | 35.2; $\pm$;0.3 | 4.6; $\pm$;1.1 | 3.1; $\pm$;0.8 | 35.8; $\pm$;0.2 |
| BEATs | E | 64.3; $\pm$;0.9 | 55.4; $\pm$;0.9 | 31.4; $\pm$;0.2 | 11.4; $\pm$;1.2 | 7.0; $\pm$;0.7 | 34.7; $\pm$;0.2 |
| EAT (CLS Token) | E | 50.2; $\pm$;0.9 | 42.1; $\pm$;1.1 | 24.7; $\pm$;0.3 | 1.9; $\pm$;0.9 | 1.8; $\pm$;0.6 | 22.6; $\pm$;0.2 |
| *Baselines* | | | | | | | |
| Log-Mel (M) | S | 57.7; $\pm$;0.6 | 47.3; $\pm$;0.6 | 34.2; $\pm$;0.3 | 3.5; $\pm$;1.2 | 2.6; $\pm$;0.9 | 33.0; $\pm$;0.2 |
| VAE Latent (VC) | E | 58.2; $\pm$;0.7 | 49.5; $\pm$;0.6 | 34.3; $\pm$;0.3 | 3.8; $\pm$;1.4 | 2.8; $\pm$;0.9 | 25.9; $\pm$;0.3 |

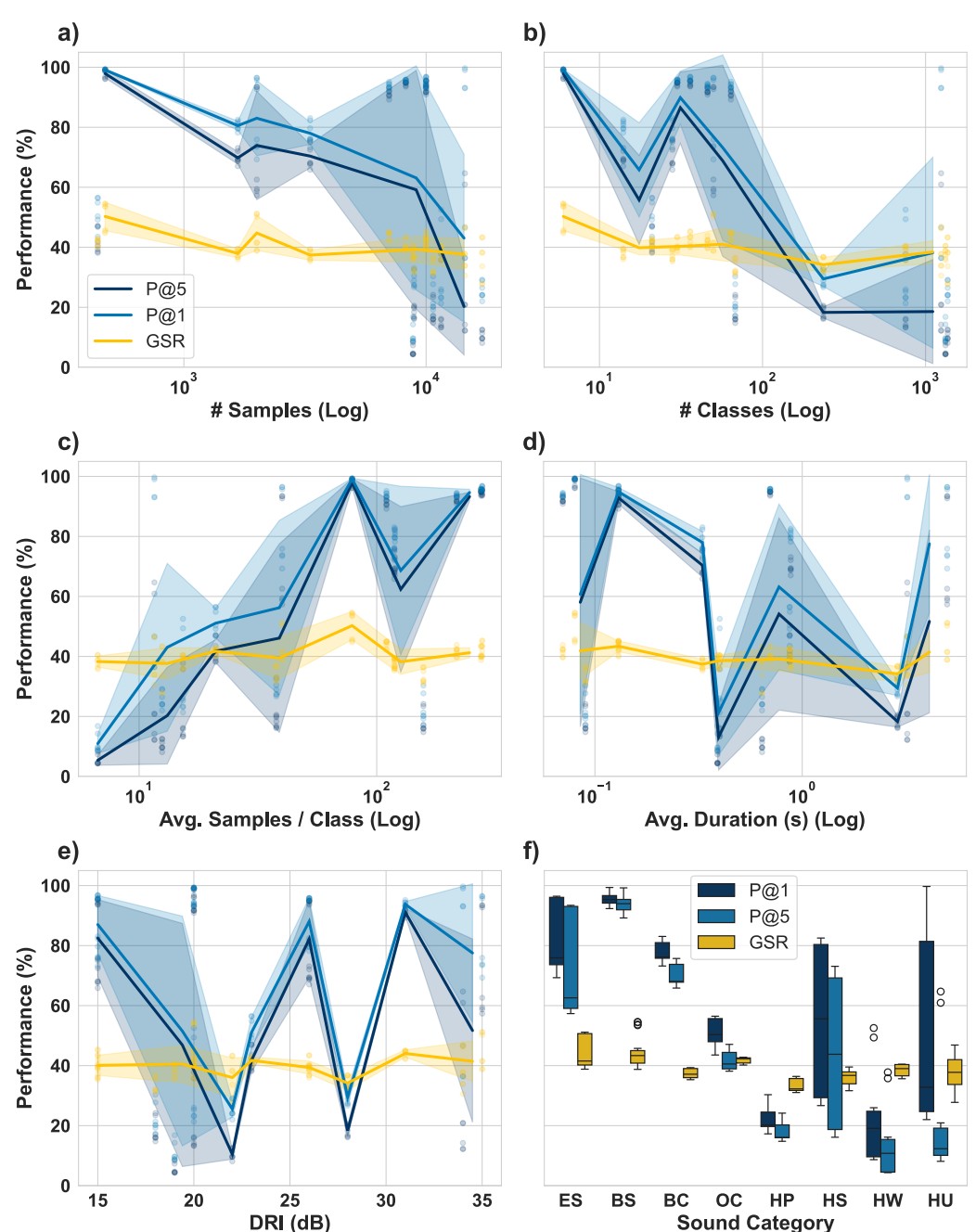

Figure 2: **Performance trends of top 5% of configurations across VocSim subsets.** (a-e) Each point represents a subset's performance. Trend lines show the mean performance against key subset properties. Local metrics like **P@1 and P@5 (blue lines) are highly sensitive to dataset structure**, degrading significantly as the number of classes increases (b) while improving with more samples per class (c). In contrast, the global metric **GSR (yellow line) remains remarkably stable** across these conditions, suggesting it captures a more intrinsic property of the embedding geometry. (f) Boxplots show performance distributions broken down by sound category, highlighting strengths (e.g., on ES) and weaknesses (e.g., on HU).

**Intrinsic Geometry vs. Local Retrieval Accuracy.** Analysis of performance trends reveals a fundamental difference between our local and global metrics (Figure 2). Local neighborhood

coherence (P@k) is highly sensitive to the structural properties of each subset; it degrades significantly as the number of samples in a subset or the number of classes increases (Fig 2a-b) and improves with more samples per class (Fig 2c). This confirms its role as a measure of local retrieval accuracy in a given task structure. In contrast, the Global Separation Rate (GSR) remains remarkably stable across these same conditions, indicating it captures a more intrinsic and task-agnostic property of the embedding's geometric organization. Performance also varies by sound category (Fig 2f), with embeddings performing best on classes with consistent acoustic signatures like Environmental Sounds (ES) and struggling with the high intra-class variance of Human Words (HW) and Utterances (HU).

**The Generalization Gap Revealed by GSR.**   The Global Separation Rate (GSR) quantifies a critical generalization gap. While the top model achieves a strong GSR of 41.7% on public data, this performance drops only moderately to 39.4% on the hidden OOD data. However, the true significance of this drop is revealed by calibrating against an empirical random baseline (Appendix G). On public data, the model's score represents a substantial **16.9-point (micro-averaged) lift** over random chance, confirming that it has learned meaningful class structure. On the hidden OOD data, this lift collapses to just **5.8 points (micro-averaged)**. This demonstrates that despite a superficially high absolute score, the model's ability to organize the novel classes is only marginally better than a random assignment, a finding that explains the sharp drop in local retrieval (P@k).

**Interpreting P@k and GSR jointly.**   P@k reflects local neighborhood purity and is sensitive to class structure (e.g., number of classes, samples per class). GSR, calibrated by permutation baselines, offers a geometry-aware measure of global separability that is more stable across structural shifts and better aligned with worst-case boundary integrity. This stability of GSR's global signal extends to annotation noise, where it degrades more gracefully than the volatile local signal of P@k (see Appendix H.1). Comparing scores to a random permutation baseline is therefore useful to verify that a given raw GSR is truly above chance for that subset, but *within* each subset we still care about, and rank models by, the highest raw GSR (and P@k), rather than by lift-over-random. We also avoid interpreting absolute GSR values as directly comparable across different subsets, since their permutation baselines differ.

## 6   VALIDATION: VOCSIM SCORES PREDICT REAL-WORLD UTILITY

Having established VocSim as a robust measure of an embedding's foundational quality, we now investigate a key hypothesis: does strong performance on this zero-shot task predict utility in real-world applications? We test this by deploying the top-performing embeddings as fixed, off-the-shelf feature extractors in three distinct domains: unsupervised data exploration, alignment with non-human biological perception, and challenging supervised bioacoustic classification.

**Unsupervised data exploration.**   Embeddings that achieve high P@k and GSR form coherent neighborhoods and clear inter-class boundaries, enabling label-free exploration. Applying UMAP (McInnes et al., 2020) followed by HDBSCAN (McInnes & Healy, 2017) across VocSim subsets, Whisper (EWMTF D100) yields the best unsupervised weighted purity on average (40% public; 11.4% blind). Absolute blind-set purity is modest due to many low-shot classes and strong OOD shifts, but the relative ranking mirrors zero-shot retrieval and GSR, supporting VocSim as an early diagnostic for unsupervised pipelines in new acoustic domains (details in Appendix M.1).

**Alignment with avian perception.**   We test biological plausibility by predicting zebra finch triplet judgments (Zandberg et al., 2024). Frozen Whisper achieves 80.9% accuracy on the high-consistency subset, approaching reported inter-bird agreement (80–90%). This suggests that large, speech-trained encoders learn acoustic dimensions that transfer to non-human perceptual spaces without any bird-specific finetuning. Evaluation protocol and baselines are summarized in Table 25 and detailed in Appendix M.2.

**Downstream bioacoustics with fixed features.**   In a fixed-feature setting, Whisper embeddings paired with a high-frequency spectrogram frontend yield strong performance on mouse USV tasks (Goffinet et al., 2021): 99.49% Top-1 for strain classification and 53.1% Top-1 for individual identity, substantially above prior fixed-feature baselines. This demonstrates that the geometry favored by VocSim metrics can translate into practical gains on challenging bioacoustic classification when paired with an appropriate frontend (Appendix M.3, M.4, M.5).

**State-of-the-art on the HEAR Benchmark.**   We validated our embeddings on seven HEAR benchmark tasks (Turian et al., 2022), following the official K-fold linear probing protocol. Our **Whisper**

**EWMTF D100** model establishes new state-of-the-art (SOTA) results, consistently outperforming strong baselines like CLAP D100 and top leaderboard models such as GURA and RedRice (Turian, 2022; Dinkel et al., 2023). For example, it achieves **79.3%** accuracy on CREMA-D emotion recognition (Cao et al., 2014) and **98.6%** on Speech Commands (Warden, 2018), surpassing previous SOTA scores by +4.1% and +1.0%, respectively. This confirms that representations successful on our VocSim benchmark are also SOTA general-purpose features for diverse audio tasks (see Appendix M.6).

## 7 DISCUSSION AND CONCLUSION

This paper introduced VocSim, a training-free benchmark that provides a controlled and interpretable measure of zero-shot content identity in single-source audio. By aggregating 19 diverse corpora, VocSim enables a cross-source evaluation that reveals both what current embeddings capture well and where they fail to generalize.

**A practical blueprint for zero-shot similarity.** Our results provide a simple recipe for strong zero-shot audio similarity: use a large frozen encoder (e.g., Whisper), apply explicit time–F pooling (F = frequency for spectrograms; feature/channel for hidden states), and optionally compress with label-free, per-subset PCA. Spearman distance is a robust default for high-dimensional pooled features. This recipe is effective and efficient, requiring no finetuning or labels, and produces compact vectors suitable for indexing and retrieval.

**Interpreting P@k and GSR.** P@k reflects local neighborhood purity and is sensitive to class structure (e.g., number of classes, samples per class). GSR, calibrated by permutation baselines, offers a geometry-aware measure of global separability that is more stable across structural shifts and better aligned with worst-case boundary integrity. This stability of GSR's global signal extends to annotation noise, where it degrades more gracefully than the volatile local signal of P@k (see Appendix H.1). Comparing scores to a random permutation baseline is therefore crucial for correctly interpreting generalization across datasets with different geometric properties, a principle that underpins our main finding.

**A generalization ceiling on OOD speech.** On hidden OOD low-resource speech, we observe a sharp drop in P@k and a small GSR lift-over-random. This indicates that current embeddings retain structured geometry but fail to align it with novel class boundaries. Overcoming this ceiling likely requires pretraining objectives that explicitly reward cross-domain content identity, stronger invariances to speaker/channel/recording conditions, and better coverage of low-resource phonotactics.

**Predictive validity beyond the benchmark.** VocSim scores predict utility across three applications. First, the top embeddings' structure (high P@k and GSR) yields superior unsupervised clustering purity in a label-free setting. Second, they achieve strong alignment with avian perceptual similarity (80.9% triplet accuracy), approaching inter-bird agreement. Third, the same fixed-feature embeddings produce state-of-the-art results on two mouse USV tasks when combined with a high-frequency frontend. Finally, under the official HEAR linear-probe protocol, our Whisper EWMTF D100 embedding achieves state-of-the-art performance on several tasks, further highlighting transferability.

**Threats to validity and scope.** VocSim intentionally focuses on single-source audio to isolate content representation. This excludes polyphonic mixtures and high-level semantic tasks (e.g., intent or function). Filtering of frequent tokens in large public corpora mitigates extreme Zipfian skew and is not applied to hidden sets to preserve natural distributions. Bird syllables are labeled per individual to reflect the lack of cross-individual label semantics. Standardizing to 16 kHz trades some high-frequency detail for compatibility; our USV results show that appropriate frontends can recover information beyond this band. Our comparisons involve heterogeneous encoders (architecture, size, supervision, and pretraining corpora), and VocSim is an evaluation suite of off-the-shelf frozen features rather than a causal analysis of training regimes. We therefore evaluate all models under identical, training-free conditions and avoid attributing gains to any single factor; capacity-controlled ablations are orthogonal to our goal.

**Outlook.** We release VocSim as a living benchmark with code and a leaderboard, inviting evaluation of new embedding families and pooling schemes. The empirical success of permutation-calibrated GSR suggests incorporating lift-aware objectives during pretraining. Beyond audio, the methodology—single-source focus and permutation-calibrated global separability—may inform training-free evaluation in other modalities.

ETHICS STATEMENT

Throughout the paper, we use the terms "blind test sets" and "hidden OOD sets" interchangeably to refer to the two non-public, out-of-distribution low-resource speech subsets (Shipibo–Conibo and Chintang). To uphold data sovereignty and our ethical commitments, these datasets are evaluated via a secure, server-side process where neither the raw audio nor the labels are publicly exposed. This approach adheres to the IRB-approved protocols and informed consent under which the data were collected, respecting their status under community custodianship and the principles of the Nagoya Protocol. In contrast, all other datasets in the benchmark are publicly available under their original licenses (Appendix D.4). This governance model allows for rigorous out-of-distribution evaluation while protecting the rights of the data communities.

REPRODUCIBILITY STATEMENT

We release code for preprocessing, feature extraction, distance computation, and evaluation (P@k, GSR), with deterministic pipelines and fixed seeds for stochastic components (e.g., UMAP, permutation baselines). Configuration files reproduce all tables and figures; processed datasets and a public leaderboard are provided (Appendix B).

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

## LLM Usage

Large Language Models were used as a writing aid and code assistant throughout the preparation of this manuscript. Specific uses included: (1) Rephrasing sentences and paragraphs to improve clarity and flow. (2) Correcting grammar and spelling. (3) Suggesting alternative phrasings for technical concepts. (4) Providing code snippet suggestions. The core scientific ideas, experimental design, and interpretation of results are entirely our own. LLMs were not used for research ideation or generating substantive content.

## A   Appendix: Supplemental Details

This appendix provides supplemental information to support the main text. Appendix B details the public URLs for all code, data, and leaderboards. Appendix D details the sources and preprocessing of the 19 corpora aggregated in VocSim and provides justification for our aggregation methodology. Appendix F provides detailed descriptions of our evaluation metrics, including justification for the GSR metric's design and robustness, our empirical baseline methodology, and an analysis of metric correlations. Full, detailed results for all configurations are provided in Appendix P, followed by further details on applications and benchmark extensibility.

## B   Code, Data, and Leaderboard

All the code to reproduce the results of the paper is available at the following url: `https://github.com/anonymoussubmission0000/vocsim`. The code contains feature extraction, model training (AE/VAE), distance computation, benchmark evaluation (P@k, GSR), clustering analysis, and application benchmarks (avian perception, mouse classification).

The VocSim dataset is available at the following url: `https://huggingface.co/datasets/anonymous-submission000/vocsim`.

A leaderboard for comparing results on the VocSim benchmark will be published after review. The leaderboard will rank submissions primarily by their performance on the blind test sets, reporting P@1, P@5, and GSR as separate columns rather than collapsing them into a single scalar score. This design reflects our view that local retrieval (P@k) and global separability (GSR) are complementary aspects of embedding quality; for the configurations evaluated so far, the same Whisper-based model is top-ranked across all three metrics on the blind sets.

The processed Avian Perception dataset is available at: `https://huggingface.co/datasets/anonymous-submission000/avian-perception-benchmark`.

The processed Mouse Strain dataset is available at: `https://huggingface.co/datasets/anonymous-submission000/mouse-strain-classification-benchmark`.

The processed Mouse Identity dataset is available at: `https://huggingface.co/datasets/anonymous-submission000/mouse-identity-classification-benchmark`.

## C   Glossary of Terms and Abbreviations

This section provides a comprehensive legend for all abbreviations and technical terms used throughout the paper to ensure clarity.

Table 3: Comprehensive glossary of abbreviations.

| Abbreviation | Description |
|---|---|
| **Dataset Subset Categories** | |
| BS | **B**ird **S**yllables: Individual, stereotyped acoustic units from birdsong. |
| BC | **B**ird **C**alls: Shorter, often simpler vocalizations than song syllables. |
| OC | **O**tter **C**alls: Vocalizations from giant otters. |
| HP | **H**uman **P**hones: The smallest units of sound in human speech (e.g., /a/, /k/). |
| HW | **H**uman **W**ords: Spoken words from various speakers and contexts. |
| HU | **H**uman **U**tterances: Complete spoken phrases or sentences. |
| HS | **H**uman **S**ounds: Non-verbal human vocalizations (e.g., coughs, laughter). |
| ES | **E**nvironmental **S**ounds: Sounds from the environment (e.g., siren, rain). |
| **Base Feature Extractors (Models)** | |
| EW | **E**ncoder of **W**hisper (large-v3 model). |
| W2V | **W**av**2Vec** 2.0 (Base model). |
| WLM | **W**av**LM** (Large model). |
| CLP | **CL**A**P** (Contrastive Language-Audio Pre-training) model. |
| EAT | **E**fficient **A**udio **T**ransformer; we use its utterance-level [CLS] token embedding. |
| BEATs | Transformer-based audio encoder pretrained on AudioSet-2M with masked audio modeling. |
| MAE | Masked **A**uto**E**ncoder (AudioMAE). |
| VC / AC | Our custom-trained per-subset **V**ariational Autoen**c**oder / **A**utoen**c**oder. |
| M | Baseline Log-**M**el spectrograms. |
| CC | **C**odebook **C**odes from the EnCodec neural audio codec (concatenated codebook indices). |
| **Pooling and Post-Processing Variants** | |
| MTF | **M**ean over **T**ime and **M**ean over **F**. (F = frequency for spectrograms; feature/channel for hidden states.) EWMTF concatenates these two vectors. |
| TF | **T** first time-step and **F** first frequency/feature. EWTF concatenates these two vectors. |
| T / F | Use only the first time-step vector (T) or the first frequency/feature trajectory (F). |
| ET / EF | For Whisper/WhisperSeg: ET = first frame embedding; EF = values of the first feature/channel across time. |
| EWTF | Whisper encoder features concatenating ET and EF (first-time + first-F). |
| EWMTF | Whisper encoder features concatenating mean-over-time and mean-over-F. |
| ETF / EMTF | Analogous to EWTF/EWMTF for WhisperSeg. |
| D100 / D30 | PCA to 100 or 30 dimensions (label-free, per-subset). |

*Note: For spectrograms, F denotes frequency (Mel bins); for hidden-state sequences, F denotes the feature/channel axis.*

| | |
|---|---|
| **Metrics and Technical Terms** | |
| P@k | **P**recision@**k**: A local metric measuring neighborhood purity. |
| GSR | **G**lobal **S**eparation **R**ate: Our global metric for class separability, normalized to [0,1] (reported as %). |
| NID | **N**earest **I**nter-class **D**istance: The distance to the closest point of a different class. |
| Avg_ID | **Avg**. **I**ntra-class **D**istance: The average distance to all points of the same class. |
| OOD | **O**ut-**o**f-**D**istribution: Data from a different distribution than the training/public data. |
| DRI | **D**ynamic **R**ange **I**ndex: A heuristic for the signal-to-noise ratio within a clip. |

# D DATASET DETAILS

This section provides specifics about the datasets aggregated within VocSim and the preprocessing steps applied.

## D.1 ON THE METHODOLOGY OF BENCHMARK AGGREGATION

VocSim's construction by aggregating multiple corpora is a deliberate methodological choice and a principled strategy shared by many of the most influential benchmarks in machine learning, which are designed to test robust generalization across diverse tasks and conditions. Prominent examples span multiple domains: in natural language processing, benchmarks like GLUE (Wang et al., 2019), SuperGLUE (Wang et al., 2020), and the Massive Text Embedding Benchmark (MTEB) (Enevoldsen et al., 2025) aggregate numerous existing datasets to form a comprehensive evaluation suite. In computer vision, the WILDS collection (Koh et al., 2021) curates datasets specifically to test in-the-wild generalization. This is also a standard practice in the audio domain itself, with both the HEAR

(Turian et al., 2022) and SUPERB (Yang et al., 2021) benchmarks being composed of aggregated tasks from various sources.

Creating a novel, monolithic dataset with VocSim's engineered variability would be prohibitively difficult and resource-intensive. Such an effort would require an immense, multi-disciplinary data collection campaign spanning both pristine lab environments and noisy, uncontrolled field settings; capturing a vast dynamic range of signal durations from milliseconds to seconds; and encompassing fundamentally different vocal production systems, from human phonetics to the avian syrinx. Therefore, aggregation is the only feasible and principled approach to construct a benchmark that can systematically probe the limits of generalization in the way VocSim is designed to do.

## D.2 Pre-training Data Overlap Analysis

To ensure a fair and transparent zero-shot evaluation, we conducted a systematic audit to identify potential overlaps between VocSim's source corpora and the known pre-training datasets of the foundation models we benchmarked. The analysis distinguishes between confirmed overlap (source explicitly listed in training data), likely overlap (e.g., a common public dataset likely scraped for a web-scale corpus that lacks a manifest), and no evidence of overlap. The results are summarized in Table 4.

Table 4: Pre-training data overlap matrix for key models and VocSim's public subsets.

| VocSim Subset | | Foundation Model | | | | | | |
|---|---|---|---|---|---|---|---|---|
| ID | Source Type | Whisper-L-v3 | WavLM-Large | Wav2Vec 2.0 | CLAP | AudioMAE | EAT-Large | BEATs |
| *Animal Vocalizations* | | | | | | | | |
| BC, BS1–5, OC1 | Bird/Otter calls | ○ | ○ | ○ | ○ | ○ | ○ | ○ |
| *Environmental Sounds* | | | | | | | | |
| ES1 | ESC-50 (env. sounds) | ∼ | ○ | ○ | ● | ● | ● | ● |
| *Human Speech & Non-Speech* | | | | | | | | |
| HP, HW1 | LibriSpeech-derived | ∼ | ● | ● | ○ | ○ | ○ | ○ |
| HW2, HU2 | AMI Meeting Corpus | ∼ | ○ | ○ | ○ | ○ | ○ | ○ |
| HU1 | VCTK Corpus | ∼ | ○ | ○ | ○ | ○ | ○ | ○ |
| HS1, HS2 | Vocal imitations (AudioSet-like) | ∼ | ○ | ○ | ● | ● | ● | ● |

**Legend:**
● **Confirmed overlap (explicitly listed in pre-training corpus)**
∼ **Likely overlap (indirect/content-based)**
○ **No evidence of overlap**

**Wav2Vec 2.0:**

- **Base Model**: Trained on LibriSpeech (Panayotov et al., 2015) (960 hours), leading to confirmed overlap with VocSim subsets using LibriSpeech-derived data.
- **Large Model**: Utilizes Libri-Light (Kahn et al., 2020) (53,200 hours), which also overlaps with LibriSpeech-derived subsets.

**WavLM:** Involves LibriSpeech, Libri-Light, GigaSpeech (Chen et al., 2021), and VoxPopuli (Wang et al., 2021), causing likely overlap with any English-based speech datasets within VocSim.

**Whisper and WhisperSeg** Pre-trained on a broad multilingual dataset ( 680,000 hours, including 563,000 hours of English) scraped from the web, suggesting potential overlap with diverse human speech subsets. WhisperSeg's fine-tuning with animal vocalizations introduces specialized capabilities but does not use specific subsets.

**CLAP** Likely overlap in environmental sounds like ESC-50, given its use of AudioSet and data from sources such as Freesound (Font et al., 2013), which include public audio clips.

**AudioMAE, BEATs, and EAT:** All these models were trained using AudioSet-2M, creating likely overlaps with environmental sounds represented in VocSim from general audio sources.
**EnCodec:** Utilizes datasets like DNS Challenge (Dubey et al., 2023), Common Voice Ardila et al. (2020), AudioSet, and others, suggesting potential, though indirect, overlap with general audio categories.

### D.3 Comparison to Existing Benchmarks

Table 5: Comparison of VocSim with other major audio benchmarks and datasets. VocSim is uniquely positioned as a diagnostic tool for the intrinsic, zero-shot geometric quality of single-source audio representations, complementing benchmarks that focus on downstream adaptability, complex scene analysis, or high-level semantic tasks like music tagging.

| Resource | Primary Purpose | Evaluation Paradigm | Key Challenge |
|---|---|---|---|
| *Evaluation Benchmarks for Model Adaptability* | | | |
| SUPERB Yang et al. (2021) / HEAR Turian et al. (2022) | Benchmark Model Adaptability | Fine-Tuning or Linear Probing | Adapting a frozen or fine-tuned encoder to succeed on a wide suite of supervised downstream tasks (e.g., ASR, SID). |
| BirdSet (Rauch et al., 2025) / BIRB (Hamer et al., 2023) / BEANS (Hagiwara et al., 2023) | Bioacoustic Scene Analysis | Supervised (Weak) | Detecting calls in polyphonic soundscapes under domain shift. |
| DCASE Challenge Mesaros et al. (2018) | Evaluate Scene Analysis Systems | Supervised Training (System Competition) | Detecting and classifying overlapping sound events within complex, polyphonic real-world acoustic scenes. |
| **VocSim (Ours)** | **Benchmark Intrinsic Representation** | **Training-Free (Zero-Shot)** | **Generalizing acoustic signature matching across diverse, single-source sounds with varying recording conditions.** |
| *Large-Scale Datasets for Training/Evaluation* | | | |
| AudioSet Gemmeke et al. (2017) | Train/Evaluate Classifiers | Supervised Training | Multi-label classification of events in complex, often polyphonic, video soundtracks. |
| WavCaps Mei et al. (2024), LAION-Audio Wu et al. (2024), Fusion-Audio Chen et al. (2025) | Pre-train Foundation Models | N/A (Training Data) | Learning robust multimodal representations from vast quantities of noisy, unstructured audio-text pairs scraped from the web. |
| MagnaTagATune Law et al. (2009) | Train/Evaluate Music Taggers | Supervised Training | Predicting high-level semantic and musical tags (e.g., "rock", "guitar", "fast") from complex polyphonic music clips. |

### D.4 VocSim Aggregation Sources

**BC − The Vocal Repertoire of the Domesticated Zebra Finch (Elie).** Source: Elie & Theunissen (2020), based on Elie & Theunissen (2015). Contains 3,433 calls from 50 birds across 11 types. License: **CC BY 4.0**. URL: `https://doi.org/10.6084/m9.figshare.11905533.v1`.

**BS1 − Deep Audio Segmenter Dataset (DAS).** Source: Clemens (2021), used for "Steinfath et al. (2021). Contains 473 syllables from 1 bird across 6 types. License:

**CC0 1.0**. URL: `https://data.goettingen-research-online.de/citation?persistentId=doi:10.25625/ZXJJJY`.

**BS2 − Clustered Subset of "Benchmarking Nearest Neighbor Retrieval..." (Tomka).** Source: Tomka, Tomas et al. (2024). Contains 48,411 vocalizations from 4 birds across 36 types. License: **CC BY 4.0**. URL: `http://hdl.handle.net/20.500.11850/673918`.

**BS3 − Bengalese Finch Song Repository (Nicholson).** Source: Nicholson et al. (2022), used by Cohen et al. (2022). Contains >245,000 syllables from 4 birds. License: **CC BY 4.0**. URL: `https://figshare.com/articles/dataset/Bengalese_Finch_song_repository/4805749`.

**BS4 − Automated annotation of birdsong with a neural network that segments spectrograms.** Source: Cohen et al. (2022). Contains Bengalese finch syllables. License: **CC BY 4.0**. URL: `https://doi.org/10.7554/eLife.63853`.

**BS5 − Labeled songs of domestic canary M1-2016-spring (Serinus canaria)** Source: Giraudon et al. (2021). Contains canary syllables. License: **CC BY 4.0**. URL: `https://doi.org/10.5281/zenodo.6521932`.

**ES1 − ESC50: Dataset for Environmental Sound Classification.** Source: Piczak (2015). Contains 2,000 recordings across 50 classes (40 clips/class). License: **CC BY 4.0**. URL: `https://doi.org/10.1145/2733373.2806390`.

**HP, HW1 − LibriSpeech Corpus w/ Alignments.** Source: Core data from Panayotov et al. (2015), with alignments generated via MFA McAuliffe et al. (2017). Contains segmented phones (HP) and words (HW1) derived from read English speech. License: CC-BY-4.0. URL: `https://huggingface.co/datasets/gilkeyio/librispeech-alignments`.

**HU1 − CSTR VCTK Corpus.** Source: Yamagishi et al. (2019). Contains segmented utterances (HU1) from English speakers with various accents. License: CC-BY-4.0. URL: `https://huggingface.co/datasets/CSTR-Edinburgh/vctk`.

**HS2 − Vocal Sketch & Vocal Imitation Set (VocImSet).** Source: Cartwright et al. (2018); Kim & Pardo (2018), used in Cartwright & Pardo (2015); Kim & Pardo (2018). Contains 10,690 vocal imitations after filtering. Licenses: **Open** (Vocal Sketch), **CC BY 4.0** (VocImSet). URLs: `https://doi.org/10.5281/zenodo.1251982`, `https://doi.org/10.5281/zenodo.1340763`.

**HS1, HW2, HU2 − AMI Meeting Corpus.** Source: Carletta et al. (2005). Contains 100 hrs of meetings with segmented vocal sounds (HS1), words (HW2), utterances (HU2). License: **CC-BY-4.0**. URL: `https://groups.inf.ed.ac.uk/ami/corpus/`.

**HW3, HU3 − Shipibo-Conibo Language Corpus (ACQDIV).** Source: Stoll & Bickel (2013). Contains 75,000 transcribed utterances (words HW3, utterances HU3). Status: **Blind Test Set (Non-public)**. URL: `https://www.acqdiv.uzh.ch/en/resources.html`.

**HW4, HU4 − Chintang Language Corpus (ACQDIV).** Source: Stoll & Bickel (2013). Contains >1M transcribed words (words HW4, utterances HU4) from 90 sessions used here. Status: **Blind Test Set (Non-public)**. URL: `https://www.acqdiv.uzh.ch/en/resources.html`.

**OC1 − The Vocal Repertoire of Adult and Neonate Giant Otters (Pteronura brasiliensis).** Source: "Mumm & Knörnschild (2014). Contains 441 recordings across 21 call types. License: **CC BY 4.0**. URL: `https://doi.org/10.1371/journal.pone.0112562`.

D.5    VOCSIM PREPROCESSING

The following steps were applied after aggregating the source datasets:

1. **Resampling:** All audio waveforms were resampled to 16 kHz using 'torchaudio'.

2. **Outlier Removal:** For each subset, audio samples with durations in the top 2% (98th percentile) were removed to exclude extreme outliers that might skew results.

3. **Minimum Class Size:** Classes (defined by the 'label' column) containing fewer than six samples were removed. This ensures that metrics like P@k (especially k=5) and GSR are meaningful and that classes have sufficient examples for robust comparison.

4. **Subset Size Capping (for AE/VAE training):** If a subset contained more than 10,000 samples after the above steps, a random selection of classes was performed to bring the sample count to approximately 10,000-17,000, while maintaining class distributions as much as possible. This step was primarily to manage training time for the subset-specific AE/VAE models. The benchmark evaluation itself uses the data after steps 1-3.

5. **Frequency Filtering (for some sources):** The removal of top-frequency classes is applied only to large, public speech corpora (e.g., LibriSpeech, AMI) to mitigate extreme class imbalances where a few stop words would otherwise dominate the dataset. The blind test sets, sourced from low-resource languages, do not exhibit such extreme distributions and are therefore used in their entirety to ensure a realistic evaluation scenario, as detailed below:

    - **BC (Elie Finch Calls):** Calls '10.wav', '16.wav' removed (considered outliers/non-standard).
    - **BS5 (Canary Syllables):** Classes 'TRASH', 'SIL', 'call' removed.
    - **VCTK and LibriSpeech (HP/HW1/HU1):** No phone classes removed (HP). Top 50 most frequent words removed from HW1. Top 12 most frequent utterances removed from HU1. This addresses extreme class imbalance common in speech corpora.
    - **AMI (HS1/HW2/HU2):** Top 3 vocal sounds (laugh, other, cough) removed from HS1. Top 25 words removed from HW2. Top 35 utterances removed from HU2. Again, this mitigates extreme frequency imbalances.

6. **Unique Labeling for Birds:** For birdsong subsets (BS1-BS5), labels were made unique per individual bird (e.g., $bird1\_syllA$, $bird2\_syllA$) as syllable labels often lack consistent acoustic meaning across different individuals. This ensures comparisons are made within an individual's repertoire unless explicitly comparing across birds (not the primary focus here).

7. **Final Format:** The processed data was structured and saved using the Hugging Face 'datasets' library format.

The overall preprocessing pipeline is summarized in Algorithm 1.

### D.6 JUSTIFICATION FOR ASYMMETRIC PREPROCESSING OF SPEECH CORPORA

Our preprocessing pipeline includes a step to filter out the most frequent words and utterances from large, public English speech corpora (LibriSpeech, AMI) but does not apply this filtering to the low-resource blind test sets (Shipibo-Conibo, Chintang). This is a deliberate methodological choice to ensure a meaningful evaluation.

Large, general-language corpora exhibit a Zipfian distribution where a few common function words (e.g., "the," "a," "is") account for a massive fraction of the data. Without filtering, the benchmark would disproportionately reward models for recognizing these few, acoustically varied, and often short tokens, rather than testing performance across a broader vocabulary. This filtering ensures the task remains a challenging test of content identity over a diverse set of words.

The blind test sets, being smaller and sourced from documentation of low-resource languages, do not suffer from this same pathological imbalance. Their word frequency distribution is a natural property of the collected corpus. Applying a frequency filter here would be inappropriate, as it would artificially alter the nature of the data and remove what might be scientifically relevant, frequent vocalizations. This asymmetric approach ensures that both the public and blind evaluations are as fair and representative as possible for their respective data types.

---

**Algorithm 1** Data Preprocessing Procedure

---

**Require:** Aggregated dataset $D$ containing multiple subsets.
**Ensure:** Processed dataset $D_{proc}$.
1: $D_{proc} \leftarrow \emptyset$
2: **for** each subset $S$ in $D$ **do**
3:     Resample all audio in $S$ to 16 kHz.
4:     Remove samples with duration $> 98$th percentile duration in $S$.
5:     Identify classes $C_S$ in $S$.
6:     Remove classes $c \in C_S$ where $|c| < 6$.
7:     Apply source-specific frequency/class filtering (e.g., LibriSpeech-aligned, VCTK, AMI).
8:     **if** $S$ is a birdsong subset **then**
9:         Make labels unique per bird.          ▷ e.g., '$bird1\_syllA$'
10:    Let $S_{filtered}$ be the resulting subset.
11:    **if** $|S_{filtered}| > 17000$ **then**         ▷ Optional step for AE/VAE training efficiency
12:        Rank classes in $S_{filtered}$ by size descending.
13:        Uniformly select classes to form $S'_{filtered}$ with $|S'_{filtered}| \approx 10k - 17k$.
14:        $S_{final} \leftarrow S'_{filtered}$
15:    **else**
16:        $S_{final} \leftarrow S_{filtered}$
17:    Add $S_{final}$ to $D_{proc}$.
18: **return** $D_{proc}$

---

### D.7 VAE AND AE TRAINING DETAILS

Our evaluation includes custom unsupervised models to serve as strong, domain-specific baselines. All models were trained without access to any labels.

**Per-Subset Models (Domain-Specific Baselines)** The primary custom baselines reported in our results (labeled **VC** for VAE and **AC** for AE) were trained on a per-subset basis. For each of the 19 subsets in VocSim, including the blind test sets, a separate AE and VAE model was trained exclusively on the unlabeled audio from that specific subset. This approach does not create a single generalist model; instead, it establishes a strong, domain-specific performance baseline for each task. This allows us to directly quantify the benefit of large-scale pre-training by comparing foundation models against a simple unsupervised model that is perfectly adapted to the target domain's acoustic statistics.

**VAE Training** The VAE compresses $128 \times 128$ log-Mel spectrogram patches into a 32-dimensional Gaussian latent $(\mu, \sigma^2)$ and reconstructs them via a symmetric decoder. Its loss is the negative Evidence Lower Bound (ELBO), based on Goffinet et al. (2021):

$$\mathcal{L}_{\text{VAE}} = \underbrace{\mathbb{E}_{q(z|x)}[-\log p(x|z)]}_{\text{reconstruction error}} + \underbrace{D_{\text{KL}}\big[q(z|x) \,\|\, \mathcal{N}(0, I)\big]}_{\text{latent regularization}}.$$

Training hyperparameters:

- Optimizer: Adam, $\alpha = 1 \times 10^{-3}$, betas $(0.9, 0.999)$
- Batch size: 64 spectrogram chunks (50% overlap)
- Epochs: up to 50, with early stopping after 10 epochs without ELBO improvement
- Spectrogram frontend: 512-sample FFT, 256-sample hop, 128-band Mel filter, log-scaling

Because the KL term enforces a smoothly varying latent space, reconstructions preserve overall patterns but exhibit modest smoothing of fine spectral detail (Figure 3).

**AE Training** The AE encodes full-length Mel spectrograms into a 256-dimensional bottleneck and decodes them back. Its objective is pure reconstruction with L1 loss plus a small sparsity penalty on the bottleneck, based on Best et al. (2023):

$$\mathcal{L}_{\text{AE}} = \|X - \widehat{X}\|_1 + 0.01 \|Z\|_1.$$

Training hyperparameters:

- Optimizer: AdamW, $\alpha = 3 \times 10^{-4}$, weight decay $1 \times 10^{-2}$
- Batch size: 128 full-spectrograms
- Epochs: up to 50, with ReduceLROnPlateau (factor 0.5, patience 5) and early stopping after 10 epochs without loss improvement
- Mixed-precision training enabled for GPU

Because it optimizes only reconstruction fidelity, the AE reproduces fine spectro-temporal details very closely, resulting in sharp, high-fidelity outputs (Figure 4).

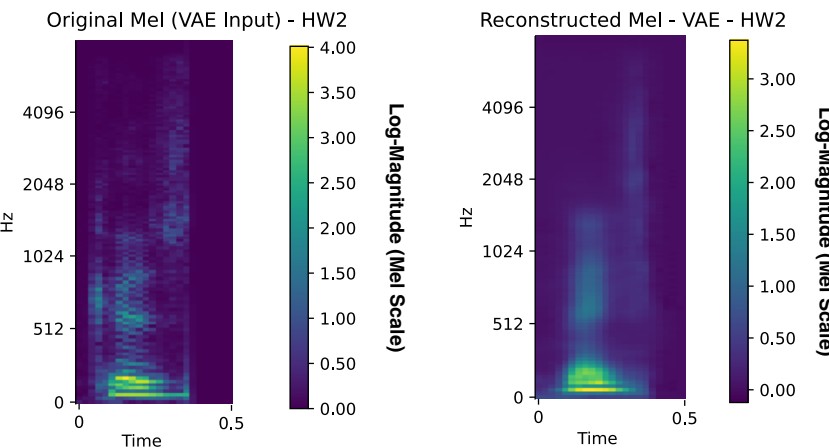

Figure 3: VAE reconstruction (HW2). **Left:** Original log-Mel input. **Right:** Reconstructed output, showing smoothness due to KL regularization.

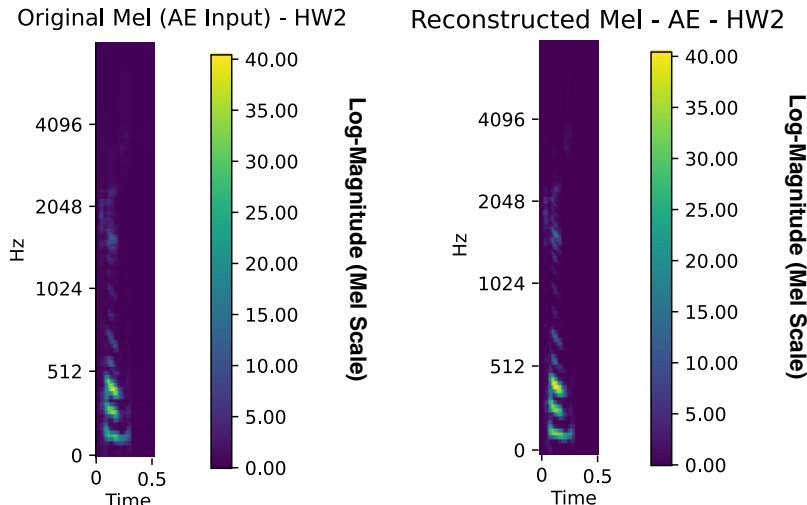

Figure 4: AE reconstruction (HW2). **Left:** Original log-Mel input. **Right:** Decoded output, closely matching fine spectral details.

### D.8   FEATURE EXTRACTION PROCEDURE

All audio clips in VocSim were first resampled to 16 kHz (except where models internally require a different rate) and amplitude-normalized. We then extracted a variety of feature embeddings, some

learned directly on VocSim, others borrowed from large pretrained networks, and, where necessary, applied pooling or dimensionality reduction to obtain fixed-length vectors.

**Log-Mel Spectrograms (M)**   We compute 128-band log-Mel spectrograms with a 512-sample FFT window and 256-sample hop (Hann window), followed by $\log(1 + x)$ compression (Davis & Mermelstein, 1980). Each clip yields a $128 \times T$ time-frequency matrix, which we flatten to a single vector for distance calculations.

**VAE Embeddings (VC)**   Using our convolutional variational autoencoder trained on all VocSim sounds (Appendix D.7), we split each clip into overlapping $128 \times 128$-frame spectrogram patches. Each patch is mapped to a 32-dimensional latent mean vector by the encoder. We obtain 32xT-D representation per clip (Goffinet et al., 2021).

**AE Embeddings (AC)**   Our convolutional autoencoder compresses full-length Mel spectrograms into a 256-D bottleneck. After feeding a clip's spectrogram through the encoder, yielding a 256xT-D vector per clip (Best et al., 2023).

**EAT (EAT)**   We include the Efficient Audio Transformer (EAT) (Chen et al., 2024), a recent spectrogram-based model, as a strong baseline. We use the `worstchan/EAT-large` version fine-tuned on AudioSet-2M. The model processes a normalized log-Mel spectrogram of the input audio. To obtain a single, fixed-length vector representing the entire clip, we take the output embedding of the special `[CLS]` token from the final Transformer layer. This provides a 1024-dimensional utterance-level representation.

**BEATs (BEATs)**   We use the BEATs model (Chen et al., 2023), specifically the `BEATs_iter3_plus_AS2M.pt` checkpoint pretrained on AudioSet-2M. The model internally processes raw audio by first computing a 128-bin log-Mel spectrogram (fbank), which is then normalized using the official preset mean (15.41663) and standard deviation (6.55582). This spectrogram is divided into patches, which are fed into a Transformer encoder. We extract the output from the final encoder layer, which produces a sequence of frame-level embeddings that are then handled by our standard pooling and optional PCA pipeline.

**Whisper Encoder (EW)**   We employ the encoder of OpenAI's Whisper model (`openai/whisper-large-v3`) (Radford et al., 2022). Input audio is padded or truncated to its required 30-second context window. The encoder produces hidden states of shape $[1500 \times 1280]$ (T $\times$ D). We do not run external VAD. The `transformers` implementation uses attention masks so self-attention attends only to valid (non-padded) frames during encoding. In pooling, we evaluate both **Mean (incl. pad)** and **Mean (mask-excluding)**; our main tables use the same default throughout the paper, and the ablations report both variants side-by-side.

**1. Encoder Self-Attention:** The `transformers` library implementation correctly uses an attention mask. This ensures that when the model computes the representation for a valid audio frame, its self-attention mechanism *only* attends to other valid audio frames and ignores the padded regions. This is critical for generating a meaningful contextual representation of the audio content itself.

**2. Post-Hoc Pooling:** Our simple post-hoc pooling operations (e.g., mean over the time axis) are applied to the full $[1500 \times 1280]$ output sequence without an explicit mask. The output vectors corresponding to padded input frames are not zero; they are valid "padding embeddings" that the model learns to produce for silent or non-audio inputs. Including these vectors in a simple mean pooling operation has a mild regularizing effect, pulling the final vector slightly towards a neutral, "no-audio" representation based on the clip's duration. While masked pooling (which would involve explicitly excluding these padding embeddings from the mean calculation) is a valid alternative, our approach serves as a simple, common, and empirically effective heuristic for creating fixed-length representations from the variable-length encoder outputs.

**WhisperSeg Encoder (E)**   A CTranslate2-converted Whisper variant with a custom voice-activity detection front end (Gu et al., 2023) processes each clip into a sequence of $[750 \times 1280]$ (T $\times$ D) embeddings. This model is tuned for animal sounds and delivers frame-level features comparable

to standard Whisper. Unless otherwise noted, we use the same default mean variant as in our main tables and report both **Mean (incl. pad)** and **Mean (mask-excluding)** in ablations.

**Wav2Vec 2.0 (W2V) and WavLM (WLM)**    We extract contextualized features from pretrained Wav2Vec 2.0 (768-D per frame) (Baevski et al., 2020) and WavLM (768-D per frame) (Chen et al., 2022). Raw waveforms are fed directly, and the final Transformer layer's activations are used as our sequence features.

**CLAP (CLP)**    From the Contrastive Language–Audio Pretraining model (Elizalde et al., 2022), we take the final audio embedding (typically 512-D) produced for each clip after audio–text joint training on web data.

**AudioMAE (MAE)**    A masked-spectrogram autoencoder (Huang et al., 2023) yields a fixed 768×512 feature map per clip. We flatten this map into a single vector.

**EnCodec Codes (CC)**    Using a pretrained neural audio codec (Défossez et al., 2022), we extract discrete codebook indices (e.g., 8 codebooks × $T$ frames), which we concatenate into a long integer-valued vector representation. We include EnCodec as a representative discrete-token baseline to situate neural codec approaches within our framework. Our primary metrics operate in continuous vector spaces to probe embedding geometry directly; a comprehensive, training-free evaluation tailored to discrete sequences (e.g., token-level edit or alignment distances) entails different design choices and is orthogonal to VocSim's focus on continuous content-identity geometry. We therefore treat EnCodec as a reference point rather than a dedicated sequence-similarity track.

For all encoder models, we extract features from the final Transformer layer's activations, a standard practice for obtaining the most contextually-rich representations in frozen-feature evaluations. For models like EAT and CLAP that already produce a single utterance-level vector per clip (e.g., the [CLS] token in EAT or the final audio embedding in CLAP), we treat this as the fixed-length embedding and optionally apply PCA, without applying any additional temporal pooling.

**Derived Fixed-Length Vectors**    For models that produce a sequence of embeddings, we adopt a canonical shape of [T x D] (Time frames x Feature dimensions). We derive fixed-length vectors using one or more of the following explicit pooling operations:

- **mean_time**: Mean pooling across the time axis (axis 0), yielding a single vector of length **D**.
- **mean_feat**: Mean pooling across the feature axis (axis 1), yielding a vector of length **T**.
- **first_time**: Taking the full feature vector from the first time frame (t=0), yielding a vector of length **D**.
- **first_feat**: Taking the activation values of the first feature dimension (d=0) across all time frames, yielding a vector of length **T**.
- **Concatenated Pooling**: Concatenating vectors derived from two of the above methods, such as concat_mean_time_feat (length D+T) or concat_first_time_feat (length D+T).

After pooling, we optionally apply Principal Component Analysis (PCA) fit on each VocSim subset to produce the final embeddings. Below are two concise tables: one for the base extractors and one for the pooled+PCA variants. Unless otherwise noted, we use **Mean (incl. pad)** as the default pooling for sequential encoders to obtain fixed-length vectors.

Table 6: Base Feature Extractors and Their Raw Output Shapes (canonical $[T \times D]$ format)

| Short | Extractor | Raw Output Shape [T x D] |
|---|---|---|
| M | Log-Mel spectrogram | [T x 128] |
| VC | VAE latent mean | [T x 32] |
| AC | AE bottleneck | [T x 256] |
| EW | Whisper encoder | [1500 x 1280] |
| E | WhisperSeg encoder | [750 x 1280] |
| WLM | WavLM encoder | [T x 768] |
| W2V | Wav2Vec 2.0 encoder | [T x 768] |
| EAT | EAT [CLS] token | [1024] |
| CLP | CLAP audio embedding | [512] (Already fixed-length) |
| MAE | AudioMAE masked-spectrogram features | [512 x 768] |
| CC | EnCodec discrete codes | [T x 8] |

Table 7: Derived Fixed-Length Embeddings via Pooling and PCA

| Base (Shape [T x D]) | Pooling / Variant (Explicit Name) | PCA Dims | Final Length |
|---|---|---|---|
| **Whisper (EW)** [1500x1280] | first_time (ET) | – | 1280 |
| | + PCA (ET D30/D100) | 30/100 | 30 / 100 |
| | first_feat (EF) | – | 1500 |
| | + PCA (EF D30/D100) | 30/100 | 30 / 100 |
| | concat_first_time_feat (EWTF) | – | 2780 |
| | + PCA (EWTF D30/D100) | 30/100 | 30 / 100 |
| | concat_mean_time_feat (EWMTF) | – | 2780 |
| | + PCA (EWMTF D30/D100) | 30/100 | 30 / 100 |
| **WhisperSeg (E)** [750x1280] | first_time (ET) | – | 1280 |
| | + PCA (ET D30/D100) | 30/100 | 30 / 100 |
| | first_feat (EF) | – | 750 |
| | + PCA (EF D30/D100) | 30/100 | 30 / 100 |
| | concat_first_time_feat (ETF) | – | 2030 |
| | + PCA (ETF D30/D100) | 30/100 | 30 / 100 |
| | concat_mean_time_feat (EMTF) | – | 2030 |
| | + PCA (EMTF D30/D100) | 30/100 | 30 / 100 |
| **CLAP / MAE / CC / WavLM / W2V** | + PCA (D30/D100) | 30/100 | 30 / 100 |

**Notes:**

- $T$ = number of time frames;

- "–" under PCA indicates the raw pooling (no dimensionality reduction).

- "first_row_col" concatenates the first time-step and first F; "mean_row_col" concatenates the mean over time and mean over F. F is frequency for spectrograms and feature/channel for hidden-state sequences.

# E NOTES ON SPEARMAN DISSIMILARITY

Spearman's rank-based dissimilarity is computationally heavier than Cosine or Euclidean, but offers two advantages in our setting: (i) reduced sensitivity to feature scaling and marginal distributions induced by pooling and PCA; and (ii) mitigation of hubness in high-D spaces by emphasizing relative order rather than absolute magnitudes ("Radovanović et al., 2010). Its consistent gains across foundation models suggest rank-order relationships are a good fit for content-identity geometry.

# F BENCHMARK METHODS DETAILS

Our evaluation pipeline is deterministic: for a given dataset, feature extractor, and distance metric, the resulting distance matrix and all subsequent benchmark scores are identical on every run. Therefore, run-to-run stochasticity is zero, and we report point estimates. This section details the algorithms for the metrics used in our analysis.

## F.1 PRECISION@K (P@K) ALGORITHM

Precision@k is a standard local metric that measures the purity of an item's immediate neighborhood. It is a direct and intuitive measure of the local coherence of the embedding space.

1. **Input:** A square pairwise distance matrix $D$ of size $N \times N$, a list of class labels $L$ of length $N$, and a set of integers $k$ (e.g., $\{1, 5\}$).

2. **Data Filtering:** Items with no valid labels are excluded from the evaluation.

3. **Per-Point Precision Calculation:** For each evaluable data point $i$:

   3.a. Let $C(i)$ be the class of point $i$.

   3.b. Identify the set $N_k(i)$ of the $k$ nearest neighbors to point $i$ (excluding $i$ itself) based on the distances in $D$.

   3.c. Count the number of neighbors in $N_k(i)$ that share the same class as point $i$.

   $$\text{CorrectNeighbors}_k(i) = |\{j \mid j \in N_k(i), C(j) = C(i)\}|$$

4. **Aggregation:** The final P@k score is the total number of correct neighbors across all points, divided by the total number of neighbors considered ($N_{\text{evaluable}} \times k$). This is equivalent to the average proportion of correct neighbors.

   $$\text{P@k} = \frac{\sum_i \text{CorrectNeighbors}_k(i)}{N_{\text{evaluable}} \times k}$$

5. **Output:** The final P@k score, a value in $[0, 1]$.

## F.2 GLOBAL SEPARATION RATE (GSR) ALGORITHM

The Global Separation Rate (GSR) is a robust, point-wise global metric that averages the local separation of every point in the dataset. It provides a more continuous and outlier-resistant measure than binary, percentile-based methods.

The algorithm is implemented as follows:

1. **Input:** A square pairwise distance matrix $D$ of size $N \times N$, a list of class labels $L$ of length $N$, and a minimum class size parameter `min_class_size` (e.g., 2).

2. **Data Filtering:** Items with no valid labels or belonging to classes with fewer than `min_class_size` samples are excluded from the evaluation.

3. **Per-Point Score Calculation:** For each evaluable data point $i$:

3.a. Let $C(i)$ be the class of point $i$.

3.b. Find the **Average Intra-class Distance (Avg_ID)**: The mean of distances from point $i$ to all other points within its own class.

$$\text{Avg\_ID}_i = \text{Mean}\left(\{d(i,j) \mid j \neq i, C(j) = C(i)\}\right)$$

3.c. Find the **Nearest Inter-class Distance (NID)**: The distance from point $i$ to the closest point from any other class.

$$\text{NID}_i = \min_{k:C(k)\neq C(i)} d(i,k)$$

3.d. Calculate the **Local Separation Score** for point $i$, which ranges from -1 (total overlap) to +1 (perfect local separation).

$$\text{Local\_Score}_i = \frac{\text{NID}_i - \text{Avg\_ID}_i}{\text{NID}_i + \text{Avg\_ID}_i + \epsilon}$$

4. **Calculate Final GSR Score:** The final score is the average of all local scores, normalized to the range [0, 1].

$$\text{GSR}_{\text{norm}} = \frac{1}{2}\left(\frac{\sum_i \text{Local\_Score}_i}{\text{Number of evaluable points}} + 1\right)$$

This score is then multiplied by 100 to be presented as a percentage in all result tables.

$$\text{GSR} = \text{GSR}_{\text{norm}} \times 100$$

5. **Output:** The final normalized GSR score.

### F.3 CLASS SEPARATION RATIO (CSR) ALGORITHM

The Class Separation Ratio (CSR) is a point-wise metric, similar in spirit to GSR, but it compares the nearest inter-class distance to the *furthest* intra-class distance. It evaluates how well a class is separated from others relative to its own internal spread.

1. **Input:** A square pairwise distance matrix $D$ of size $N \times N$, a list of class labels $L$ of length $N$, and a minimum class size parameter `min_class_size`.

2. **Data Filtering:** Items with no valid labels or belonging to classes with fewer than `min_class_size` samples are excluded.

3. **Per-Point Score Calculation:** For each evaluable data point $i$:

3.a. Let $C(i)$ be the class of point $i$.

3.b. Find the **Maximum Intra-class Distance (MID)**: The distance from point $i$ to the *furthest* point within its own class.

$$\text{MID}_i = \max_{j:j\neq i, C(j)=C(i)} d(i,j)$$

3.c. Find the **Nearest Inter-class Distance (NID)**: The distance from point $i$ to the *closest* point from any other class.

$$\text{NID}_i = \min_{k:C(k)\neq C(i)} d(i,k)$$

3.d. Calculate the **Local Class Separation Score** for point $i$, ranging from -1 to +1.

$$\text{Local\_CSR}_i = \frac{\text{NID}_i - \text{MID}_i}{\text{NID}_i + \text{MID}_i + \epsilon}$$

4. **Aggregation:** The local scores are first averaged within each class. The final score is the weighted average of these per-class scores, weighted by class size.

$$\text{CSR}_{\text{raw}} = \frac{\sum_{c\in\text{Classes}} |c| \times \text{Mean}(\{\text{Local\_CSR}_i \mid i \in c\})}{\text{Total number of evaluable points}}$$

5. **Normalization:** The final score is normalized to the range [0, 1], where 1.0 is best.

$$\text{CSR} = \frac{1}{2}(\text{CSR}_{\text{raw}} + 1)$$

### F.4 F-VALUE BENCHMARK (CS) ALGORITHM

The F-Value, which we abbreviate as CS (Class Separation), is a class-pair-wise metric that measures the ratio of inter-class separation to intra-class compactness. A higher score indicates better separation.

1. **Input:** A distance matrix $D$ and labels $L$.
2. **Per-Class Statistics:** For each valid class $C_i$ (with at least `min_class_size` samples):
   - Calculate the **Average Intra-class Distance**:
   $$\text{AvgIntra}(C_i) = \text{Mean}(\{d(a, b) \mid a, b \in C_i, a \neq b\})$$
3. **Per-Class-Pair Calculation:** For every ordered pair of distinct classes $(C_i, C_j)$:
   - Calculate the **Average Inter-class Distance**:
   $$\text{AvgInter}(C_i, C_j) = \text{Mean}(\{d(a, b) \mid a \in C_i, b \in C_j\})$$
   - Calculate a separation ratio, where a larger value indicates better separation between the class pair.
   $$S_{ij} = \frac{\text{AvgInter}(C_i, C_j)}{\text{AvgIntra}(C_i) + \epsilon}$$
   - To create a bounded score in the range [0, 1] where higher is better, we transform this ratio:
   $$F_{\text{transformed},ij} = \frac{S_{ij}}{1 + S_{ij}}$$
4. **Aggregation:** The final CS score is the mean of all transformed F-Values over all $M \times (M - 1)$ ordered class pairs.

### F.5 CLASS SEPARATION CONFUSION FRACTION (CSCF) ALGORITHM

CSCF is an intuitive, class-pair-wise metric that counts the fraction of "confused" class pairs. A lower raw score is better.

1. **Input:** A distance matrix $D$ and labels $L$.
2. **Per-Class Statistics:** As with the F-Value, calculate $\text{AvgIntra}(C_i)$ for each class $C_i$.
3. **Per-Class-Pair Calculation:** For every ordered pair of distinct classes $(C_i, C_j)$:
   - Calculate $\text{AvgInter}(C_i, C_j)$.
   - A **confusion event** occurs if the average distance between the classes is smaller than the average internal distance of the anchor class $C_i$.
   $$\text{IsConfused}(i, j) = \begin{cases} 1 & \text{if } \text{AvgInter}(C_i, C_j) < \text{AvgIntra}(C_i) \\ 0 & \text{otherwise} \end{cases}$$
4. **Aggregation:** The final CSCF score is the total number of confusion events divided by the total number of ordered class pairs.
   $$\text{CSCF} = \frac{\sum_{i \neq j} \text{IsConfused}(i, j)}{M \times (M - 1)}$$

### F.6 CLUSTERING PURITY BENCHMARK ALGORITHM

This benchmark evaluates how well an embedding's intrinsic structure aligns with the ground-truth labels in a completely unsupervised setting.

1. **Data Preparation:** Start with the fixed-length embeddings for all samples in a subset.
2. **Dimensionality Reduction (UMAP):** Project the high-dimensional embeddings into a 2D space using UMAP (McInnes et al., 2020). This step helps preserve both local and global structure in a space that is more amenable to density-based clustering.

3. **Clustering (HDBSCAN):** Apply the HDBSCAN algorithm (McInnes & Healy, 2017) to the 2D UMAP projection. HDBSCAN is used for its ability to find clusters of varying shapes and densities, and for its robustness in identifying points that do not belong to any cluster (noise).

4. **Weighted Purity Calculation:** The quality of the resulting clusters is measured against the ground-truth labels.

   4.a. For each cluster discovered by HDBSCAN (excluding noise points), identify the majority ground-truth class among its members.

   4.b. The purity of that cluster is the fraction of its members belonging to that majority class.

   4.c. The final **Weighted Purity** is the size-weighted average of these individual cluster purities:

$$\text{Purity}_{\text{weighted}} = \frac{\sum_j |C_j| \times \text{purity}(C_j)}{\sum_j |C_j|}$$

   where $C_j$ is the set of items in the $j$-th cluster found by HDBSCAN.

**Silhouette Score**   The Silhouette Score is a metric that quantifies the quality of clusters by measuring how similar an object is to its own group (cohesion) compared to other groups (separation). In our benchmark, we compute this score not on clusters discovered by an algorithm, but directly on the ground-truth classes. This "supervised" use of the metric serves as a well-established measure of the geometric coherence of the true classes within the embedding space. For each data point, it compares its average distance to other points in its own class with its average distance to points in the nearest neighboring class. A high score indicates that the ground-truth classes are dense and well-separated.

1. **Intra-Cluster Cohesion ($a(i)$):** The average distance between point $i$ and all other points within the same ground-truth class. A low value for $a(i)$ indicates that the point is well-matched to its own cluster.

2. **Inter-Cluster Separation ($b(i)$):** The average distance from point $i$ to all points in the single nearest neighboring cluster (i.e., the cluster that is closest to point $i$, to which $i$ does not belong). A high value for $b(i)$ indicates that the point is well-separated from neighboring clusters.

The silhouette score for point $i$ is then given by the formula:

$$s(i) = \frac{b(i) - a(i)}{\max\{a(i), b(i)\}}$$

The overall score is the mean of $s(i)$ for all points. The score ranges from -1 to +1, where a value near +1 indicates dense and well-separated clusters, a value near 0 indicates overlapping clusters, and a value near -1 suggests that points have likely been assigned to the wrong cluster.

### F.7 P@K BASELINE CALCULATION

To rigorously evaluate whether the observed Precision@k (P@k) scores reflect true class structure or are simply an artifact of the embedding's intrinsic geometry, we established an empirical random baseline using permutation tests. This procedure allows us to determine the P@k score that would be expected by chance for a given embedding space and to test the statistical significance of the observed score.

**Procedure**   The test was conducted for the top-performing configuration (Whisper 'EWMTF D100' embeddings with Spearman distance) on each of the 19 VocSim subsets.

1. **Calculate Observed Score:** For a given subset, the pairwise distance matrix is computed once. The true P@k scores (for k=1 and k=5) are then calculated using this matrix and the ground-truth labels.

2. **Create Null Distribution:** The core of the method involves creating a null distribution of scores that could occur by chance. To do this, we hold the distance matrix fixed and randomly shuffle the ground-truth labels 1,000 times. For each of these permutations, we

recalculate the P@k scores. This process generates a distribution of P@k scores expected under the null hypothesis that there is no relationship between the embedding geometry and the class labels.

3. **Calculate Statistics:** From this null distribution, we compute:

- **The Baseline Mean P@k:** The average of the 1,000 permuted P@k scores, which serves as our empirical random baseline.
- **The 95% Confidence Interval (CI):** The range containing 95% of the permuted scores (from the 2.5th to the 97.5th percentile), indicating the expected spread of random scores.
- **The p-value:** The proportion of permuted scores that were greater than or equal to the originally observed P@k score. A low p-value (e.g., $p < 0.001$) indicates that the observed score is highly unlikely to have occurred by chance.

This analysis provides a robust statistical foundation for interpreting the P@k results. The aggregated results are summarized in Table 8 and Table 9.

Table 8: Empirical Permutation Test for P@1 Significance. Results compare the observed P@1 of the top-performing embedding against a baseline derived from 1000 label permutations per subset.

| Set Type | Observed P@1 (Mean %) | Baseline P@1 (Mean %) | Baseline 95% CI (Mean) | p < 0.001 (Count) |
|---|---|---|---|---|
| Public | 66.80 | 5.80 | [5.36, 6.13] | 15/15 |
| Blind | 11.45 | 0.92 | [0.70, 1.18] | 4/4 |

*Note: Baseline for EWMTF D100 (PCA) with Spearman distance.*

Table 9: Empirical Permutation Test for P@5 Significance. Results compare the observed P@5 of the top-performing embedding against a baseline derived from 1000 label permutations per subset.

| Set Type | Observed P@5 (Mean %) | Baseline P@5 (Mean %) | Baseline 95% CI (Mean) | p < 0.001 (Count) |
|---|---|---|---|---|
| Public | 57.35 | 5.80 | [5.62, 6.01] | 15/15 |
| Blind | 7.67 | 0.91 | [0.80, 1.02] | 4/4 |

*Note: Baseline for EWMTF D100 (PCA) with Spearman distance.*

# G  GSR Baseline Calculation

To rigorously evaluate whether the observed Global Separation Rate (GSR) scores reflect true class structure or are simply an artifact of the embedding's intrinsic geometry, we established an empirical random baseline using permutation tests. This procedure allows us to determine the GSR score that would be expected by chance for a given embedding space and to test the statistical significance of the observed score.

**Procedure**  The test was conducted for the top-performing configuration (Whisper 'EWMTF D100' embeddings with Spearman distance) on each of the 19 VocSim subsets.

1. **Calculate Observed Score:** For a given subset, the pairwise distance matrix is computed once. The true GSR score is then calculated using this matrix and the ground-truth labels.

2. **Create Null Distribution:** The core of the method involves creating a null distribution of scores that could occur by chance. To do this, we hold the distance matrix fixed and randomly shuffle the ground-truth labels 1,000 times. For each of these permutations, we recalculate the GSR score. This process generates a distribution of GSR scores expected under the null hypothesis that there is no relationship between the embedding geometry and the class labels.

3. **Calculate Statistics:** From this null distribution, we compute:

- **The Baseline Mean GSR:** The average of the 1,000 permuted GSR scores, which serves as our empirical random baseline.
- **The 95% Confidence Interval (CI):** The range containing 95% of the permuted scores (from the 2.5th to the 97.5th percentile), indicating the expected spread of random scores.
- **The p-value:** The proportion of permuted scores that were greater than or equal to the originally observed GSR score. A low p-value (e.g., $p < 0.001$) indicates that the observed score is highly unlikely to have occurred by chance.

This analysis provides a robust statistical foundation for interpreting the GSR results. The aggregated results are summarized in Table 10.

Table 10: Empirical Permutation Test for GSR Significance. Results compare the observed GSR of the top-performing embedding against a baseline derived from 1000 label permutations per subset.

| Set Type | Observed GSR (Mean %) | Baseline GSR (Mean %) | Baseline 95% CI (Mean) | p < 0.001 (Count) |
|---|---|---|---|---|
| Public | 41.76 | 24.90 | [24.82, 24.98] | 15/15 |
| Blind | 39.52 | 33.74 | [33.69, 33.80] | 4/4 |

*Note: Baseline for EWMTF D100 (PCA) with Spearman distance.*

## G.1 INTERPRETATION OF METRIC CORRELATIONS

The correlation matrix (Table 11) reveals the relationships between different evaluation metrics. For this analysis, all metrics were transformed such that **higher scores indicate better performance** (e.g., CSCF becomes $1 - \text{CSCF}_{\text{raw}}$). The correlations are calculated using Spearman's $\rho$ and are averaged across all public VocSim subsets.

Table 11: Spearman Rank Correlation ($\rho$) of Key Performance Metrics on Public Subsets. All metrics are transformed such that higher values indicate better performance.

| | GSR | Silhouette | P@1 | P@5 | CSR | CS | CSCF |
|---|---|---|---|---|---|---|---|
| GSR | 1.00 | 0.82 | 0.77 | 0.83 | 0.95 | 0.15 | 0.77 |
| Silhouette | 0.82 | 1.00 | 0.80 | 0.85 | 0.72 | 0.40 | 0.82 |
| P@1 | 0.77 | 0.80 | 1.00 | 0.95 | 0.71 | 0.46 | 0.85 |
| P@5 | 0.83 | 0.85 | 0.95 | 1.00 | 0.75 | 0.49 | 0.90 |
| CSR | 0.95 | 0.72 | 0.71 | 0.75 | 1.00 | -0.05 | 0.74 |
| CS | 0.15 | 0.40 | 0.46 | 0.49 | -0.05 | 1.00 | 0.32 |
| CSCF | 0.77 | 0.82 | 0.85 | 0.90 | 0.74 | 0.32 | 1.00 |

To generate the correlation matrix in Table 11, we followed a two-step process. First, for each feature-distance configuration (e.g., 'EWMTF D100 - Spearman'), we calculated its average score on each metric (P@1, GSR, etc.) across all 15 public VocSim subsets. This yielded a single summary value per metric for each of the configurations evaluated. Second, we computed the Spearman rank correlation ($\rho$) between the vectors of these summary scores. This analysis reveals how the ranking of different embedding methods according to one metric corresponds to their ranking according to another.

**High Correlations ($\rho > 0.8$):** The analysis reveals a cluster of highly inter-correlated metrics, suggesting they measure a similar underlying quality of embedding geometry.

- **GSR vs. CSR (0.95):** This very high correlation is expected. Both metrics assess the integrity of class boundaries relative to internal class spread, with GSR operating on a point-wise basis and CSR on a class-wise basis. Their strong agreement confirms they capture the same fundamental property.
- **P@1 vs. P@5 (0.95):** This is also an intuitive result. An embedding that correctly identifies the single nearest neighbor (high P@1) is highly likely to have multiple correct neighbors within its top five (high P@5).

- **Silhouette, P@5, and CSCF ($\rho \geq 0.82$):** These three metrics exhibit strong positive correlations with each other and with GSR. This indicates that embeddings with good cluster cohesion versus separation (Silhouette) also tend to have pure local neighborhoods (P@5) and well-separated class averages (CSCF).

**Moderate Correlations** ($0.7 < \rho < 0.8$): This group shows solid relationships, reinforcing the connections between different aspects of a well-structured embedding space.

- **Boundary Metrics (GSR/CSR) vs. Neighborhood Purity (P@k):** The correlations here range from 0.71 to 0.83. This demonstrates that embeddings with clear, well-defined class boundaries (high GSR/CSR) reliably produce pure local neighborhoods where a point's nearest neighbors share its class.
- **Boundary Metrics (GSR/CSR) vs. Centroid Separation (CSCF):** With correlations around 0.74 to 0.77, the data shows a strong tendency for embeddings with sharp class boundaries to also have well-separated class averages.

**Low or Near-Zero Correlations:** The most significant insight comes from the CS (Class Separation / F-Value) metric, which behaves largely independently from the other metrics. This highlights that CS measures a distinct geometric property.

- **Consensus Metrics (GSR, CSR, P@k, Silhouette, CSCF):** This large group is moderately to highly inter-correlated, collectively rewarding embeddings that produce distinct, internally consistent clusters with sharp boundaries and pure local neighborhoods.
- **Outlier Metric (CS):** This metric is based on the ratio of the average distance between all inter-class pairs to all intra-class pairs. It is sensitive to the global placement of clusters' "centers of mass" but less so to the integrity of their boundaries.

This distinction explains the uniquely low correlations involving CS:

- **CS vs. CSR ($\rho = -0.05$):** This near-zero correlation is a crucial finding. It demonstrates that achieving sharp class boundaries (high CSR) has no systematic relationship with maximizing the average distance between clusters (high transformed CS). An embedding can produce extremely tight, compact clusters (which is favorable for CSR) while these clusters remain geometrically close to one another, resulting in a modest CS score.
- **CS vs. All Other Metrics ($\rho \approx 0.15 - 0.49$):** CS shows only weak positive correlations with the entire consensus group. This indicates that knowing an embedding's CS score is a poor predictor of its performance on metrics that measure boundary integrity (GSR, CSR), local neighborhood purity (P@k), or cluster cohesion (Silhouette).

In summary, the analysis reveals a strong agreement among six key performance metrics. The CS metric stands apart, measuring a different aspect of embedding quality that is not strongly correlated with the properties measured by the rest.

## G.2 DETAILS ON GLOBAL SEPARATION RATE (GSR)

The formulation of GSR establishes a theoretical neutral point at 50%, corresponding to the mathematical case where the nearest inter-class distance ('NID') equals the average intra-class distance ('Avg_ID'). However, this theoretical point does not represent the performance of a random baseline on a real-world embedding space. Because GSR compares a *minimum* ('NID') with an *average* ('Avg_ID'), its expected value on a structured but randomly labeled space is non-obvious and depends entirely on the embedding's geometry.

A high GSR score requires both clear class boundaries (a large 'NID') and high intra-class compactness (a small 'Avg_ID'). Its significance is therefore best measured by its improvement over an **empirical random baseline**, which must be calculated for each dataset via permutation testing (Appendix G). As our results show, this baseline varies (e.g., 24.9 % for public sets vs. 33.7 % for blind sets), as it is sensitive to the intrinsic geometric structure of the point cloud for each data subset.

This formulation gives GSR two key advantages over the Silhouette Score. First, by using the distance to the single nearest inter-class neighbor (NID), GSR provides a much stricter, **point-wise** test of the

class **boundary** for every single sample. Second, this reliance on the NID makes GSR inherently more robust to the non-convex manifold shapes common in audio, as it does not assume a coherent "neighboring cluster." While each local score is sensitive to outliers, the final GSR metric achieves robustness by averaging thousands of these scores across the dataset, yielding a stable, global measure of class separability.

# H    ABLATION STUDIES AND ROBUSTNESS

**Methodological Robustness.**    Our findings are supported by rigorous validations detailed in this appendix. An exhaustive sweep of all 32 Whisper encoder layers reveals remarkable performance stability across model depth, validating our standard use of the final layer. Further ablations confirm our conclusions are insensitive to the choice of pooling strategy and that our results are strongly corroborated by alternative clustering metrics (NMI, Purity, ARI). This comprehensive validation ensures our results reflect intrinsic embedding properties, not artifacts of our experimental design.

## H.1    ROBUSTNESS TO LABEL NOISE

To evaluate the robustness of our primary metrics to potential annotation errors, we performed a label-noise sensitivity analysis. We systematically introduced noise by randomly flipping a fraction $y\%$ of the class labels, for $y \in \{1, 5, 10, 20\}$, and then recomputed GSR and P@1. The experiment was run on three representative subsets (HP, BC, ES1) using the embeddings from our top-performing models (Whisper, CLAP, and WavLM). The results, shown in Figure 5, demonstrate that GSR is considerably more robust to label noise than P@1. As a local metric, P@1 degrades sharply as even a small fraction of incorrect labels corrupts the immediate neighborhoods of many points. In contrast, GSR, which averages a global signal across all points, degrades more smoothly. For instance, with the Whisper model on the BC subset, a 10% label noise rate causes P@1's performance to drop by 19.2% from its baseline, whereas GSR's performance decreases by only 8.8%. This trend holds across all tested models and subsets. This analysis validates that GSR is not merely a proxy for the mislabeled rate; it provides a more stable measure of an embedding's geometric integrity that is less susceptible to moderate levels of annotation noise, a common challenge in real-world datasets.

## H.2    SEQUENCE POOLING ABLATION

To test sensitivity to the pooling method used to aggregate frame-level features, we compared six strategies for the Whisper encoder. The evaluation was run on three representative subsets, sampling their top 100 classes with 6 clips each. **Main-results default used throughout the paper: Whisper EWMTF D100 with Spearman distance**, where EWMTF = concat(mean_time [mask-excluding] and mean_F), with label-free per-subset PCA to 100D. The strategies are:

- **Mean (mask-excluding):** Mean over valid frames only.
- **Mean (incl. pad):** Mean over all output frames (including padding embeddings).
- **Max Pooling:** Element-wise maximum over frames.
- **First Time:** Only the first frame embedding.
- **Last Time:** Only the last frame embedding.
- **Attention Pooling:** Magnitude-weighted average of frame embeddings.

Both mean variants are shown below for direct comparison.

**Results.**    Table 12 shows that while performance varies slightly, the overall conclusions are stable. Masked mean and max pooling, which consider the entire sequence, are effective and consistent. Time-only strategies (first/last) generally underperform. The relatively narrow performance range across methods indicates that our main findings are not an artifact of a specific pooling choice. Best scores within each subset/column are highlighted in **bold**.

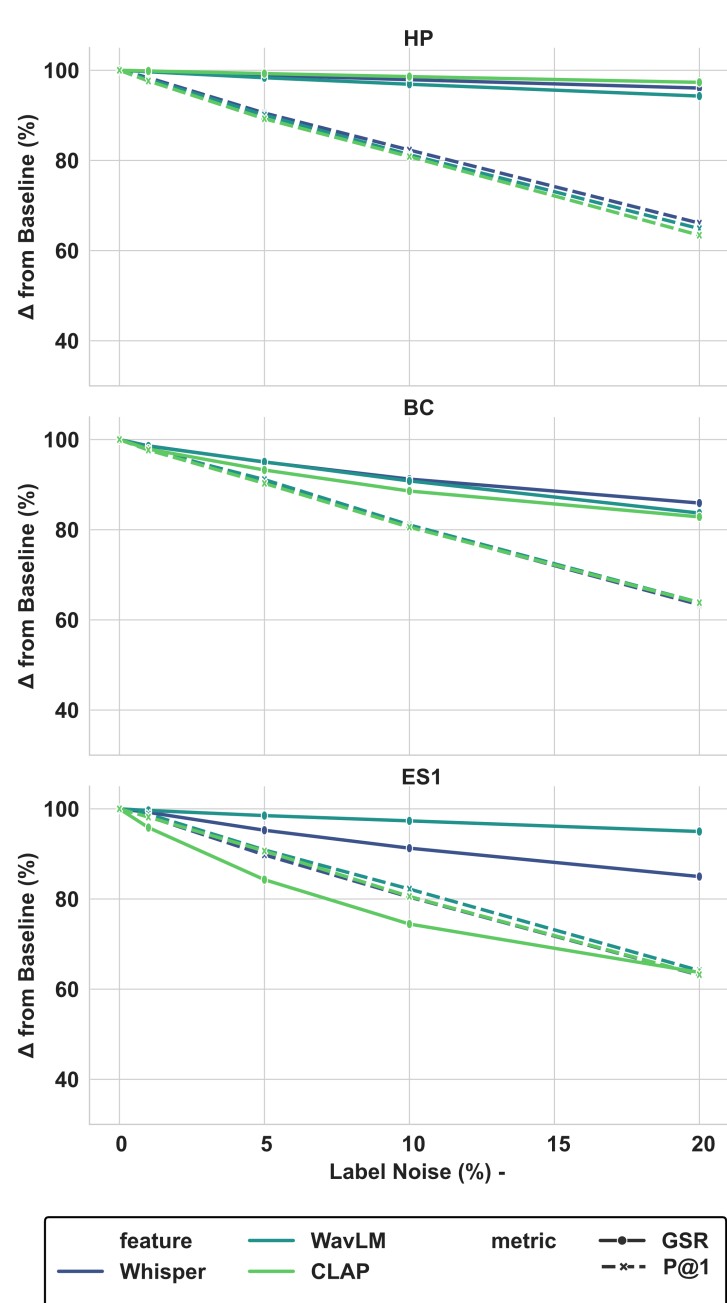

Figure 5: **Robustness to Label Noise: GSR vs. P@1.** Performance of GSR and P@1 as a function of the percentage of randomly flipped labels. Scores are normalized to the performance at 0% noise (100%) to show relative degradation. Across all models (Whisper, CLAP, WavLM) and subsets (HP, BC, ES1), GSR shows a much more graceful decay compared to the steep drop-off of P@1, highlighting its superior robustness to annotation errors.

## H.3 SEQUENCE-AWARE DISTANCE VIA DTW RE-RANKING

We also tested whether temporal pooling discards important sequence information by implementing a sequence-aware baseline using Dynamic Time Warping (DTW).

Table 12: Pooling ablation results per subset (P@k and GSR shown as percentages).

| Subset | Method | P@1 (%) | P@5 (%) | GSR (%) |
|---|---|---|---|---|
| HP | Masked Mean (mask-excl.) | 14.46 | 9.41 | **39.78** |
| | Mean (incl. pad) | **15.9** | **12.1** | 39.4 |
| | Max Pool | 5.15 | 3.97 | 31.60 |
| | First Time | 5.64 | 4.17 | 20.70 |
| | Last Time | 3.92 | 3.14 | 14.37 |
| | Attention Pool | 5.88 | 3.77 | 19.53 |
| BC | Masked Mean (mask-excl.) | 41.07 | 22.02 | **40.50** |
| | Mean (incl. pad) | **65.5** | **58.9** | 36.5 |
| | Max Pool | 42.26 | 24.76 | 37.44 |
| | First Time | 27.98 | 18.21 | 29.69 |
| | Last Time | 10.71 | 11.31 | 22.24 |
| | Attention Pool | 26.79 | 18.45 | 28.78 |
| BS1 | Masked Mean (mask-excl.) | 97.22 | 89.44 | 61.64 |
| | Mean (incl. pad) | **99.2** | **96.1** | 53.3 |
| | Max Pool | 97.22 | 86.67 | **62.17** |
| | First Time | 97.22 | 82.78 | 60.29 |
| | Last Time | 94.44 | 68.33 | 48.70 |
| | Attention Pool | 94.44 | 78.89 | 57.82 |

**Methodology.** We evaluated this baseline on five representative VocSim subsets (HP, BC, BS3, ES1, HU2), each sampled to include 5 clips from the 50 most frequent classes. The process was as follows:

1. **Feature Extraction:** We extracted frame-level features $[T \times 1280]$ from the final layer of a frozen Whisper-large-v3 encoder.

2. **Preprocessing:** Each sequence was truncated based on its true audio duration to remove padding, each frame was L2-normalized, dimensionality was reduced to $D=64$ using label-free PCA, and the sequence was temporally subsampled with a stride of 3.

3. **Candidate Shortlisting:** For efficiency, we first computed a full distance matrix using Spearman distance on the mean-pooled, PCA-reduced vectors.

4. **DTW Re-ranking:** For each query, we identified the top $M=200$ nearest candidates from the pooled distance matrix. We then computed the true sequence distance for only these candidate pairs using multi-dimensional DTW with a Sakoe–Chiba band (radius $r=0.1$) and path-length normalization.

This re-ranking approach preserves the zero-shot protocol by modifying only the distance function, while keeping the encoder frozen and computation tractable.

**Results.** As shown in Table 13, DTW re-ranking did not improve performance on average, and in most cases, it significantly degraded both local precision (P@k) and global separation (GSR). On very short clips (e.g., HP, BS3), the effective sequence length after subsampling was often too short ($\sim$1–2 frames) for alignment to be meaningful. On longer clips (ES1), DTW slightly improved the global metric (GSR +1.1) but drastically hurt local precision (P@1 –20.8). These results strongly suggest that our simple pooling of contextualized frame embeddings is a highly effective and efficient strategy for this task.

Table 13: Average results of the DTW re-ranking ablation across five representative subsets.

| Method | P@1 (%) | P@5 (%) | GSR (%) |
|---|---|---|---|
| Pooled (Spearman) | 38.57 | 22.87 | 41.45 |
| DTW Re-rank (M=200) | 23.66 | 16.16 | 38.59 |
| **Delta (DTW − Pooled)** | **−14.91** | **−6.70** | **−2.86** |

**Per-Subset Breakdown and Computational Cost.** The full per-subset results are provided in Table 14. The average time for a single DTW comparison was 0.6 ms on an NVIDIA RTX 3090 GPU, with a total computation time of ~2.1 minutes for the entire experiment.

Table 14: Per-subset results for the DTW re-ranking ablation. Metrics are P@1 / P@5 / GSR, all in percent.

| Subset | Pooled Baseline | DTW Re-rank | Delta (DTW - Pooled) |
|--------|-----------------|-------------|----------------------|
| HP  | 14.8 /  9.3 / 36.8 |  7.2 /  5.5 / 33.4 | $-7.6$ / $-3.8$ / $-3.4$ |
| BC  | 35.7 / 19.9 / 41.4 | 12.9 /  7.6 / 38.1 | $-22.8$ / $-12.3$ / $-3.3$ |
| BS3 | 67.1 / 42.3 / 47.0 | 66.2 / 45.2 / 43.4 | $-0.9$ / $+2.9$ / $-3.6$ |
| ES1 | 47.6 / 29.4 / 42.7 | 26.8 / 17.8 / 43.8 | $-20.8$ / $-11.6$ / $+1.1$ |
| HU2 | 27.6 / 13.5 / 39.3 |  5.2 /  4.6 / 34.2 | $-22.4$ / $-8.9$ / $-5.1$ |

### H.4 CLUSTERING-BASED EVALUATION: NMI, PURITY, AND ARI

To provide an alternative, clustering-based view of representation quality, we followed the protocol of prior work like HuBERT Hsu et al. (2021). We ran k-means clustering (with k set to the number of true classes) on the frozen embeddings of our top models. The resulting clusters were then evaluated against the ground-truth labels using Normalized Mutual Information (NMI), Purity, and Adjusted Rand Index (ARI).

**Results.** The results, summarized in Table 15, corroborate the findings from our primary P@k and GSR metrics. The top-performing embeddings according to our main metrics, **Whisper and CLAP**, also achieve the highest scores here, yielding clusters that align well with the true class structure. This demonstrates that the strong geometric separation captured by our main metrics translates directly to meaningful and coherent clusters in a fully unsupervised setting.

Table 15: Average clustering metrics across three representative subsets (HP, BC, BS3).

| Configuration (Average) | NMI (%) | Purity (%) | ARI (%) |
|-------------------------|---------|------------|---------|
| Whisper EWMTF D100 | **64.73** | **57.59** | **31.05** |
| CLAP D100          | 63.95    | 55.34     | 28.74   |
| WavLM D100         | 59.86    | 49.43     | 22.45   |

### H.5 LAYER DEPENDENCE ANALYSIS

To assess the sensitivity of our results to the choice of encoder layer, we performed an exhaustive sweep of all 32 layers of the Whisper encoder on sampled versions of all 19 VocSim subsets.

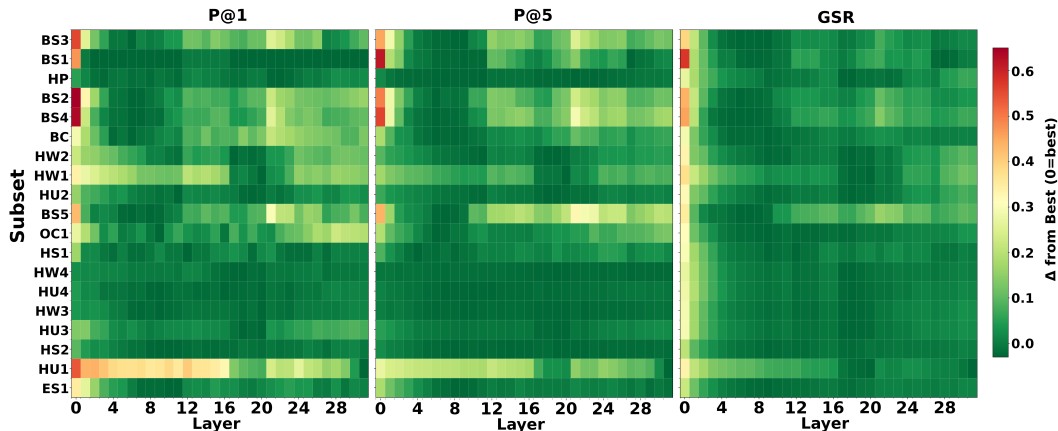

Figure 6: Heatmaps showing the performance drop ($\Delta$) from the best-performing layer (0=best, red=worse) for each metric across all 32 Whisper layers on the sampled subsets. The deltas are consistently small, indicating low sensitivity to layer choice.

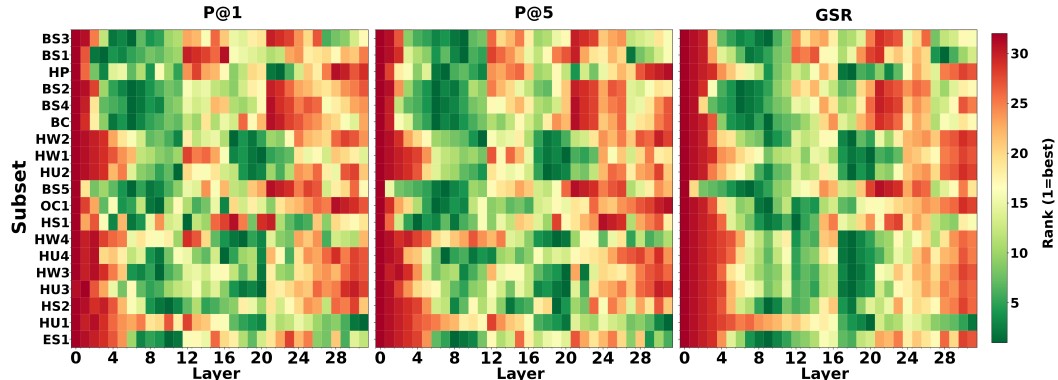

Figure 7: Layer rankings (1=best, 32=worst) for each subset and metric. While middle layers often rank highly, the overall pattern is consistent across metrics.

**Results.** Our analysis reveals that performance is remarkably stable across the encoder's depth. As shown in Figure 6, the performance drop ($\Delta$) from the best-performing layer is uniformly small across all subsets and metrics. While middle layers (approximately 8-20) often rank highest (Figure 7), the potential performance gain from an optimal layer choice is marginal. A comprehensive evaluation that sweeps across every layer for every model and distance metric would be computationally prohibitive and is beyond the scope of this paper's primary contribution. Given that the performance differences are minor, our choice to use the standard final layer for all encoders is a practical and well-justified decision for establishing a consistent benchmark protocol.

# I  COMPUTATIONAL COST AND FEASIBILITY

To address the practical feasibility of using different embeddings, we analyzed the computational requirements for feature extraction. While large foundation models like Whisper offer the best performance, they are computationally intensive and best suited for post-hoc analysis in a server or lab environment rather than real-time, resource-constrained deployment. Our analysis, shown in Table 16, provides a guide to the trade-offs between zero-shot performance and computational cost. The success of PCA compression on large model embeddings offers a valuable pathway to creating smaller, more efficient representations for downstream tasks.

Table 16: Computational analysis of benchmarked models. MACs were estimated using the 'fvcore' library for a 1-second, 16kHz audio input. Peak memory is for inference with batch size 1 on an NVIDIA RTX 3090 GPU. Model names and parameter counts are verified against their official sources.

| Model Pipeline | Parameters (M) | MACs (G/s) | Peak Memory (GB) |
|---|---|---|---|
| *Large-Scale Pretrained Models* | | | |
| Whisper-L-v3 | 635.05 | 953.06 | 6.41 |
| WavLM-Large | 206.30 | 377.19 | 8.10 |
| Wav2Vec2-Base | 89.65 | 201.65 | 8.28 |
| AudioMAE | 85.25 | 43.71 | 0.38 |
| CLAP | 68.55 | 14.94 | 0.83 |
| *Smaller & Custom Models* | | | |
| EnCodec (24kHz) | 7.43 | 44.73 | 0.50 |
| Paper VAE (Custom) | 8.73 | 0.79 | 0.19 |
| Paper AE (Custom) | 1.96 | 1.35 | 0.10 |
| *Baselines* | | | |
| Mel Spectrogram | 0.00 | 0.00 | Minimal |

## J  FULL PERFORMANCE TREND VISUALIZATIONS

The main text analyzes performance trends using the top 5% of configurations for clarity (Figure 2). This appendix provides the corresponding visualizations for broader selections of the data.

Figure 8 illustrates the performance trends for the top 50% of all evaluated feature–distance configurations. Each subplot shows the relationship between performance (P@1, P@5, and GSR) and a specific structural property of the VocSim subsets.

Figure 9 presents the identical analysis but includes all (100%) of the configurations, incorporating the full performance range from the best models down to the weakest baselines.

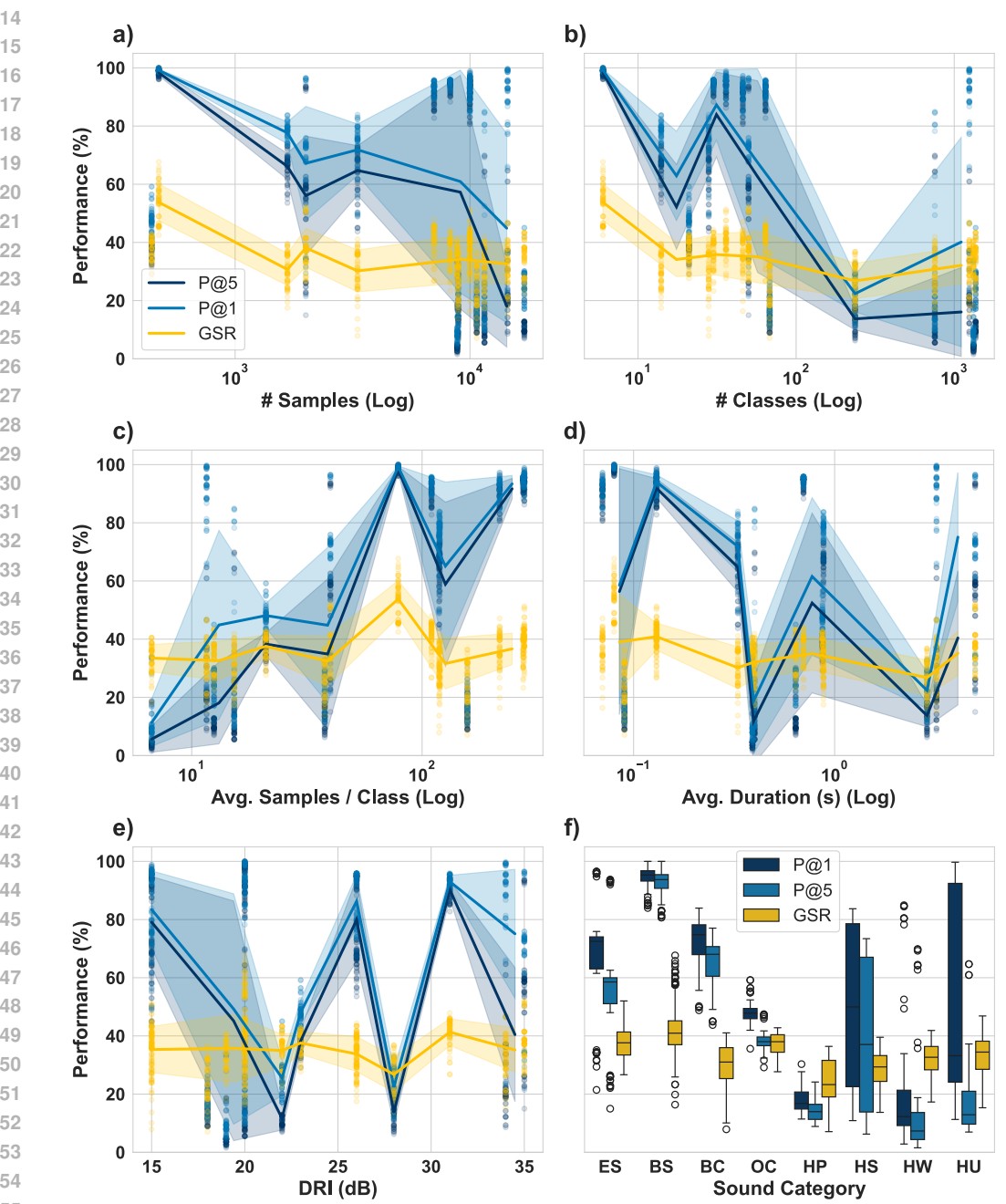

Figure 8: **Generalization trends for the top 50% of configurations.** The patterns observed, such as the sensitivity of P@k to class structure and the stability of GSR, are consistent with the analysis of the top 5% in the main text, albeit with more variance due to the inclusion of less optimal configurations.

## K    EVALUATING CUSTOM MODELS ON VOCSIM

VocSim's evaluation framework is designed to accommodate new audio embedding methods in three main steps: (1) implement a compatible feature extractor, (2) register it in the VocSim configuration, and (3) execute the existing zero-shot pipeline.

**1. Define a Custom Extractor**    Create a class that adheres to the VocSim extractor interface:

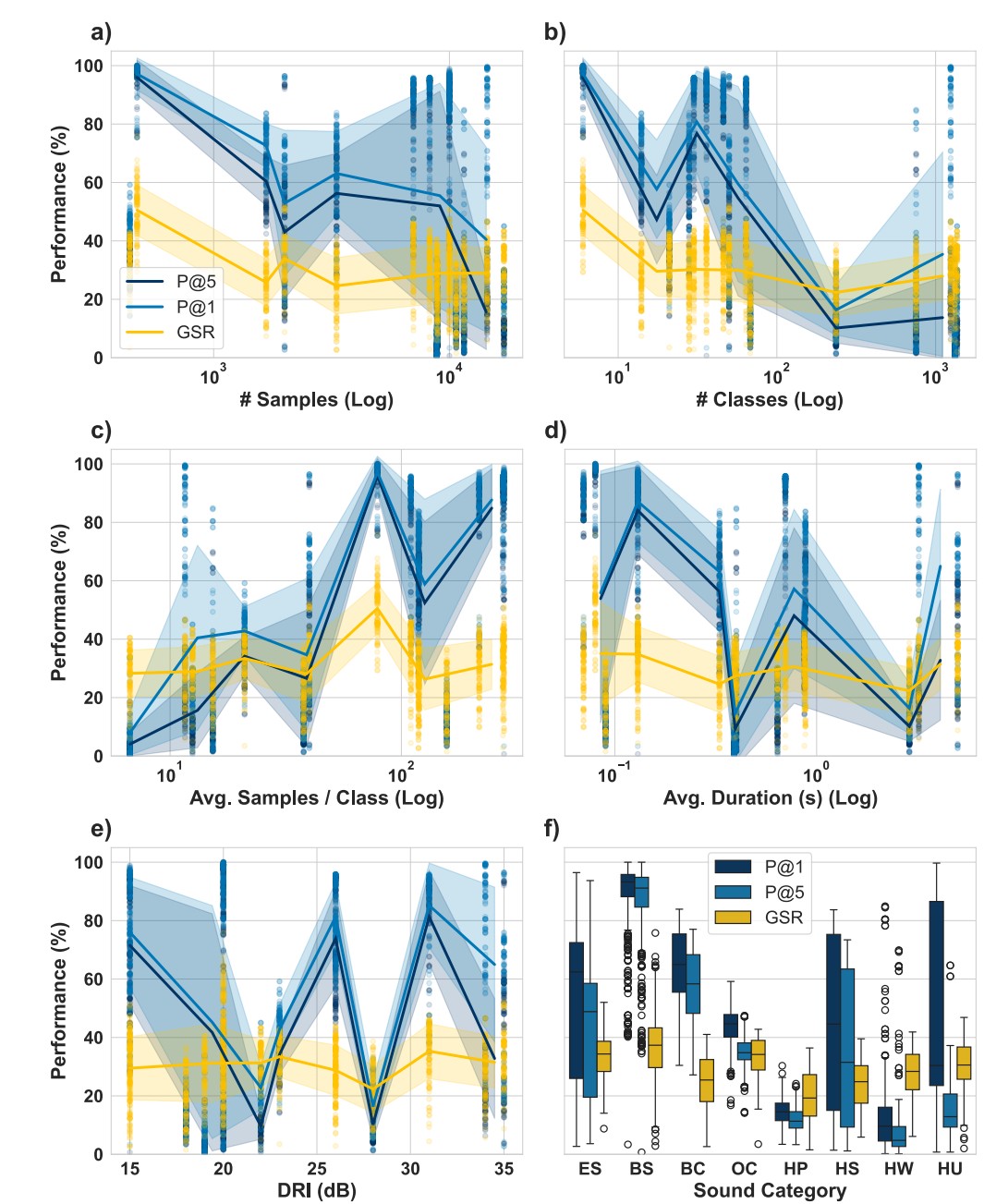

Figure 9: **Generalization trends for all (100%) configurations.** While the inclusion of all baseline and suboptimal methods introduces significant noise and lowers the average performance, the fundamental trends remain visible. This supports our decision to focus on the top 5% for a clearer presentation in the main text.

- It must accept as input a raw waveform (array or tensor) and its sampling rate.

- It must output either

  - A fixed-length vector (1D embedding), or

  - A sequence of frame/patch embeddings (2D array of shape [frames × dim]).

- Optionally, it may produce higher-dimensional structures (e.g., codebook indices, spectrogram patches) that VocSim's pooling routines can reduce.

- The extractor's constructor should handle loading any pretrained weights or model files and placing the model on the correct device.

**2. Configure VocSim to Use the Extractor**   In your VocSim YAML configuration:

- Add an entry under `feature_extractors` with
    - A unique `name` and `short_name` for reporting,
    - The Python import path and class name of your extractor,
    - Any constructor parameters (e.g., model checkpoint path),
    - A list of distance metrics to compute (e.g., `cosine`, `euclidean`).
- To define pooled or PCA-compressed variants, add additional entries that reference the base extractor, specify pooling (e.g., `mean_time`, `first_frame`), and the target PCA dimension.

**3. Run the Zero-Shot Pipeline**   Invoke the VocSim runner with the updated configuration, selecting the steps you wish to execute. This will:

1. *Extract features:* Apply all registered extractors to each VocSim subset.
2. *Compute distances:* Build pairwise distance matrices for each feature–metric pair.
3. *Evaluate benchmarks:* Compute P@1, P@5, and GSR (and any additional benchmarks configured).

**4. Inspect and Share Results**   After completion, VocSim produces CSV/JSON summaries of all metrics per feature and subset. Compare your custom model's scores against existing baselines to assess its zero-shot generalization. We encourage publishing these results on the public VocSim leaderboard to facilitate community comparison and progress.

**5. Blind-Test set Evaluation via GitHub Pull Request**   To include your extractor in the blind-test evaluation, please fork the VocSim repository, add your feature-extractor implementation and updated YAML configuration, and submit a pull request. Once we verify that your code adheres to the interface and configuration requirements, we will run the blind test subsets on your behalf and merge your extractor into VocSim. We will also update the Leaderboard to reflect your model's performance.

## L   PRINCIPLED SCOPE AND LIMITATIONS

VocSim's design involves deliberate, principled choices to ensure a focused and interpretable evaluation. Its primary scope is to measure zero-shot content identity in single-source audio. This focus means other important aspects of audio understanding are explicitly outside its scope.

- **Polyphony and Scene Analysis:** By design, VocSim does not evaluate performance on overlapping sources. This is a deliberate choice to isolate core representation quality from the distinct challenge of source separation.
- **Abstract Semantics and Music:** The benchmark focuses on classes with strong acoustic-to-label consistency. It excludes tasks requiring high-level cultural inference (e.g., music genre) or functional reasoning where acoustic variance within a class is extremely high.
- **Metric Properties:** Our metrics provide a powerful but specific view. P@k is purely local. GSR is a strict, global, worst-case measure for each point; an embedding might have a moderate GSR but still exhibit useful class-average separation, a property not captured by this metric.

**Unifying Content Identity via Acoustic Signatures.**   We acknowledge the conceptual heterogeneity of class labels across the aggregated subsets, which span phones, words, individual-specific animal syllables, and environmental sound categories. This diversity is a deliberate feature of VocSim's design, intended to test an embedding's ability to resolve content identity under a hierarchy of real-world invariances. The unifying principle across all tasks is the matching of a consistent **spectro-temporal profile**, or acoustic signature. For a phone, this signature is highly stereotyped. For a word, the core

signature must be identified across variations in speaker identity, prosody, and coarticulation. For a bird syllable labeled 'bird1_syllA', the task is to recognize that specific acoustic form within the bird's repertoire, correctly treating it as distinct from 'bird2_syllA', which is acoustically unrelated despite the shared arbitrary label. For an environmental sound like "siren," the signature must be robust to different siren types and recording conditions. By collapsing these tasks, VocSim does not propose a single, narrow definition of content; rather, it evaluates an embedding's fundamental ability to form coherent geometric clusters based on the acoustic evidence of the signal's shape, which is the foundational requirement for any higher-level audio understanding.

**Preprocessing and Labeling Schemes details.**   Our preprocessing choices, such as frequency filtering and individual-specific bird syllable labeling, are necessary interventions to ensure a fair and meaningful evaluation. The frequency filtering applied to public speech corpora was a targeted correction for extreme class imbalances not present in the low-resource blind sets. For instance, public corpora like LibriSpeech contain stop words that appear orders of magnitude more frequently than other vocabulary, and without filtering, the benchmark would devolve into a test of recognizing a few common words. The blind sets did not exhibit this pathological imbalance and were therefore used in their entirety.

Similarly, labeling bird syllables as unique per individual (e.g., 'bird1_syllA' vs. 'bird2_syllA') is a methodological necessity. Syllable labels in ornithology are often arbitrary annotations ('A', 'B', 'C') that do not imply any acoustic similarity across different individuals. Treating them as a single class would be scientifically invalid, as it would force a model to merge acoustically distinct vocalizations, creating incorrect and unlearnable intra-class variance. Our approach correctly frames the task as identifying consistent acoustic signatures within a given bird's repertoire, a valid and challenging form of within-category identity matching.

**Standardization of Audio Sample Rate.**   The decision to resample all audio to 16 kHz is a pragmatic and standard practice for evaluating general-purpose audio models. This ensures compatibility with the majority of influential pretrained architectures, including Whisper and Wav2Vec 2.0, which are designed for and expect this sample rate. While we acknowledge that this may discard discriminative high-frequency information for certain animal vocalizations (e.g., harmonics in birdsong above 8 kHz), this is a known trade-off in building a general-purpose benchmark. Crucially, all models are evaluated under these identical conditions, ensuring that their relative performance remains a valid and fair comparison of their ability to process standard-resolution audio. Our downstream validation on mouse ultrasonic vocalizations (USVs), where we successfully employ a custom high-frequency frontend, further demonstrates our awareness of this issue and validates that the top-performing embeddings are powerful feature extractors when paired with an appropriate, domain-specific frontend.

## M   APPLICATIONS: FURTHER DETAILS

### M.1   UNSUPERVISED CLUSTERING

To understand whether embeddings naturally partition sounds by their true categories without any label supervision, we apply a two-stage unsupervised pipeline, UMAP for dimensionality reduction followed by HDBSCAN for clustering, and evaluate the resulting clusters against the ground-truth labels using weighted purity.

**Procedure**

1. **Data Preparation.**
   - Remove any samples lacking valid class labels.
   - If using raw embeddings, replace any NaNs or infinities with finite values (e.g., global min/max) to ensure numerical stability.
   - If using a precomputed distance matrix, confirm its diagonal is zero.

2. **Dimensionality Reduction (UMAP).**
   - Project high-dimensional features (or precomputed distances) into 2D (or another low-dimensional space) using UMAP (McInnes et al., 2020).

- UMAP preserves local and global structure, facilitating clustering in a compact space.

3. **Clustering (HDBSCAN).**

- Run HDBSCAN (McInnes & Healy, 2017) on the UMAP embedding to discover clusters of variable shape and density.
- Parameters such as `min_cluster_size` ensure clusters have meaningful support; points not belonging to any dense region are labeled as noise.

4. **Weighted Purity Calculation.**

- For each non-noise cluster, identify the majority true class among its members.
- Compute the cluster's purity as the fraction of members belonging to that majority class.
- The *weighted purity* is the size-weighted average of these cluster purities:

$$\text{Purity}_{\text{weighted}} = \frac{\sum_j |C_j| \times \text{purity}(C_j)}{\sum_j |C_j|},$$

where $C_j$ is the set of items in cluster $j$.

A high weighted purity (near 1) indicates that the embedding space, when clustered without labels, aligns closely with the true class structure. Conversely, low purity suggests that same-class items are scattered across multiple clusters or mixed with other classes, revealing weaknesses in the embedding's global organization. This unsupervised clustering benchmark complements local metrics (such as P@k) by evaluating the global geometry of the embedding space.

## M.2 Alignment with Avian Perceptual Similarity

This benchmark tests whether zero-shot audio embeddings mirror zebra finches' own judgments of song-syllable similarity, using behavioral data from Zandberg et al. (2024) (Zandberg et al., 2024). In that study, finches performed a two-alternative-forced-choice (2AFC) task, associating each probe syllable $X$ with one of two training sounds ($A$ or $B$), and also yielded derived triplets $(A, P, N)$ where $P$ was judged closer to anchor $A$ than $N$. Their measurements establish an empirical ceiling ($\approx$80–90% accuracy) based on bird–bird consistency.

We evaluate our embeddings against the high-consistency subset of these finch judgements in two tasks:

**Probe (2AFC) Task** For each trial $(X, A, B)$ with bird decision $D \in \{A, B\}$:

1. Look up distances $d(X, A)$ and $d(X, B)$ in the precomputed distance matrix.

2. The model "chooses" the closer sound.

3. *Accuracy* is the fraction of trials where model choice matches $D$.

4. A binomial test (chance=50%) checks significance.

5. Optionally, compute Spearman's $\rho$ and Kendall's $\tau$ between the signed distance difference $d(X, A) - d(X, B)$ and the bird's choice encoded as $+1$ (chose $A$) or $-1$ (chose $B$).

**Triplet Task** For each derived triplet $(A, P, N)$ drawn from high-consistency trials:

1. Retrieve $d(A, P)$ and $d(A, N)$ from the distance matrix.

2. The model "agrees" if $d(A, P) < d(A, N)$.

3. *Accuracy* is the percentage of triplets where the inequality holds.

4. A binomial test assesses significance against 50% chance.

5. Optionally, correlate $d(A, P) - d(A, N)$ with a constant bird-choice indicator (e.g., +1 for each triplet) to quantify rank-order alignment.

In Zandberg et al. (2024), zebra finches themselves agree on the same similarity judgment only about 80–90% of the time, both when different birds are compared (inter-bird consistency, around 80%) and when the same bird is retested on identical probes (intra-bird consistency, up to about 90%). These figures set a practical "ceiling" for any model attempting to mimic avian perception:

- An **80%** model accuracy matches the average agreement level one bird's choices have with another's, indicating the model performs as well as a typical zebra finch on this task.
- A **90%** model accuracy approaches the repeatability of a single bird's own judgments, and so represents a near-maximal alignment with avian perception given the natural variability in behavior.

Thus, a computational embedding that achieves ~80% accuracy is within the expected range of bird–bird agreement, while pushing toward 90% suggests the model captures nearly all of the reliably perceived distinctions.

### M.3 METHODOLOGY FOR ULTRASONIC VOCALIZATION (USV) ANALYSIS

The state-of-the-art results for the mouse USV tasks were achieved by feeding the standard Whisper encoder model a specialized input representation suitable for high-frequency audio.

**Spectrogram Generation Method.** The log-Mel spectrogram was computed directly from the original 250 kHz audio waveforms. This was accomplished by dynamically adjusting the Short-Time Fourier Transform (STFT) parameters to be appropriate for the high sample rate, thereby preserving the spectral information in the ultrasonic range (>20 kHz).

**Implementation.** In our framework, this high-frequency spectrogram generation is implemented within the 'WhisperSegExtractor'. The embeddings used for the downstream classification tasks reported in the main text were generated using this extractor.

### M.4 DOWNSTREAM CLASSIFICATION: MOUSE STRAIN

To assess the practical value of our embeddings in bioacoustic applications, we predict the genetic strain of laboratory mice from their ultrasonic vocalizations (USVs) (Van Segbroeck et al., 2017; Goffinet et al., 2021). This task tests whether embeddings capture strain-specific acoustic cues beyond generic similarity.

**Dataset and Preprocessing** We use a publicly available USV dataset in which each syllable is labeled by mouse strain (e.g., C57 vs. DBA) and the identity of the individual mouse. Audio is segmented into individual syllables and preprocessed as described in Appendix M.3 to generate fixed-length embeddings (e.g., Whisper-based, VAE latents, log-Mel+PCA).

**Classification Protocol** To ensure a rigorous evaluation of generalization to unseen individuals and prevent data leakage, syllables from the same mouse never appear in both training and testing sets of a fold. We employ **group-stratified 5-fold cross-validation** by mouse identity. For each embedding set, we evaluate three off-the-shelf classifiers:

- Standardize features per fold (zero mean, unit variance).
- **k-Nearest Neighbors** (k=3,10,30).
- **Random Forest** (max depth=10,15,20; balanced class weights).
- **Multi-Layer Perceptron** (one hidden layer, L2 regularization $\alpha \in \{0.1, 0.01, 0.001\}$).

Hyperparameters are selected by grid search within each training fold.

**Metrics and Baselines** We report mean Top-1 and Top-5 accuracies (±standard deviation) across folds. As baselines, we reproduce results for spectrogram+PCA and VAE latents from Goffinet et al. (2021) under the same evaluation protocol. Higher accuracy signals that the embedding encodes subtle spectral and temporal markers distinctive of mouse strains.

### M.5 DOWNSTREAM CLASSIFICATION: MOUSE IDENTITY

We further evaluate embeddings by testing whether they capture fine-grained individual signatures in mouse ultrasonic vocalizations (USVs). Each syllable is labeled by the emitting mouse's identity (36 individuals), making this a challenging multi-class task that probes subtle, consistent vocal traits.

**Dataset and Preprocessing**   We use a publicly available USV dataset in which each syllable is tagged with the individual mouse identity. Audio is segmented into discrete syllables and preprocessed as described in Appendix M.3 to generate fixed-length embeddings (e.g., Whisper-based, Mel+PCA, VAE latents).

**Classification Setup**

- **Classifier:** A Multi-Layer Perceptron (MLP) with one or two hidden layers. We sweep L2 regularization strengths ($\alpha \in \{0.01, 0.001, 0.0001\}$) and hidden-layer sizes (e.g., 400 or [200,200] neurons).

- **Cross-Validation:** For the closed-set task of identifying a mouse from a known set of 36 individuals, we used 5-fold stratified cross-validation. This strategy partitions the syllables for each mouse across the folds, ensuring that the training set in each fold contains examples from every identity, and the test set contains held-out syllables from those same identities. This approach correctly tests the model's ability to learn a discriminative signature for each of the known mice and classify new vocalizations from that same set, which is the appropriate methodology for a closed-set identification task.

- **Feature Scaling:** Within each training fold, embeddings are standardized to zero mean and unit variance; the same scaling is applied to the test fold.

**Performance Metrics**

- *Top-1 Accuracy*: Percentage of syllables correctly assigned to their true individual.

- *Top-5 Accuracy*: Fraction of cases where the correct identity appears among the classifier's top five predictions.

**Baselines for Comparison**   We compare against the results from Goffinet et al. (2021) (Goffinet et al., 2021), who evaluated spectrogram+PCA, MUPET, and VAE latent features under similar MLP settings. These published accuracies serve as direct reference points. High Top-1 and Top-5 accuracies, substantially above the chance level of $1/36 \approx 2.8\%$, indicate that embeddings encode idiosyncratic vocal characteristics unique to individual mice.

### M.6   HEAR BENCHMARK EVALUATION DETAILS

To provide extensive external validation for our findings, we evaluated our top-performing embeddings (Whisper EWMTF D100 and CLAP D100) on a suite of tasks from the HEAR 2021 benchmark (Turian et al., 2022). This tests the hypothesis that embeddings scoring highly on VocSim's zero-shot similarity tasks are also effective as general-purpose frozen feature extractors for standard downstream classification.

**Methodology.**   We strictly followed the official HEAR evaluation protocol. For each task, we used the provided K-fold cross-validation splits. For each fold, we trained a linear logistic regression classifier on the training splits and evaluated on the test split. The embeddings were first standardized using a 'StandardScaler' fit on the training data. The regularization parameter 'C' for the logistic regression was chosen from {0.01, 0.1, 1.0, 10.0} by evaluating performance on a 20% validation split held out from the training data. For tasks with official train/validation/test splits (e.g., Speech Commands), we tuned 'C' on the validation set and reported final accuracy on the test set after retraining on the combined train+validation sets.

**Results.**   The results, summarized in Table 17, demonstrate state-of-the-art performance for the Whisper EWMTF D100 embedding. It consistently outperforms the strong CLAP D100 baseline and meets or exceeds the top scores from specialized systems on the official HEAR leaderboard. For Speech Commands (5h), we followed the official train/validation/test split, so the accuracy is reported as a single value.

Table 17: Detailed performance on HEAR benchmark tasks. We report mean accuracy (%) $\pm$ standard deviation over the official K-fold splits.

| HEAR Task | Folds | Whisper D100 | CLAP D100 | HEAR SOTA | SOTA Model Ref. |
|---|---|---|---|---|---|
| Beijing Opera Tian et al. (2014) | 5 | **97.65 $\pm$ 4.13** | 97.03 $\pm$ 3.95 | 97.46 | OpenL3 (Cramer et al., 2019) |
| GTZAN Music/Speech Tzanetakis (1999) | 10 | **99.23 $\pm$ 2.31** | **99.23 $\pm$ 2.31** | **99.23** | CP-JKU (Koutini et al., 2022) |
| Gunshot Triangulation Cooper & Shaw (2020) | 7 | **97.92 $\pm$ 3.34** | 94.05 $\pm$ 5.83 | 94.94 | OpenL3 (Cramer et al., 2019) |
| Mridangam Stroke Akshay Anantapadmanabhan et al. (2020a) | 5 | 96.86 $\pm$ 0.93 | 96.42 $\pm$ 0.47 | **97.53** | GURA (Turian, 2022) |
| Mridangam Tonic Akshay Anantapadmanabhan et al. (2020b) | 5 | 89.71 $\pm$ 0.76 | 87.73 $\pm$ 1.04 | **96.55** | RedRice (Dinkel et al., 2023) |
| CREMA-D Emotion Cao et al. (2014) | 5 | **79.28 $\pm$ 0.56** | 58.42 $\pm$ 0.95 | 75.21 | GURA (Turian, 2022) |
| Speech Commands (5h) Warden (2018) | TVT | **98.61** | 68.24 | 97.63 | IUT-CSE (Turian, 2022) |

## N    COMPUTE RESOURCES

All experiments were run on a single workstation with an NVIDIA RTX 3090 GPU (24 GB VRAM), an AMD Ryzen 9 5950X CPU, and 128 GB DDR4 RAM. Reproducing the full pipeline end-to-end requires roughly 6 days ($\approx 144$ h) of wall-clock time, broken down as: *Feature extraction* ($\approx 72$ h), *Distance matrix computation* ($\approx 48$ h), and *Benchmark evaluations (P@k, GSR, clustering, applications)* ($\approx 24$ h).

## O    LIMITATIONS AND FUTURE EXTENSIONS.

VocSim excludes polyphonic mixtures by design; complementary training-free benchmarks could target mixture-aware similarity or separation-invariant matching. Our PCA adaptation is deliberately simple; future work may compare other label-free normalizations (e.g., whitening, ICA) under a training-free constraint. Finally, expanding single-source coverage (e.g., isolated musical notes, percussion) would further probe cross-mechanism generalization.

## P    FULL RESULTS

Table 18: P@1 Results Across Subsets and Distances (↑ better)

| Method | Dist | BC | BS1 | BS2 | BS3 | BS4 | BS5 | ES1 | HP | HS1 | HS2 | HU1 | HU2 | HU3 | HU4 | HW1 | HW2 | HW3 | HW4 | OC1 | Avg | Avg (Blind) |
|---|---|---|---|---|---|---|---|---|---|---|---|---|---|---|---|---|---|---|---|---|---|---|
| EWMTF D100 | S | 76.2 | 99.2 | 95.8 | 93.6 | 93.4 | 95.1 | 75.9 | **30.3** | 79.5 | 30.5 | **99.7** | 22.0 | 16.3 | **13.3** | 49.4 | 14.3 | **8.6** | 7.6 | 46.7 | **66.8** | **11.5** |
| EWMTF | S | 65.5 | 99.2 | 92.2 | 89.0 | 93.4 | 95.1 | 75.9 | 15.9 | 78.7 | 20.0 | **99.7** | 22.0 | 16.3 | **13.3** | 15.1 | 14.3 | **8.6** | 7.6 | 46.7 | 61.5 | **11.5** |
| BEATs | E | 75.5 | 99.4 | 95.8 | 91.7 | 92.3 | 95.7 | 95.1 | 17.2 | 83.6 | 40.4 | 55.9 | 20.7 | **17.0** | 13.1 | 31.1 | 10.1 | 7.6 | **7.9** | **60.5** | 64.3 | 11.4 |
| BEATs | C | 75.6 | 99.4 | 95.8 | 91.7 | 92.4 | 95.6 | 95.5 | 17.2 | **86.4** | **40.7** | 55.7 | 25.6 | 16.4 | 12.9 | 31.2 | 10.0 | 7.4 | 7.3 | **60.5** | 64.9 | 11.0 |
| EWTF D100 | S | 80.5 | 98.5 | 97.2 | 95.3 | 95.6 | 95.8 | 74.8 | 18.9 | 77.6 | 30.0 | 92.3 | 25.8 | 15.3 | 10.6 | 19.4 | 9.3 | 7.8 | 6.1 | 51.5 | 64.2 | 9.9 |
| EWTF | S | 73.3 | 98.5 | 97.2 | 92.4 | 95.6 | 95.8 | 74.8 | 11.4 | 77.4 | 21.7 | 92.3 | 25.8 | 15.3 | 10.6 | 9.7 | 9.3 | 7.8 | 6.1 | 51.5 | 61.8 | 9.9 |
| EMTF | S | 61.7 | 97.5 | 95.4 | 84.9 | 92.3 | 95.0 | 69.3 | 15.7 | 76.4 | 15.1 | 99.0 | 26.5 | 15.4 | 10.0 | 16.1 | 16.8 | 8.2 | 5.9 | 43.5 | 60.3 | 9.9 |
| EMTF D100 | S | 73.2 | 98.5 | 95.4 | 92.9 | 92.3 | 95.0 | 69.3 | 27.7 | 78.2 | 26.7 | 99.0 | 26.5 | 15.4 | 10.0 | 52.5 | 16.8 | 8.2 | 5.9 | 43.5 | 65.8 | 9.9 |
| EWT | S | 73.2 | 98.5 | 97.2 | 92.1 | 95.7 | 95.8 | 74.1 | 11.6 | 76.9 | 22.4 | 91.9 | 27.7 | 14.9 | 10.0 | 9.8 | 9.3 | 7.8 | 6.4 | 49.2 | 61.7 | 9.8 |
| EWT D100 | S | 79.7 | 98.5 | 97.0 | 95.4 | 95.7 | 95.8 | 74.1 | 18.7 | 79.4 | 29.4 | 91.9 | 27.7 | 14.9 | 10.0 | 19.1 | 9.3 | 7.8 | 6.4 | 49.2 | 64.1 | 9.8 |
| EWTF D100 | E | 78.2 | 99.4 | 96.8 | 95.0 | 95.5 | 95.8 | 73.1 | 15.6 | 80.5 | 29.4 | 88.6 | 23.7 | 13.5 | 11.2 | 17.1 | 7.9 | 7.2 | 5.6 | 47.2 | 62.9 | 9.4 |
| EWTF | E | 75.0 | 99.4 | 95.7 | 93.5 | 95.5 | 95.8 | 73.1 | 11.7 | 79.5 | 24.4 | 88.6 | 23.7 | 13.5 | 11.2 | 10.2 | 7.9 | 7.2 | 5.6 | 47.2 | 61.4 | 9.4 |
| EWTF D100 | C | 79.2 | 99.4 | 96.7 | 95.0 | 95.5 | 95.7 | 72.5 | 15.8 | 80.0 | 29.3 | 88.5 | 24.6 | 13.4 | 11.1 | 17.2 | 7.9 | 7.2 | 5.8 | 47.8 | 63.0 | 9.4 |
| EWTF | C | 75.8 | 99.6 | 97.1 | 93.4 | 95.5 | 95.7 | 72.5 | 11.9 | 79.5 | 24.6 | 88.5 | 24.6 | 13.4 | 11.1 | 10.4 | 7.9 | 7.2 | 5.8 | 47.8 | 61.6 | 9.4 |
| EWT D100 | C | 78.7 | 99.4 | 96.6 | 94.7 | 95.2 | 95.8 | 72.4 | 15.7 | 79.6 | 28.7 | 88.0 | 27.3 | 13.3 | 11.1 | 17.1 | 7.9 | 7.0 | 5.9 | 47.8 | 63.0 | 9.3 |
| EWT | C | 75.3 | 99.4 | 97.1 | 93.3 | 95.2 | 95.7 | 72.4 | 11.8 | 79.8 | 24.1 | 88.0 | 27.3 | 13.3 | 11.1 | 10.3 | 7.9 | 7.0 | 5.9 | 47.8 | 61.7 | 9.3 |
| EWT D100 | E | 78.2 | 99.4 | 96.8 | 94.8 | 95.3 | 95.8 | 73.1 | 15.3 | 80.2 | 28.7 | 88.1 | 18.1 | 13.2 | 10.9 | 17.0 | 7.9 | 6.8 | 5.8 | 47.4 | 62.4 | 9.2 |
| EWT | E | 74.6 | 99.4 | 95.8 | 93.2 | 95.3 | 95.8 | 73.1 | 11.6 | 78.9 | 24.0 | 88.1 | 18.1 | 13.2 | 10.9 | 10.1 | 7.9 | 6.8 | 5.8 | 47.4 | 60.9 | 9.2 |
| ETF D100 | S | 77.7 | 98.9 | 95.8 | 94.4 | 93.7 | 95.7 | 74.3 | 21.9 | 78.2 | 28.4 | 95.4 | 29.7 | 14.3 | 8.7 | 25.2 | 10.9 | 7.3 | 5.1 | 48.5 | 64.6 | 8.9 |
| ETF | S | 69.4 | 98.9 | 95.8 | 89.9 | 93.7 | 95.7 | 74.3 | 15.3 | 76.7 | 20.3 | 95.4 | 29.7 | 14.3 | 8.7 | 11.5 | 10.9 | 7.3 | 5.1 | 48.5 | 61.8 | 8.9 |
| ET | S | 68.6 | 98.9 | 91.4 | 94.5 | 93.9 | 95.7 | 74.0 | 14.7 | 76.3 | 20.0 | 95.3 | 29.6 | 14.0 | 9.0 | 11.6 | 10.6 | 7.2 | 5.2 | 48.5 | 61.6 | 8.8 |
| ET D100 | S | 76.8 | 98.9 | 95.6 | 94.5 | 93.9 | 95.6 | 74.0 | 22.1 | 78.7 | 27.8 | 95.3 | 29.6 | 14.0 | 9.0 | 25.3 | 10.6 | 7.2 | 5.2 | 48.5 | 64.5 | 8.8 |
| EWMTF D100 | C | 68.3 | 98.9 | 93.9 | 91.1 | 91.3 | 94.3 | 69.3 | 20.0 | 79.8 | 23.1 | 95.3 | 28.0 | 11.3 | 10.8 | 26.7 | 6.0 | 5.8 | 5.9 | 46.9 | 62.2 | 8.4 |
| EWMTF | C | 64.9 | 98.7 | 94.6 | 88.7 | 91.3 | 94.3 | 69.3 | 14.0 | 79.2 | 18.8 | 95.3 | 28.0 | 11.3 | 10.8 | 13.2 | 6.0 | 5.8 | 5.9 | 46.9 | 60.8 | 8.4 |
| EW | C | 64.9 | 99.6 | 97.1 | 88.7 | 91.3 | 93.2 | 69.3 | 14.0 | 79.2 | 18.8 | 95.3 | 28.0 | 11.3 | 10.8 | 13.2 | 6.0 | 5.8 | 5.9 | 46.9 | 60.4 | 8.4 |
| EW | E | 65.0 | 99.6 | 94.0 | 88.6 | 91.3 | 93.2 | 67.8 | 13.5 | 78.6 | 18.4 | 95.8 | 17.2 | 11.0 | 10.5 | 12.3 | 6.0 | 5.8 | 5.5 | 45.8 | 59.1 | 8.2 |
| EWMTF D100 | E | 67.8 | 99.2 | 94.0 | 90.9 | 91.3 | 94.1 | 67.8 | 19.7 | 78.6 | 22.6 | 95.8 | 17.2 | 11.0 | 10.5 | 25.9 | 6.0 | 5.8 | 5.5 | 45.8 | 61.1 | 8.2 |
| EWMTF | E | 65.0 | 99.2 | 94.0 | 88.6 | 91.3 | 94.1 | 67.8 | 13.5 | 78.6 | 18.4 | 95.8 | 17.2 | 11.0 | 10.5 | 12.3 | 6.0 | 5.8 | 5.5 | 45.8 | 59.2 | 8.2 |
| ETF D100 | C | 75.9 | 99.2 | 95.6 | 93.9 | 93.5 | 95.8 | 73.0 | 21.0 | 80.1 | 27.5 | 92.9 | 29.4 | 13.7 | 7.6 | 23.5 | 9.7 | 6.5 | 4.9 | 49.4 | 64.0 | 8.2 |
| ETF | C | 71.6 | 99.2 | 95.6 | 91.1 | 93.5 | 95.8 | 73.0 | 16.8 | 78.3 | 22.1 | 92.9 | 29.4 | 13.7 | 7.6 | 13.6 | 9.7 | 6.5 | 4.9 | 49.4 | 62.1 | 8.2 |
| ET | C | 71.1 | 99.2 | 95.4 | 91.0 | 93.3 | 95.8 | 72.5 | 16.6 | 78.4 | 21.9 | 92.8 | 29.2 | 13.7 | 7.6 | 13.6 | 9.5 | 6.4 | 4.7 | 46.9 | 61.8 | 8.1 |
| ET D100 | C | 75.4 | 99.2 | 95.4 | 93.8 | 93.3 | 95.6 | 72.5 | 20.9 | 79.8 | 27.0 | 92.8 | 29.2 | 13.7 | 7.6 | 23.4 | 9.5 | 6.4 | 4.7 | 46.9 | 63.6 | 8.1 |
| CLP D100 | S | 83.0 | 98.9 | 96.9 | 94.4 | 95.2 | 95.8 | 95.7 | 21.3 | 81.3 | 33.0 | 46.6 | 24.1 | 13.5 | 7.9 | 25.3 | 8.9 | 5.8 | 5.2 | 54.9 | 63.7 | 8.1 |
| CLP | S | 76.1 | 98.9 | 96.9 | 91.2 | 95.2 | 95.9 | 95.7 | 15.9 | 81.6 | 26.0 | 46.6 | 24.1 | 13.5 | 7.9 | 17.8 | 8.9 | 5.8 | 5.2 | 54.9 | 61.7 | 8.1 |
| CLP D100 | C | 83.0 | 99.4 | 96.4 | 94.5 | 94.2 | 95.7 | **96.5** | 20.0 | 83.6 | 32.7 | 36.2 | 24.1 | 12.3 | 8.9 | 26.0 | 9.3 | 5.5 | 4.9 | 59.2 | 63.4 | 7.9 |
| CLP | C | 79.8 | 99.2 | 95.3 | 93.0 | 94.2 | **95.9** | **96.5** | 17.7 | 83.7 | 28.9 | 36.2 | 24.1 | 12.3 | 8.9 | 21.3 | 9.3 | 5.5 | 4.9 | 59.2 | 62.3 | 7.9 |
| ETF D100 | E | 76.0 | 99.2 | 95.5 | 93.9 | 93.4 | 95.8 | 73.8 | 20.2 | 80.6 | 27.5 | 93.2 | 29.2 | 12.5 | 7.5 | 23.3 | 9.7 | 6.3 | 5.0 | 50.3 | 64.1 | 7.8 |
| ETF | E | 71.6 | 99.2 | 96.2 | 91.3 | 93.4 | 95.8 | 73.8 | 16.5 | 76.5 | 22.4 | 93.2 | 29.2 | 12.5 | 7.5 | 13.4 | 9.7 | 6.3 | 5.0 | 50.3 | 62.1 | 7.8 |
| ET D100 | E | 75.5 | 99.2 | 95.3 | 93.7 | 93.2 | 95.6 | 73.5 | 20.2 | 79.6 | 27.2 | 93.0 | 28.9 | 12.3 | 7.6 | 23.1 | 9.6 | 6.3 | 4.9 | 49.2 | 63.8 | 7.8 |
| ET | E | 71.7 | 99.2 | 95.3 | 91.0 | 93.2 | 95.8 | 73.5 | 16.3 | 77.7 | 22.1 | 93.0 | 28.9 | 12.3 | 7.6 | 13.3 | 9.6 | 6.3 | 4.9 | 49.2 | 62.0 | 7.8 |
| CLP D100 | E | 82.3 | 99.4 | 96.6 | 94.5 | 94.4 | 95.7 | **96.5** | 19.2 | 82.5 | 32.4 | 36.4 | 24.0 | 11.7 | 8.6 | 26.0 | 8.7 | 5.3 | 4.4 | 56.5 | 63.0 | 7.5 |
| CLP | E | 79.7 | 99.4 | 96.7 | 93.3 | 94.4 | **95.9** | **96.5** | 17.3 | 82.0 | 28.9 | 36.4 | 24.0 | 11.7 | 8.6 | 21.3 | 8.7 | 5.3 | 4.4 | 56.5 | 62.1 | 7.5 |
| EWTF D30 | C | 75.8 | 99.6 | 95.8 | 93.4 | 94.1 | 95.2 | 67.3 | 11.9 | 79.5 | 24.6 | 64.8 | 20.9 | 9.9 | 10.9 | 10.4 | 4.5 | 4.7 | 4.4 | 45.4 | 58.9 | 7.5 |
| EWT D30 | C | 75.3 | 99.4 | 95.8 | 93.3 | 94.1 | 95.2 | 67.6 | 11.8 | 79.8 | 24.1 | 64.1 | 27.3 | 9.9 | 10.7 | 10.3 | 4.6 | 4.8 | 4.4 | 44.9 | 59.2 | 7.4 |
| EMTF | C | 62.1 | 98.1 | 92.5 | 85.1 | 89.1 | 93.9 | 62.6 | 14.6 | 75.1 | 14.4 | 80.6 | 30.1 | 10.5 | 8.9 | 15.0 | 7.6 | 5.4 | 4.0 | 45.8 | 57.8 | 7.2 |
| E | C | 62.1 | 98.1 | 3.4 | 85.1 | 89.1 | 94.7 | 62.6 | 14.6 | 75.1 | 14.4 | 80.6 | 30.1 | 10.5 | 8.9 | 15.0 | 7.6 | 5.4 | 4.0 | 45.8 | 51.9 | 7.2 |
| EMTF D100 | C | 66.3 | 98.7 | 92.5 | 88.9 | 89.1 | 93.9 | 62.6 | 20.1 | 76.3 | 18.6 | 80.6 | 30.1 | 10.5 | 8.9 | 29.6 | 7.6 | 5.4 | 4.0 | 45.8 | 60.0 | 7.2 |
| EWT D30 | E | 74.6 | 99.4 | 95.8 | 93.2 | 94.3 | 95.3 | 67.5 | 11.6 | 78.9 | 24.0 | 65.5 | 18.1 | 9.9 | 9.3 | 10.1 | 4.5 | 4.8 | 4.6 | 42.6 | 58.4 | 7.1 |
| EWTF D30 | E | 75.0 | 99.4 | 95.7 | 93.5 | 94.3 | 95.2 | 67.8 | 11.7 | 79.5 | 24.4 | 66.1 | 19.0 | 10.0 | 9.1 | 10.2 | 4.4 | 4.8 | 4.4 | 43.5 | 58.7 | 7.1 |
| EMTF | E | 61.5 | 99.2 | 93.6 | 84.8 | 89.7 | 93.7 | 61.6 | 13.8 | 74.0 | 13.8 | 80.5 | 19.3 | 10.1 | 8.9 | 13.8 | 7.9 | 5.2 | 3.9 | 42.0 | 56.6 | 7.0 |
| E | E | 61.5 | 99.2 | 95.3 | 84.8 | 89.7 | 94.7 | 61.6 | 13.8 | 74.0 | 13.8 | 80.5 | 19.3 | 10.1 | 8.9 | 13.8 | 7.9 | 5.2 | 3.9 | 42.0 | 56.8 | 7.0 |
| EMTF D100 | E | 66.4 | 99.2 | 93.0 | 88.9 | 89.7 | 93.7 | 61.6 | 19.9 | 74.2 | 18.0 | 80.5 | 19.3 | 10.1 | 8.9 | 29.3 | 7.9 | 5.2 | 3.9 | 42.0 | 58.9 | 7.0 |
| EWTF D30 | S | 73.3 | 98.9 | 95.4 | 92.4 | 93.1 | 94.3 | 65.1 | 11.4 | 77.4 | 21.7 | 59.0 | 20.2 | 9.5 | 9.3 | 9.7 | 4.1 | 4.2 | 3.7 | 47.8 | 57.6 | 6.7 |
| EWT D30 | S | 73.2 | 98.7 | 95.4 | 92.1 | 93.0 | 94.5 | 64.9 | 11.6 | 76.9 | 22.4 | 60.8 | 27.7 | 9.6 | 8.2 | 9.8 | 4.1 | 4.2 | 4.2 | 43.5 | 57.9 | 6.6 |
| ET D30 | C | 71.1 | 99.2 | 93.2 | 91.0 | 91.0 | 94.8 | 68.1 | 16.6 | 78.4 | 21.9 | 76.7 | 29.2 | 9.5 | 7.7 | 13.6 | 5.2 | 4.5 | 3.5 | 44.4 | 59.6 | 6.3 |
| ETF D30 | C | 71.6 | 99.2 | 93.2 | 91.1 | 91.3 | 94.8 | 68.8 | 16.8 | 78.3 | 22.1 | 77.2 | 29.4 | 9.6 | 7.7 | 13.6 | 5.4 | 4.4 | 3.4 | 47.4 | 60.0 | 6.3 |
| CLP D30 | C | 79.8 | 99.2 | 95.3 | 93.0 | 92.7 | 95.5 | 95.8 | 17.7 | 83.7 | 28.9 | 23.7 | 24.1 | 9.7 | 6.9 | 21.3 | 6.9 | 4.6 | 4.0 | 55.3 | 60.9 | 6.3 |
| EWMTF D30 | C | 64.9 | 98.7 | 92.4 | 88.7 | 89.4 | 93.0 | 63.0 | 14.0 | 79.2 | 18.8 | 85.7 | 28.0 | 7.6 | 8.6 | 13.2 | 3.0 | 3.4 | 4.8 | 44.0 | 58.4 | 6.1 |
| ETF D30 | E | 71.6 | 99.2 | 93.4 | 91.3 | 91.6 | 95.1 | 68.8 | 16.5 | 76.5 | 22.4 | 77.9 | 29.2 | 9.3 | 7.0 | 13.4 | 5.1 | 4.4 | 3.3 | 44.4 | 59.7 | 6.0 |
| EWMTF D30 | S | 65.5 | 97.7 | 92.6 | 89.0 | 89.0 | 92.3 | 63.7 | 15.9 | 78.7 | 20.0 | 86.1 | 22.0 | 8.5 | 7.6 | 15.1 | 3.5 | 3.6 | 4.2 | 42.9 | 58.3 | 6.0 |
| ET D30 | E | 71.7 | 99.2 | 93.2 | 91.0 | 91.3 | 95.1 | 68.3 | 16.3 | 77.7 | 22.1 | 77.3 | 28.9 | 9.4 | 6.9 | 13.3 | 5.1 | 4.1 | 3.2 | 42.6 | 59.5 | 5.9 |
| CLP D30 | E | 79.7 | 99.4 | 95.4 | 93.3 | 93.0 | 95.5 | 96.0 | 17.3 | 82.0 | 28.9 | 23.9 | 24.0 | 9.7 | 6.2 | 21.3 | 6.7 | 3.9 | 3.8 | 52.4 | 60.6 | 5.9 |
| CLP D30 | S | 76.1 | 98.9 | 94.0 | 91.2 | 91.6 | 94.9 | 94.8 | 15.9 | 81.6 | 26.0 | 22.1 | 24.1 | 8.7 | 7.0 | 17.8 | 5.3 | 3.6 | 3.8 | 48.5 | 58.9 | 5.8 |
| EWMTF D30 | E | 65.0 | 99.2 | 92.5 | 88.6 | 89.6 | 92.9 | 61.5 | 13.5 | 78.6 | 18.4 | 86.7 | 17.2 | 6.6 | 8.7 | 12.3 | 2.8 | 3.1 | 4.3 | 43.8 | 57.5 | 5.7 |
| EMTF D30 | C | 62.1 | 98.1 | 90.4 | 85.1 | 86.2 | 92.5 | 56.7 | 14.6 | 75.1 | 14.4 | 60.1 | 30.1 | 8.3 | 7.9 | 15.0 | 3.5 | 3.2 | 3.0 | 43.8 | 55.2 | 5.6 |
| ET D30 | S | 68.6 | 98.3 | 91.4 | 89.4 | 89.5 | 94.2 | 65.8 | 14.7 | 76.3 | 20.0 | 70.7 | 29.6 | 8.3 | 7.3 | 11.6 | 4.0 | 3.7 | 3.0 | 43.5 | 57.9 | 5.6 |
| ETF D30 | S | 69.4 | 98.3 | 91.8 | 89.9 | 89.4 | 94.2 | 65.0 | 15.3 | 76.7 | 20.3 | 70.9 | 29.7 | 8.6 | 7.1 | 11.5 | 4.0 | 3.6 | 2.7 | 45.4 | 58.1 | 5.5 |
| EMTF D30 | E | 61.5 | 98.5 | 90.8 | 84.8 | 86.8 | 92.3 | 56.1 | 13.8 | 74.0 | 13.8 | 60.4 | 19.3 | 8.0 | 7.6 | 13.8 | 3.4 | 3.1 | 3.1 | 40.4 | 54.0 | 5.5 |
| EMTF D30 | S | 61.7 | 97.5 | 90.4 | 84.9 | 85.2 | 92.1 | 56.5 | 15.7 | 76.4 | 15.1 | 68.3 | 26.5 | 7.5 | 6.4 | 16.1 | 4.0 | 3.7 | 3.7 | 43.3 | 55.6 | 5.3 |
| WLM | S | 48.8 | 98.9 | 87.8 | 84.0 | 90.0 | 93.2 | 34.4 | 24.1 | 73.0 | 10.9 | 99.4 | 11.3 | 6.8 | 6.9 | 78.3 | 33.7 | 3.0 | 4.4 | 50.1 | 61.2 | 5.3 |
| WLM D100 | S | 58.2 | 98.9 | 92.3 | 89.1 | 90.0 | 92.1 | 34.4 | 26.3 | 75.7 | 14.8 | 99.4 | 42.0 | 6.8 | 6.9 | 84.5 | 33.7 | 3.0 | 4.4 | 50.1 | 65.4 | 5.3 |
| WLM | C | 50.3 | 99.2 | 89.2 | 85.5 | 89.5 | 93.3 | 34.4 | 25.1 | 74.8 | 12.3 | 99.2 | **45.0** | 5.5 | 7.6 | 80.3 | 33.9 | 2.4 | 4.1 | 49.0 | 64.1 | 4.9 |
| WLM D100 | C | 56.6 | 99.2 | 92.2 | 89.0 | 89.5 | 92.2 | 34.4 | 26.3 | 75.8 | 14.7 | 99.2 | **45.0** | 5.5 | 7.6 | **84.9** | 33.9 | 2.4 | 4.1 | 49.0 | 65.5 | 4.9 |
| WLM D100 | E | 55.6 | 99.4 | 92.2 | 89.1 | 89.6 | 92.2 | 34.0 | 26.3 | 74.5 | 14.9 | 99.2 | 30.1 | 4.9 | 7.2 | 84.7 | 33.1 | 2.5 | 3.9 | 46.3 | 64.1 | 4.6 |
| WLM | E | 49.8 | 99.2 | 89.3 | 85.7 | 89.6 | 93.4 | 34.0 | 24.9 | 74.0 | 12.4 | 99.2 | 30.1 | 4.9 | 7.2 | 80.5 | 33.1 | 2.5 | 3.9 | 46.3 | 62.8 | 4.6 |
| M | C | **83.9** | **100.0** | 98.8 | **97.2** | 95.6 | 95.0 | 31.8 | 22.2 | 69.5 | 12.5 | 98.7 | 24.6 | 5.8 | 6.2 | 29.6 | 10.2 | 3.1 | 3.3 | 50.8 | 61.4 | 4.6 |

Table 18: (Continued) P@1 Results Across Subsets and Distances (↑ better)

| Method | Dist | BC | BS1 | BS2 | BS3 | BS4 | BS5 | ES1 | HP | HS1 | HS2 | HU1 | HU2 | HU3 | HU4 | HW1 | HW2 | HW3 | HW4 | OC1 | Avg | Avg (Blind) |
|---|---|---|---|---|---|---|---|---|---|---|---|---|---|---|---|---|---|---|---|---|---|---|
| M | E | 82.4 | 99.8 | **98.9** | 97.2 | **95.9** | 95.1 | 31.4 | 21.3 | 66.9 | 12.4 | 98.5 | 23.5 | 5.1 | 5.8 | 31.2 | 10.5 | 2.5 | 2.9 | 49.2 | 60.9 | 4.1 |
| VC | C | 77.1 | **100.0** | 97.5 | 96.1 | 94.7 | 95.6 | 34.5 | 17.8 | 70.8 | 14.4 | 88.1 | 23.7 | 3.9 | 7.0 | 22.3 | 8.2 | 2.2 | 3.2 | 46.7 | 59.2 | 4.1 |
| VC | E | 76.8 | **100.0** | 97.6 | 95.9 | 94.3 | 95.4 | 35.1 | 17.4 | 67.4 | 14.5 | 81.3 | 19.2 | 4.0 | 6.0 | 21.3 | 7.9 | 2.1 | 3.1 | 48.1 | 58.2 | 3.8 |
| AC | S | 80.0 | **100.0** | 98.0 | 93.1 | 93.8 | 94.7 | 34.2 | 12.0 | 69.3 | 15.3 | 98.6 | 21.1 | 4.8 | 5.6 | 15.5 | 5.7 | 2.3 | 1.8 | 35.6 | 57.8 | 3.6 |
| WLM D30 | C | 50.3 | 99.2 | 89.2 | 85.5 | 86.8 | 90.8 | 30.2 | 25.1 | 74.8 | 12.3 | 94.6 | **45.0** | 3.5 | 5.9 | 80.3 | 27.8 | 1.8 | 3.3 | 45.1 | 62.5 | 3.6 |
| M | S | 78.2 | **100.0** | 97.6 | 96.0 | 94.1 | 93.6 | 21.4 | 14.6 | 67.5 | 11.1 | 98.5 | 22.1 | 3.9 | 5.0 | 19.4 | 8.1 | 2.3 | 2.7 | 43.5 | 57.7 | 3.5 |
| WLM D30 | E | 49.8 | 99.2 | 89.3 | 85.7 | 86.7 | 90.8 | 29.4 | 24.9 | 74.0 | 12.4 | 95.2 | 30.1 | 3.2 | 5.7 | 80.5 | 27.9 | 1.7 | 2.8 | 45.4 | 61.4 | 3.4 |
| WLM D30 | S | 48.8 | 98.9 | 87.8 | 84.0 | 83.8 | 88.6 | 26.7 | 24.1 | 73.0 | 10.9 | 91.3 | 11.3 | 3.4 | 4.7 | 78.3 | 25.6 | 1.7 | 3.2 | 45.6 | 58.6 | 3.2 |
| EF D100 | C | 58.7 | 98.5 | 91.3 | 88.3 | 87.7 | 91.8 | 23.8 | 14.8 | 65.9 | 7.3 | 61.4 | 19.1 | 4.1 | 5.1 | 10.1 | 2.1 | 1.2 | 2.1 | 41.0 | 50.8 | 3.1 |
| EF | C | 56.1 | 98.3 | 91.3 | 85.1 | 87.7 | 91.8 | 23.8 | 12.5 | 66.8 | 6.9 | 61.4 | 19.1 | 4.1 | 5.1 | 7.2 | 2.1 | 1.2 | 2.1 | 41.0 | 50.1 | 3.1 |
| EF | S | 55.6 | 97.5 | 88.4 | 84.3 | 88.2 | 92.3 | 19.1 | 11.6 | 65.2 | 5.8 | 65.0 | 21.7 | 3.1 | 5.1 | 6.7 | 2.6 | 1.5 | 2.5 | 38.8 | 49.5 | 3.0 |
| EF D100 | S | 58.9 | 97.5 | 92.2 | 89.5 | 88.2 | 92.3 | 19.1 | 17.0 | 66.5 | 6.2 | 65.0 | 21.7 | 3.1 | 5.1 | 13.2 | 2.6 | 1.5 | 2.5 | 38.8 | 51.2 | 3.0 |
| AC | C | 77.4 | **100.0** | 97.8 | 93.3 | 93.1 | 94.2 | 33.3 | 11.2 | 68.3 | 13.6 | 94.2 | 20.4 | 3.5 | 4.4 | 15.1 | 5.1 | 1.8 | 1.9 | 35.1 | 56.8 | 2.9 |
| AC | E | 77.3 | **100.0** | 97.7 | 93.3 | 93.1 | 94.1 | 29.8 | 11.0 | 61.5 | 11.6 | 90.1 | 13.1 | 3.6 | 4.3 | 14.6 | 4.9 | 1.8 | 1.9 | 35.6 | 55.2 | 2.9 |
| EF | E | 55.0 | 98.7 | 91.9 | 84.8 | 87.9 | 91.8 | 20.8 | 12.3 | 64.6 | 6.8 | 61.2 | 19.1 | 3.6 | 4.7 | 6.8 | 2.0 | 1.1 | 2.1 | 40.1 | 49.6 | 2.9 |
| EF D100 | E | 57.6 | 98.7 | 91.2 | 88.3 | 87.9 | 91.8 | 20.8 | 14.5 | 64.7 | 7.0 | 61.2 | 19.1 | 3.6 | 4.7 | 9.8 | 2.0 | 1.1 | 2.1 | 40.1 | 50.3 | 2.9 |
| EF D30 | E | 55.0 | 98.5 | 88.8 | 84.8 | 85.6 | 90.5 | 23.2 | 12.3 | 64.6 | 6.8 | 37.5 | 19.1 | 3.3 | 4.7 | 6.8 | 1.6 | 1.0 | 1.8 | 37.9 | 47.5 | 2.7 |
| EF D30 | S | 55.6 | 97.5 | 88.4 | 84.3 | 84.3 | 89.9 | 19.4 | 11.6 | 65.2 | 5.8 | 30.7 | 21.7 | 2.7 | 4.8 | 6.7 | 1.6 | 1.2 | 1.8 | 37.2 | 46.7 | 2.6 |
| EF D30 | C | 56.1 | 98.3 | 88.9 | 85.1 | 85.4 | 90.4 | 24.1 | 12.5 | 66.8 | 6.9 | 37.4 | 19.1 | 3.4 | 4.2 | 7.2 | 1.6 | 1.0 | 1.7 | 37.6 | 47.8 | 2.6 |
| VC | S | 64.5 | 99.6 | 93.9 | 92.4 | 91.5 | 92.0 | 25.0 | 13.3 | 63.8 | 7.7 | 58.7 | 19.2 | 2.6 | 4.1 | 12.0 | 3.5 | 1.0 | 1.8 | 39.7 | 51.8 | 2.4 |
| EWF | S | 56.0 | 98.9 | 92.8 | 86.1 | 87.9 | 88.8 | 21.0 | 10.0 | 67.1 | 5.4 | 89.9 | 19.3 | 2.3 | 3.7 | 3.9 | 1.4 | 1.2 | 1.8 | 42.4 | 51.4 | 2.3 |
| EWF D100 | S | 61.1 | 98.9 | 92.8 | 90.5 | 87.9 | 88.8 | 21.0 | 14.4 | 64.3 | 6.0 | 89.9 | 19.3 | 2.3 | 3.7 | 8.1 | 1.4 | 1.2 | 1.8 | 42.4 | 52.4 | 2.3 |
| MAE | S | 43.8 | 98.7 | 85.2 | 81.1 | 74.8 | 78.1 | 64.6 | 7.0 | 62.5 | 6.9 | 77.1 | 0.8 | 1.9 | 4.1 | 2.2 | 0.9 | 0.8 | 1.9 | 34.7 | 47.9 | 2.2 |
| MAE D100 | S | 50.6 | 98.7 | 85.2 | 85.7 | 74.8 | 80.0 | 64.6 | 9.2 | 64.3 | 11.4 | 77.1 | - | 1.9 | 4.1 | 3.5 | 0.9 | 0.8 | 1.9 | 34.7 | 53.2 | 2.2 |
| W2V D100 | S | 36.0 | 82.0 | 61.1 | 46.6 | 47.6 | 74.4 | 20.1 | 6.5 | 61.4 | 7.1 | 99.3 | 22.5 | 2.3 | 4.2 | 56.7 | 12.6 | 0.9 | 1.3 | 27.7 | 44.1 | 2.2 |
| W2V | S | 32.8 | 75.5 | 56.7 | 41.4 | 47.6 | 74.2 | 20.1 | 5.2 | 59.2 | 5.5 | 99.3 | 22.5 | 2.3 | 4.2 | 39.7 | 12.6 | 0.9 | 1.3 | 27.7 | 41.3 | 2.2 |
| EAT | S | 64.0 | 98.7 | 91.1 | 88.5 | 88.3 | 92.0 | 48.9 | 10.8 | 67.5 | 12.4 | 76.1 | 17.1 | 2.7 | 3.7 | 7.1 | 1.4 | 0.8 | 1.4 | 35.4 | 53.3 | 2.2 |
| EAT | C | 63.4 | 99.2 | 91.1 | 88.1 | 87.7 | 92.1 | 48.8 | 10.4 | 68.3 | 12.6 | 75.7 | 17.1 | 3.0 | 3.7 | 7.0 | 1.4 | 0.8 | 1.4 | 33.8 | 53.1 | 2.2 |
| EWF | C | 55.9 | 98.9 | 88.0 | 86.4 | 85.6 | 88.2 | 25.4 | 10.1 | 65.9 | 6.2 | 75.9 | 17.4 | 2.6 | 3.2 | 4.4 | 1.1 | 1.2 | 1.2 | 42.4 | 50.1 | 2.1 |
| EWF D100 | C | 58.1 | 98.5 | 90.9 | 89.3 | 85.6 | 88.2 | 25.4 | 12.4 | 65.9 | 6.7 | 75.9 | 17.4 | 2.6 | 3.2 | 6.0 | 1.1 | 1.2 | 1.2 | 42.4 | 50.9 | 2.1 |
| EWF D30 | C | 55.9 | 98.5 | 88.0 | 86.4 | 81.6 | 86.3 | 26.2 | 10.1 | 65.9 | 6.2 | 55.0 | 16.8 | 2.2 | 3.7 | 4.4 | 1.0 | 1.1 | 1.1 | 43.5 | 48.4 | 2.0 |
| W2V | C | 33.6 | 78.9 | 56.9 | 42.0 | 45.2 | 74.8 | 20.1 | 5.3 | 61.2 | 5.7 | 99.3 | 24.4 | 1.6 | 4.1 | 42.8 | 11.6 | 1.0 | 1.2 | 26.8 | 41.9 | 2.0 |
| W2V D100 | C | 35.0 | 81.2 | 59.6 | 44.9 | 45.2 | 74.6 | 20.1 | 6.0 | 62.7 | 7.0 | 99.3 | 24.4 | 1.6 | 4.1 | 55.7 | 11.6 | 1.0 | 1.2 | 26.8 | 43.6 | 2.0 |
| EAT | E | 60.6 | 98.3 | 90.3 | 87.9 | 86.3 | 91.0 | 43.9 | 9.9 | 65.4 | 10.5 | 54.4 | 16.7 | 2.6 | 3.1 | 5.6 | 0.9 | 0.7 | 1.2 | 32.0 | 50.2 | 1.9 |
| EWF | E | 55.1 | 98.9 | 87.9 | 86.2 | 85.3 | 88.1 | 22.4 | 9.8 | 65.0 | 6.3 | 75.8 | 17.3 | 2.2 | 2.9 | 4.3 | 0.9 | 1.2 | 1.4 | 42.9 | 49.7 | 1.9 |
| EWF D100 | E | 58.0 | 98.9 | 90.7 | 89.0 | 85.3 | 88.1 | 22.4 | 12.0 | 62.9 | 6.6 | 75.8 | 17.3 | 2.2 | 2.9 | 6.0 | 0.9 | 1.2 | 1.4 | 42.9 | 50.4 | 1.9 |
| MAE | E | 43.5 | 97.0 | 82.7 | 80.9 | 71.0 | 80.9 | 59.3 | 6.6 | 63.3 | 7.3 | 61.5 | 2.5 | 1.7 | 3.7 | 2.3 | 0.8 | 0.8 | 1.4 | 30.4 | 46.0 | 1.9 |
| MAE D100 | E | 47.7 | 98.1 | 82.7 | 86.7 | 71.0 | 76.8 | 59.3 | 8.0 | 63.4 | 10.3 | 61.5 | 2.5 | 1.7 | 3.7 | 3.1 | 0.8 | 0.8 | 1.4 | 30.4 | 46.8 | 1.9 |
| MAE | C | 44.0 | 97.7 | 77.4 | 80.9 | 71.4 | 80.8 | 62.3 | 6.6 | 64.5 | 7.4 | 62.2 | 16.3 | 1.9 | 3.5 | 2.3 | 0.7 | 0.7 | 1.5 | 30.2 | 47.0 | 1.9 |
| MAE D100 | C | 48.8 | 97.7 | 82.8 | 87.0 | 71.4 | 77.2 | 62.3 | 8.2 | 65.6 | 10.3 | 62.2 | 16.3 | 1.9 | 3.5 | 3.4 | 0.7 | 0.7 | 1.5 | 30.2 | 48.3 | 1.9 |
| EWF D30 | S | 56.0 | 98.5 | 88.0 | 86.1 | 80.7 | 85.8 | 22.0 | 10.0 | 67.1 | 5.4 | 56.5 | 16.9 | 1.5 | 3.8 | 3.9 | 0.7 | 0.9 | 1.4 | 41.3 | 47.9 | 1.9 |
| EWF D30 | E | 55.1 | 98.9 | 87.9 | 86.2 | 81.3 | 86.2 | 25.1 | 9.8 | 65.0 | 6.3 | 55.4 | 14.7 | 2.0 | 3.3 | 4.3 | 1.0 | 1.0 | 1.1 | 44.4 | 48.1 | 1.9 |
| W2V D100 | E | 34.1 | 79.9 | 59.3 | 43.7 | 44.6 | 73.7 | 18.1 | 5.9 | 59.2 | 6.6 | 99.3 | 8.5 | 1.5 | 3.6 | 52.5 | 9.5 | 0.9 | 1.2 | 27.7 | 41.5 | 1.8 |
| W2V | E | 33.5 | 78.0 | 56.8 | 40.9 | 44.6 | 73.4 | 18.1 | 5.3 | 58.8 | 5.6 | 99.3 | 8.5 | 1.5 | 3.6 | 41.2 | 9.5 | 0.9 | 1.2 | 27.7 | 40.1 | 1.8 |
| W2V D30 | S | 32.8 | 75.5 | 56.7 | 41.4 | 40.6 | 68.6 | 18.0 | 5.2 | 59.2 | 5.5 | 99.0 | 22.5 | 1.3 | 3.7 | 39.7 | 7.0 | 0.8 | 1.1 | 27.0 | 39.9 | 1.7 |
| W2V D30 | C | 33.6 | 78.9 | 56.9 | 42.0 | 41.1 | 70.3 | 18.4 | 5.3 | 61.2 | 5.7 | 99.1 | 24.4 | 1.3 | 3.8 | 42.8 | 8.3 | 0.8 | 1.0 | 27.7 | 41.0 | 1.7 |
| MAE D30 | C | 44.0 | 96.0 | 77.4 | 80.9 | 62.8 | 66.4 | 58.7 | 6.6 | 64.5 | 7.4 | 37.6 | 16.3 | 1.5 | 3.5 | 2.3 | 0.5 | 0.7 | 1.1 | 27.4 | 43.2 | 1.7 |
| MAE D30 | S | 43.8 | 95.8 | 77.3 | 81.1 | 61.4 | 66.5 | 56.5 | 7.0 | 62.5 | 6.9 | 37.6 | - | 1.3 | 3.8 | 2.2 | 0.5 | 0.6 | 1.1 | 29.7 | 44.9 | 1.7 |
| W2V D30 | E | 33.5 | 78.0 | 56.8 | 40.9 | 40.2 | 69.7 | 17.2 | 5.3 | 58.8 | 5.6 | 99.1 | 8.5 | 1.5 | 3.3 | 41.2 | 7.7 | 0.8 | 1.0 | 28.6 | 39.4 | 1.6 |
| MAE D30 | E | 43.5 | 95.6 | 77.4 | 80.9 | 62.8 | 66.2 | 56.8 | 6.6 | 63.3 | 7.3 | 37.1 | 2.5 | 1.4 | 3.4 | 2.3 | 0.6 | 0.6 | 1.1 | 28.3 | 42.1 | 1.6 |
| CC | E | 33.8 | 89.6 | 72.7 | 47.6 | 58.6 | 68.1 | 2.7 | 3.9 | 57.4 | 1.6 | 18.7 | 15.1 | 0.8 | 3.8 | 1.8 | 0.2 | 0.6 | 1.0 | 16.8 | 32.6 | 1.5 |
| CC | S | 30.4 | 90.7 | 70.0 | 40.7 | 47.5 | 61.2 | 4.2 | 3.6 | 56.5 | 1.9 | 46.4 | 15.1 | 1.0 | 3.3 | 1.7 | 0.2 | 0.4 | 0.9 | 17.5 | 32.5 | 1.4 |
| CC | C | 31.4 | 89.0 | 71.9 | 44.8 | 55.1 | 62.2 | 2.8 | 3.4 | 56.2 | 1.4 | 25.6 | 15.1 | 0.7 | 3.2 | 1.9 | 0.2 | 0.4 | 0.7 | 18.6 | 32.0 | 1.3 |

Table 19: P@5 Results Across Subsets and Distances (↑ better)

| Method | Dist | BC | BS1 | BS2 | BS3 | BS4 | BS5 | ES1 | HP | HS1 | HS2 | HU1 | HU2 | HU3 | HU4 | HW1 | HW2 | HW3 | HW4 | OC1 | Avg | Avg (Blind) |
|---|---|---|---|---|---|---|---|---|---|---|---|---|---|---|---|---|---|---|---|---|---|---|
| EWMTF D100 | S | 68.0 | 96.1 | 94.5 | 91.6 | 90.7 | 93.5 | 62.6 | **24.2** | 68.7 | 19.7 | 60.9 | 8.1 | **10.9** | **9.1** | 35.8 | 7.3 | **5.4** | **5.3** | 38.5 | **57.4** | 7.7 |
| EWMTF | S | 58.9 | 96.1 | 89.4 | 86.0 | 90.7 | 93.5 | 62.6 | 12.1 | 67.7 | 12.9 | 60.9 | 8.1 | **10.9** | **9.1** | 9.6 | 7.3 | **5.4** | **5.3** | 38.5 | 53.0 | 7.7 |
| BEATs | E | 67.1 | 97.8 | 93.8 | 89.1 | 88.6 | 94.5 | 88.5 | 13.6 | 76.3 | 24.8 | 15.3 | 9.4 | 10.0 | 8.2 | 18.8 | 4.7 | 4.6 | 5.0 | 48.3 | 55.4 | 7.0 |
| BEATs | C | 67.1 | 97.7 | 93.8 | 89.1 | 88.6 | 94.4 | 88.8 | 13.5 | 77.3 | 25.0 | 15.3 | 10.4 | 9.8 | 8.8 | 19.0 | 4.7 | 4.4 | 4.9 | **49.9** | 55.6 | 7.0 |
| EMTF | C | 55.1 | 96.2 | 93.6 | 81.3 | 89.2 | 94.0 | 57.3 | 12.7 | 64.6 | 10.0 | **64.7** | 11.4 | 10.1 | 7.6 | 10.9 | 8.3 | 5.1 | 4.3 | 38.1 | 52.5 | 6.8 |
| EMTF D100 | S | 65.8 | 96.2 | 93.6 | 90.9 | 89.2 | 94.0 | 57.3 | 23.4 | 67.0 | 16.8 | **64.7** | 11.4 | 10.1 | 7.6 | 37.8 | 8.3 | 5.1 | 4.3 | 38.1 | 57.0 | 6.8 |
| EWTF | S | 66.6 | 98.0 | 96.1 | 90.3 | 94.3 | 95.1 | 59.7 | 9.0 | 66.0 | 13.5 | 20.0 | 10.6 | 9.8 | 7.6 | 5.5 | 4.5 | 4.8 | 4.1 | 39.0 | 51.2 | 6.6 |
| EWTF D100 | S | 73.2 | 98.0 | 96.1 | 93.9 | 94.3 | 95.1 | 59.7 | 14.6 | 67.3 | 17.7 | 20.0 | 10.6 | 9.8 | 7.6 | 10.5 | 4.5 | 4.8 | 4.1 | 39.0 | 53.0 | 6.6 |
| EWT D100 | S | 72.7 | 97.5 | 96.1 | 93.8 | 94.2 | 95.0 | 60.0 | 14.4 | 66.8 | 17.4 | 19.8 | 11.4 | 9.7 | 7.3 | 10.4 | 4.5 | 4.7 | 4.0 | 38.0 | 52.8 | 6.4 |
| EWT | S | 66.5 | 97.5 | 96.1 | 90.0 | 94.2 | 95.1 | 60.0 | 9.2 | 66.2 | 13.6 | 19.8 | 11.4 | 9.7 | 7.3 | 5.5 | 4.5 | 4.7 | 4.0 | 38.0 | 51.2 | 6.4 |
| EWTF D100 | C | 71.9 | 99.1 | 95.3 | 93.2 | 93.6 | 95.0 | 58.7 | 11.7 | 69.0 | 17.2 | 19.2 | 9.7 | 8.7 | 7.1 | 9.0 | 3.6 | 4.1 | 3.8 | 39.3 | 52.4 | 5.9 |
| EWTF | C | 69.0 | 98.9 | 95.7 | 91.3 | 93.6 | 95.0 | 58.7 | 9.0 | 67.8 | 14.7 | 19.2 | 9.7 | 8.7 | 7.1 | 5.7 | 3.6 | 4.1 | 3.8 | 39.3 | 51.4 | 5.9 |
| ETF | C | 61.8 | 97.1 | 94.4 | 87.6 | 91.6 | 94.8 | 61.1 | 12.3 | 64.5 | 12.5 | 22.2 | 12.9 | 8.9 | 6.6 | 7.3 | 5.0 | 4.3 | 3.7 | 38.8 | 50.9 | 5.9 |
| ETF D100 | S | 69.9 | 97.1 | 94.4 | 92.8 | 91.6 | 94.8 | 61.1 | 17.9 | 66.9 | 17.0 | 22.2 | 12.9 | 8.9 | 6.6 | 15.5 | 5.0 | 4.3 | 3.7 | 38.8 | 53.2 | 5.9 |
| EWT | C | 68.7 | 98.8 | 95.7 | 91.1 | 93.5 | 95.0 | 58.5 | 9.0 | 67.6 | 14.5 | 19.1 | 11.2 | 8.5 | 6.9 | 5.6 | 3.6 | 4.1 | 3.8 | 38.0 | 51.3 | 5.8 |
| EWT D100 | C | 71.7 | 99.1 | 95.2 | 93.0 | 93.5 | 95.0 | 58.5 | 11.6 | 69.0 | 17.0 | 19.1 | 11.2 | 8.5 | 6.9 | 9.0 | 3.6 | 4.1 | 3.8 | 38.0 | 52.3 | 5.8 |
| ET D100 | S | 69.0 | 96.7 | 94.1 | 92.6 | 91.1 | 94.7 | 60.6 | 17.9 | 65.9 | 16.7 | 22.0 | 12.9 | 8.8 | 6.3 | 15.3 | 5.0 | 4.4 | 3.7 | 37.9 | 52.8 | 5.8 |
| ET | S | 61.8 | 97.1 | 89.4 | 92.6 | 91.1 | 94.8 | 60.6 | 12.2 | 64.6 | 12.3 | 22.0 | 12.9 | 8.8 | 6.3 | 7.3 | 5.0 | 4.4 | 3.7 | 37.9 | 50.8 | 5.8 |
| EWTF D100 | E | 70.9 | 98.9 | 95.4 | 93.1 | 93.8 | **95.1** | 58.6 | 11.3 | 69.5 | 17.3 | 19.2 | 9.5 | 8.4 | 7.2 | 9.1 | 3.5 | 3.9 | 3.7 | 37.9 | 52.2 | 5.8 |
| EWTF | E | 68.4 | 99.0 | 94.4 | 91.3 | 93.8 | **95.1** | 58.6 | 9.0 | 67.8 | 14.8 | 19.2 | 9.5 | 8.4 | 7.2 | 5.6 | 3.5 | 3.9 | 3.7 | 37.9 | 51.2 | 5.8 |
| EWT D100 | E | 70.7 | 98.8 | 95.3 | 92.9 | 93.7 | 95.0 | 58.4 | 11.2 | 69.3 | 17.0 | 19.2 | 9.1 | 8.2 | 7.2 | 9.0 | 3.5 | 4.0 | 3.6 | 37.1 | 52.0 | 5.7 |
| EWT | E | 68.3 | 99.0 | 94.3 | 91.2 | 93.7 | **95.1** | 58.4 | 8.9 | 67.6 | 14.6 | 19.2 | 9.1 | 8.2 | 7.2 | 5.6 | 3.5 | 4.0 | 3.6 | 37.1 | 51.0 | 5.7 |
| ETF | C | 65.1 | 98.9 | 93.8 | 89.3 | 90.4 | 94.8 | 59.7 | 13.1 | 66.6 | 13.5 | 21.0 | 12.8 | 8.4 | 6.5 | 8.2 | 4.4 | 3.9 | 3.5 | 41.3 | 51.5 | 5.6 |
| ETF D100 | C | 68.7 | 99.2 | 93.8 | 92.0 | 90.4 | 94.8 | 59.7 | 16.5 | 68.3 | 16.3 | 21.0 | 12.8 | 8.4 | 6.5 | 13.6 | 4.4 | 3.9 | 3.5 | 41.3 | 52.9 | 5.6 |
| ET D100 | C | 68.2 | 99.1 | 93.6 | 91.7 | 90.1 | 94.8 | 59.4 | 16.4 | 67.9 | 16.0 | 20.9 | 12.6 | 8.3 | 6.2 | 13.4 | 4.4 | 3.9 | 3.5 | 40.3 | 52.6 | 5.5 |
| ET | C | 64.4 | 99.1 | 93.6 | 89.0 | 90.1 | 94.8 | 59.4 | 13.0 | 66.8 | 13.2 | 20.9 | 12.6 | 8.3 | 6.2 | 8.1 | 4.4 | 3.9 | 3.5 | 40.3 | 51.3 | 5.5 |
| EW | C | 58.7 | 98.6 | 95.7 | 85.6 | 87.2 | 90.3 | 55.8 | 10.5 | 66.5 | 11.8 | 36.1 | 11.8 | 7.1 | 7.3 | 8.0 | 2.8 | 3.5 | 3.8 | 33.2 | 50.2 | 5.4 |
| EWMTF | C | 58.7 | 97.5 | 92.1 | 85.6 | 87.2 | 92.1 | 55.8 | 10.5 | 66.5 | 11.8 | 36.1 | 11.8 | 7.1 | 7.3 | 8.0 | 2.8 | 3.5 | 3.8 | 33.2 | 50.0 | 5.4 |
| EWMTF D100 | C | 61.0 | 97.5 | 91.4 | 87.9 | 87.2 | 92.1 | 55.8 | 14.7 | 67.1 | 14.3 | 36.1 | 11.8 | 7.1 | 7.3 | 15.5 | 2.8 | 3.5 | 3.8 | 33.2 | 51.2 | 5.4 |
| ETF D100 | E | 68.1 | 98.9 | 94.0 | 91.9 | 90.7 | 95.0 | 59.4 | 16.0 | 69.4 | 16.3 | 20.9 | 12.6 | 8.0 | 6.5 | 13.4 | 4.4 | 3.7 | 3.4 | 40.5 | 52.8 | 5.4 |
| ETF | E | 64.3 | 98.9 | 94.4 | 89.4 | 90.7 | 95.0 | 59.4 | 12.8 | 66.3 | 13.5 | 20.9 | 12.6 | 8.0 | 6.5 | 7.8 | 4.4 | 3.7 | 3.4 | 40.5 | 51.4 | 5.4 |
| ET | E | 63.9 | 98.9 | 93.8 | 89.1 | 90.3 | 95.0 | 58.7 | 12.7 | 66.8 | 13.3 | 20.8 | 12.4 | 7.9 | 6.3 | 7.8 | 4.3 | 3.7 | 3.4 | 39.3 | 51.1 | 5.3 |
| ET D100 | E | 67.6 | 98.8 | 93.8 | 91.7 | 90.3 | 95.0 | 58.7 | 15.9 | 68.9 | 16.1 | 20.8 | 12.4 | 7.9 | 6.3 | 13.3 | 4.3 | 3.7 | 3.4 | 39.3 | 52.5 | 5.3 |
| CLP D100 | C | 75.4 | 99.1 | 95.4 | 92.9 | 91.9 | 94.8 | **93.7** | 16.6 | **73.4** | 20.5 | 12.2 | 9.7 | 7.9 | 6.4 | 16.4 | 4.6 | 3.4 | 3.3 | 47.4 | 56.3 | 5.3 |
| CLP | C | 72.5 | 98.9 | 94.2 | 91.5 | 91.9 | 94.8 | **93.7** | 15.2 | 73.0 | 18.4 | 12.2 | 9.7 | 7.9 | 6.4 | 14.3 | 4.6 | 3.4 | 3.3 | 47.4 | 55.5 | 5.3 |
| CLP D100 | S | 75.7 | 96.8 | 95.6 | 93.2 | 93.0 | 94.7 | 92.7 | 17.0 | 69.3 | **20.5** | 14.2 | 9.6 | 7.8 | 6.2 | 15.8 | 4.7 | 3.5 | 3.4 | 41.7 | 55.6 | 5.2 |
| CLP | S | 69.0 | 98.1 | 95.6 | 89.5 | 93.0 | 94.8 | 92.7 | 14.0 | 70.0 | 16.6 | 14.2 | 9.6 | 7.8 | 6.2 | 11.9 | 4.7 | 3.5 | 3.4 | 41.7 | 54.4 | 5.2 |
| EWMTF | E | 58.0 | 97.7 | 91.5 | 85.3 | 87.4 | 91.8 | 55.2 | 10.2 | 68.2 | 11.5 | 37.2 | 9.3 | 6.5 | 7.1 | 7.4 | 2.7 | 3.4 | 3.6 | 32.4 | 49.7 | 5.2 |
| EWMTF D100 | E | 60.4 | 97.7 | 91.5 | 87.6 | 87.4 | 91.8 | 55.2 | 14.1 | 68.2 | 14.0 | 37.2 | 9.3 | 6.5 | 7.1 | 14.8 | 2.7 | 3.4 | 3.6 | 32.4 | 50.9 | 5.2 |
| EW | E | 58.0 | 98.6 | 91.5 | 85.3 | 87.4 | 90.3 | 55.2 | 10.2 | 68.2 | 11.5 | 37.2 | 9.3 | 6.5 | 7.1 | 7.4 | 2.7 | 3.4 | 3.6 | 32.4 | 49.7 | 5.2 |
| CLP | E | 72.5 | 99.2 | 95.4 | 91.6 | 91.9 | 94.8 | 93.4 | 14.8 | 72.1 | 18.4 | 12.2 | 9.7 | 7.3 | 6.2 | 14.0 | 4.4 | 3.2 | 3.0 | 47.1 | 55.4 | 4.9 |
| CLP D100 | E | 75.0 | 99.2 | 95.3 | 93.0 | 91.9 | 94.8 | 93.4 | 16.1 | 73.1 | 20.1 | 12.2 | 9.7 | 7.3 | 6.2 | 16.2 | 4.4 | 3.2 | 3.0 | 47.1 | 56.1 | 4.9 |
| EWTF D30 | C | 69.0 | 99.2 | 94.3 | 91.3 | 91.8 | 94.4 | 53.9 | 9.0 | 67.8 | 14.7 | 16.6 | 7.4 | 6.6 | 6.2 | 5.7 | 2.2 | 3.0 | 3.2 | 37.6 | 50.3 | 4.8 |
| EWT D30 | C | 68.7 | 99.1 | 94.2 | 91.1 | 91.5 | 94.3 | 53.5 | 9.0 | 67.6 | 14.5 | 16.5 | 11.2 | 6.6 | 6.2 | 5.6 | 2.2 | 3.0 | 3.2 | 36.6 | 50.4 | 4.7 |
| EWTF D30 | E | 68.4 | 99.0 | 94.4 | 91.3 | 93.6 | 94.5 | 53.6 | 9.0 | 67.8 | 14.8 | 16.8 | 7.0 | 6.4 | 6.4 | 5.6 | 2.2 | 2.9 | 3.1 | 37.3 | 50.2 | 4.7 |
| EWTF D30 | S | 66.6 | 98.6 | 94.0 | 90.3 | 90.7 | 93.6 | 52.0 | 9.0 | 66.0 | 13.5 | 15.9 | 7.0 | 6.5 | 6.4 | 5.5 | 2.2 | 2.9 | 3.1 | 37.2 | 49.5 | 4.7 |
| EMTF D100 | C | 57.2 | 97.2 | 89.8 | 85.0 | 84.8 | 92.5 | 48.8 | 15.6 | 63.8 | 11.1 | 33.4 | 13.6 | 6.9 | 6.0 | 18.1 | 3.6 | 3.1 | 2.9 | 35.3 | 50.0 | 4.7 |
| E | C | 53.3 | 97.0 | 0.7 | 80.7 | 84.8 | 93.0 | 48.8 | 11.5 | 63.1 | 8.9 | 33.4 | 13.6 | 6.9 | 6.0 | 9.6 | 3.6 | 3.1 | 2.9 | 35.3 | 42.5 | 4.7 |
| EMTF | C | 53.3 | 97.0 | 89.8 | 80.7 | 84.8 | 92.5 | 48.8 | 11.5 | 63.1 | 8.9 | 33.4 | 13.6 | 6.9 | 6.0 | 9.6 | 3.6 | 3.1 | 2.9 | 35.3 | 48.4 | 4.7 |
| EWT D30 | S | 66.5 | 98.4 | 94.1 | 90.0 | 90.2 | 93.5 | 52.0 | 9.2 | 66.2 | 13.6 | 16.1 | 11.4 | 6.4 | 6.4 | 5.5 | 2.2 | 2.8 | 3.2 | 36.2 | 49.7 | 4.7 |
| EWT D30 | E | 68.3 | 98.7 | 94.3 | 91.2 | 92.0 | 94.5 | 53.4 | 8.9 | 67.6 | 14.6 | 16.7 | 9.1 | 6.3 | 6.4 | 5.6 | 2.1 | 2.9 | 3.1 | 36.6 | 50.2 | 4.7 |
| CLP D30 | C | 72.5 | 98.9 | 94.2 | 91.5 | 89.9 | 94.4 | 93.2 | 15.2 | 73.0 | 18.4 | 9.3 | 9.7 | 6.9 | 5.8 | 14.3 | 3.9 | 2.8 | 2.8 | 46.5 | 55.0 | 4.6 |
| EMTF D100 | E | 56.3 | 97.4 | 90.2 | 85.2 | 85.1 | 92.4 | 48.0 | 15.2 | 64.6 | 10.9 | 33.7 | 10.7 | 6.7 | 5.6 | 17.9 | 3.5 | 2.8 | 2.7 | 34.6 | 49.7 | 4.5 |
| E | E | 53.0 | 97.4 | 93.8 | 80.7 | 85.1 | 92.9 | 48.0 | 11.1 | 63.4 | 8.5 | 33.7 | 10.7 | 6.7 | 5.6 | 9.0 | 3.5 | 2.8 | 2.7 | 34.6 | 48.4 | 4.5 |
| EMTF | E | 53.0 | 97.4 | 90.9 | 80.7 | 85.1 | 92.4 | 48.0 | 11.1 | 63.4 | 8.5 | 33.7 | 10.7 | 6.7 | 5.6 | 9.0 | 3.5 | 2.8 | 2.7 | 34.6 | 48.1 | 4.5 |
| CLP D30 | E | 72.5 | 99.2 | 94.2 | 91.6 | 90.1 | 94.4 | 93.0 | 14.8 | 72.1 | 18.4 | 9.4 | 9.7 | 6.2 | 5.7 | 14.0 | 3.6 | 2.8 | 2.6 | 45.9 | 54.9 | 4.3 |
| ET D30 | C | 64.4 | 99.1 | 91.3 | 89.0 | 87.6 | 94.0 | 54.5 | 13.0 | 66.8 | 13.2 | 18.4 | 12.6 | 6.5 | 5.3 | 8.1 | 2.6 | 2.7 | 2.7 | 38.7 | 50.2 | 4.3 |
| ETF D30 | C | 65.1 | 99.2 | 91.6 | 89.3 | 87.9 | 94.0 | 55.0 | 13.1 | 66.6 | 13.5 | 18.5 | 12.8 | 6.5 | 5.2 | 8.2 | 2.6 | 2.7 | 2.8 | 39.7 | 50.5 | 4.3 |
| EWMTF D30 | C | 58.7 | 97.5 | 89.6 | 85.6 | 85.1 | 90.5 | 51.2 | 10.5 | 66.5 | 11.8 | 25.1 | 11.8 | 5.4 | 6.3 | 8.0 | 1.6 | 2.3 | 3.1 | 32.7 | 48.4 | 4.3 |
| EWMTF D30 | S | 58.9 | 96.3 | 90.3 | 86.0 | 84.7 | 90.3 | 52.7 | 12.1 | 67.7 | 12.9 | 26.4 | 8.1 | 5.7 | 5.9 | 9.6 | 2.0 | 2.3 | 3.1 | 33.3 | 48.8 | 4.2 |
| EWMTF D30 | E | 58.0 | 97.8 | 89.6 | 85.3 | 85.3 | 90.0 | 50.3 | 10.2 | 67.6 | 11.5 | 25.7 | 9.3 | 4.9 | 6.4 | 7.4 | 1.5 | 2.3 | 3.0 | 31.6 | 48.1 | 4.1 |
| ET D30 | E | 63.9 | 98.8 | 91.4 | 89.1 | 88.0 | 94.1 | 54.3 | 12.7 | 66.8 | 13.3 | 18.6 | 12.4 | 6.0 | 5.1 | 7.8 | 2.5 | 2.6 | 2.6 | 39.1 | 50.2 | 4.1 |
| ETF D30 | E | 64.3 | 98.8 | 91.6 | 89.4 | 88.2 | 94.1 | 54.8 | 12.8 | 66.3 | 13.5 | 18.6 | 12.6 | 5.9 | 5.1 | 7.8 | 2.5 | 2.6 | 2.6 | 39.3 | 50.3 | 4.1 |
| CLP D30 | S | 69.0 | 98.1 | 92.4 | 89.5 | 88.7 | 93.8 | 92.0 | 14.0 | 70.0 | 16.6 | 8.6 | 9.6 | 5.9 | 5.1 | 11.9 | 3.1 | 2.5 | 2.8 | 41.0 | 53.2 | 4.1 |
| EMTF D30 | S | 55.1 | 96.2 | 88.2 | 81.3 | 81.0 | 90.6 | 44.7 | 12.7 | 64.6 | 10.0 | 25.2 | 11.4 | 5.5 | 5.6 | 10.9 | 2.3 | 2.2 | 2.6 | 34.8 | 47.3 | 4.0 |
| ET D30 | S | 61.8 | 97.1 | 89.4 | 87.3 | 85.9 | 93.2 | 52.2 | 12.2 | 64.6 | 12.3 | 17.7 | 12.9 | 5.8 | 5.0 | 7.3 | 2.3 | 2.5 | 2.6 | 36.4 | 48.9 | 4.0 |
| ETF D30 | S | 61.8 | 97.7 | 89.8 | 87.6 | 86.2 | 93.4 | 52.0 | 12.3 | 64.5 | 12.5 | 17.7 | 12.9 | 5.7 | 5.0 | 7.3 | 2.2 | 2.5 | 2.5 | 37.5 | 49.0 | 3.9 |
| EMTF D30 | C | 53.3 | 97.0 | 87.5 | 80.7 | 81.9 | 90.7 | 44.0 | 11.5 | 63.1 | 8.9 | 21.0 | 13.6 | 5.4 | 5.3 | 9.6 | 2.0 | 2.1 | 2.3 | 34.1 | 46.6 | 3.8 |
| WLM | S | 43.7 | 97.8 | 86.2 | 81.5 | 86.4 | 90.7 | 25.9 | 21.1 | 58.8 | 7.2 | 37.0 | 9.3 | 4.2 | 5.3 | 62.7 | 18.7 | 1.9 | 2.9 | 34.2 | 50.8 | 3.6 |
| WLM D100 | S | 50.4 | 97.1 | 90.5 | 87.0 | 86.4 | 90.4 | 25.9 | 23.1 | 62.3 | 9.3 | 37.0 | 22.6 | 4.2 | 5.3 | 69.0 | 18.7 | 1.9 | 2.9 | 34.2 | 53.6 | 3.6 |
| EMTF D30 | E | 53.0 | 97.5 | 87.8 | 80.7 | 81.9 | 90.6 | 43.3 | 11.1 | 63.4 | 8.5 | 21.0 | 10.7 | 5.2 | 4.8 | 9.0 | 1.9 | 2.0 | 2.1 | 33.3 | 46.2 | 3.5 |
| WLM | C | 44.9 | 98.6 | 87.5 | 83.0 | 85.6 | 91.3 | 26.2 | 21.5 | 60.1 | 8.0 | 30.9 | **25.9** | 3.5 | 5.2 | 64.5 | 18.2 | 1.6 | 2.7 | 37.6 | 52.3 | 3.2 |
| WLM D100 | C | 49.5 | 98.5 | 90.5 | 86.6 | 85.6 | 90.2 | 26.2 | 22.8 | 61.7 | 9.2 | 30.9 | **25.9** | 3.5 | 5.2 | 69.9 | 18.2 | 1.6 | 2.7 | 37.6 | 53.5 | 3.2 |
| WLM | E | 45.1 | 98.5 | 87.5 | 83.1 | 85.5 | 91.2 | 24.7 | 21.4 | 59.6 | 8.0 | 30.3 | 20.2 | 3.2 | 5.1 | 64.2 | 17.5 | 1.6 | 2.6 | 34.7 | 51.4 | 3.1 |
| WLM D100 | E | 49.1 | 98.5 | 90.4 | 86.6 | 85.5 | 90.2 | 24.7 | 22.7 | 62.0 | 9.0 | 30.3 | 20.2 | 3.2 | 5.1 | 69.3 | 17.5 | 1.6 | 2.6 | 34.7 | 52.7 | 3.1 |
| M | C | **77.0** | 99.9 | 98.2 | 96.4 | 94.2 | 93.1 | 22.8 | 17.1 | 56.2 | 7.2 | 23.5 | 9.0 | 3.4 | 5.0 | 17.9 | 4.7 | 1.8 | 2.2 | 37.5 | 50.3 | 3.1 |

Table 19: (Continued) P@5 Results Across Subsets and Distances (↑ better)

| Method | Dist | BC | BS1 | BS2 | BS3 | BS4 | BS5 | ES1 | HP | HS1 | HS2 | HU1 | HU2 | HU3 | HU4 | HW1 | HW2 | HW3 | HW4 | OC1 | Avg | Avg (Blind) |
|---|---|---|---|---|---|---|---|---|---|---|---|---|---|---|---|---|---|---|---|---|---|---|
| VC | C | 70.7 | 99.9 | 97.0 | 95.3 | 93.5 | 94.6 | 24.8 | 15.0 | 56.8 | 9.2 | 20.8 | 9.4 | 3.2 | 4.9 | 15.3 | 4.5 | 1.6 | 2.2 | 39.0 | 49.7 | 3.0 |
| VC | E | 70.2 | 99.8 | 96.9 | 95.2 | 93.4 | 94.3 | 26.5 | 14.8 | 56.7 | 9.2 | 19.6 | 8.5 | 3.3 | 4.6 | 14.4 | 4.5 | 1.5 | 1.9 | 38.0 | 49.5 | 2.8 |
| WLM D30 | C | 44.9 | 98.6 | 87.5 | 83.0 | 82.8 | 88.2 | 23.3 | 21.5 | 60.1 | 8.0 | 22.9 | **25.9** | 2.8 | 4.8 | 64.5 | 14.7 | 1.3 | 2.3 | 37.2 | 50.9 | 2.8 |
| WLM D30 | E | 45.1 | 98.5 | 87.5 | 83.1 | 82.8 | 88.3 | 22.9 | 21.4 | 59.6 | 8.0 | 22.8 | 20.2 | 2.7 | 4.6 | 64.2 | 14.5 | 1.2 | 2.2 | 36.1 | 50.3 | 2.7 |
| M | S | 70.7 | **100.0** | 96.7 | 94.8 | 92.2 | 91.2 | 15.0 | 11.3 | 53.6 | 6.3 | 22.8 | 7.7 | 2.7 | 4.3 | 11.1 | 3.8 | 1.5 | 2.0 | 33.0 | 47.3 | 2.6 |
| M | E | 75.5 | 99.7 | **98.2** | **96.4** | **94.4** | 93.2 | 22.1 | 17.1 | 55.1 | 6.9 | 24.3 | 8.7 | 2.6 | 4.5 | 18.8 | 4.7 | 1.5 | 1.9 | 36.9 | 50.1 | 2.6 |
| WLM D30 | S | 43.7 | 97.8 | 86.2 | 81.5 | 80.3 | 86.7 | 21.0 | 21.1 | 58.8 | 7.2 | 22.7 | 9.3 | 2.6 | 4.0 | 62.7 | 13.7 | 1.2 | 2.3 | 35.6 | 48.6 | 2.5 |
| EF D100 | C | 50.8 | 97.8 | 88.9 | 85.0 | 83.7 | 89.1 | 18.4 | 11.8 | 53.7 | 4.6 | 16.4 | 6.0 | 2.6 | 4.2 | 6.5 | 1.2 | 1.0 | 1.8 | 34.6 | 43.2 | 2.4 |
| EF | C | 48.6 | 97.5 | 88.9 | 81.3 | 83.7 | 89.1 | 18.4 | 10.1 | 53.1 | 4.6 | 16.4 | 6.0 | 2.6 | 4.2 | 5.0 | 1.2 | 1.0 | 1.8 | 34.6 | 42.6 | 2.4 |
| AC | S | 71.7 | **100.0** | 97.2 | 92.1 | 92.1 | 93.5 | 22.5 | 9.5 | 56.7 | 9.3 | 21.4 | 7.3 | 2.8 | 3.9 | 9.5 | 2.7 | 1.4 | 1.5 | 29.2 | 47.6 | 2.4 |
| EF | E | 48.3 | 98.4 | 89.5 | 80.9 | 83.7 | 88.9 | 17.2 | 9.9 | 52.9 | 4.5 | 16.2 | 6.1 | 2.6 | 4.1 | 4.7 | 1.2 | 0.9 | 1.7 | 32.8 | 42.4 | 2.3 |
| EF D100 | E | 50.0 | 98.4 | 88.7 | 84.6 | 83.7 | 88.9 | 17.2 | 11.6 | 53.7 | 4.6 | 16.2 | 6.1 | 2.6 | 4.1 | 6.1 | 1.2 | 0.9 | 1.7 | 32.8 | 42.9 | 2.3 |
| EF D30 | E | 48.3 | 97.8 | 86.4 | 80.9 | 81.1 | 87.3 | 18.2 | 9.9 | 52.9 | 4.5 | 11.7 | 6.1 | 2.5 | 4.0 | 4.7 | 1.0 | 0.7 | 1.7 | 32.0 | 41.5 | 2.2 |
| EF D100 | S | 51.8 | 95.9 | 89.6 | 87.0 | 84.5 | 89.7 | 14.7 | 13.8 | 52.1 | 4.3 | 17.0 | 7.8 | 2.4 | 3.9 | 8.6 | 1.4 | 0.9 | 1.8 | 31.3 | 43.3 | 2.2 |
| EF | S | 48.2 | 97.4 | 86.2 | 80.6 | 84.5 | 89.7 | 14.7 | 9.9 | 52.4 | 4.3 | 17.0 | 7.8 | 2.4 | 3.9 | 4.8 | 1.4 | 0.9 | 1.8 | 31.3 | 42.0 | 2.2 |
| EF D30 | C | 48.6 | 97.5 | 86.5 | 81.3 | 81.1 | 87.2 | 18.7 | 10.1 | 53.1 | 4.6 | 11.7 | 6.0 | 2.6 | 3.9 | 5.0 | 1.1 | 0.8 | 1.6 | 33.7 | 41.8 | 2.2 |
| EF D30 | S | 48.2 | 97.4 | 86.2 | 80.6 | 79.9 | 87.1 | 15.3 | 9.9 | 52.4 | 4.3 | 10.1 | 7.8 | 1.9 | 3.9 | 4.8 | 1.1 | 0.9 | 1.6 | 32.6 | 41.2 | 2.1 |
| VC | S | 57.2 | 99.0 | 91.5 | 89.7 | 88.8 | 88.8 | 17.2 | 11.3 | 51.4 | 5.2 | 14.8 | 6.4 | 2.2 | 3.7 | 8.1 | 2.0 | 0.8 | 1.5 | 30.5 | 44.1 | 2.1 |
| AC | C | 69.0 | 99.1 | 96.9 | 91.9 | 91.6 | 92.7 | 22.1 | 9.1 | 54.4 | 8.0 | 20.7 | 6.8 | 2.3 | 3.6 | 8.9 | 2.4 | 1.0 | 1.4 | 25.3 | 46.6 | 2.1 |
| EWF | C | 48.9 | 97.9 | 84.6 | 82.4 | 79.2 | 83.2 | 19.5 | 8.1 | 51.5 | 4.1 | 18.5 | 5.0 | 2.3 | 3.4 | 2.9 | 0.7 | 0.9 | 1.4 | 34.8 | 41.4 | 2.0 |
| EWF D100 | C | 50.7 | 97.8 | 87.8 | 85.5 | 79.2 | 83.2 | 19.5 | 9.1 | 51.3 | 4.3 | 18.5 | 5.0 | 2.3 | 3.4 | 3.7 | 0.7 | 0.9 | 1.4 | 34.8 | 42.1 | 2.0 |
| AC | E | 68.1 | 99.1 | 96.9 | 91.7 | 91.5 | 92.5 | 19.3 | 9.0 | 52.3 | 6.9 | 19.9 | 5.2 | 2.1 | 3.5 | 8.7 | 2.4 | 1.0 | 1.3 | 24.9 | 45.9 | 2.0 |
| EWF D100 | S | 51.9 | 95.6 | 90.7 | 87.4 | 82.8 | 84.1 | 17.5 | 11.5 | 51.2 | 3.7 | 20.5 | 6.4 | 1.8 | 3.5 | 5.1 | 0.8 | 0.8 | 1.6 | 32.9 | 42.8 | 1.9 |
| EWF | S | 48.1 | 95.6 | 90.7 | 82.3 | 82.8 | 84.1 | 17.5 | 8.2 | 50.6 | 3.7 | 20.5 | 6.4 | 1.8 | 3.5 | 3.0 | 0.8 | 0.8 | 1.6 | 32.9 | 41.8 | 1.9 |
| EWF | E | 48.3 | 98.0 | 84.5 | 82.2 | 79.0 | 82.9 | 17.6 | 8.1 | 51.7 | 4.1 | 18.4 | 5.0 | 2.1 | 3.3 | 2.8 | 0.7 | 0.9 | 1.4 | 33.7 | 41.1 | 1.9 |
| EWF D100 | E | 50.0 | 98.0 | 87.7 | 85.4 | 79.0 | 82.9 | 17.6 | 9.1 | 51.3 | 4.2 | 18.4 | 5.0 | 2.1 | 3.3 | 3.5 | 0.7 | 0.9 | 1.4 | 33.7 | 41.8 | 1.9 |
| EWF D30 | S | 48.1 | 97.2 | 84.9 | 82.3 | 73.7 | 81.3 | 17.2 | 8.2 | 50.6 | 3.7 | 15.3 | 4.6 | 1.8 | 3.4 | 3.0 | 0.6 | 0.8 | 1.4 | 32.9 | 40.2 | 1.9 |
| EAT | C | 54.2 | 97.4 | 88.7 | 85.1 | 82.7 | 89.0 | 36.5 | 8.5 | 53.3 | 7.8 | 17.5 | 4.7 | 2.1 | 3.5 | 4.7 | 0.8 | 0.7 | 1.3 | 27.3 | 43.9 | 1.9 |
| EAT | E | 50.8 | 96.7 | 87.5 | 84.6 | 79.7 | 86.9 | 32.2 | 8.2 | 50.4 | 6.4 | 13.4 | 4.5 | 1.7 | 3.4 | 3.8 | 0.6 | 0.7 | 1.3 | 26.1 | 42.1 | 1.8 |
| EAT | S | 54.4 | 97.3 | 88.9 | 85.8 | 83.2 | 89.2 | 36.4 | 8.6 | 52.7 | 7.9 | 17.5 | 4.7 | 2.0 | 3.1 | 4.7 | 0.8 | 0.7 | 1.3 | 27.6 | 44.0 | 1.8 |
| EWF D30 | E | 48.3 | 97.9 | 84.5 | 82.2 | 74.0 | 81.2 | 19.0 | 8.1 | 51.7 | 4.1 | 14.7 | 4.2 | 1.9 | 3.3 | 2.8 | 0.6 | 0.8 | 1.4 | 32.8 | 40.4 | 1.8 |
| EWF D30 | C | 48.9 | 97.8 | 84.6 | 82.4 | 74.5 | 81.6 | 20.0 | 8.1 | 51.5 | 4.1 | 14.6 | 4.6 | 2.0 | 3.3 | 2.9 | 0.6 | 0.8 | 1.3 | 33.3 | 40.6 | 1.8 |
| W2V | C | 28.7 | 77.2 | 53.7 | 37.4 | 38.4 | 68.4 | 14.5 | 4.7 | 47.3 | 4.1 | 32.2 | 9.2 | 1.5 | 3.7 | 29.6 | 5.9 | 0.7 | 1.2 | 24.2 | 31.7 | 1.8 |
| W2V D100 | C | 29.7 | 77.8 | 55.9 | 39.9 | 38.4 | 68.8 | 14.5 | 5.1 | 48.5 | 4.5 | 32.2 | 9.2 | 1.5 | 3.7 | 40.0 | 5.9 | 0.7 | 1.2 | 24.2 | 33.0 | 1.8 |
| W2V | S | 28.6 | 75.4 | 53.4 | 37.2 | 40.8 | 68.4 | 15.0 | 4.7 | 46.7 | 3.7 | 34.8 | 7.8 | 1.8 | 3.0 | 27.0 | 6.7 | 0.8 | 1.2 | 23.8 | 31.6 | 1.7 |
| W2V D100 | S | 30.4 | 76.2 | 57.5 | 41.6 | 40.8 | 69.7 | 15.0 | 5.6 | 48.8 | 4.4 | 34.8 | 7.8 | 1.8 | 3.0 | 41.3 | 6.7 | 0.8 | 1.2 | 23.8 | 33.6 | 1.7 |
| W2V | E | 28.2 | 76.8 | 53.4 | 36.9 | 37.9 | 67.1 | 12.6 | 4.5 | 46.4 | 3.9 | 31.5 | 4.7 | 1.3 | 3.6 | 27.8 | 4.8 | 0.7 | 1.1 | 23.1 | 30.6 | 1.7 |
| W2V D100 | E | 29.5 | 77.5 | 55.5 | 38.9 | 37.9 | 67.8 | 12.6 | 4.9 | 46.5 | 4.3 | 31.5 | 4.7 | 1.3 | 3.6 | 36.7 | 4.8 | 0.7 | 1.1 | 23.1 | 31.8 | 1.7 |
| W2V D30 | S | 28.6 | 75.4 | 53.4 | 37.2 | 35.3 | 64.1 | 12.9 | 4.7 | 46.7 | 3.7 | 27.5 | 7.8 | 1.4 | 3.5 | 27.0 | 4.0 | 0.6 | 1.0 | 24.6 | 30.2 | 1.6 |
| W2V D30 | C | 28.7 | 77.2 | 53.7 | 37.4 | 35.3 | 64.9 | 14.0 | 4.7 | 47.3 | 4.1 | 28.2 | 9.2 | 1.4 | 3.4 | 29.6 | 4.3 | 0.7 | 1.1 | 24.2 | 30.8 | 1.6 |
| W2V D30 | E | 28.2 | 76.8 | 53.4 | 36.9 | 34.5 | 64.4 | 13.4 | 4.5 | 46.4 | 3.9 | 28.1 | 4.7 | 1.3 | 3.4 | 27.8 | 3.9 | 0.7 | 1.0 | 23.2 | 30.0 | 1.6 |
| MAE D100 | S | 45.0 | 95.6 | 81.5 | 86.4 | 67.7 | 73.1 | 49.2 | 7.8 | 50.8 | 7.1 | 18.5 | - | 1.2 | 3.2 | 2.7 | 0.6 | 0.6 | 1.3 | 26.4 | 43.7 | 1.6 |
| MAE | S | 38.8 | 95.6 | 81.5 | 76.2 | 67.7 | 68.7 | 49.2 | 6.2 | 49.1 | 4.6 | 18.5 | 0.8 | 1.2 | 3.2 | 1.8 | 0.6 | 0.6 | 1.3 | 26.4 | 39.0 | 1.6 |
| MAE D100 | C | 41.7 | 93.9 | 78.9 | 82.6 | 64.6 | 68.9 | 48.1 | 7.1 | 50.9 | 6.1 | 15.0 | 4.3 | 1.3 | 3.4 | 2.4 | 0.5 | 0.6 | 1.1 | 24.0 | 39.3 | 1.6 |
| MAE | C | 38.5 | 93.9 | 72.6 | 75.3 | 64.6 | 72.3 | 48.1 | 6.1 | 50.0 | 4.6 | 15.0 | 4.3 | 1.3 | 3.4 | 1.8 | 0.5 | 0.6 | 1.1 | 24.0 | 38.1 | 1.6 |
| MAE | E | 38.0 | 93.8 | 78.7 | 75.0 | 64.3 | 72.4 | 44.2 | 6.1 | 49.5 | 4.6 | 14.9 | 1.5 | 1.4 | 3.2 | 1.7 | 0.5 | 0.6 | 1.1 | 23.8 | 37.9 | 1.6 |
| MAE D100 | E | 41.0 | 93.7 | 78.7 | 82.3 | 64.3 | 68.5 | 44.2 | 7.1 | 50.6 | 5.9 | 14.9 | 1.5 | 1.4 | 3.2 | 2.3 | 0.5 | 0.6 | 1.1 | 23.8 | 38.6 | 1.6 |
| MAE D30 | S | 38.8 | 92.6 | 73.1 | 76.2 | 56.1 | 58.6 | 42.5 | 6.2 | 49.1 | 4.6 | 10.8 | - | 1.2 | 3.0 | 1.8 | 0.4 | 0.5 | 1.1 | 23.0 | 38.1 | 1.4 |
| MAE D30 | C | 38.5 | 92.3 | 72.6 | 75.3 | 56.2 | 57.6 | 44.4 | 6.1 | 50.0 | 4.6 | 10.3 | 4.3 | 1.2 | 3.1 | 1.8 | 0.4 | 0.5 | 0.9 | 22.7 | 35.8 | 1.4 |
| MAE D30 | E | 38.0 | 92.5 | 72.4 | 75.0 | 56.1 | 57.2 | 42.9 | 6.1 | 49.5 | 4.6 | 10.2 | 1.5 | 1.3 | 2.9 | 1.7 | 0.4 | 0.5 | 0.9 | 21.6 | 35.3 | 1.4 |
| CC | E | 30.6 | 85.8 | 68.9 | 42.4 | 51.2 | 60.6 | 3.6 | 3.5 | 44.9 | 1.4 | 6.4 | 3.4 | 0.8 | 3.0 | 1.6 | 0.2 | 0.5 | 0.9 | 14.4 | 27.9 | 1.3 |
| CC | C | 28.3 | 85.0 | 68.5 | 40.0 | 47.5 | 54.7 | 3.6 | 3.4 | 45.3 | 1.1 | 8.2 | 3.4 | 0.8 | 2.9 | 1.5 | 0.2 | 0.4 | 0.8 | 14.6 | 27.0 | 1.2 |
| CC | S | 27.2 | 89.0 | 67.8 | 38.3 | 42.2 | 50.6 | 3.8 | 3.3 | 42.5 | 1.4 | 12.9 | 3.4 | 0.9 | 2.8 | 1.5 | 0.2 | 0.5 | 0.7 | 14.1 | 26.5 | 1.2 |

Table 20: GSR Results Across Subsets and Distances (↑ better)

| Method | Dist | BC | BS1 | BS2 | BS3 | BS4 | BS5 | ES1 | HP | HS1 | HS2 | HU1 | HU2 | HU3 | HU4 | HW1 | HW2 | HW3 | HW4 | OC1 | Avg | Avg (Blind) |
|---|---|---|---|---|---|---|---|---|---|---|---|---|---|---|---|---|---|---|---|---|---|---|
| EWMTF D100 | S | 39.3 | 53.7 | 43.5 | 42.5 | 45.1 | 43.4 | 41.5 | 36.4 | 39.5 | 36.8 | 37.8 | **43.3** | **40.3** | 39.9 | 40.4 | **40.5** | 38.9 | 38.4 | 42.3 | **41.7** | **39.4** |
| EWMTF | S | 36.5 | 53.3 | 44.5 | 39.7 | 45.1 | 43.4 | 41.5 | 32.0 | 35.7 | 32.5 | 37.8 | **43.3** | **40.3** | 39.9 | 34.8 | **40.5** | 38.9 | 38.4 | 42.3 | 40.2 | **39.4** |
| ET | E | 36.5 | 53.3 | 43.3 | 39.6 | 44.9 | 43.3 | 41.4 | 32.1 | 35.9 | 32.5 | 37.7 | 43.2 | 40.3 | 39.9 | 34.8 | 40.5 | 38.9 | 38.4 | 42.3 | 40.1 | 39.3 |
| ET D100 | E | 39.3 | 53.5 | 43.3 | 42.4 | 44.9 | 43.3 | 41.4 | **36.4** | **39.5** | 36.7 | 37.7 | 43.2 | 40.3 | 39.9 | 40.4 | 40.5 | 38.9 | 38.4 | 42.3 | 41.7 | 39.3 |
| M | E | **41.0** | 58.6 | 50.8 | **47.1** | **47.9** | 39.3 | 36.6 | 31.9 | 32.0 | 34.4 | 39.4 | 35.4 | 40.3 | 39.3 | 38.4 | 38.3 | **39.4** | 38.3 | 39.1 | 40.7 | 39.3 |
| CLP D100 | E | 38.7 | 54.6 | 45.1 | 43.9 | 44.8 | 39.9 | 51.1 | 35.0 | 38.9 | 36.6 | 33.9 | 38.2 | 39.6 | 39.1 | 39.2 | 38.8 | 38.2 | 37.4 | **42.8** | 41.4 | 38.6 |
| CLP | E | 36.0 | 54.4 | 45.3 | 42.1 | 44.8 | 40.3 | 51.1 | 31.8 | 36.0 | 33.2 | 33.9 | 38.2 | 39.6 | 39.1 | 36.3 | 38.8 | 38.2 | 37.4 | **42.8** | 40.3 | 38.6 |
| EWTF D100 | E | 38.3 | 56.3 | 44.5 | 43.2 | 46.1 | 41.7 | 41.4 | 32.9 | 38.7 | 35.9 | 34.5 | 38.4 | 39.8 | 38.8 | 37.1 | 38.7 | 38.4 | 37.2 | 41.6 | 40.6 | 38.6 |
| EWTF | E | 35.8 | 57.5 | 43.1 | 41.0 | 46.1 | 41.7 | 41.4 | 28.5 | 35.3 | 31.7 | 34.5 | 38.4 | 39.8 | 38.8 | 31.3 | 38.7 | 38.4 | 37.2 | 41.6 | 39.1 | 38.6 |
| EWT D100 | E | 38.2 | 56.0 | 44.4 | 43.0 | 45.9 | 41.7 | 41.4 | 32.9 | 38.8 | 35.8 | 34.4 | 41.7 | 39.8 | 38.8 | 37.1 | 38.6 | 38.3 | 37.1 | 41.5 | 40.8 | 38.5 |
| EWT | E | 35.8 | 57.5 | 43.0 | 40.8 | 45.9 | 41.7 | 41.4 | 28.5 | 35.4 | 31.6 | 34.4 | 41.7 | 39.8 | 38.8 | 31.3 | 38.6 | 38.3 | 37.1 | 41.5 | 39.3 | 38.5 |
| EMTF | S | 24.7 | 42.5 | 40.4 | 28.9 | 42.5 | 40.1 | 39.0 | 19.7 | 26.2 | 20.2 | **46.9** | 33.6 | 37.7 | 37.4 | 25.3 | 37.5 | 35.1 | 35.0 | 40.2 | 33.8 | 36.3 |
| EMTF D100 | S | 36.0 | 44.7 | 40.4 | 39.7 | 42.5 | 40.1 | 39.0 | 31.6 | 36.8 | 32.4 | **46.9** | 33.6 | 37.7 | 37.4 | 40.3 | 37.5 | 35.1 | 35.0 | 40.2 | 38.8 | 36.3 |
| CLP | S | 27.3 | 45.6 | 41.8 | 34.2 | 44.0 | 30.7 | 50.2 | 21.2 | 28.7 | 21.6 | 30.6 | 27.8 | 37.3 | 37.5 | 25.5 | 35.8 | 35.2 | 34.8 | 41.5 | 33.8 | 36.2 |
| CLP D100 | S | 37.1 | 45.3 | 41.8 | 41.3 | 44.0 | 38.7 | 50.2 | 32.2 | 38.3 | 31.9 | 30.6 | 27.8 | 37.3 | 37.5 | 35.6 | 35.8 | 35.2 | 34.8 | 41.5 | 38.1 | 36.2 |
| ETF | E | 24.3 | 45.7 | 28.5 | 29.8 | 42.1 | 39.7 | 38.8 | 18.5 | 26.3 | 20.3 | 46.3 | 27.7 | 37.2 | 37.3 | 23.6 | 36.3 | 34.7 | 34.4 | 40.2 | 32.5 | 35.9 |
| ETF D100 | E | 35.3 | 45.7 | 40.0 | 39.8 | 42.1 | 39.7 | 38.8 | 31.0 | 36.7 | 31.6 | 46.3 | 27.7 | 37.2 | 37.3 | 39.5 | 36.3 | 34.7 | 34.4 | 40.2 | 38.1 | 35.9 |
| WLM D100 | E | 26.2 | 53.5 | 42.8 | 34.4 | 37.0 | 35.0 | 37.0 | 24.1 | 33.6 | 33.7 | 40.2 | 40.8 | 36.5 | 37.0 | **41.9** | 37.6 | 34.7 | 35.0 | 37.5 | 37.0 | 35.8 |
| WLM | E | 20.4 | 53.0 | 39.2 | 29.4 | 37.0 | 40.6 | 37.0 | 19.7 | 29.4 | 27.3 | 40.2 | 40.8 | 36.5 | 37.0 | 38.2 | 37.6 | 34.7 | 35.0 | 37.5 | 35.2 | 35.8 |
| EMTF D100 | E | 33.1 | 49.0 | 39.5 | 37.3 | 40.9 | 36.3 | 37.1 | 31.6 | 32.6 | 30.4 | 39.0 | 38.0 | 36.6 | 36.1 | 38.4 | 36.6 | 35.2 | 34.2 | 38.0 | 37.2 | 35.5 |
| EMTF | E | 30.3 | 49.0 | 40.6 | 34.5 | 40.9 | 36.3 | 37.1 | 27.6 | 29.6 | 26.7 | 39.0 | 38.0 | 36.6 | 36.1 | 33.3 | 36.6 | 35.2 | 34.2 | 38.0 | 35.8 | 35.5 |
| E | E | 30.3 | 49.0 | 43.3 | 34.5 | 40.9 | **44.3** | 37.1 | 27.6 | 29.6 | 26.7 | 39.0 | 38.0 | 36.6 | 36.1 | 33.3 | 36.6 | 35.2 | 34.2 | 38.0 | 36.5 | 35.5 |
| ETF | S | 27.4 | 45.4 | 40.9 | 31.9 | 43.7 | 40.7 | 38.5 | 20.8 | 26.9 | 21.0 | 33.2 | 38.2 | 36.7 | 36.6 | 23.4 | 36.1 | 34.6 | 34.1 | 40.6 | 33.9 | 35.5 |
| ETF D100 | S | 36.7 | 45.4 | 40.9 | 40.6 | 43.7 | 40.7 | 38.5 | 31.7 | 37.0 | 31.4 | 33.2 | 38.2 | 36.7 | 36.6 | 35.6 | 36.1 | 34.6 | 34.1 | 40.6 | 38.0 | 35.5 |
| ET | S | 27.4 | 45.4 | 33.9 | 31.9 | 43.4 | 40.5 | 38.4 | 20.8 | 27.0 | 20.9 | 33.1 | 37.4 | 36.7 | 36.6 | 23.3 | 36.1 | 34.6 | 34.0 | 40.4 | 33.9 | 35.5 |
| ET D100 | S | 36.8 | 45.4 | 40.7 | 40.6 | 43.4 | 40.5 | 38.4 | 31.7 | 37.1 | 31.4 | 33.1 | 37.4 | 36.7 | 36.6 | 35.6 | 36.1 | 34.6 | 34.0 | 40.4 | 37.9 | 35.5 |
| CLP D30 | E | 36.0 | 55.5 | 43.7 | 42.1 | 43.0 | 37.7 | 51.0 | 31.8 | 36.0 | 33.2 | 30.1 | 38.2 | 36.5 | 35.9 | 36.3 | 35.8 | 34.7 | 33.9 | 40.9 | 39.4 | 35.2 |
| EWMTF | E | 31.4 | 54.2 | 37.5 | 35.0 | 40.8 | 35.0 | 37.1 | 23.3 | 31.8 | 27.1 | 40.8 | 34.8 | 36.3 | 35.8 | 28.1 | 32.0 | 34.9 | 33.9 | 38.0 | 35.1 | 35.2 |
| EWMTF D100 | E | 33.7 | 54.2 | 37.5 | 37.3 | 40.8 | 35.0 | 37.1 | 27.2 | 31.8 | 30.7 | 40.8 | 34.8 | 36.3 | 35.8 | 33.4 | 32.0 | 34.9 | 33.9 | 38.0 | 36.3 | 35.2 |
| EW | E | 31.4 | 55.9 | 37.5 | 35.0 | 40.8 | 35.0 | 37.1 | 23.3 | 31.8 | 27.1 | 40.8 | 34.8 | 36.3 | 35.8 | 28.1 | 32.0 | 34.9 | 33.9 | 38.0 | 35.2 | 35.2 |
| ETF D30 | E | 36.5 | 54.7 | 41.2 | 39.7 | 43.0 | 41.3 | 38.8 | 32.0 | 35.7 | 32.5 | 32.2 | **43.3** | 36.1 | 36.2 | 34.8 | 35.6 | 34.3 | 34.2 | 39.9 | 38.7 | 35.2 |
| ET D30 | E | 36.5 | 54.4 | 41.0 | 39.6 | 42.8 | 41.2 | 38.7 | 32.1 | 35.9 | 32.5 | 32.1 | 43.2 | 36.1 | 36.2 | 34.8 | 35.6 | 34.3 | 34.2 | 39.8 | 38.7 | 35.2 |
| EF | E | 28.0 | 51.8 | 42.3 | 35.2 | 42.2 | 36.9 | 39.9 | 24.9 | 25.9 | 24.9 | 36.9 | 35.3 | 35.6 | 35.6 | 30.3 | 34.4 | 34.1 | 33.7 | 38.3 | 35.1 | 34.8 |
| EF D100 | E | 33.7 | 51.9 | 41.2 | 39.4 | 42.2 | 36.9 | 39.9 | 30.0 | 31.6 | 31.0 | 36.9 | 35.3 | 35.6 | 35.6 | 35.6 | 34.4 | 34.1 | 33.7 | 38.3 | 37.2 | 34.8 |
| EWT | S | 26.9 | 45.6 | 41.4 | 34.1 | 45.2 | 39.6 | 37.6 | 18.8 | 26.8 | 19.7 | 28.5 | 35.3 | 36.1 | 35.9 | 19.3 | 34.2 | 34.0 | 33.0 | 40.0 | 32.9 | 34.7 |
| BEATs | E | 24.0 | 58.6 | 31.2 | 17.9 | 29.0 | 32.0 | 46.9 | 9.1 | 34.8 | 30.0 | 27.6 | 28.8 | 36.2 | 35.5 | 33.2 | 29.9 | 34.5 | 32.7 | 38.8 | 31.4 | 34.7 |
| EWT D100 | S | 35.7 | 45.6 | 41.3 | 41.5 | 45.2 | 39.6 | 37.6 | 29.7 | 36.4 | 29.8 | 28.5 | 35.3 | 36.1 | 35.9 | 30.9 | 34.2 | 34.0 | 33.0 | 40.0 | 36.8 | 34.7 |
| EWTF D100 | S | 35.7 | 45.6 | 41.4 | 41.5 | 45.4 | 39.5 | 37.6 | 29.7 | 36.4 | 29.9 | 28.7 | 32.9 | 36.1 | 35.9 | 30.9 | 34.2 | 34.0 | 32.9 | 40.0 | 36.6 | 34.7 |
| EWTF | S | 26.9 | 45.6 | 41.4 | 34.0 | 45.4 | 39.5 | 37.6 | 18.8 | 26.8 | 19.8 | 28.7 | 32.9 | 36.1 | 35.9 | 19.2 | 34.2 | 34.0 | 32.9 | 40.0 | 32.7 | 34.7 |
| EWTF D30 | E | 35.8 | 57.5 | 43.1 | 41.0 | 44.1 | 39.6 | 38.5 | 28.5 | 35.3 | 31.7 | 28.7 | 33.6 | 35.9 | 35.2 | 31.3 | 33.8 | 34.1 | 33.2 | 39.3 | 37.5 | 34.6 |
| EWT D30 | E | 35.8 | 57.2 | 43.0 | 40.8 | 43.9 | 39.8 | 38.5 | 28.5 | 35.4 | 31.6 | 28.6 | 41.7 | 35.8 | 35.2 | 31.3 | 33.8 | 34.1 | 33.1 | 39.2 | 37.9 | 34.6 |
| EWF | E | 28.1 | 52.4 | 31.8 | 32.5 | 38.2 | 35.7 | 38.3 | 21.5 | 25.5 | 22.7 | 33.3 | 33.1 | 34.9 | 34.3 | 23.9 | 30.8 | 33.2 | 31.9 | 37.9 | 32.4 | 33.6 |
| EWF D100 | E | 32.6 | 52.4 | 35.9 | 36.4 | 38.2 | 35.7 | 38.3 | 26.4 | 30.9 | 28.7 | 33.3 | 33.1 | 34.9 | 34.3 | 30.7 | 30.8 | 33.2 | 31.9 | 37.9 | 34.8 | 33.6 |
| WLM | S | 10.1 | 46.6 | 29.3 | 19.8 | 30.8 | 35.2 | 34.9 | 8.7 | 21.3 | 16.1 | 37.0 | 24.2 | 34.1 | 34.8 | 28.2 | 31.7 | 31.9 | 31.9 | 33.4 | 27.1 | 33.2 |
| WLM D100 | S | 18.0 | 46.6 | 35.9 | 31.9 | 30.8 | 30.2 | 34.9 | 15.9 | 31.4 | 28.5 | 37.0 | 36.0 | 34.1 | 34.8 | 36.6 | 31.7 | 31.9 | 31.9 | 33.4 | 31.9 | 33.2 |
| EF D100 | S | 31.6 | 44.8 | 37.9 | 38.4 | 39.7 | 35.9 | 38.8 | 30.1 | 33.7 | 30.2 | 31.8 | 29.0 | 33.9 | 34.9 | 33.2 | 31.2 | 31.8 | 31.9 | 39.0 | 35.0 | 33.2 |
| EF | S | 18.7 | 44.0 | 28.7 | 28.0 | 39.7 | 35.9 | 38.8 | 16.4 | 21.2 | 15.7 | 31.8 | 29.0 | 33.9 | 34.9 | 19.0 | 31.2 | 31.8 | 31.9 | 39.0 | 29.1 | 33.2 |
| M | S | 35.9 | 65.8 | 50.2 | 44.2 | 44.2 | 29.9 | 30.5 | 17.5 | 22.9 | 25.7 | 29.1 | 25.7 | 34.0 | 33.4 | 27.0 | 29.6 | 32.8 | 31.6 | 34.0 | 34.2 | 33.0 |
| AC | S | 32.4 | 64.8 | 47.6 | 40.4 | 41.1 | 35.8 | 29.2 | 22.8 | 30.6 | 24.9 | 35.0 | 28.8 | 35.5 | 32.8 | 31.7 | 30.0 | 33.1 | 30.4 | 35.6 | 35.4 | 32.9 |
| M | C | 35.7 | 67.7 | **51.3** | 44.4 | 45.5 | 31.7 | 27.9 | 20.0 | 24.9 | 25.2 | 28.3 | 26.5 | 34.9 | 32.8 | 28.4 | 30.1 | 33.0 | 30.6 | 29.8 | 34.5 | 32.8 |
| BEATs | C | 22.6 | 59.5 | 30.4 | 17.1 | 28.2 | 30.9 | 44.5 | 7.9 | 32.5 | 28.4 | 26.6 | 26.9 | 34.1 | 33.4 | 31.9 | 28.3 | 32.5 | 30.7 | 37.9 | 30.2 | 32.7 |
| W2V D100 | E | 18.8 | 39.9 | 25.8 | 28.8 | 29.6 | 29.5 | 34.0 | 21.2 | 28.0 | 29.3 | 41.2 | 34.0 | 33.3 | 33.9 | 37.9 | 33.2 | 31.0 | 31.2 | 34.2 | 31.0 | 32.4 |
| W2V | E | 12.4 | 37.8 | 22.0 | 26.2 | 29.8 | 33.6 | 34.0 | 17.0 | 22.3 | 21.9 | 41.2 | 34.0 | 33.3 | 33.9 | 33.0 | 33.2 | 31.0 | 31.2 | 34.2 | 28.8 | 32.4 |
| EMTF D30 | E | 30.3 | 48.9 | 37.1 | 34.5 | 38.7 | 33.5 | 34.3 | 27.6 | 29.6 | 26.7 | 33.7 | 38.0 | 32.9 | 32.7 | 33.3 | 32.0 | 31.2 | 30.5 | 35.7 | 34.3 | 31.8 |
| EWMTF D30 | E | 31.4 | 54.5 | 35.4 | 35.0 | 38.7 | 32.6 | 34.1 | 23.3 | 29.1 | 27.1 | 35.6 | 34.8 | 32.7 | 32.4 | 28.1 | 27.0 | 31.1 | 30.2 | 35.8 | 33.5 | 31.6 |
| W2V D100 | S | 13.2 | 38.4 | 22.5 | 23.9 | 22.5 | 25.5 | 34.4 | 18.5 | 28.5 | 28.1 | 39.4 | 30.1 | 33.5 | 33.0 | 34.5 | 30.0 | 30.0 | 29.3 | 32.2 | 28.1 | 31.4 |
| W2V | S | 5.5 | 31.2 | 12.6 | 16.0 | 22.5 | 29.7 | 34.4 | 8.7 | 16.4 | 12.9 | 39.4 | 30.1 | 33.5 | 33.0 | 24.5 | 30.0 | 30.0 | 29.3 | 32.2 | 23.1 | 31.4 |
| EWF D100 | S | 30.8 | 44.7 | 36.2 | 36.0 | 36.7 | 33.9 | 38.7 | 27.8 | 32.0 | 27.4 | 27.7 | 23.0 | 31.9 | 32.8 | 28.9 | 27.5 | 29.5 | 29.7 | 38.7 | 32.7 | 31.0 |
| EWF | S | 18.6 | 44.7 | 36.2 | 24.4 | 36.7 | 33.9 | 38.7 | 12.8 | 20.1 | 13.5 | 27.7 | 23.0 | 31.9 | 32.8 | 12.4 | 27.5 | 29.5 | 29.7 | 38.7 | 27.3 | 31.0 |
| CC | C | 23.0 | 32.7 | 23.4 | 13.8 | 18.9 | 31.1 | 39.1 | 10.7 | 26.2 | 30.3 | 40.6 | 25.3 | 29.6 | 33.3 | 28.8 | 26.0 | 28.4 | 31.5 | 35.0 | 27.0 | 30.7 |
| WLM D30 | E | 20.4 | 53.0 | 39.2 | 29.4 | 34.4 | 30.4 | 31.2 | 19.7 | 29.4 | 27.3 | 31.0 | 40.8 | 31.1 | 32.6 | 38.2 | 33.9 | 28.7 | 29.7 | 32.1 | 32.7 | 30.5 |
| ETF | C | 25.0 | 55.6 | 37.8 | 30.5 | 39.9 | 36.7 | 33.4 | 18.8 | 23.8 | 18.8 | 26.5 | 36.7 | 31.4 | 31.2 | 22.1 | 31.5 | 28.9 | 28.2 | 35.4 | 31.5 | 30.0 |
| ETF D100 | C | 30.1 | 58.5 | 37.8 | 35.7 | 39.9 | 36.7 | 33.4 | 25.0 | 30.3 | 25.3 | 26.5 | 36.7 | 31.4 | 31.2 | 31.4 | 31.5 | 28.9 | 28.2 | 35.4 | 34.3 | 30.0 |
| ET | C | 25.1 | 58.0 | 37.4 | 30.4 | 39.6 | 36.4 | 33.3 | 18.4 | 24.1 | 18.7 | 26.3 | 36.5 | 31.4 | 31.2 | 22.1 | 31.5 | 29.0 | 28.2 | 35.4 | 31.5 | 29.9 |
| ET D100 | C | 30.1 | 58.0 | 37.4 | 35.6 | 39.6 | 36.4 | 33.3 | 25.1 | 30.5 | 25.2 | 26.3 | 36.5 | 31.4 | 31.2 | 31.4 | 31.5 | 29.0 | 28.2 | 35.4 | 34.2 | 29.9 |
| EF D30 | E | 28.0 | 52.4 | 37.1 | 35.2 | 38.3 | 31.3 | 34.5 | 24.9 | 25.9 | 24.9 | 36.6 | 35.3 | 30.2 | 30.3 | 29.1 | 28.3 | 28.1 | | 34.1 | 32.8 | 29.2 |
| CLP | C | 23.8 | 61.9 | 38.1 | 34.4 | 39.2 | 30.3 | **52.0** | 18.3 | 24.5 | 20.0 | 20.5 | 27.3 | 30.4 | 29.7 | 24.3 | 29.0 | 27.9 | 26.8 | 36.5 | 32.0 | 28.7 |
| CLP D100 | C | 28.5 | 60.5 | 40.6 | 37.8 | 39.2 | 30.4 | **52.0** | 23.1 | 29.3 | 25.5 | 20.5 | 27.3 | 30.4 | 29.7 | 29.4 | 29.0 | 27.9 | 26.8 | 36.5 | 34.0 | 28.7 |
| EWTF | C | 24.1 | 60.2 | 40.0 | 32.7 | 42.0 | 33.8 | 33.1 | 14.0 | 23.3 | 17.8 | 21.4 | 27.8 | 30.5 | 29.3 | 17.3 | 28.5 | 28.0 | 26.3 | 34.1 | 30.0 | 28.5 |
| EWTF D100 | C | 28.5 | 62.5 | 39.8 | 36.8 | 42.0 | 33.7 | 33.1 | 20.1 | 29.2 | 24.0 | 21.4 | 27.8 | 30.5 | 29.3 | 25.9 | 28.5 | 28.0 | 26.3 | 34.1 | 32.5 | 28.5 |
| EWT | C | 24.1 | 59.7 | 40.0 | 32.5 | 41.7 | 33.7 | 33.1 | 14.1 | 23.4 | 17.6 | 21.2 | 34.3 | 30.4 | 29.2 | 17.3 | 28.5 | 28.0 | 26.2 | 33.9 | 30.3 | 28.5 |
| EWT D100 | C | 28.4 | 61.8 | 39.5 | 36.6 | 41.7 | 33.7 | 33.1 | 20.1 | 29.3 | 23.8 | 21.2 | 34.3 | 30.4 | 29.2 | 25.9 | 28.5 | 28.0 | 26.2 | 33.9 | 32.8 | 28.5 |
| MAE | E | 20.7 | 45.4 | 28.2 | 24.3 | 28.6 | 36.2 | 37.7 | 16.2 | 24.9 | 22.0 | 32.7 | 37.0 | 29.4 | 29.3 | 24.7 | 29.2 | 27.5 | 26.7 | 36.2 | 29.6 | 28.2 |
| MAE D100 | E | 25.8 | 44.0 | 28.2 | 29.4 | 28.6 | 27.3 | 37.7 | 20.7 | 30.3 | 28.9 | 32.7 | 37.0 | 29.4 | 29.3 | 30.3 | 29.2 | 27.5 | 26.7 | 36.2 | 31.1 | 28.2 |
| EWF D30 | E | 28.1 | 52.4 | 31.8 | 32.5 | 33.9 | 30.5 | 32.6 | 21.5 | 25.5 | 22.7 | 27.1 | 24.3 | 29.3 | 29.0 | 23.9 | 24.6 | 27.4 | 26.0 | 34.0 | 29.7 | 27.9 |
| MAE D100 | S | 23.1 | 43.3 | 24.7 | 27.9 | 22.9 | 26.0 | 37.1 | 17.6 | 29.0 | 25.6 | 29.7 | - | 28.1 | 28.8 | 26.8 | 25.3 | 25.2 | 25.2 | 35.0 | 28.1 | 26.8 |

Table 20: (Continued) GSR Results Across Subsets and Distances (↑ better)

| Method | Dist | BC | BS1 | BS2 | BS3 | BS4 | BS5 | ES1 | HP | HS1 | HS2 | HU1 | HU2 | HU3 | HU4 | HW1 | HW2 | HW3 | HW4 | OC1 | Avg | Avg (Blind) |
|---|---|---|---|---|---|---|---|---|---|---|---|---|---|---|---|---|---|---|---|---|---|---|
| MAE | S | 12.2 | 43.3 | 24.7 | 15.7 | 22.9 | 24.0 | 37.1 | 8.4 | 17.1 | 12.7 | 29.7 | - | 28.1 | 28.8 | 15.1 | 25.3 | 25.2 | 25.2 | 35.0 | 23.1 | 26.8 |
| VC | E | 30.7 | 64.0 | 47.5 | 42.6 | 46.1 | 38.2 | 28.4 | 20.9 | 27.2 | 24.3 | 27.2 | 26.2 | 27.9 | 26.8 | 28.3 | 28.8 | 25.3 | 23.5 | 33.5 | 34.3 | 25.9 |
| W2V D30 | E | 12.4 | 37.8 | 22.0 | 26.2 | 24.6 | 25.0 | 29.3 | 17.0 | 22.3 | 21.9 | 32.9 | 34.0 | 26.3 | 28.2 | 33.0 | 28.1 | 24.0 | 24.7 | 28.4 | 26.3 | 25.8 |
| CLP D30 | S | 27.3 | 45.6 | 37.3 | 34.2 | 37.2 | 30.9 | 50.3 | 21.2 | 28.7 | 21.6 | 19.2 | 27.8 | 27.0 | 27.0 | 25.5 | 25.2 | 24.0 | 23.7 | 34.0 | 31.1 | 25.5 |
| WLM D100 | C | 13.7 | 55.2 | 37.4 | 23.3 | 26.8 | 25.9 | 26.7 | 11.2 | 24.4 | 22.3 | 30.4 | 37.2 | 26.7 | 27.4 | 34.1 | 28.4 | 23.2 | 23.8 | 29.0 | 28.4 | 25.3 |
| WLM | C | 7.9 | 55.3 | 32.1 | 16.4 | 26.8 | 34.3 | 26.7 | 7.1 | 17.5 | 13.7 | 30.4 | 37.2 | 26.7 | 27.4 | 27.0 | 28.4 | 23.2 | 23.8 | 29.0 | 26.0 | 25.3 |
| AC | E | 33.8 | 55.6 | 46.7 | 41.6 | 44.1 | 35.8 | 25.6 | 28.6 | 24.4 | 22.2 | 36.0 | 30.1 | 27.4 | 23.6 | 34.7 | 33.3 | 26.2 | 22.1 | 32.1 | 35.0 | 24.8 |
| ET D30 | S | 27.4 | 45.1 | 33.9 | 32.1 | 35.8 | 34.0 | 29.9 | 20.8 | 27.0 | 20.9 | 20.1 | 37.4 | 26.0 | 26.7 | 23.3 | 24.5 | 23.3 | 23.2 | 32.9 | 29.7 | 24.8 |
| ETF D30 | S | 27.4 | 45.1 | 34.4 | 31.9 | 36.1 | 34.7 | 29.8 | 20.8 | 26.9 | 21.0 | 20.3 | 38.2 | 26.1 | 26.5 | 23.4 | 24.4 | 23.3 | 23.3 | 32.7 | 29.8 | 24.8 |
| EMTF D30 | S | 24.7 | 42.5 | 32.2 | 28.9 | 32.7 | 30.7 | 29.1 | 19.7 | 26.2 | 20.2 | 28.6 | 33.6 | 26.2 | 26.4 | 25.3 | 23.8 | 22.7 | 22.7 | 31.0 | 28.6 | 24.5 |
| EWTF D30 | S | 26.9 | 46.7 | 36.8 | 34.0 | 38.6 | 32.6 | 29.4 | 18.8 | 26.8 | 19.8 | 16.4 | 21.7 | 26.0 | 26.0 | 19.2 | 22.8 | 23.1 | 22.5 | 32.3 | 28.2 | 24.4 |
| EWT D30 | S | 26.9 | 46.3 | 36.6 | 34.1 | 38.5 | 33.0 | 29.5 | 18.8 | 26.8 | 19.7 | 16.3 | 35.3 | 25.8 | 26.0 | 19.3 | 22.8 | 23.1 | 22.4 | 32.0 | 29.1 | 24.3 |
| EWMTF D30 | S | 24.3 | 46.9 | 32.0 | 29.8 | 33.1 | 30.2 | 29.1 | 18.5 | 26.3 | 20.3 | 28.2 | 27.7 | 25.3 | 25.7 | 23.6 | 21.3 | 21.8 | 22.1 | 31.3 | 28.2 | 23.7 |
| EMTF | C | 16.8 | 48.4 | 30.1 | 22.1 | 32.5 | 25.8 | 26.6 | 13.1 | 16.1 | 12.0 | 28.8 | 28.1 | 25.4 | 24.5 | 20.0 | 25.2 | 23.1 | 21.8 | 27.8 | 24.9 | 23.7 |
| EMTF D100 | C | 20.9 | 48.7 | 30.1 | 26.6 | 32.5 | 25.8 | 26.6 | 18.0 | 20.0 | 16.3 | 28.8 | 28.1 | 25.4 | 24.5 | 28.0 | 25.2 | 23.1 | 21.8 | 27.8 | 26.9 | 23.7 |
| E | C | 16.8 | 48.4 | - | 22.1 | 32.5 | 38.5 | 26.6 | 13.1 | 16.1 | 12.0 | 28.8 | 28.1 | 25.4 | 24.5 | 20.0 | 25.2 | 23.1 | 21.8 | 27.8 | 25.4 | 23.7 |
| EW | C | 18.0 | 61.2 | 40.0 | 23.1 | 32.4 | 23.3 | 27.0 | 8.6 | 15.9 | 12.6 | 31.4 | 22.7 | 25.1 | 24.4 | 13.1 | 18.2 | 22.9 | 21.8 | 27.7 | 25.0 | 23.5 |
| EWMTF | C | 18.0 | 58.3 | 27.4 | 23.1 | 32.4 | 23.3 | 27.0 | 8.6 | 15.9 | 12.6 | 31.4 | 22.7 | 25.1 | 24.4 | 13.1 | 18.2 | 22.9 | 21.8 | 27.7 | 24.0 | 23.5 |
| EWMTF D100 | C | 21.4 | 58.0 | 27.6 | 26.7 | 32.4 | 23.3 | 27.0 | 12.4 | 19.4 | 16.9 | 31.4 | 22.7 | 25.1 | 24.4 | 19.9 | 18.2 | 22.9 | 21.8 | 27.7 | 25.7 | 23.5 |
| CLP D30 | C | 23.8 | 61.9 | 38.1 | 34.4 | 35.6 | 26.5 | 51.2 | 18.3 | 24.5 | 20.0 | 15.3 | 27.3 | 24.9 | 24.3 | 24.3 | 23.8 | 22.0 | 21.1 | 32.6 | 30.5 | 23.1 |
| ETF D30 | C | 25.0 | 60.0 | 33.6 | 30.5 | 35.6 | 32.4 | 28.3 | 18.3 | 23.8 | 18.8 | 17.9 | 36.7 | 24.2 | 24.6 | 22.1 | 23.2 | 21.4 | 21.3 | 30.5 | 29.1 | 22.9 |
| ET D30 | C | 25.1 | 59.4 | 33.2 | 30.4 | 35.3 | 32.1 | 28.2 | 18.4 | 24.1 | 18.7 | 17.8 | 36.5 | 24.2 | 24.6 | 22.1 | 23.2 | 21.4 | 21.3 | 30.5 | 29.0 | 22.9 |
| MAE D30 | E | 20.7 | 42.0 | 23.0 | 24.3 | 23.5 | 20.6 | 32.5 | 16.2 | 24.9 | 22.0 | 23.3 | 37.0 | 23.9 | 24.1 | 24.7 | 23.8 | 21.9 | 21.4 | 31.5 | 26.0 | 22.8 |
| EAT | E | 17.9 | 53.9 | 31.6 | 28.5 | 29.2 | 25.6 | 25.7 | 11.4 | 18.3 | 21.9 | 24.7 | 16.8 | 23.3 | 23.8 | 19.7 | 17.0 | 21.5 | 21.8 | 28.4 | 24.7 | 22.6 |
| EF D100 | C | 21.2 | 54.5 | 33.7 | 30.1 | 34.7 | 25.9 | 31.6 | 16.4 | 18.7 | 17.2 | 25.5 | 23.0 | 23.8 | 23.8 | 23.7 | 22.2 | 21.5 | 21.1 | 28.5 | 27.1 | 22.5 |
| EF | C | 13.5 | 54.8 | 33.7 | 23.4 | 34.7 | 25.9 | 31.6 | 10.7 | 12.2 | 10.0 | 25.5 | 23.0 | 23.8 | 23.8 | 16.1 | 22.2 | 21.5 | 21.1 | 28.5 | 24.4 | 22.5 |
| EWTF D30 | C | 24.1 | 64.4 | 37.3 | 32.7 | 37.9 | 29.8 | 27.9 | 14.0 | 23.3 | 17.8 | 13.8 | 20.2 | 23.8 | 23.0 | 17.3 | 20.6 | 21.1 | 19.9 | 29.5 | 27.4 | 21.9 |
| EWT D30 | C | 24.1 | 63.6 | 37.0 | 32.5 | 37.6 | 30.0 | 27.9 | 14.1 | 23.4 | 17.6 | 13.7 | 34.3 | 23.7 | 22.9 | 17.3 | 20.5 | 21.1 | 19.9 | 29.3 | 28.2 | 21.9 |
| EF D30 | S | 18.7 | 44.0 | 28.7 | 28.0 | 31.1 | 23.7 | 26.9 | 16.4 | 21.2 | 15.7 | 18.7 | 29.0 | 22.1 | 23.4 | 19.0 | 18.2 | 19.7 | 19.7 | 27.9 | 24.5 | 21.2 |
| WLM D30 | S | 10.1 | 46.6 | 29.3 | 19.8 | 24.4 | 22.3 | 23.2 | 8.7 | 21.3 | 16.1 | 19.1 | 24.2 | 22.0 | 23.7 | 28.2 | 23.7 | 19.1 | 19.8 | 22.1 | 22.6 | 21.2 |
| EWF | C | 14.1 | 52.9 | 19.7 | 20.0 | 28.0 | 24.0 | 29.9 | 7.4 | 11.4 | 8.1 | 19.8 | 19.7 | 22.8 | 22.3 | 9.1 | 17.1 | 20.2 | 18.7 | 27.8 | 20.6 | 21.0 |
| EWF D100 | C | 19.8 | 54.4 | 25.1 | 25.7 | 28.0 | 24.0 | 29.9 | 12.1 | 17.5 | 14.3 | 19.8 | 19.7 | 22.8 | 22.3 | 16.6 | 17.1 | 20.2 | 18.7 | 27.8 | 23.4 | 21.0 |
| W2V D100 | C | 6.4 | 33.1 | 12.1 | 15.2 | 16.0 | 17.7 | 22.7 | 8.2 | 17.0 | 16.6 | 32.2 | 28.5 | 22.0 | 23.1 | 30.0 | 24.1 | 18.9 | 19.1 | 24.6 | 20.3 | 20.8 |
| W2V | C | 2.7 | 29.3 | 8.5 | 12.1 | 16.0 | 23.8 | 22.7 | 4.9 | 10.3 | 8.6 | 32.2 | 28.5 | 22.0 | 23.1 | 21.6 | 24.1 | 18.9 | 19.1 | 24.6 | 18.0 | 20.8 |
| EWF D30 | S | 18.6 | 44.4 | 25.2 | 24.4 | 24.8 | 23.4 | 26.2 | 12.8 | 20.1 | 13.5 | 16.1 | 13.1 | 20.2 | 20.7 | 12.4 | 13.2 | 17.4 | 16.7 | 27.7 | 21.1 | 18.8 |
| EWMTF D30 | C | 18.0 | 58.3 | 24.5 | 23.1 | 28.8 | 19.8 | 22.3 | 8.6 | 15.9 | 12.6 | 22.9 | 22.7 | 19.5 | 19.4 | 13.1 | 12.2 | 17.3 | 16.7 | 23.9 | 21.8 | 18.2 |
| EMTF D30 | C | 16.8 | 48.4 | 26.2 | 22.1 | 28.5 | 21.7 | 22.2 | 13.1 | 16.1 | 12.0 | 26.0 | 28.1 | 19.7 | 19.4 | 20.0 | 18.3 | 17.3 | 16.4 | 23.9 | 22.5 | 18.2 |
| W2V D30 | S | 5.5 | 31.2 | 12.6 | 16.0 | 13.5 | 15.8 | 22.1 | 8.7 | 16.4 | 12.9 | 22.2 | 30.1 | 19.2 | 20.1 | 24.5 | 19.8 | 15.8 | 15.6 | 20.5 | 18.1 | 17.7 |
| CC | S | 2.6 | 38.0 | 18.3 | 9.7 | 4.4 | 2.9 | 42.7 | 1.5 | 7.4 | 17.0 | 28.9 | 2.0 | 16.4 | 19.2 | 11.5 | 11.2 | 17.6 | 17.3 | 3.5 | 13.4 | 17.6 |
| WLM D30 | C | 7.9 | 55.3 | 32.1 | 16.4 | 22.8 | 20.0 | 18.1 | 7.1 | 17.5 | 13.7 | 16.4 | 37.2 | 18.1 | 19.9 | 27.0 | 21.9 | 14.8 | 16.1 | 19.7 | 22.2 | 17.2 |
| EF D30 | C | 13.5 | 54.8 | 27.0 | 23.4 | 27.7 | 17.5 | 22.4 | 10.7 | 12.2 | 10.0 | 16.1 | 23.0 | 16.0 | 16.5 | 16.1 | 15.0 | 13.9 | 13.8 | 21.0 | 20.7 | 15.1 |
| MAE D30 | S | 12.2 | 38.4 | 14.1 | 15.7 | 13.2 | 12.6 | 25.8 | 8.4 | 17.1 | 12.7 | 13.0 | - | 16.1 | 17.0 | 15.1 | 14.0 | 13.1 | 13.6 | 23.8 | 16.9 | 14.9 |
| MAE | C | 7.3 | 38.9 | 9.6 | 10.1 | 14.9 | 24.2 | 29.4 | 4.5 | 10.7 | 7.9 | 19.6 | 26.3 | 16.1 | 15.9 | 10.4 | 15.5 | 13.7 | 12.7 | 25.9 | 17.0 | 14.6 |
| MAE D100 | C | 12.2 | 38.9 | 14.8 | 15.8 | 14.9 | 13.6 | 29.4 | 7.7 | 16.9 | 14.9 | 19.6 | 26.3 | 16.1 | 15.9 | 16.6 | 15.5 | 13.7 | 12.7 | 25.9 | 18.9 | 14.6 |
| EWF D30 | C | 14.1 | 54.2 | 19.7 | 20.0 | 21.4 | 16.6 | 20.6 | 7.4 | 11.4 | 8.1 | 12.1 | 9.9 | 15.1 | 15.2 | 9.1 | 10.0 | 12.8 | 11.6 | 21.6 | 17.1 | 13.7 |
| W2V D30 | C | 2.7 | 29.3 | 8.5 | 12.1 | 10.5 | 12.0 | 15.7 | 4.9 | 10.3 | 8.6 | 19.0 | 28.5 | 12.7 | 14.7 | 21.6 | 16.3 | 10.4 | 10.9 | 15.8 | 14.4 | 12.2 |
| AC | C | 21.2 | 59.1 | 42.4 | 33.5 | 37.2 | 24.0 | 15.6 | 13.8 | 9.9 | 9.5 | 25.5 | 15.8 | 14.6 | 10.3 | 22.0 | 20.0 | 12.6 | 8.9 | 19.4 | 24.6 | 11.6 |
| EAT | S | 8.4 | 53.6 | 25.2 | 20.5 | 21.5 | 15.8 | 15.5 | 3.1 | 8.1 | 11.0 | 14.7 | 7.6 | 12.4 | 12.5 | 10.4 | 8.1 | 10.4 | 10.3 | 18.7 | 16.2 | 11.4 |
| EAT | C | 7.6 | 53.9 | 23.6 | 17.0 | 18.9 | 15.1 | 14.8 | 2.5 | 7.6 | 10.4 | 13.4 | 7.2 | 11.8 | 11.9 | 0.0 | 7.8 | 9.1 | 9.8 | 9.7 | 18.3 | 15.2 | 10.8 |
| VC | C | 15.8 | **75.8** | 45.3 | 35.4 | 40.9 | 27.0 | 14.1 | 6.9 | 12.3 | 9.2 | 11.5 | 10.4 | 12.6 | 11.4 | 13.3 | 13.3 | 9.8 | 8.2 | 19.3 | 23.4 | 10.5 |
| MAE D30 | C | 7.3 | 35.6 | 9.6 | 10.1 | 9.3 | 7.1 | 21.3 | 4.5 | 10.7 | 7.9 | 9.0 | 26.3 | 10.0 | 10.0 | 10.4 | 9.7 | 8.1 | 7.6 | 18.3 | 13.1 | 8.9 |
| VC | S | 10.4 | 62.0 | 26.1 | 21.3 | 28.1 | 19.5 | 8.8 | 3.7 | 10.2 | 5.9 | 5.7 | 5.1 | 8.7 | 8.1 | 6.6 | 6.1 | 5.7 | 4.9 | 15.5 | 15.7 | 6.9 |

Table 21: CSR Results Across Subsets and Distances (↑ better)

| Method | Dist | BC | BS1 | BS2 | BS3 | BS4 | BS5 | ES1 | HP | HS1 | HS2 | HU1 | HU2 | HU3 | HU4 | HW1 | HW2 | HW3 | HW4 | OC1 | Avg | Avg (Blind) |
|---|---|---|---|---|---|---|---|---|---|---|---|---|---|---|---|---|---|---|---|---|---|---|
| ETF | E | - | 42.7 | 33.4 | 28.1 | 32.6 | 28.5 | 34.8 | 24.9 | - | 26.6 | 33.2 | - | 36.6 | 35.8 | 29.6 | 37.4 | 35.1 | 34.3 | 36.8 | 33.1 | 35.4 |
| ETF D100 | E | - | 42.1 | 31.8 | 31.5 | 32.6 | 28.5 | 34.8 | 30.0 | - | 31.9 | 33.2 | - | 36.6 | 35.8 | 36.3 | 37.4 | 35.1 | 34.3 | 36.8 | 34.3 | 35.4 |
| ET D100 | E | - | 41.9 | 31.6 | 31.4 | 32.4 | 28.3 | 34.6 | 30.0 | - | 31.8 | 33.1 | - | 36.5 | 35.8 | 36.3 | 37.4 | 35.1 | 34.3 | 36.8 | 34.2 | 35.4 |
| ET | E | - | 42.7 | 31.6 | 28.0 | 32.4 | 28.3 | 34.6 | 25.0 | - | 26.6 | 33.1 | - | 36.5 | 35.8 | 29.6 | 37.4 | 35.1 | 34.3 | 36.8 | 33.0 | 35.4 |
| EWTF D100 | E | - | 40.5 | 31.4 | 30.5 | 33.5 | 26.8 | 35.1 | 26.1 | - | - | 30.2 | - | 36.0 | 34.8 | 33.4 | 35.6 | 34.6 | 33.0 | 36.2 | 33.2 | 34.6 |
| EWTF | E | - | 40.3 | 29.1 | 27.6 | 33.5 | 26.8 | 35.1 | 21.3 | - | 26.1 | 30.2 | - | 36.0 | 34.8 | 26.7 | 35.6 | 34.6 | 33.0 | 36.2 | 31.7 | 34.6 |
| EWT | E | - | 40.3 | 28.9 | 27.4 | 33.3 | 26.6 | 35.0 | 21.3 | - | 26.0 | 30.1 | - | 36.0 | 34.8 | 26.7 | 35.6 | 34.5 | 33.0 | 36.1 | 31.6 | 34.6 |
| EWT D100 | E | - | 40.2 | 31.2 | 30.3 | 33.3 | 26.6 | 35.0 | 26.1 | - | 31.2 | 30.1 | - | 36.0 | 34.8 | 33.4 | 35.6 | 34.5 | 33.0 | 36.1 | 33.0 | 34.6 |
| CLP D100 | E | - | 41.7 | 32.0 | 31.7 | 31.9 | 26.2 | 42.4 | 27.7 | - | - | 29.0 | - | 35.5 | 34.7 | 34.0 | 34.8 | 34.1 | 32.9 | 37.1 | 33.7 | 34.3 |
| CLP | E | - | 41.8 | 32.4 | 28.8 | 31.9 | 26.8 | 42.4 | 24.2 | - | 26.6 | 29.0 | - | 35.5 | 34.7 | 30.6 | 34.8 | 34.1 | 32.9 | 37.1 | 32.7 | 34.3 |
| CC | E | 24.0 | 33.1 | 27.3 | 20.5 | 23.6 | 29.6 | 41.1 | 16.3 | 24.4 | 31.4 | 41.1 | 26.5 | 33.3 | 34.9 | 32.9 | 31.8 | 32.2 | 33.3 | 34.1 | 30.1 | 33.4 |
| M | E | - | 48.4 | 40.1 | 36.8 | 35.5 | 24.5 | 25.8 | 17.7 | - | - | 32.2 | - | 34.5 | 30.9 | 30.2 | 32.4 | 33.1 | - | 31.9 | 32.4 | 32.9 |
| CLP D100 | S | 30.2 | 38.0 | 32.4 | 31.4 | 31.8 | 28.3 | 39.6 | 26.4 | 32.0 | 26.7 | 26.2 | 20.3 | 33.3 | 33.6 | 30.4 | 31.8 | 31.2 | 30.7 | 36.3 | 31.1 | 32.2 |
| CLP | S | 18.3 | 33.6 | 32.4 | 20.4 | 31.8 | 13.2 | 39.6 | 14.5 | 20.5 | 15.3 | 26.2 | 20.3 | 33.3 | 33.6 | 19.0 | 31.8 | 31.2 | 30.7 | 36.3 | 26.4 | 32.2 |
| EMTF | S | 16.3 | 30.8 | 31.1 | 17.3 | 30.3 | 28.7 | 31.8 | 13.6 | 18.7 | 14.4 | 39.5 | 25.4 | 33.5 | 33.3 | 18.8 | 33.3 | 31.0 | 30.8 | 35.1 | 27.0 | 32.2 |
| EMTF D100 | S | 29.1 | 37.5 | 31.1 | 30.3 | 30.3 | 28.7 | 31.8 | 25.9 | 30.8 | 27.2 | 39.5 | 25.4 | 33.5 | 33.3 | 34.2 | 33.3 | 31.0 | 30.8 | 35.1 | 31.5 | 32.2 |
| EWMTF | S | 16.1 | 38.1 | 13.0 | 17.7 | 30.4 | 29.2 | 31.2 | 12.7 | 18.7 | 14.2 | 39.0 | 20.4 | 33.0 | 33.2 | 17.5 | 32.2 | 30.6 | 30.1 | 34.9 | 25.9 | 31.7 |
| EWMTF D100 | S | 28.4 | 38.1 | 30.8 | 29.8 | 30.4 | 29.2 | 31.2 | 25.4 | 30.7 | 26.3 | 39.0 | 20.4 | 33.0 | 33.2 | 33.7 | 32.2 | 30.6 | 30.1 | 34.9 | 30.9 | 31.7 |
| ETF D100 | S | 30.0 | 38.0 | 31.7 | 30.8 | 30.5 | 29.1 | 31.3 | 26.3 | 31.3 | 26.7 | 28.1 | 33.4 | 32.8 | 33.0 | 30.9 | 32.4 | 30.9 | 30.2 | 35.5 | 31.2 | 31.7 |
| ETF | S | 18.7 | 38.0 | 31.7 | 19.1 | 30.5 | 29.1 | 31.3 | 14.6 | 19.6 | 15.3 | 28.1 | 33.4 | 32.8 | 33.0 | 17.8 | 32.4 | 30.9 | 30.2 | 35.5 | 27.5 | 31.7 |
| ET | S | 18.7 | 38.0 | 21.0 | 30.8 | 30.4 | 29.0 | 31.3 | 14.6 | 19.6 | 15.3 | 28.0 | 32.6 | 32.7 | 33.0 | 17.8 | 32.4 | 30.8 | 30.2 | 35.3 | 27.5 | 31.7 |
| ET D100 | S | 30.1 | 38.0 | 31.6 | 30.8 | 30.4 | 29.0 | 31.3 | 26.4 | 31.3 | 26.7 | 28.0 | 32.6 | 32.7 | 33.0 | 30.9 | 32.4 | 30.8 | 30.2 | 35.3 | 31.1 | 31.7 |
| WLM D100 | E | - | 41.6 | 28.8 | 24.1 | 23.2 | 22.4 | 31.3 | 17.2 | - | - | 34.4 | - | 32.0 | 32.0 | 33.1 | 32.0 | 30.2 | 30.1 | 30.2 | 29.5 | 31.1 |
| WLM | E | - | 39.5 | 23.2 | 18.5 | 23.2 | 28.1 | 31.3 | 13.5 | - | - | 34.4 | - | 32.0 | 32.0 | 28.4 | 32.0 | 30.2 | 30.1 | 30.2 | 28.4 | 31.1 |
| EWT D100 | S | 28.7 | 37.8 | 31.7 | 30.7 | 31.9 | 28.6 | 30.5 | 24.7 | 30.8 | 25.4 | 24.1 | 30.7 | 32.2 | 32.2 | 26.9 | 30.6 | 30.2 | 29.2 | 34.7 | 30.1 | 31.0 |
| EWT | S | 17.8 | 37.8 | 31.7 | 19.7 | 31.9 | 28.6 | 30.5 | 13.2 | 19.6 | 14.4 | 24.1 | 30.7 | 32.2 | 32.2 | 14.8 | 30.6 | 30.2 | 29.2 | 34.7 | 26.5 | 31.0 |
| EWTF D100 | S | 28.7 | 37.9 | 31.7 | 30.8 | 32.0 | 28.6 | 30.6 | 24.7 | 30.8 | 25.5 | 24.3 | 28.4 | 32.2 | 32.2 | 26.9 | 30.6 | 30.2 | 29.2 | 34.8 | 30.0 | 30.9 |
| EWTF | S | 17.8 | 37.9 | 31.7 | 19.8 | 32.0 | 28.6 | 30.6 | 13.2 | 19.5 | 14.5 | 24.3 | 28.4 | 32.2 | 32.2 | 14.8 | 30.6 | 30.2 | 29.2 | 34.8 | 26.4 | 30.9 |
| ET D30 | E | - | 41.5 | 28.5 | 28.0 | 29.5 | 23.9 | 30.7 | 25.0 | - | 26.6 | 27.0 | - | 31.4 | 31.5 | 29.6 | 31.8 | 29.6 | 29.3 | 33.4 | 29.8 | 30.5 |
| ETF D30 | E | - | 41.7 | 28.6 | 28.1 | 29.6 | 24.1 | 30.8 | 24.9 | - | 26.6 | 27.1 | - | 31.4 | 31.5 | 29.6 | 31.8 | 29.6 | 29.2 | 33.4 | 29.9 | 30.4 |
| CLP D30 | E | - | 41.5 | 29.4 | 28.8 | 29.0 | 23.1 | 40.8 | 24.2 | - | 26.6 | 24.9 | - | 31.8 | 31.0 | 30.6 | 31.4 | 30.0 | 28.9 | 34.4 | 30.4 | 30.4 |
| EWTF D30 | E | - | 40.3 | 29.1 | 27.6 | 30.4 | 23.6 | 31.0 | 21.3 | - | 26.1 | 24.1 | - | 31.2 | 30.5 | 26.7 | 30.1 | 29.5 | 28.2 | 33.1 | 28.9 | 29.9 |
| EWT D30 | E | - | 39.9 | 28.9 | 27.4 | 30.3 | 23.5 | 31.0 | 21.3 | - | 26.0 | 24.1 | - | 31.2 | 30.5 | 26.7 | 30.1 | 29.5 | 28.3 | 33.0 | 28.8 | 29.9 |
| WLM D100 | S | 13.5 | 37.5 | 26.4 | 22.7 | 17.8 | 20.7 | 29.6 | 12.8 | 26.6 | 24.5 | 29.8 | 25.7 | 30.9 | 31.3 | 25.2 | 25.6 | 28.7 | 28.4 | 28.5 | 25.6 | 29.8 |
| WLM | S | 6.3 | 33.0 | 16.8 | 10.2 | 17.8 | 20.0 | 29.6 | 5.7 | 15.8 | 11.8 | 29.8 | 37.9 | 30.9 | 31.3 | 16.5 | 25.6 | 28.7 | 28.4 | 28.5 | 22.3 | 29.8 |
| EF D100 | S | 25.6 | 37.8 | 29.2 | 29.6 | 29.0 | 27.2 | 33.7 | 24.8 | 28.7 | 26.0 | 27.1 | 20.1 | 30.7 | 31.4 | 28.5 | 28.0 | 28.6 | 28.3 | 34.0 | 28.9 | 29.8 |
| EF | S | 12.2 | 32.1 | 16.8 | 16.9 | 29.0 | 27.2 | 33.7 | 11.4 | 15.4 | 11.5 | 27.1 | 20.1 | 30.7 | 31.4 | 14.1 | 28.0 | 28.6 | 28.3 | 34.0 | 23.6 | 29.8 |
| EMTF D100 | E | - | 35.7 | 25.6 | 24.5 | 26.9 | 18.6 | 25.2 | 22.0 | - | - | 30.4 | - | 30.5 | 29.6 | 31.2 | 31.6 | 29.2 | 27.3 | 29.6 | 27.8 | 29.1 |
| E | E | - | 35.7 | 31.6 | 21.6 | 26.9 | 34.7 | 25.2 | 18.2 | - | 17.0 | 30.4 | - | 30.5 | 29.6 | 26.0 | 31.6 | 29.2 | 27.3 | 29.6 | 27.8 | 29.1 |
| EMTF | E | - | 35.7 | 27.0 | 21.6 | 26.9 | 18.6 | 25.2 | 18.2 | - | 17.0 | 30.4 | - | 30.5 | 29.6 | 26.0 | 31.6 | 29.2 | 27.3 | 29.6 | 26.5 | 29.1 |
| W2V D100 | S | 10.2 | 31.9 | 16.9 | 18.2 | 16.2 | 18.6 | 29.5 | 15.2 | 24.5 | 24.3 | 32.7 | 20.6 | 30.6 | 29.8 | 28.0 | 26.4 | 27.2 | 26.3 | 27.8 | 23.9 | 28.5 |
| W2V | S | 3.5 | 22.3 | 7.4 | 9.9 | 16.2 | 17.9 | 29.5 | 6.2 | 12.3 | 9.5 | 32.7 | 20.6 | 30.6 | 29.8 | 17.2 | 26.4 | 27.2 | 26.3 | 27.8 | 19.7 | 28.5 |
| EF | E | - | 39.2 | 29.4 | 22.3 | 29.9 | 24.0 | 33.4 | 15.2 | - | 18.0 | 31.5 | - | 30.4 | 28.5 | 24.2 | 30.1 | 28.1 | 26.3 | 30.1 | 27.6 | 28.3 |
| EF D100 | E | - | 39.0 | 28.0 | 27.1 | 29.9 | 24.0 | 33.4 | 19.6 | - | 24.0 | 31.5 | - | 30.4 | 28.5 | 29.7 | 30.1 | 28.1 | 26.3 | 30.1 | 28.7 | 28.3 |
| EWMTF | E | - | 37.3 | 23.2 | 21.2 | 27.5 | 20.2 | 24.8 | 15.0 | - | 17.5 | 32.7 | - | 30.2 | 28.1 | 20.7 | 26.7 | 28.6 | 25.8 | 29.1 | 25.5 | 28.2 |
| EWMTF D100 | E | - | 37.3 | 23.2 | 23.4 | 27.5 | 20.2 | 24.8 | 18.3 | - | - | 32.7 | - | 30.2 | 28.1 | 25.7 | 26.7 | 28.6 | 25.8 | 29.1 | 26.8 | 28.2 |
| EW | E | - | 39.1 | 23.2 | 21.2 | 27.5 | 20.2 | 24.8 | 15.0 | - | 17.5 | 32.7 | - | 30.2 | 28.1 | 20.7 | 26.7 | 28.6 | 25.8 | 29.1 | 25.7 | 28.2 |
| EWF D100 | S | 25.0 | 37.5 | 27.6 | 27.3 | 27.0 | 26.0 | 33.5 | 22.9 | 27.1 | 23.5 | 23.1 | 14.9 | 28.9 | 29.6 | 24.9 | 24.7 | 26.4 | 26.4 | 33.7 | 26.8 | 27.8 |
| EWF | S | 12.4 | 37.5 | 27.6 | 14.3 | 27.0 | 26.0 | 33.5 | 8.9 | 14.6 | 9.8 | 23.1 | 14.9 | 28.9 | 29.6 | 9.2 | 24.7 | 26.4 | 26.4 | 33.7 | 22.6 | 27.8 |
| AC | S | 23.2 | 45.3 | 28.5 | 22.7 | 21.3 | 20.6 | 20.2 | 16.4 | 22.0 | 18.9 | 28.8 | 23.5 | 30.8 | 27.0 | 26.1 | 25.7 | 28.3 | 24.7 | 28.8 | 25.4 | 27.7 |
| M | S | 22.7 | 47.4 | 32.6 | 28.8 | 25.7 | 13.7 | 22.9 | 8.3 | 13.2 | 17.9 | 20.6 | 15.6 | 28.7 | 28.0 | 18.3 | 23.0 | 27.5 | 25.9 | 23.9 | 23.4 | 27.5 |
| EWF | E | - | 36.4 | 18.7 | 19.9 | 26.9 | 24.6 | 29.5 | 13.8 | - | 16.0 | 27.3 | - | 30.0 | 27.2 | 19.0 | 26.8 | 28.0 | 24.5 | 29.7 | 24.9 | 27.4 |
| EWF D100 | E | - | 36.3 | 22.3 | 23.6 | 26.9 | 24.6 | 29.5 | 17.9 | - | 21.5 | 27.3 | - | 30.0 | 27.2 | 25.5 | 26.8 | 28.0 | 24.5 | 29.7 | 26.4 | 27.4 |
| M | C | 26.9 | 46.1 | 30.9 | 27.5 | 23.9 | 14.9 | 17.1 | 11.4 | 15.6 | 17.6 | 20.3 | 19.6 | 29.9 | 26.7 | 20.2 | 24.4 | 27.5 | 24.2 | 21.9 | 23.5 | 27.1 |
| W2V D100 | E | - | 28.9 | 15.4 | 18.2 | 19.3 | 18.8 | 25.6 | 11.7 | - | - | 34.3 | - | 27.7 | 27.8 | 30.3 | 28.1 | 25.1 | 25.0 | 26.3 | 24.2 | 26.4 |
| W2V | E | - | 26.3 | 12.3 | 16.0 | 19.3 | 22.4 | 25.6 | 9.2 | - | 14.5 | 34.3 | - | 27.7 | 27.8 | 24.6 | 28.1 | 25.1 | 25.0 | 26.3 | 22.8 | 26.4 |
| WLM D30 | E | - | 39.5 | 23.2 | 18.5 | 18.9 | 17.1 | 23.6 | 13.5 | - | - | 25.2 | - | 26.1 | 27.3 | 28.4 | 27.6 | 23.7 | 24.3 | 24.1 | 24.1 | 25.4 |
| ET | C | 15.0 | 38.4 | 21.6 | 15.2 | 21.1 | 15.5 | 23.7 | 12.1 | 17.0 | 12.9 | 20.7 | 30.9 | 26.4 | 26.6 | 16.2 | 26.9 | 24.3 | 23.5 | 27.7 | 21.9 | 25.2 |
| ET D100 | C | 20.4 | 38.4 | 21.6 | 20.5 | 21.1 | 15.5 | 23.7 | 18.8 | 24.1 | 19.5 | 20.7 | 30.9 | 26.4 | 26.6 | 25.8 | 26.9 | 24.3 | 23.5 | 27.7 | 24.0 | 25.2 |
| ETF D100 | C | 20.4 | 38.9 | 21.8 | 20.6 | 21.2 | 15.8 | 23.8 | 18.7 | 23.8 | 19.5 | 20.9 | 31.2 | 26.5 | 26.6 | 25.8 | 26.9 | 24.3 | 23.5 | 27.7 | 24.1 | 25.2 |
| ETF | C | 15.0 | 35.9 | 21.8 | 15.3 | 21.2 | 15.8 | 23.8 | 12.0 | 16.8 | 13.0 | 20.9 | 31.2 | 26.5 | 26.6 | 16.2 | 26.9 | 24.3 | 23.5 | 27.7 | 21.8 | 25.2 |
| EMTF D30 | E | - | 34.9 | 23.0 | 21.6 | 24.4 | 16.1 | 22.1 | 18.2 | - | 17.0 | 25.1 | - | 26.4 | 26.8 | 24.9 | 25.3 | 26.9 | 23.3 | 26.9 | 23.9 | 25.1 |
| EWMTF D30 | E | - | 36.9 | 21.2 | 21.2 | 25.1 | 18.1 | 21.5 | 15.0 | - | 17.5 | 27.5 | - | 26.2 | 24.6 | 20.7 | 21.8 | 24.5 | 22.0 | 26.7 | 23.2 | 24.3 |
| MAE D100 | S | 18.9 | 35.8 | 17.9 | 19.5 | 15.3 | 20.1 | 30.9 | 14.4 | 24.6 | 21.9 | 24.9 | - | 25.6 | 25.9 | 23.4 | 22.9 | 22.6 | 22.4 | 31.0 | 23.2 | 24.1 |
| MAE | S | 8.4 | 35.8 | 17.9 | 8.3 | 15.3 | 14.8 | 30.9 | 5.8 | 12.6 | 9.4 | 24.9 | 50.0 | 25.6 | 25.9 | 11.5 | 22.9 | 22.6 | 22.4 | 31.0 | 20.8 | 24.1 |
| CC | C | 14.9 | 20.1 | 12.4 | 6.9 | 10.6 | 19.2 | 33.0 | 6.3 | 16.9 | 21.2 | 35.6 | 18.4 | 23.0 | 26.5 | 22.1 | 21.0 | 21.3 | 24.3 | 26.8 | 20.0 | 23.8 |
| EWTF | C | 13.4 | 32.9 | 19.9 | 14.6 | 22.3 | 14.2 | 24.1 | 9.1 | 16.4 | 12.5 | 16.8 | 22.6 | 25.5 | 24.4 | 12.9 | 24.2 | 23.4 | 21.5 | 26.3 | 19.8 | 23.7 |
| EWTF D100 | C | 17.7 | 35.8 | 20.3 | 18.7 | 22.3 | 14.2 | 24.1 | 14.4 | 22.9 | 18.8 | 16.8 | 22.6 | 25.5 | 24.4 | 21.5 | 24.2 | 23.4 | 21.5 | 26.3 | 21.9 | 23.7 |
| EWT D100 | C | 17.7 | 35.0 | 20.0 | 18.5 | 22.2 | 14.1 | 24.1 | 14.5 | 23.0 | 18.7 | 16.6 | 28.9 | 25.4 | 24.4 | 21.5 | 24.2 | 23.4 | 21.4 | 26.1 | 22.1 | 23.6 |
| EWT | C | 13.4 | 32.2 | 19.9 | 14.5 | 22.2 | 14.1 | 24.1 | 9.1 | 16.6 | 12.4 | 16.6 | 28.9 | 25.4 | 24.4 | 12.9 | 24.2 | 23.4 | 21.4 | 26.1 | 20.1 | 23.6 |
| CLP D100 | C | 17.6 | 40.0 | 21.5 | 19.3 | 20.0 | 14.2 | 36.8 | 15.7 | 21.2 | 18.7 | 14.9 | 19.7 | 25.0 | 24.4 | 22.3 | 23.6 | 22.8 | 21.5 | 28.2 | 22.5 | 23.4 |
| CLP | C | 13.2 | 39.0 | 17.6 | 15.4 | 20.0 | 12.7 | 36.8 | 11.3 | 16.3 | 13.3 | 14.9 | 19.7 | 25.0 | 24.4 | 17.0 | 23.6 | 22.8 | 21.5 | 28.2 | 20.7 | 23.4 |
| EF D30 | E | - | 37.8 | 22.7 | 22.3 | 24.0 | 17.9 | 25.7 | 15.2 | - | 18.0 | 24.5 | 28.1 | 24.4 | 22.9 | 24.2 | 24.5 | 22.0 | 20.6 | 25.2 | 23.5 | 22.5 |
| MAE | E | - | 32.4 | 16.6 | 13.3 | 18.1 | 26.5 | 27.5 | 9.4 | - | 15.1 | 25.2 | - | 23.8 | 22.6 | 18.9 | 25.0 | 21.6 | 20.3 | 28.8 | 21.6 | 22.1 |
| MAE D100 | E | - | 30.0 | 16.6 | 17.2 | 18.1 | 18.0 | 27.5 | 12.7 | - | 21.4 | 25.2 | - | 23.8 | 22.6 | 24.6 | 25.0 | 21.6 | 20.3 | 28.8 | 22.1 | 22.1 |
| EWF D30 | E | - | 35.4 | 18.7 | 19.9 | 22.1 | 18.9 | 22.0 | 13.8 | - | 16.0 | 20.8 | - | 24.0 | 21.8 | 19.0 | 20.6 | 22.0 | 18.8 | 25.2 | 21.2 | 21.6 |

Table 21: (Continued) CSR Results Across Subsets and Distances (↑ better)

| Method | Dist | BC | BS1 | BS2 | BS3 | BS4 | BS5 | ES1 | HP | HS1 | HS2 | HU1 | HU2 | HU3 | HU4 | HW1 | HW2 | HW3 | HW4 | OC1 | Avg | Avg (Blind) |
|---|---|---|---|---|---|---|---|---|---|---|---|---|---|---|---|---|---|---|---|---|---|---|
| WLM D100 | C | 8.5 | 39.8 | 15.9 | 12.4 | 10.2 | 11.3 | 19.8 | 8.0 | 20.3 | 17.8 | 23.5 | 32.7 | 22.6 | 23.2 | 21.4 | 21.6 | 19.1 | 19.7 | 23.0 | 19.5 | 21.1 |
| WLM | C | 4.1 | 36.5 | 11.0 | 7.0 | 10.2 | 19.6 | 19.8 | 4.3 | 13.0 | 9.5 | 23.5 | 32.7 | 22.6 | 23.2 | 14.7 | 21.6 | 19.1 | 19.7 | 23.0 | 17.6 | 21.1 |
| CLP D30 | S | 18.3 | 33.6 | 22.5 | 20.4 | 20.1 | 16.1 | 33.8 | 14.5 | 20.5 | 15.3 | 14.5 | 20.3 | 21.8 | 21.9 | 19.0 | 20.2 | 19.1 | 18.6 | 26.1 | 20.9 | 20.3 |
| VC | E | - | 47.7 | 31.3 | 25.8 | 28.0 | 21.3 | 18.4 | 12.1 | - | - | 20.4 | - | 21.9 | 19.9 | 20.1 | 23.1 | 19.0 | - | 24.4 | 23.8 | 20.3 |
| ET D30 | S | 18.7 | 33.3 | 21.0 | 19.3 | 19.1 | 15.1 | 20.5 | 14.6 | 19.6 | 15.3 | 15.2 | 32.6 | 20.9 | 21.6 | 17.8 | 20.0 | 18.5 | 18.3 | 25.1 | 20.3 | 19.8 |
| ETF D30 | S | 18.7 | 33.4 | 21.1 | 19.1 | 19.2 | 15.3 | 20.3 | 14.6 | 19.6 | 15.3 | 15.3 | 33.4 | 21.0 | 21.4 | 17.8 | 20.0 | 18.6 | 18.3 | 24.9 | 20.4 | 19.8 |
| W2V D30 | E | - | 26.3 | 12.3 | 16.0 | 14.6 | 14.6 | 20.2 | 9.2 | - | 14.5 | 25.2 | - | 20.4 | 21.7 | 24.6 | 22.1 | 18.0 | 18.3 | 20.1 | 18.6 | 19.6 |
| EWTF D30 | S | 17.8 | 34.3 | 22.4 | 19.8 | 20.6 | 16.1 | 20.1 | 13.2 | 19.5 | 14.5 | 12.5 | 16.5 | 20.8 | 20.8 | 14.8 | 18.5 | 18.4 | 17.6 | 24.6 | 19.1 | 19.4 |
| EWT D30 | S | 17.8 | 34.1 | 22.3 | 19.7 | 20.7 | 16.3 | 20.1 | 13.2 | 19.6 | 14.4 | 12.3 | 30.7 | 20.7 | 20.8 | 14.8 | 18.5 | 18.3 | 17.6 | 24.3 | 19.8 | 19.4 |
| EMTF D30 | S | 16.3 | 30.8 | 19.3 | 17.3 | 17.6 | 14.6 | 19.8 | 13.6 | 18.7 | 14.4 | 21.1 | 25.4 | 20.9 | 21.0 | 18.8 | 19.1 | 17.8 | 17.6 | 23.4 | 19.3 | 19.3 |
| EWMTF D30 | S | 16.1 | 33.6 | 19.5 | 17.7 | 18.6 | 16.1 | 19.5 | 12.7 | 18.7 | 14.2 | 21.0 | 20.4 | 20.1 | 20.4 | 17.5 | 17.1 | 17.2 | 17.1 | 23.6 | 19.0 | 18.7 |
| E | C | 7.8 | 28.0 | - | 10.0 | 14.7 | 23.5 | 15.5 | 7.5 | 10.0 | 6.4 | 18.9 | 18.1 | 19.1 | 18.5 | 13.3 | 19.5 | 17.4 | 15.9 | 18.6 | 15.7 | 17.7 |
| EMTF D100 | C | 10.5 | 29.5 | 14.2 | 13.2 | 14.7 | 7.9 | 15.5 | 11.2 | 13.2 | 9.4 | 18.9 | 18.1 | 19.1 | 18.5 | 20.3 | 19.5 | 17.4 | 15.9 | 18.6 | 16.1 | 17.7 |
| EMTF | C | 7.8 | 28.0 | 14.2 | 10.0 | 14.7 | 7.9 | 15.5 | 7.5 | 10.0 | 6.4 | 18.9 | 18.1 | 19.1 | 18.5 | 13.3 | 19.5 | 17.4 | 15.9 | 18.6 | 14.8 | 17.7 |
| ET D30 | C | 15.0 | 36.4 | 16.6 | 15.2 | 16.2 | 10.5 | 17.8 | 12.1 | 17.0 | 12.9 | 12.8 | 30.9 | 18.8 | 19.2 | 16.2 | 18.3 | 16.5 | 16.2 | 21.7 | 17.9 | 17.7 |
| ETF D30 | C | 15.0 | 37.0 | 16.7 | 15.3 | 16.3 | 10.6 | 17.9 | 12.0 | 16.8 | 13.0 | 12.1 | 31.2 | 18.8 | 19.2 | 16.2 | 18.3 | 16.5 | 16.2 | 21.7 | 18.0 | 17.7 |
| EWMTF | C | 8.9 | 30.4 | 9.7 | 9.8 | 15.0 | 9.0 | 15.2 | 4.7 | 9.8 | 6.8 | 21.7 | 13.7 | 19.0 | 18.3 | 8.0 | 13.0 | 17.3 | 15.9 | 18.8 | 14.0 | 17.6 |
| EW | C | 8.9 | 35.2 | 19.9 | 9.8 | 15.0 | 9.0 | 15.2 | 4.7 | 9.8 | 6.8 | 21.7 | 13.7 | 19.0 | 18.3 | 8.0 | 13.0 | 17.3 | 15.9 | 18.8 | 14.7 | 17.6 |
| EWMTF D100 | C | 11.3 | 31.3 | 13.2 | 12.3 | 15.0 | 9.0 | 15.2 | 7.2 | 12.7 | 9.8 | 21.7 | 13.7 | 19.0 | 18.3 | 13.1 | 13.0 | 17.3 | 15.9 | 18.8 | 15.2 | 17.6 |
| CLP D30 | C | 13.2 | 39.0 | 17.6 | 15.4 | 15.8 | 10.6 | 33.2 | 11.3 | 16.3 | 13.3 | 10.3 | 19.7 | 19.2 | 18.7 | 17.0 | 18.3 | 16.8 | 15.8 | 23.4 | 18.2 | 17.6 |
| EF D100 | C | 11.8 | 35.2 | 16.6 | 15.7 | 17.1 | 11.7 | 24.0 | 10.1 | 13.0 | 11.5 | 19.0 | 14.7 | 18.5 | 18.1 | 17.6 | 17.7 | 16.3 | 15.6 | 19.0 | 17.0 | 17.1 |
| EF | C | 6.3 | 32.8 | 16.6 | 10.1 | 17.1 | 11.7 | 24.0 | 6.0 | 7.7 | 6.0 | 19.0 | 14.7 | 18.5 | 18.1 | 10.9 | 17.7 | 16.3 | 15.6 | 19.0 | 15.2 | 17.1 |
| WLM D30 | S | 6.3 | 33.0 | 16.8 | 10.2 | 9.4 | 9.4 | 16.7 | 5.7 | 15.8 | 11.8 | 13.7 | 37.9 | 18.0 | 19.2 | 16.5 | 17.1 | 15.3 | 15.7 | 16.1 | 16.0 | 17.0 |
| MAE D30 | E | - | 27.2 | 12.4 | 13.3 | 13.1 | 12.5 | 21.1 | 9.4 | - | 15.1 | 16.6 | - | 18.4 | 17.7 | 18.9 | 19.5 | 16.4 | 15.4 | 23.8 | 16.9 | 17.0 |
| EF D30 | S | 12.2 | 32.1 | 16.8 | 16.9 | 15.8 | 12.5 | 19.9 | 11.4 | 15.4 | 11.5 | 13.8 | 20.1 | 17.9 | 18.8 | 14.1 | 14.6 | 15.7 | 15.4 | 21.0 | 16.6 | 17.0 |
| EWTF D30 | C | 13.4 | 34.6 | 17.0 | 14.6 | 17.7 | 10.4 | 17.9 | 9.1 | 16.4 | 12.5 | 9.9 | 14.8 | 18.3 | 17.6 | 12.9 | 16.2 | 16.2 | 14.9 | 20.9 | 16.1 | 16.8 |
| EWT D30 | C | 13.4 | 33.7 | 16.7 | 14.5 | 17.6 | 10.4 | 18.0 | 9.1 | 16.6 | 12.4 | 9.9 | 28.9 | 18.3 | 17.6 | 12.9 | 16.2 | 16.2 | 14.9 | 20.7 | 16.7 | 16.7 |
| W2V D100 | C | 3.2 | 22.5 | 5.9 | 8.3 | 7.5 | 8.9 | 16.0 | 5.5 | 13.0 | 12.0 | 24.4 | 23.9 | 17.7 | 18.8 | 21.8 | 19.5 | 15.0 | 15.2 | 18.5 | 14.6 | 16.7 |
| W2V | C | 1.2 | 18.7 | 3.6 | 6.0 | 7.5 | 13.9 | 16.0 | 3.1 | 7.0 | 5.6 | 24.4 | 23.9 | 17.7 | 18.8 | 13.6 | 19.5 | 15.0 | 15.2 | 18.5 | 13.1 | 16.7 |
| EWF | C | 7.4 | 26.6 | 8.2 | 8.5 | 13.6 | 12.4 | 21.8 | 4.2 | 7.2 | 4.8 | 13.8 | 12.2 | 17.9 | 17.1 | 6.0 | 13.2 | 15.7 | 13.8 | 18.8 | 12.8 | 16.1 |
| EWF D100 | C | 11.5 | 30.9 | 11.8 | 12.5 | 13.6 | 12.4 | 21.8 | 7.4 | 11.9 | 9.2 | 13.8 | 12.2 | 17.9 | 17.1 | 11.9 | 13.2 | 15.7 | 13.8 | 18.8 | 14.6 | 16.1 |
| AC | E | - | 37.7 | 27.6 | 23.9 | 27.4 | 18.0 | 13.7 | 17.2 | - | 11.3 | 27.2 | - | 17.3 | 13.4 | 25.1 | 26.7 | 15.7 | - | 20.1 | 21.5 | 15.5 |
| EWF D30 | S | 12.4 | 32.6 | 14.7 | 14.3 | 13.7 | 13.4 | 19.0 | 8.9 | 14.6 | 9.8 | 11.6 | 9.7 | 16.5 | 16.7 | 9.2 | 10.6 | 13.9 | 13.1 | 20.5 | 14.5 | 15.0 |
| W2V D30 | S | 3.5 | 22.3 | 7.4 | 9.9 | 7.3 | 8.5 | 16.2 | 6.2 | 12.3 | 9.5 | 15.9 | 20.6 | 15.7 | 16.2 | 17.2 | 15.4 | 12.7 | 12.4 | 14.9 | 12.9 | 14.3 |
| CC | S | 2.0 | 22.3 | 8.7 | 4.0 | 1.3 | 0.9 | 40.0 | 0.8 | 5.5 | 12.9 | 21.2 | 1.3 | 13.4 | 14.8 | 7.0 | 8.9 | 13.9 | 12.9 | 2.1 | 10.2 | 13.8 |
| WLM D30 | C | 4.1 | 36.5 | 11.0 | 7.0 | 6.5 | 6.4 | 11.2 | 4.3 | 13.0 | 9.5 | 11.4 | 32.7 | 14.0 | 15.3 | 14.7 | 15.1 | 11.1 | 12.0 | 13.4 | 13.1 | 13.1 |
| EWMTF D30 | C | 8.9 | 30.4 | 10.9 | 9.8 | 12.1 | 7.0 | 11.2 | 4.7 | 9.8 | 6.8 | 14.7 | 13.7 | 13.9 | 13.7 | 8.0 | 8.2 | 12.3 | 11.4 | 15.3 | 11.7 | 12.8 |
| EMTF D30 | C | 7.8 | 28.0 | 11.2 | 10.0 | 11.6 | 5.8 | 11.8 | 7.5 | 10.0 | 6.4 | 12.5 | 18.1 | 13.9 | 13.7 | 13.3 | 13.3 | 12.2 | 11.3 | 14.9 | 12.3 | 12.8 |
| MAE D30 | S | 8.4 | 27.3 | 7.9 | 8.3 | 6.3 | 8.0 | 17.9 | 5.8 | 12.6 | 9.4 | 9.5 | - | 13.2 | 13.8 | 11.5 | 11.4 | 10.6 | 10.8 | 18.3 | 11.7 | 12.1 |
| MAE | C | 3.9 | 21.3 | 3.9 | 3.4 | 5.9 | 12.9 | 21.2 | 2.7 | 7.3 | 5.0 | 13.8 | 19.8 | 12.5 | 11.9 | 7.0 | 12.1 | 10.3 | 9.2 | 19.8 | 10.7 | 11.0 |
| MAE D100 | C | 7.2 | 21.3 | 7.0 | 6.3 | 5.9 | 8.1 | 21.2 | 5.0 | 12.6 | 10.5 | 13.8 | 19.8 | 12.5 | 11.9 | 12.3 | 12.1 | 10.3 | 9.2 | 19.8 | 11.9 | 11.0 |
| EF D30 | C | 6.3 | 32.8 | 10.7 | 10.1 | 10.3 | 6.0 | 14.1 | 6.0 | 7.7 | 6.0 | 10.5 | 14.7 | 11.3 | 11.5 | 10.9 | 11.0 | 9.7 | 9.4 | 12.8 | 11.2 | 10.5 |
| EWF D30 | C | 7.4 | 29.3 | 8.2 | 8.5 | 8.8 | 6.9 | 12.0 | 4.2 | 7.2 | 4.8 | 7.5 | 6.0 | 10.9 | 10.8 | 6.0 | 7.1 | 9.1 | 7.9 | 12.9 | 9.2 | 9.7 |
| W2V D30 | C | 1.2 | 18.7 | 3.6 | 6.0 | 4.0 | 4.7 | 9.8 | 3.1 | 7.0 | 5.6 | 12.6 | 23.9 | 9.2 | 10.8 | 13.6 | 11.6 | 7.5 | 7.8 | 10.0 | 9.0 | 8.8 |
| VC | C | 7.1 | 46.9 | 19.5 | 12.8 | 14.2 | 7.8 | 7.1 | 2.9 | 6.7 | 5.2 | 6.9 | 5.8 | 8.3 | 6.8 | 7.1 | 8.3 | 6.1 | 4.7 | 10.7 | 10.3 | 6.5 |
| MAE D30 | C | 3.9 | 17.3 | 3.9 | 3.4 | 2.9 | 3.8 | 12.9 | 2.7 | 7.3 | 5.0 | 5.6 | 19.8 | 7.2 | 6.9 | 7.0 | 7.0 | 5.7 | 5.1 | 12.5 | 7.4 | 6.2 |
| AC | C | 7.1 | 29.2 | 13.8 | 10.6 | 14.2 | 5.6 | 5.2 | 4.4 | 2.9 | 2.9 | 15.8 | 7.5 | 6.9 | 4.0 | 11.1 | 12.7 | 5.3 | 3.2 | 8.6 | 9.0 | 4.9 |
| VC | S | 5.5 | 35.3 | 11.0 | 7.6 | 10.5 | 7.0 | 5.0 | 1.7 | 5.8 | 3.5 | 3.6 | 3.1 | 6.1 | 5.3 | 3.7 | 4.0 | 3.8 | 3.1 | 9.8 | 7.1 | 4.6 |

Table 22: CS Results Across Subsets and Distances (↓ better)

| Method | Dist | BC | BS1 | BS2 | BS3 | BS4 | BS5 | ES1 | HP | HS1 | HS2 | HU1 | HU2 | HU3 | HU4 | HW1 | HW2 | HW3 | HW4 | OC1 | Avg | Avg (Blind) |
|---|---|---|---|---|---|---|---|---|---|---|---|---|---|---|---|---|---|---|---|---|---|---|
| EWMTF D30 | C | 35.8 | 20.2 | 28.5 | 31.0 | 22.2 | 29.3 | 34.7 | 46.2 | 44.4 | 43.2 | 32.5 | 41.2 | **44.9** | 47.0 | 45.0 | 44.7 | **47.2** | **46.8** | 38.5 | 38.1 | **46.5** |
| EWTF D30 | C | 36.2 | 18.9 | 23.1 | 25.7 | 18.6 | 25.2 | 36.5 | 47.3 | 44.5 | 45.7 | 43.2 | 43.9 | 45.0 | 47.3 | 46.4 | 45.4 | 47.5 | 47.0 | 38.7 | 38.2 | 46.7 |
| EWMTF | C | 35.8 | 20.2 | 29.6 | 31.0 | 23.3 | 30.2 | 36.1 | 46.2 | 44.4 | 43.2 | 32.4 | 41.2 | 45.3 | 47.2 | 45.0 | 44.6 | 47.3 | 47.1 | 39.5 | 38.4 | 46.7 |
| EWMTF D100 | C | 36.6 | 21.2 | 29.2 | 31.7 | 23.3 | 30.2 | 36.1 | 46.3 | 44.7 | 43.7 | 32.4 | 41.2 | 45.3 | 47.2 | 44.5 | 44.6 | 47.3 | 47.1 | 39.5 | 38.5 | 46.7 |
| EW | C | 35.8 | 20.8 | 26.5 | 31.0 | 23.3 | 36.5 | 36.1 | 46.2 | 44.4 | 43.2 | 32.4 | 41.2 | 45.3 | 47.2 | 45.0 | 44.6 | 47.3 | 47.1 | 39.5 | 38.6 | 46.7 |
| EWT D30 | C | 36.3 | 19.1 | 23.2 | 25.8 | 18.8 | 25.2 | 36.6 | 47.3 | 44.6 | 45.9 | 43.3 | 46.2 | 45.0 | 47.4 | 46.5 | 45.5 | 47.5 | 47.1 | 38.8 | 38.4 | 46.7 |
| CLP D30 | C | 36.9 | 25.0 | 22.4 | 25.8 | 20.4 | 28.4 | **22.2** | 43.8 | 42.4 | 43.2 | 42.9 | 42.8 | 45.3 | 47.3 | 40.2 | 41.3 | 47.4 | 47.3 | 36.7 | 36.9 | 46.8 |
| AC | C | **32.5** | 16.5 | 23.9 | 25.1 | 16.8 | 30.0 | 36.8 | 43.8 | **40.7** | **42.3** | 36.2 | 38.9 | 45.4 | 46.9 | 40.7 | 39.3 | 47.7 | 47.3 | 36.3 | 36.2 | 46.8 |
| EMTF D30 | C | 36.5 | 26.4 | 30.0 | 34.8 | 24.2 | 24.5 | 37.0 | 45.4 | 46.0 | 43.7 | 32.3 | 40.5 | 45.2 | 47.3 | 42.9 | 43.0 | 47.5 | 47.3 | 38.8 | 38.6 | 46.8 |
| ETF D30 | C | 38.5 | 24.7 | 28.4 | 31.4 | 22.1 | **23.1** | 35.8 | 45.8 | 45.1 | 45.2 | 41.9 | 45.8 | 45.1 | 47.7 | 44.8 | 44.1 | 47.5 | 47.5 | 37.8 | 39.1 | 47.0 |
| ET D30 | C | 38.8 | 25.0 | 28.7 | 31.5 | 22.4 | 23.3 | 35.9 | 45.9 | 45.1 | 45.3 | 42.1 | 45.8 | 45.2 | 47.7 | 44.9 | 44.2 | 47.6 | 47.5 | 38.0 | 39.2 | 47.0 |
| EMTF | C | 36.5 | 26.4 | 31.1 | 34.8 | 25.6 | 25.6 | 38.1 | 45.4 | 46.0 | 43.7 | **32.1** | 40.5 | 45.6 | 47.6 | 42.9 | 43.2 | 47.6 | 47.6 | 39.8 | 39.0 | 47.1 |
| EMTF D100 | C | 37.2 | 27.8 | 31.1 | 35.5 | 25.6 | 25.6 | 38.1 | 45.7 | 46.2 | 44.1 | **32.1** | 40.5 | 45.6 | 47.6 | 42.6 | 43.2 | 47.6 | 47.6 | 39.8 | 39.1 | 47.1 |
| E | C | 36.5 | 26.4 | 21.5 | 34.8 | 25.6 | 37.8 | 38.1 | 45.4 | 46.0 | 43.7 | **32.1** | 40.5 | 45.6 | 47.6 | 42.9 | 43.2 | 47.6 | 47.6 | 39.8 | 40.1 | 47.1 |
| EWTF D100 | C | 38.1 | 21.3 | 24.9 | 27.8 | 21.4 | 27.8 | 39.1 | 47.6 | 45.5 | 46.6 | 43.2 | 45.0 | 45.8 | 47.8 | 46.5 | 46.0 | 47.8 | 47.6 | 40.7 | 39.5 | 47.3 |
| EWTF | C | 36.2 | 22.3 | 26.5 | 25.7 | 21.4 | 27.8 | 39.1 | 47.3 | 44.5 | 45.7 | 43.2 | 45.0 | 45.8 | 47.8 | 46.4 | 46.0 | 47.8 | 47.6 | 40.7 | 39.3 | 47.3 |
| EF D30 | C | 36.5 | 23.1 | 25.5 | 32.8 | 21.0 | 28.3 | 41.3 | 44.9 | 46.8 | 45.8 | 36.5 | 42.4 | 46.4 | 47.4 | 43.1 | 42.2 | 47.8 | 47.5 | 37.1 | 38.8 | 47.3 |
| EWT D100 | C | 38.2 | 21.4 | 25.0 | 27.8 | 21.5 | 27.7 | 39.2 | 47.6 | 45.6 | 46.7 | 43.4 | 46.2 | 45.9 | 47.9 | 46.6 | 46.0 | 47.8 | 47.6 | 40.8 | 39.6 | 47.3 |
| EWT | C | 36.3 | 22.3 | 26.5 | 25.8 | 21.5 | 27.8 | 39.2 | 47.3 | 44.6 | 45.9 | 43.4 | 46.2 | 45.9 | 47.9 | 46.5 | 46.0 | 47.8 | 47.6 | 40.8 | 39.4 | 47.3 |
| CLP D100 | C | 38.6 | 27.3 | 25.1 | 28.1 | 22.8 | 30.6 | 26.8 | 44.8 | 43.4 | 44.4 | 43.6 | 42.8 | 46.0 | 47.8 | 41.6 | 42.5 | 47.8 | 47.7 | 38.9 | 38.4 | 47.3 |
| CLP | C | 36.9 | 25.0 | 22.4 | 25.8 | 22.8 | 31.2 | 26.8 | 43.8 | 42.4 | 43.2 | 43.6 | 42.8 | 46.0 | 47.8 | 40.2 | 42.5 | 47.8 | 47.7 | 38.9 | 37.8 | 47.3 |
| VC | C | 35.0 | **9.5** | **14.5** | **16.6** | **12.1** | 25.2 | 37.5 | **39.8** | 44.6 | 44.6 | 41.7 | **37.8** | 46.8 | **46.8** | 35.1 | 36.3 | 48.0 | 47.8 | **35.8** | **34.5** | 47.4 |
| EWMTF D30 | S | 42.7 | 38.8 | 33.9 | 35.6 | 29.5 | 34.0 | 38.7 | 47.1 | 45.7 | 45.2 | 38.0 | 44.2 | 46.1 | 47.9 | 43.8 | 44.9 | 47.9 | 47.7 | 43.7 | 41.9 | 47.4 |
| EWTF D30 | S | 41.7 | 39.1 | 31.4 | 32.0 | 24.9 | 31.3 | 39.3 | 47.7 | 45.3 | 46.5 | 44.1 | 45.2 | 46.2 | 48.0 | 46.7 | 46.1 | 47.8 | 47.8 | 43.2 | 41.8 | 47.5 |
| EWT D30 | S | 41.9 | 39.2 | 31.7 | 31.9 | 25.1 | 31.4 | 39.3 | 47.7 | 45.2 | 46.5 | 44.2 | 46.5 | 46.3 | 48.1 | 46.7 | 46.1 | 47.9 | 47.8 | 43.1 | 41.9 | 47.5 |
| ETF D100 | C | 40.5 | 27.9 | 30.7 | 33.3 | 25.0 | 27.1 | 38.3 | 46.6 | 45.9 | 46.1 | 42.1 | 45.8 | 46.1 | 48.2 | 45.2 | 45.0 | 47.9 | 48.0 | 40.3 | 40.5 | 47.5 |
| ETF | C | 38.5 | 29.2 | 30.7 | 31.4 | 25.0 | 27.1 | 38.3 | 45.8 | 45.1 | 45.2 | 42.1 | 45.8 | 46.1 | 48.2 | 44.8 | 45.0 | 47.9 | 48.0 | 40.3 | 40.2 | 47.5 |
| ET | C | 38.8 | 28.1 | 30.9 | 31.5 | 25.3 | 27.1 | 38.4 | 45.9 | 45.1 | 45.3 | 42.3 | 45.8 | 46.1 | 48.2 | 44.9 | 45.1 | 48.0 | 48.0 | 40.4 | 40.3 | 47.6 |
| ET D100 | C | 40.7 | 28.1 | 30.9 | 33.4 | 25.3 | 27.2 | 38.4 | 46.7 | 45.9 | 46.2 | 42.3 | 45.8 | 46.1 | 48.2 | 45.3 | 45.1 | 48.0 | 48.0 | 40.4 | 40.6 | 47.6 |
| CLP D30 | S | 41.6 | 40.3 | 29.9 | 33.7 | 28.4 | 33.2 | 25.8 | 46.1 | 43.7 | 45.2 | 44.1 | 43.0 | 46.5 | 48.0 | 42.6 | 43.7 | 47.9 | 48.0 | 42.5 | 40.7 | 47.6 |
| ETF D30 | S | 42.2 | 40.4 | 34.8 | 35.7 | 28.2 | 28.5 | 39.5 | 46.9 | 45.6 | 46.4 | 43.0 | 46.2 | 46.4 | 48.2 | 45.5 | 45.4 | 48.0 | 48.1 | 42.8 | 42.2 | 47.6 |
| EMTF D30 | S | 43.6 | 41.8 | 35.7 | 38.7 | 31.0 | 31.2 | 40.5 | 46.9 | 46.1 | 46.0 | 38.1 | 44.0 | 46.4 | 48.1 | 43.6 | 44.6 | 48.0 | 48.0 | 43.3 | 42.4 | 47.6 |
| ET D30 | S | 42.3 | 40.5 | 35.2 | 35.9 | 28.6 | 29.0 | 39.5 | 46.9 | 45.7 | 46.5 | 44.2 | 46.2 | 46.4 | 48.2 | 45.5 | 45.4 | 48.0 | 48.1 | 42.7 | 42.3 | 47.7 |
| EWF D30 | C | 37.4 | 21.5 | 28.5 | 32.3 | 25.3 | 33.2 | 40.8 | 45.9 | 45.2 | 45.6 | 35.9 | 40.7 | 46.8 | 47.8 | 45.1 | 43.4 | 48.1 | 47.9 | 37.4 | 39.4 | 47.7 |
| EF | C | 36.5 | 23.1 | 28.9 | 32.8 | 25.1 | 31.2 | 43.9 | 44.9 | 46.8 | 45.8 | 38.9 | 42.4 | 47.1 | 47.9 | 43.1 | 43.4 | 48.2 | 48.0 | 39.1 | 39.8 | 47.8 |
| EF D100 | C | 38.5 | 25.9 | 28.9 | 35.0 | 25.1 | 31.2 | 43.9 | 45.4 | 47.1 | 46.4 | 38.9 | 42.4 | 47.1 | 47.9 | 44.1 | 43.4 | 48.2 | 48.0 | 39.1 | 40.3 | 47.8 |
| VC | S | 39.3 | 17.7 | 20.3 | 23.3 | 17.9 | 29.5 | 40.4 | 45.9 | 45.6 | 45.9 | 42.3 | 40.3 | 47.4 | 47.4 | 38.4 | 39.4 | 48.4 | 48.1 | 38.6 | 37.5 | 47.8 |
| EWMTF D100 | S | 46.1 | 46.5 | 39.1 | 40.0 | 36.3 | 40.7 | 43.3 | 47.7 | 47.1 | 47.0 | 36.3 | 44.2 | 46.9 | 48.4 | 42.7 | 44.9 | 48.2 | 48.3 | 46.8 | 44.2 | 48.0 |
| EWMTF | S | 42.7 | 46.5 | 32.5 | 35.6 | 36.3 | 40.7 | 43.3 | 47.1 | 45.7 | 45.2 | 36.3 | 44.2 | 46.9 | 48.4 | 43.8 | 44.9 | 48.2 | 48.3 | 46.8 | 43.3 | 48.0 |
| EWMTF D30 | E | - | 31.6 | 37.5 | 39.1 | 33.1 | 37.8 | 41.2 | 47.7 | - | 46.1 | 39.3 | - | 47.2 | 48.4 | 47.1 | 47.0 | 48.4 | 48.3 | 43.7 | 42.7 | 48.1 |
| EWF D100 | C | 39.1 | 22.9 | 30.0 | 33.8 | 27.4 | 35.7 | 43.5 | 46.2 | 45.6 | 46.1 | 37.9 | 42.7 | 47.4 | 48.3 | 45.5 | 44.3 | 48.5 | 48.2 | 39.3 | 40.7 | 48.1 |
| EWF | C | 37.4 | 23.2 | 28.5 | 32.3 | 27.4 | 35.7 | 43.5 | 45.9 | 45.2 | 45.6 | 37.9 | 42.7 | 47.4 | 48.3 | 45.1 | 44.3 | 48.5 | 48.2 | 39.3 | 40.3 | 48.1 |
| AC | E | - | 28.4 | 34.5 | 35.0 | 28.7 | 37.6 | 40.6 | 46.7 | - | 45.2 | 40.8 | - | 47.4 | 48.2 | 45.1 | 44.4 | 48.8 | - | 42.0 | 40.9 | 48.1 |
| WLM D30 | C | 35.1 | 20.4 | 19.1 | 28.0 | 14.8 | 25.3 | 40.8 | 42.3 | 46.7 | 45.5 | 41.2 | 44.6 | 47.7 | 48.1 | **28.1** | **36.2** | 48.6 | 48.4 | 38.9 | 36.8 | 48.2 |
| EWTF D30 | E | - | 31.2 | 34.2 | 35.9 | 30.7 | 35.1 | 42.5 | 48.5 | - | 47.6 | 45.8 | - | 47.3 | 48.5 | 48.0 | 47.5 | 48.6 | 48.4 | 43.7 | 42.7 | 48.2 |
| EWT D30 | E | - | 31.3 | 34.3 | 35.9 | 30.8 | 35.1 | 42.6 | 48.5 | - | 47.7 | 45.9 | - | 47.3 | 48.5 | 48.0 | 47.5 | 48.6 | 48.4 | 43.8 | 42.8 | 48.2 |
| EWMTF | E | - | 32.5 | 38.1 | 39.1 | 34.0 | 38.4 | 42.1 | 47.7 | - | 46.1 | 39.4 | - | 47.4 | 48.5 | 47.1 | 47.0 | 48.5 | 48.4 | 44.2 | 43.0 | 48.2 |
| EWMTF D100 | E | - | 32.5 | 38.1 | 39.6 | 34.0 | 38.4 | 42.1 | 47.8 | - | - | 39.4 | - | 47.4 | 48.5 | 46.9 | 47.0 | 48.5 | 48.4 | 44.2 | 42.9 | 48.2 |
| EW | E | - | 32.0 | 38.1 | 39.1 | 34.0 | 42.5 | 42.1 | 47.7 | - | 46.1 | 39.4 | - | 47.4 | 48.5 | 47.1 | 47.0 | 48.5 | 48.4 | 44.2 | 43.0 | 48.2 |
| CLP D30 | E | - | 35.7 | 33.8 | 36.1 | 32.0 | 36.9 | 33.9 | 46.7 | - | 46.3 | 45.9 | - | 47.4 | 48.6 | 44.7 | 45.4 | 48.6 | 48.5 | 42.7 | 42.1 | 48.3 |
| EMTF D30 | E | - | 36.4 | 38.7 | 41.5 | 34.8 | 34.1 | 42.6 | 47.4 | - | 46.3 | 39.2 | - | 47.4 | 48.6 | 46.2 | 46.2 | 48.7 | 48.5 | 43.9 | 43.1 | 48.3 |
| EWTF D100 | S | 45.8 | 46.7 | 37.8 | 38.6 | 33.9 | 40.1 | 44.0 | 48.4 | 46.8 | 47.9 | 44.0 | 46.5 | 47.4 | 48.7 | 46.9 | 46.9 | 48.4 | 48.7 | 46.8 | 45.0 | 48.3 |
| EWTF | S | 41.7 | 46.7 | 37.8 | 32.0 | 33.9 | 40.1 | 44.0 | 47.7 | 45.3 | 46.5 | 44.0 | 46.5 | 47.4 | 48.7 | 46.7 | 46.9 | 48.4 | 48.7 | 46.8 | 44.2 | 48.3 |
| EMTF D100 | S | 46.6 | 47.3 | 41.0 | 42.0 | 37.7 | 40.0 | 44.3 | 47.8 | 47.3 | 47.5 | 36.4 | 44.0 | 47.4 | 48.8 | 42.9 | 45.0 | 48.4 | 48.7 | 46.7 | 44.7 | 48.3 |
| EMTF | S | 43.6 | 41.8 | 41.0 | 38.7 | 37.7 | 40.0 | 44.3 | 46.9 | 46.1 | 46.0 | 36.4 | 44.0 | 47.4 | 48.8 | 43.6 | 45.0 | 48.4 | 48.7 | 46.7 | 43.9 | 48.3 |
| EWT | S | 41.9 | 46.7 | 37.8 | 31.9 | 34.1 | 40.1 | 44.0 | 47.7 | 45.2 | 46.5 | 44.1 | 46.5 | 47.4 | 48.8 | 46.7 | 46.9 | 48.4 | 48.6 | 46.8 | 44.2 | 48.3 |
| EWT D100 | S | 45.9 | 46.7 | 38.0 | 38.7 | 34.1 | 40.2 | 44.0 | 48.4 | 46.9 | 47.9 | 44.1 | 46.5 | 47.4 | 48.8 | 46.9 | 46.9 | 48.4 | 48.6 | 46.8 | 45.0 | 48.3 |
| ETF D30 | E | - | 35.3 | 37.9 | 39.6 | 33.0 | 33.4 | 42.2 | 47.7 | - | 47.4 | 45.2 | - | 47.3 | 48.8 | 47.2 | 46.8 | 48.6 | 48.7 | 43.2 | 43.3 | 48.3 |
| ET D30 | E | - | 35.5 | 38.0 | 39.7 | 33.2 | 33.6 | 42.2 | 47.8 | - | 47.4 | 45.3 | - | 47.3 | 48.8 | 47.2 | 46.9 | 48.7 | 48.7 | 43.3 | 43.3 | 48.4 |
| VC | E | - | 24.0 | 28.0 | 29.1 | 25.0 | 34.7 | 41.8 | 44.5 | - | - | 44.0 | - | 48.1 | 48.3 | 42.1 | 42.4 | 48.8 | - | 42.2 | 38.8 | 48.4 |
| W2V D30 | C | 35.6 | 32.5 | 30.6 | 36.6 | 26.0 | 31.5 | 44.9 | 46.9 | 46.9 | 46.7 | 39.0 | 45.1 | 48.0 | 47.9 | 36.7 | 41.2 | 48.8 | 48.9 | 40.0 | 40.7 | 48.4 |
| E | E | - | 37.3 | 39.5 | 41.5 | 35.8 | 43.2 | 43.2 | 47.4 | - | 46.3 | 39.3 | - | 47.6 | 48.7 | 46.2 | 46.4 | 48.7 | 48.7 | 44.5 | 44.0 | 48.4 |
| EMTF D100 | E | - | 37.3 | 39.5 | 42.0 | 35.8 | 35.1 | 43.2 | 47.6 | - | - | 39.3 | - | 47.6 | 48.7 | 46.1 | 46.4 | 48.7 | 48.7 | 44.5 | 43.4 | 48.4 |
| EMTF | E | - | 37.3 | 40.1 | 41.5 | 35.8 | 35.1 | 43.2 | 47.4 | - | 46.3 | 39.3 | - | 47.6 | 48.7 | 46.2 | 46.4 | 48.7 | 48.7 | 44.5 | 43.6 | 48.4 |
| ETF | S | 42.2 | 47.0 | 40.1 | 35.7 | 36.1 | 39.7 | 44.0 | 46.9 | 45.6 | 46.4 | 43.1 | 46.2 | 47.6 | 48.9 | 45.5 | 46.5 | 48.6 | 48.8 | 46.7 | 44.5 | 48.4 |
| ETF D100 | S | 46.2 | 47.0 | 40.1 | 40.6 | 36.1 | 39.7 | 44.0 | 48.2 | 47.0 | 47.9 | 43.1 | 46.2 | 47.6 | 48.9 | 46.0 | 46.5 | 48.6 | 48.8 | 46.7 | 45.2 | 48.4 |
| ET D100 | S | 46.1 | 47.0 | 40.3 | 40.9 | 36.4 | 39.8 | 44.0 | 48.2 | 47.0 | 47.9 | 43.2 | 46.2 | 47.6 | 48.9 | 46.0 | 46.5 | 48.6 | 48.7 | 46.7 | 45.3 | 48.5 |
| ET | S | 42.3 | 47.0 | 35.2 | 40.9 | 36.4 | 39.7 | 44.0 | 46.9 | 45.7 | 46.5 | 43.2 | 46.2 | 47.6 | 48.9 | 45.5 | 46.5 | 48.6 | 48.8 | 46.7 | 44.6 | 48.5 |
| EF D30 | E | - | 33.9 | 35.5 | 40.0 | 32.0 | 36.6 | 45.0 | 47.0 | - | 47.5 | 42.1 | 45.7 | 47.9 | 48.6 | 46.3 | 45.6 | 48.8 | 48.6 | 42.7 | 43.2 | 48.5 |
| CC | S | 34.3 | 24.8 | 24.2 | 33.9 | 19.9 | 30.7 | 49.0 | 45.8 | 46.5 | 47.3 | 41.0 | 39.0 | 48.4 | 47.8 | 43.1 | 43.9 | 49.0 | 48.7 | 37.2 | 39.7 | 48.5 |
| EWTF D100 | E | - | 33.2 | 35.6 | 37.4 | 33.0 | 37.1 | 44.1 | 48.7 | - | - | 45.9 | - | 47.7 | 48.8 | 48.1 | 47.8 | 48.8 | 48.7 | 45.0 | 43.3 | 48.5 |
| EWTF | E | - | 31.2 | 34.2 | 35.9 | 33.0 | 37.1 | 44.1 | 48.5 | - | 47.6 | 45.9 | - | 47.7 | 48.8 | 48.0 | 47.8 | 48.8 | 48.7 | 45.0 | 43.3 | 48.5 |
| EWT D100 | E | - | 33.2 | 35.7 | 37.4 | 33.1 | 37.1 | 44.2 | 48.7 | - | 48.2 | 46.0 | - | 47.8 | 48.8 | 48.1 | 47.9 | 48.8 | 48.7 | 45.0 | 43.7 | 48.5 |
| EWT | E | - | 31.2 | 34.3 | 35.9 | 33.1 | 37.1 | 44.2 | 48.5 | - | 47.7 | 46.0 | - | 47.8 | 48.8 | 48.0 | 47.9 | 48.8 | 48.7 | 45.0 | 43.3 | 48.5 |
| CLP | E | - | 37.5 | 36.2 | 36.1 | 33.9 | 39.0 | 37.0 | 46.7 | - | 46.3 | 46.4 | - | 47.8 | 48.8 | 44.7 | 46.1 | 48.8 | 48.8 | 44.0 | 43.0 | 48.6 |
| CLP D100 | E | - | 37.3 | 35.8 | 37.7 | 33.9 | 38.6 | 37.0 | 47.2 | - | - | 46.4 | - | 47.8 | 48.8 | 45.5 | 46.1 | 48.8 | 48.8 | 44.0 | 42.9 | 48.6 |

Table 22: (Continued) CS Results Across Subsets and Distances (↓ better)

| Method | Dist | BC | BS1 | BS2 | BS3 | BS4 | BS5 | ES1 | HP | HS1 | HS2 | HU1 | HU2 | HU3 | HU4 | HW1 | HW2 | HW3 | HW4 | OC1 | Avg | Avg (Blind) |
|---|---|---|---|---|---|---|---|---|---|---|---|---|---|---|---|---|---|---|---|---|---|---|
| WLM D100 | C | 38.3 | 24.5 | 22.1 | 30.5 | 17.5 | 28.7 | 42.8 | 43.9 | 47.3 | 46.7 | 41.4 | 44.6 | 48.2 | 48.5 | 31.0 | 38.4 | 48.9 | 48.7 | 42.1 | 38.6 | 48.6 |
| WLM | C | 35.1 | 20.4 | 19.1 | 28.0 | 17.5 | 35.8 | 42.8 | 42.3 | 46.7 | 45.5 | 41.4 | 44.6 | 48.2 | 48.5 | **28.1** | 38.4 | 48.9 | 48.7 | 42.1 | 38.0 | 48.6 |
| CLP D100 | S | 45.9 | 46.9 | 38.7 | 40.7 | 37.1 | 41.0 | 36.4 | 48.1 | 46.3 | 47.4 | 46.1 | 43.0 | 48.0 | 48.9 | 45.6 | 46.4 | 48.8 | 48.9 | 46.8 | 44.8 | 48.6 |
| CLP | S | 41.6 | 40.3 | 38.7 | 33.7 | 37.1 | 31.5 | 36.4 | 46.1 | 43.7 | 45.2 | 46.1 | 43.0 | 48.0 | 48.9 | 42.6 | 46.4 | 48.8 | 48.9 | 46.8 | 42.8 | 48.6 |
| EF D30 | S | 43.8 | 40.7 | 34.1 | 37.8 | 28.8 | 35.1 | 45.7 | 47.3 | 47.2 | 47.8 | 43.4 | 43.4 | 48.3 | 48.5 | 45.1 | 45.7 | 49.0 | 48.7 | 43.4 | 43.3 | 48.6 |
| ETF D100 | E | - | 37.6 | 39.4 | 40.9 | 35.3 | 36.6 | 43.7 | 48.2 | - | 47.9 | 45.4 | - | 47.9 | 49.0 | 47.5 | 47.4 | 48.9 | 48.9 | 44.8 | 44.3 | 48.7 |
| ETF | E | - | 38.5 | 40.8 | 39.6 | 35.3 | 36.6 | 43.7 | 47.7 | - | 47.4 | 45.4 | - | 47.9 | 49.0 | 47.2 | 47.4 | 48.9 | 48.9 | 44.8 | 44.3 | 48.7 |
| ET | E | - | 38.5 | 39.5 | 39.7 | 35.5 | 36.6 | 43.7 | 47.8 | - | 47.4 | 45.4 | - | 47.9 | 49.0 | 47.2 | 47.4 | 48.9 | 49.0 | 44.9 | 44.3 | 48.7 |
| ET D100 | E | - | 37.7 | 39.5 | 41.0 | 35.5 | 36.7 | 43.7 | 48.2 | - | 47.9 | 45.4 | - | 47.9 | 49.0 | 47.5 | 47.4 | 48.9 | 49.0 | 44.9 | 44.4 | 48.7 |
| EWF D30 | E | - | 32.4 | 37.3 | 39.8 | 34.9 | 40.1 | 44.6 | 47.6 | - | 47.4 | 41.3 | - | 48.2 | 48.8 | 47.3 | 46.3 | 49.0 | 48.8 | 42.9 | 43.5 | 48.7 |
| WLM D30 | S | 42.3 | 39.4 | 28.9 | 32.8 | 20.1 | 31.3 | 44.9 | 44.1 | 47.4 | 46.9 | 42.3 | 48.0 | 48.3 | 48.7 | 30.0 | 37.3 | 49.0 | 49.0 | 42.6 | 40.7 | 48.7 |
| M | C | 43.3 | 24.1 | 26.3 | 29.6 | 22.9 | 34.4 | 40.1 | 44.2 | 46.5 | 47.5 | 40.9 | 43.5 | 48.5 | 48.6 | 42.2 | 44.0 | 49.0 | 48.8 | 41.7 | 40.3 | 48.7 |
| EWF D30 | S | 44.5 | 40.3 | 33.9 | 37.3 | 33.1 | 38.2 | 45.4 | 47.7 | 46.9 | 47.8 | 42.3 | 45.3 | 48.2 | 48.8 | 46.6 | 46.6 | 49.0 | 49.0 | 43.5 | 43.9 | 48.7 |
| EF D100 | E | - | 36.0 | 38.0 | 41.5 | 35.3 | 39.0 | 46.6 | 47.4 | - | 47.9 | 43.7 | 45.7 | 48.4 | 48.9 | 46.8 | 46.4 | 49.0 | 48.8 | 44.0 | 44.3 | 48.8 |
| EF | E | - | 36.4 | 38.7 | 40.0 | 35.3 | 39.0 | 46.6 | 47.0 | - | 47.5 | 43.7 | 45.7 | 48.4 | 48.9 | 46.3 | 46.4 | 49.0 | 48.8 | 44.0 | 44.2 | 48.8 |
| W2V | C | 35.6 | 32.5 | 30.6 | 36.6 | 29.1 | 38.9 | 45.7 | 46.9 | 46.9 | 46.7 | 40.4 | 45.1 | 48.5 | 48.5 | 36.7 | 43.0 | 49.0 | 49.1 | 42.3 | 41.7 | 48.8 |
| W2V D100 | C | 37.4 | 34.3 | 32.3 | 38.2 | 29.1 | 33.7 | 45.7 | 47.3 | 47.4 | 47.2 | 40.4 | 45.1 | 48.5 | 48.5 | 38.8 | 43.0 | 49.0 | 49.1 | 42.3 | 42.0 | 48.8 |
| M | S | 41.6 | 26.3 | 29.8 | 33.5 | 24.9 | 34.3 | 45.0 | 45.1 | 45.4 | 47.0 | 40.4 | 42.1 | 48.6 | 48.6 | 42.6 | 43.9 | 49.1 | 48.9 | 41.9 | 41.0 | 48.8 |
| WLM D30 | E | - | 30.4 | 28.8 | 34.3 | 23.4 | 33.0 | 44.5 | 45.1 | - | - | 44.2 | - | 48.6 | 48.9 | 36.9 | 42.3 | 49.2 | 49.1 | 43.3 | 40.1 | 48.9 |
| EWF D100 | E | - | 33.7 | 38.6 | 40.8 | 36.6 | 41.8 | 46.3 | 47.8 | - | 47.7 | 42.7 | - | 48.6 | 49.1 | 47.6 | 46.8 | 49.2 | 49.0 | 44.0 | 44.4 | 49.0 |
| EWF | E | - | 33.9 | 37.3 | 39.8 | 36.6 | 41.8 | 46.3 | 47.6 | - | 47.4 | 42.7 | - | 48.6 | 49.1 | 47.3 | 46.8 | 49.2 | 49.0 | 44.0 | 44.2 | 49.0 |
| W2V D30 | E | - | 39.9 | 38.3 | 42.0 | 34.9 | 38.0 | 46.9 | 48.0 | - | 48.0 | 42.4 | - | 48.7 | 48.8 | 42.3 | 44.9 | 49.2 | 49.3 | 43.8 | 44.1 | 49.0 |
| AC | S | 44.0 | 22.2 | 30.9 | 30.4 | 23.1 | 36.7 | 43.4 | 47.6 | 44.9 | 47.2 | 44.6 | 46.3 | 48.7 | 49.0 | 46.3 | 46.1 | 49.4 | 49.1 | 46.0 | 41.9 | 49.1 |
| CC | C | 42.8 | 37.2 | 36.1 | 41.0 | 34.0 | 40.6 | 48.7 | 47.7 | 47.6 | 48.1 | 45.7 | 44.7 | 49.0 | 48.7 | 46.3 | 46.7 | 49.4 | 49.3 | 44.6 | 44.7 | 49.1 |
| W2V D30 | S | 43.6 | 42.7 | 39.0 | 43.2 | 37.4 | 37.8 | 47.1 | 48.7 | 48.4 | 47.7 | 41.6 | 43.6 | 48.7 | 49.0 | 39.7 | 43.2 | 49.3 | 49.4 | 44.8 | 44.5 | 49.1 |
| M | E | - | 34.9 | 35.4 | 38.5 | 31.5 | 37.8 | 43.2 | 46.8 | - | - | 44.1 | - | 49.0 | 49.1 | 45.8 | 46.6 | 49.3 | - | 45.1 | 42.7 | 49.1 |
| WLM D100 | E | - | 33.9 | 32.0 | 36.7 | 26.5 | 36.2 | 45.9 | 46.2 | - | - | 44.3 | - | 48.9 | 49.1 | 39.0 | 43.6 | 49.3 | 49.2 | 44.9 | 41.7 | 49.1 |
| WLM | E | - | 30.4 | 28.8 | 34.3 | 26.5 | 40.2 | 45.9 | 45.1 | - | - | 44.3 | - | 48.9 | 49.1 | 36.9 | 43.6 | 49.3 | 49.2 | 44.9 | 41.1 | 49.1 |
| WLM D100 | S | 46.4 | 45.9 | 37.2 | 39.8 | 28.0 | 39.6 | 47.4 | 47.0 | 47.9 | 48.3 | 42.2 | 43.8 | 48.9 | 49.2 | 33.7 | 40.4 | 49.4 | 49.3 | 46.3 | 43.7 | 49.2 |
| WLM | S | 42.3 | 39.4 | 28.9 | 32.8 | 28.0 | 35.3 | 47.4 | 44.1 | 47.4 | 46.9 | 42.2 | 48.0 | 48.9 | 49.2 | 30.0 | 40.4 | 49.4 | 49.3 | 46.3 | 41.9 | 49.2 |
| MAE D30 | C | 42.3 | 29.0 | 30.2 | 30.1 | 22.5 | 42.5 | 38.4 | 46.0 | 47.3 | 47.1 | 41.1 | 47.7 | 48.6 | 49.5 | 47.0 | 46.7 | 49.3 | 49.4 | 43.8 | 42.0 | 49.2 |
| W2V D100 | E | - | 41.1 | 39.6 | 43.1 | 37.4 | 39.9 | 47.5 | 48.3 | - | - | 43.6 | - | 49.0 | 49.1 | 43.5 | 45.9 | 49.4 | 49.4 | 45.1 | 44.8 | 49.2 |
| W2V | E | - | 39.9 | 38.3 | 42.0 | 37.4 | 42.0 | 47.5 | 48.0 | - | 48.0 | 43.6 | - | 49.0 | 49.1 | 42.3 | 45.9 | 49.4 | 49.4 | 45.1 | 44.8 | 49.2 |
| EF D100 | S | 47.2 | 47.1 | 41.0 | 43.1 | 38.4 | 42.7 | 48.2 | 48.5 | 48.3 | 48.9 | 45.9 | 43.4 | 49.1 | 49.2 | 46.5 | 47.5 | 49.5 | 49.3 | 47.3 | 46.4 | 49.3 |
| EF | S | 43.8 | 40.7 | 34.1 | 37.8 | 38.4 | 42.7 | 48.2 | 47.3 | 47.2 | 47.8 | 45.9 | 43.4 | 49.1 | 49.2 | 45.1 | 47.5 | 49.5 | 49.3 | 47.3 | 45.0 | 49.3 |
| EWF D100 | S | 47.4 | 46.9 | 40.4 | 42.3 | 39.2 | 44.0 | 48.0 | 48.7 | 48.0 | 48.9 | 44.8 | 43.4 | 49.0 | 49.4 | 47.2 | 47.9 | 49.4 | 49.4 | 47.3 | 46.4 | 49.3 |
| EWF | S | 44.5 | 46.9 | 40.4 | 37.3 | 39.2 | 44.0 | 48.0 | 47.7 | 46.9 | 47.8 | 44.8 | 43.4 | 49.0 | 49.4 | 46.6 | 47.9 | 49.4 | 49.4 | 47.3 | 45.8 | 49.3 |
| MAE D100 | C | 43.1 | 31.1 | 31.4 | 31.3 | 24.6 | 43.0 | 40.9 | 46.3 | 47.5 | 47.5 | 42.4 | 47.7 | 48.8 | 49.6 | 47.4 | 47.2 | 49.3 | 49.5 | 45.1 | 42.8 | 49.3 |
| MAE | C | 42.3 | 31.1 | 30.2 | 30.1 | 24.6 | 45.0 | 40.9 | 46.0 | 47.3 | 47.1 | 42.4 | 47.7 | 48.8 | 49.6 | 47.0 | 47.2 | 49.3 | 49.5 | 45.1 | 42.7 | 49.3 |
| MAE D30 | S | 45.9 | 40.9 | 36.7 | 36.4 | 30.4 | 42.9 | 41.9 | 47.7 | 47.5 | 47.8 | 43.9 | - | 48.9 | 49.3 | 47.7 | 47.6 | 49.5 | 49.5 | 45.7 | 44.5 | 49.3 |
| CC | E | 43.8 | 42.3 | 41.7 | 44.3 | 38.6 | 43.1 | 49.2 | 48.6 | 47.7 | 48.7 | 47.5 | 46.0 | 49.3 | 49.2 | 47.8 | 47.9 | 49.6 | 49.6 | 46.3 | 46.4 | 49.4 |
| MAE D30 | E | - | 37.1 | 37.0 | 37.5 | 30.9 | 45.5 | 43.6 | 47.5 | - | 48.3 | 44.4 | - | 49.2 | 49.6 | 48.3 | 48.1 | 49.5 | 49.6 | 46.6 | 44.5 | 49.5 |
| MAE | S | 45.9 | 46.7 | 40.3 | 36.4 | 37.1 | 45.4 | 45.3 | 47.7 | 47.5 | 47.8 | 45.4 | 50.0 | 49.2 | 49.5 | 47.7 | 48.5 | 49.6 | 49.7 | 47.8 | 46.2 | 49.5 |
| MAE D100 | S | 47.7 | 46.7 | 40.3 | 40.3 | 37.1 | 44.4 | 45.3 | 48.6 | 47.9 | 48.5 | 45.4 | - | 49.2 | 49.5 | 48.4 | 48.5 | 49.6 | 49.7 | 47.8 | 46.4 | 49.5 |
| W2V D100 | S | 47.4 | 47.5 | 44.9 | 46.8 | 44.2 | 43.3 | 48.6 | 49.3 | 48.4 | 48.8 | 43.0 | 43.6 | 49.3 | 49.6 | 41.9 | 45.6 | 49.6 | 49.7 | 47.6 | 46.8 | 49.5 |
| W2V | S | 43.6 | 42.7 | 39.0 | 43.2 | 44.2 | 37.6 | 48.6 | 48.7 | 48.4 | 47.7 | 43.0 | 43.6 | 49.3 | 49.6 | 39.7 | 45.6 | 49.6 | 49.7 | 47.6 | 45.3 | 49.5 |
| MAE | E | - | 40.0 | 38.2 | 37.5 | 33.1 | 47.1 | 45.0 | 47.5 | - | 48.3 | 45.4 | - | 49.3 | 49.6 | 48.3 | 48.4 | 49.6 | 49.7 | 47.4 | 45.3 | 49.5 |
| MAE D100 | E | - | 38.7 | 38.2 | 38.6 | 33.1 | 45.9 | 45.0 | 47.8 | - | 48.6 | 45.4 | - | 49.3 | 49.6 | 48.5 | 48.4 | 49.6 | 49.7 | 47.4 | 45.2 | 49.5 |

Table 23: CSCF Results Across Subsets and Distances (↓ better)

| Method | Dist | BC | BS1 | BS2 | BS3 | BS4 | BS5 | ES1 | HP | HS1 | HS2 | HU1 | HU2 | HU3 | HU4 | HW1 | HW2 | HW3 | HW4 | OC1 | Avg | Avg (Blind) |
|---|---|---|---|---|---|---|---|---|---|---|---|---|---|---|---|---|---|---|---|---|---|---|
| EWMTF D100 | S | **0.1** | **0.0** | 0.2 | 0.8 | **0.0** | 0.1 | 0.0 | 0.2 | 0.0 | **0.4** | 0.0 | 9.4 | **5.5** | 11.6 | **0.0** | 0.4 | **8.1** | **8.9** | 0.2 | 2.4 | **8.6** |
| EWMTF | S | 1.5 | **0.0** | 1.2 | 2.3 | **0.0** | 0.1 | 0.0 | 1.3 | 2.7 | 1.3 | **0.0** | 9.4 | **5.5** | 11.6 | 1.7 | 0.4 | **8.1** | **8.9** | 0.2 | 3.0 | **8.6** |
| EMTF | S | 1.1 | **0.0** | 0.2 | 2.1 | **0.0** | 0.1 | 0.0 | 0.7 | 3.8 | 1.7 | 0.0 | 6.7 | 9.7 | **8.3** | 1.3 | 0.4 | 10.3 | 10.0 | 0.2 | 3.0 | 9.6 |
| EMTF D100 | S | 0.4 | **0.0** | 0.2 | 1.5 | **0.0** | 0.1 | 0.0 | **0.2** | 0.0 | 0.7 | 0.0 | 6.7 | 9.7 | **8.3** | 0.4 | 0.4 | 10.3 | 10.0 | 0.2 | 2.6 | 9.6 |
| EWTF D100 | S | 0.3 | **0.0** | 0.1 | 1.2 | **0.0** | 0.1 | 0.0 | 0.2 | 0.0 | 0.5 | 0.0 | 2.7 | 8.9 | 13.8 | 1.8 | 2.3 | 9.1 | 10.4 | 0.2 | 2.7 | 10.6 |
| EWTF | S | 0.7 | **0.0** | 0.1 | 1.7 | **0.0** | 0.1 | 0.0 | 1.3 | **0.0** | 1.6 | 0.0 | 2.7 | 8.9 | 13.8 | 8.7 | 2.3 | 9.1 | 10.4 | 0.2 | 3.2 | 10.6 |
| EWT D100 | S | 0.3 | **0.0** | 0.2 | 1.0 | **0.0** | 0.1 | 0.0 | 0.3 | **0.0** | 0.5 | 0.0 | 4.9 | 8.8 | 13.8 | 1.8 | 2.4 | 9.3 | 10.5 | 0.2 | 2.9 | 10.6 |
| EWT | S | 0.8 | **0.0** | 0.1 | 1.5 | **0.0** | 0.1 | 0.0 | 1.4 | **0.0** | 1.6 | 0.0 | 4.9 | 8.8 | 13.8 | 8.7 | 2.4 | 9.3 | 10.5 | 0.2 | 3.4 | 10.6 |
| ETF | S | 0.7 | **0.0** | 0.1 | 2.4 | **0.0** | 0.1 | 0.0 | 0.7 | 6.6 | 1.6 | 0.0 | 3.5 | 8.6 | 12.0 | 3.1 | 1.3 | 10.4 | 13.8 | **0.0** | 3.4 | 11.2 |
| ETF D100 | S | **0.1** | **0.0** | 0.1 | 2.5 | **0.0** | 0.1 | 0.0 | 0.2 | 1.1 | 0.7 | 0.0 | 3.5 | 8.6 | 12.0 | 0.4 | 1.3 | 10.4 | 13.8 | **0.0** | 2.9 | 11.2 |
| ET D100 | S | 0.3 | **0.0** | 0.1 | 2.5 | **0.0** | 0.1 | 0.0 | 0.2 | 1.1 | 0.9 | 0.0 | 3.7 | 9.0 | 11.8 | 0.4 | 1.3 | 10.8 | 13.4 | **0.0** | 2.9 | 11.3 |
| ET | S | 0.7 | **0.0** | 0.9 | 2.5 | **0.0** | 0.1 | 0.0 | 0.7 | 8.2 | 1.9 | 0.0 | 3.7 | 9.0 | 11.8 | 3.1 | 1.3 | 10.8 | 13.4 | **0.0** | 3.6 | 11.3 |
| ETF D100 | C | 0.4 | **0.0** | 0.8 | 2.8 | 0.0 | 0.1 | 0.2 | 1.3 | 2.2 | 1.8 | 0.2 | 4.9 | 11.8 | 15.1 | 0.9 | 2.7 | 13.4 | 15.7 | 1.4 | 4.0 | 14.0 |
| ETF | C | 0.9 | **0.0** | 0.8 | 3.2 | 0.0 | 0.1 | 0.2 | 2.2 | 6.0 | 2.3 | 0.2 | 4.9 | 11.8 | 15.1 | 3.6 | 2.7 | 13.4 | 15.7 | 1.4 | 4.5 | 14.0 |
| EWTF | C | 1.6 | **0.0** | 0.2 | 2.5 | 0.0 | 0.1 | 0.0 | 3.7 | 5.5 | 2.1 | 0.4 | 3.9 | 10.6 | 17.2 | 9.8 | 4.6 | 13.8 | 14.9 | 1.4 | 4.9 | 14.1 |
| EWTF D100 | C | 1.2 | **0.0** | 0.2 | 2.3 | 0.0 | 0.1 | 0.0 | 2.2 | 2.2 | 1.4 | 0.4 | 3.9 | 10.6 | 17.2 | 3.3 | 4.6 | 13.8 | 14.9 | 1.4 | 4.2 | 14.1 |
| ET D100 | C | 0.4 | **0.0** | 0.7 | 2.9 | 0.0 | 0.1 | 0.2 | 1.2 | 2.2 | 1.9 | 0.2 | 6.0 | 12.0 | 15.5 | 0.9 | 2.8 | 13.4 | 15.9 | 1.7 | 4.1 | 14.2 |
| ET | C | 0.7 | **0.0** | 0.7 | 3.2 | 0.0 | 0.1 | 0.2 | 2.0 | 6.0 | 2.4 | 0.2 | 6.0 | 12.0 | 15.5 | 3.7 | 2.8 | 13.4 | 15.9 | 1.7 | 4.6 | 14.2 |
| EWT | C | 1.5 | **0.0** | 0.2 | 2.5 | 0.0 | 0.1 | 0.0 | 3.7 | 4.9 | 2.2 | 0.4 | 8.9 | 10.8 | 17.5 | 10.0 | 4.6 | 13.9 | 14.9 | 1.7 | 5.1 | 14.3 |
| EWT D100 | C | 1.2 | **0.0** | 0.2 | 2.3 | 0.0 | 0.1 | 0.0 | 2.2 | 2.2 | 1.5 | 0.4 | 8.9 | 10.8 | 17.5 | 3.4 | 4.6 | 13.9 | 14.9 | 1.7 | 4.5 | 14.3 |
| EWMTF D30 | S | 1.5 | **0.0** | 0.2 | 2.3 | 0.1 | 0.1 | 0.0 | 1.3 | 2.7 | 1.3 | 0.1 | 9.4 | 9.0 | 19.3 | 1.7 | 4.8 | 15.2 | 13.7 | 0.5 | 4.4 | 14.3 |
| CLP D100 | S | **0.1** | **0.0** | **0.0** | 0.9 | **0.0** | 0.1 | 0.0 | 0.4 | 1.1 | 0.7 | 0.4 | 5.8 | 12.5 | 17.7 | 0.7 | 1.8 | 14.4 | 14.3 | 0.2 | 3.7 | 14.7 |
| CLP | S | 0.4 | **0.0** | **0.0** | 2.9 | **0.0** | 0.2 | **0.0** | 0.9 | 2.7 | 1.7 | 0.4 | 5.8 | 12.5 | 17.7 | 1.3 | 1.8 | 14.4 | 14.3 | 0.2 | 4.1 | 14.7 |
| EWTF D30 | S | 0.7 | **0.0** | 0.2 | 1.7 | 0.0 | 0.1 | 0.0 | 1.3 | **0.0** | 1.6 | 1.4 | 6.0 | 12.6 | 17.1 | 8.7 | 7.3 | 14.6 | 14.7 | 0.5 | 4.7 | 14.7 |
| ETF D30 | S | 0.7 | **0.0** | 0.8 | 2.4 | 0.0 | 0.1 | 0.0 | 0.7 | 6.6 | 1.6 | 0.0 | 3.5 | 12.4 | 15.7 | 3.1 | 5.2 | 15.1 | 16.5 | 0.2 | 4.5 | 14.9 |
| EWT D30 | S | 0.8 | **0.0** | 0.3 | 1.5 | 0.0 | 0.1 | 0.0 | 1.4 | **0.0** | 1.6 | 1.3 | 4.9 | 12.5 | 17.8 | 8.7 | 7.3 | 15.3 | 15.6 | 0.7 | 4.7 | 15.3 |
| EMTF D30 | S | 1.1 | **0.0** | 1.0 | 2.1 | 0.1 | 0.1 | 0.1 | 0.7 | 3.8 | 1.7 | 0.1 | 6.7 | 12.9 | 15.0 | 1.3 | 3.9 | 16.5 | 16.8 | 0.2 | 4.4 | 15.3 |
| ET D30 | S | 0.7 | **0.0** | 0.9 | 1.9 | 0.0 | 0.1 | 0.0 | 0.7 | 8.2 | 1.9 | 0.0 | 3.7 | 12.9 | 17.0 | 3.1 | 5.2 | 15.5 | 17.4 | 0.2 | 4.8 | 15.7 |
| CLP | C | 1.1 | **0.0** | 0.2 | 3.2 | 0.1 | 0.2 | **0.0** | 2.0 | 4.9 | 2.5 | 3.2 | 6.0 | 12.7 | 19.1 | 2.1 | 3.0 | 14.9 | 16.4 | 0.7 | 4.9 | 15.8 |
| CLP D100 | C | 0.9 | **0.0** | 0.2 | 2.9 | 0.1 | 0.1 | 0.0 | 1.7 | 4.4 | 2.3 | 3.2 | 6.0 | 12.7 | 19.1 | 1.5 | 3.0 | 14.9 | 16.4 | 0.7 | 4.7 | 15.8 |
| CLP D30 | S | 0.4 | **0.0** | 0.1 | 2.9 | **0.0** | 0.1 | 0.0 | 0.9 | 2.7 | 1.7 | 2.2 | 5.8 | 13.6 | 20.0 | 1.3 | 2.8 | 15.2 | 16.1 | 0.7 | 4.6 | 16.2 |
| ETF D30 | C | 0.9 | **0.0** | 1.0 | 3.2 | 0.1 | 0.2 | 0.2 | 2.2 | 6.0 | 2.3 | 1.7 | 4.9 | 13.9 | 18.4 | 3.6 | 5.9 | 16.2 | 17.8 | 1.2 | 5.3 | 16.6 |
| EWMTF | C | 6.7 | **0.0** | 0.9 | 2.9 | 0.3 | 0.5 | 0.5 | 6.3 | 5.5 | 4.9 | 0.8 | 10.2 | 12.6 | 19.7 | 12.0 | 12.9 | 17.3 | 16.8 | 2.9 | 7.0 | 16.6 |
| EWMTF D100 | C | 6.3 | **0.0** | 0.5 | 2.9 | 0.3 | 0.5 | 0.5 | 5.2 | 4.4 | 4.5 | 0.8 | 10.2 | 12.6 | 19.7 | 5.8 | 12.9 | 17.3 | 16.8 | 2.9 | 6.5 | 16.6 |
| EW | C | 6.7 | **0.0** | 0.2 | 2.9 | 0.3 | 1.6 | 0.5 | 6.3 | 5.5 | 4.9 | 0.8 | 10.2 | 12.6 | 19.7 | 12.0 | 12.9 | 17.3 | 16.8 | 2.9 | 7.1 | 16.6 |
| CLP D30 | C | 1.1 | **0.0** | 0.2 | 3.2 | 0.1 | 0.1 | 0.0 | 2.0 | 4.9 | 2.5 | 4.6 | 6.0 | 13.2 | 20.3 | 2.1 | 3.6 | 15.9 | 17.3 | 0.7 | 5.2 | 16.7 |
| EWTF D30 | C | 1.6 | **0.0** | 0.3 | 2.5 | 0.1 | 0.1 | 0.1 | 3.7 | 5.5 | 2.1 | 2.7 | 6.3 | 13.3 | 20.2 | 9.8 | 8.5 | 17.1 | 17.0 | 1.9 | 5.9 | 16.9 |
| ET D30 | C | 0.7 | **0.0** | 1.0 | 3.2 | 0.1 | 0.2 | 0.2 | 2.0 | 6.0 | 2.4 | 1.7 | 6.0 | 14.2 | 19.1 | 3.7 | 5.9 | 16.3 | 18.1 | 1.4 | 5.4 | 16.9 |
| EWT D30 | C | 1.5 | **0.0** | 0.3 | 2.5 | 0.1 | 0.1 | 0.1 | 3.7 | 4.9 | 2.2 | 2.8 | 8.9 | 13.5 | 20.5 | 10.0 | 8.6 | 17.1 | 17.3 | 1.7 | 6.1 | 17.1 |
| EWMTF D30 | C | 6.7 | **0.0** | 0.6 | 2.9 | 0.3 | 0.6 | 0.7 | 6.3 | 5.5 | 4.9 | 2.8 | 10.2 | 14.3 | 21.9 | 12.0 | 17.7 | 20.2 | 18.5 | 3.1 | 7.9 | 18.8 |
| EWTF | E | 2.6 | **0.0** | 0.2 | 3.2 | 0.0 | 0.1 | 0.0 | 16.6 | 2.7 | 3.4 | 0.4 | 6.0 | 13.1 | 24.3 | 15.7 | 9.0 | 19.0 | 21.9 | 2.9 | 7.4 | 19.6 |
| EWTF D100 | E | 4.9 | **0.0** | 0.2 | 3.2 | 0.0 | 0.1 | 0.0 | 22.8 | 1.1 | 3.0 | 0.4 | 6.0 | 13.1 | 24.3 | 9.8 | 9.0 | 19.0 | 21.9 | 2.9 | 7.5 | 19.6 |
| EWT D100 | E | 4.8 | **0.0** | 0.2 | 3.3 | 0.0 | 0.1 | 0.0 | 25.2 | 1.6 | 3.3 | 0.5 | 10.3 | 13.3 | 24.6 | 10.1 | 9.4 | 19.2 | 22.0 | 2.9 | 7.9 | 19.8 |
| EWT | E | 2.6 | **0.0** | 0.2 | 3.4 | 0.0 | 0.1 | 0.0 | 18.1 | 3.3 | 3.6 | 0.5 | 10.3 | 13.3 | 24.6 | 16.0 | 9.4 | 19.2 | 22.0 | 2.9 | 7.9 | 19.8 |
| EMTF D100 | C | 7.7 | **0.0** | 1.4 | 3.1 | 0.2 | 1.4 | 0.7 | 4.0 | 8.2 | 6.2 | 2.3 | 7.6 | 15.9 | 22.2 | 1.3 | 5.7 | 20.2 | 21.5 | 2.1 | 6.9 | 19.9 |
| E | C | 7.8 | **0.0** | **0.0** | 3.5 | 0.2 | 2.3 | 0.7 | 5.0 | 9.3 | 6.7 | 2.3 | 7.6 | 15.9 | 22.2 | 4.1 | 5.7 | 20.2 | 21.5 | 2.1 | 7.2 | 19.9 |
| EMTF | C | 7.8 | **0.0** | 1.4 | 3.5 | 0.2 | 1.4 | 0.7 | 5.0 | 9.3 | 6.7 | 2.3 | 7.6 | 15.9 | 22.2 | 4.1 | 5.7 | 20.2 | 21.5 | 2.1 | 7.2 | 19.9 |
| EWTF D30 | E | 2.6 | **0.0** | 0.2 | 3.2 | 0.0 | 0.1 | 0.0 | 16.6 | 2.7 | 3.4 | 2.3 | 7.0 | 14.0 | 24.3 | 15.7 | 10.6 | 20.6 | 21.2 | 1.2 | 7.7 | 20.0 |
| EWT D30 | E | 2.6 | **0.0** | 0.2 | 3.4 | 0.0 | 0.1 | 0.0 | 18.1 | 3.3 | 3.6 | 2.5 | 10.3 | 14.3 | 24.1 | 16.0 | 10.9 | 20.8 | 21.4 | 1.4 | 8.1 | 20.2 |
| ETF | E | 0.4 | **0.0** | 1.1 | 3.6 | 0.0 | 0.1 | 0.2 | 3.4 | 1.6 | 3.4 | 0.1 | 6.0 | 13.8 | 25.2 | 5.4 | 4.5 | 20.3 | 22.6 | 1.0 | 5.9 | 20.5 |
| ETF D100 | E | 0.5 | **0.0** | 1.0 | 3.7 | 0.0 | 0.1 | 0.2 | 5.7 | 0.5 | 3.3 | 0.1 | 6.0 | 13.8 | 25.2 | 2.0 | 4.5 | 20.3 | 22.6 | 1.0 | 5.8 | 20.5 |
| ET D100 | E | 0.7 | **0.0** | 1.0 | 3.7 | 0.0 | 0.1 | 0.2 | 6.0 | 1.1 | 3.4 | 0.2 | 6.4 | 13.9 | 25.5 | 2.1 | 4.8 | 20.7 | 23.2 | 1.2 | 6.0 | 20.8 |
| ET | E | 0.4 | **0.0** | 1.0 | 3.5 | 0.0 | 0.1 | 0.2 | 3.5 | 2.2 | 3.5 | 0.2 | 6.4 | 13.9 | 25.5 | 5.5 | 4.8 | 20.7 | 23.2 | 1.2 | 6.1 | 20.8 |
| ETF D30 | E | 0.4 | **0.0** | 1.0 | 3.6 | 0.0 | 0.1 | 0.1 | 3.4 | 1.6 | 3.4 | 1.2 | 6.0 | 15.6 | 24.9 | 5.4 | 6.8 | 21.4 | 22.3 | 1.0 | 6.2 | 21.1 |
| CLP D30 | E | 0.9 | **0.0** | 0.2 | 3.5 | 0.0 | 0.1 | 0.0 | 4.0 | 1.6 | 3.9 | 3.9 | 5.3 | 16.2 | 24.0 | 1.8 | 4.3 | 20.3 | 23.9 | 0.7 | 6.0 | 21.1 |
| ET D30 | E | 0.4 | **0.0** | 1.1 | 3.5 | 0.0 | 0.1 | 0.1 | 3.5 | 2.2 | 3.5 | 1.3 | 6.4 | 15.8 | 25.1 | 5.5 | 7.0 | 21.7 | 22.6 | 1.0 | 6.4 | 21.3 |
| EMTF D30 | C | 7.8 | **0.0** | 1.4 | 3.5 | 0.4 | 1.7 | 0.9 | 5.0 | 9.3 | 6.7 | 5.4 | 7.6 | 17.8 | 23.5 | 4.1 | 9.6 | 22.4 | 23.0 | 2.9 | 8.1 | 21.7 |
| CLP | E | 0.9 | **0.0** | 0.3 | 3.5 | 0.0 | 0.3 | 0.0 | 4.0 | 1.6 | 3.9 | 3.1 | 5.3 | 17.1 | 24.8 | 1.8 | 4.7 | 21.1 | 24.8 | 0.7 | 6.2 | 22.0 |
| CLP D100 | E | 0.9 | **0.0** | 0.3 | 3.5 | 0.0 | 0.2 | 0.0 | 4.1 | 1.6 | 5.2 | 3.1 | 5.3 | 17.1 | 24.8 | 2.1 | 4.7 | 21.1 | 24.8 | 0.7 | 6.3 | 22.0 |
| WLM D100 | S | 2.1 | **0.0** | 1.0 | 3.0 | 0.7 | 0.1 | 0.1 | 0.4 | 1.1 | 1.7 | 0.0 | 32.6 | 21.3 | 23.0 | 0.0 | **0.2** | 23.2 | 21.0 | 0.2 | 6.9 | 22.1 |
| WLM | S | 5.0 | **0.0** | 1.2 | 3.6 | 0.7 | 4.6 | 0.1 | 0.9 | 3.3 | 3.5 | 0.0 | 3.7 | 21.3 | 23.0 | 0.1 | **0.2** | 23.2 | 21.0 | 0.2 | 6.1 | 22.1 |
| EF D100 | S | 0.8 | **0.0** | 0.2 | 3.6 | **0.0** | 0.1 | 0.8 | 0.2 | **0.0** | 2.6 | 0.3 | 12.7 | 24.2 | 17.6 | 0.8 | 5.3 | 26.9 | 22.4 | **0.0** | 6.2 | 22.8 |
| EF | S | 1.9 | **0.0** | 0.8 | 3.6 | **0.0** | 0.1 | 0.8 | 1.1 | 2.2 | 4.5 | 0.3 | 12.7 | 24.2 | 17.6 | 4.3 | 5.3 | 26.9 | 22.4 | **0.0** | 6.8 | 22.8 |
| EWF D100 | S | 0.8 | **0.0** | 0.2 | 2.5 | **0.0** | 0.1 | 0.5 | 0.3 | 0.5 | 1.9 | 0.1 | 13.0 | 19.9 | 29.3 | 1.6 | 6.2 | 24.3 | 24.0 | 0.2 | 6.6 | 24.4 |
| EWF | S | 2.8 | **0.0** | 0.2 | 3.2 | **0.0** | 0.1 | 0.5 | 2.0 | 2.7 | 4.1 | 0.1 | 13.0 | 19.9 | 29.3 | 9.2 | 6.2 | 24.3 | 24.0 | 0.2 | 7.5 | 24.4 |
| EF D30 | S | 1.9 | **0.0** | 0.8 | 3.6 | 0.0 | 0.2 | 1.6 | 1.1 | 2.2 | 4.5 | 2.8 | 12.7 | 24.7 | 16.7 | 4.3 | 9.6 | 28.6 | 26.2 | 1.4 | 7.8 | 25.6 |
| EWMTF D100 | E | 8.3 | **0.0** | 0.4 | 4.7 | 0.0 | 0.6 | 3.1 | 10.5 | 4.9 | 10.5 | 0.3 | 8.7 | 19.9 | 30.0 | 9.1 | 15.3 | 27.2 | 28.5 | 8.6 | 10.0 | 26.4 |
| EWMTF | E | 7.1 | **0.0** | 0.4 | 4.8 | 0.0 | 0.6 | 3.1 | 11.0 | 4.9 | 10.3 | 0.7 | 8.7 | 19.9 | 30.0 | 14.2 | 15.3 | 27.2 | 28.5 | 8.6 | 10.3 | 26.4 |
| EW | E | 7.1 | **0.0** | 0.4 | 4.8 | 0.0 | 0.8 | 3.1 | 11.0 | 4.9 | 10.3 | 0.7 | 8.7 | 19.9 | 30.0 | 14.2 | 15.3 | 27.2 | 28.5 | 8.6 | 10.3 | 26.4 |
| E | E | 15.1 | **0.0** | 1.0 | 6.8 | 0.2 | 1.4 | 2.9 | 9.0 | 6.6 | 11.3 | 1.5 | 6.9 | 21.4 | 29.0 | 7.6 | 10.4 | 27.3 | 27.9 | 8.1 | 10.2 | 26.4 |
| EMTF D100 | E | 10.6 | **0.0** | 1.5 | 6.5 | 0.2 | 0.8 | 2.9 | 9.1 | 6.0 | 11.4 | 1.5 | 6.9 | 21.4 | 29.0 | 4.3 | 10.4 | 27.3 | 27.9 | 8.1 | 9.8 | 26.4 |
| EMTF | E | 15.1 | **0.0** | 1.6 | 6.8 | 0.2 | 0.8 | 2.9 | 9.0 | 6.6 | 11.3 | 1.5 | 6.9 | 21.4 | 29.0 | 7.6 | 10.4 | 27.3 | 27.9 | 8.1 | 10.2 | 26.4 |
| WLM D100 | C | 8.2 | **0.0** | 2.3 | 4.2 | 2.1 | 2.0 | 4.0 | 1.2 | 3.3 | 4.9 | 0.1 | 0.6 | 24.2 | 26.5 | 0.0 | 0.5 | 28.0 | 28.1 | 1.2 | 7.4 | 26.7 |
| WLM | C | 8.5 | **0.0** | 2.3 | 4.5 | 2.1 | 4.0 | 4.0 | 1.6 | 4.4 | 5.9 | 0.1 | 0.6 | 24.2 | 26.5 | 0.1 | 0.5 | 28.0 | 28.1 | 1.2 | 7.7 | 26.7 |
| WLM D30 | S | 5.0 | **0.0** | 1.2 | 3.6 | 2.0 | 0.8 | 1.2 | 0.9 | 3.3 | 3.5 | 1.3 | 3.7 | 25.3 | 29.0 | 0.1 | 0.7 | 27.2 | 27.1 | 0.5 | 7.2 | 27.2 |
| EWMTF D30 | E | 7.1 | **0.0** | 0.4 | 4.8 | 0.1 | 0.6 | 3.1 | 11.0 | 4.9 | 10.3 | 1.6 | 8.7 | 21.0 | 30.6 | 14.2 | 18.3 | 28.3 | 29.0 | 7.9 | 10.6 | 27.2 |

Table 23: (Continued) CSCF Results Across Subsets and Distances (↓ better)

| Method | Dist | BC | BS1 | BS2 | BS3 | BS4 | BS5 | ES1 | HP | HS1 | HS2 | HU1 | HU2 | HU3 | HU4 | HW1 | HW2 | HW3 | HW4 | OC1 | Avg | Avg (Blind) |
|---|---|---|---|---|---|---|---|---|---|---|---|---|---|---|---|---|---|---|---|---|---|---|
| EMTF D30 | E | 15.1 | **0.0** | 1.5 | 6.8 | 0.2 | 0.9 | 2.7 | 9.0 | 6.6 | 11.3 | 2.9 | 6.9 | 23.1 | 29.4 | 7.6 | 13.3 | 28.7 | 28.7 | 8.1 | 10.7 | 27.5 |
| EWF D30 | S | 2.8 | **0.0** | 0.2 | 3.2 | 0.0 | **0.1** | 1.3 | 2.0 | 2.7 | 4.1 | 1.9 | 9.5 | 24.4 | 32.7 | 9.2 | 12.6 | 27.8 | 28.8 | 1.0 | 8.6 | 28.4 |
| EF D100 | C | 6.9 | **0.0** | 1.7 | 4.0 | 0.2 | 0.8 | 5.1 | 5.5 | 6.0 | 7.9 | 4.8 | 15.2 | 26.3 | 29.5 | 4.5 | 11.6 | 31.0 | 29.6 | 5.5 | 10.3 | 29.1 |
| EF | C | 7.3 | **0.0** | 1.7 | 4.2 | 0.2 | 0.8 | 5.1 | 6.1 | 7.7 | 8.4 | 4.8 | 15.2 | 26.3 | 29.5 | 7.0 | 11.6 | 31.0 | 29.6 | 5.5 | 10.6 | 29.1 |
| EWF D100 | C | 7.3 | **0.0** | 0.9 | 4.0 | 0.3 | 0.5 | 4.5 | 5.7 | 9.9 | 7.8 | 4.6 | 16.2 | 25.1 | 32.3 | 10.3 | 16.1 | 29.1 | 31.2 | 3.3 | 11.0 | 29.4 |
| EWF | C | 8.1 | **0.0** | 0.9 | 4.2 | 0.3 | 0.5 | 4.5 | 6.1 | 11.5 | 8.4 | 4.6 | 16.2 | 25.1 | 32.3 | 14.4 | 16.1 | 29.1 | 31.2 | 3.3 | 11.4 | 29.4 |
| WLM D30 | C | 8.5 | **0.0** | 2.3 | 4.5 | 2.7 | 2.5 | 4.7 | 1.6 | 4.4 | 5.9 | 2.0 | 0.6 | 26.5 | 30.8 | 0.1 | 1.0 | 30.4 | 30.7 | 2.1 | 8.5 | 29.6 |
| EF D30 | C | 7.3 | **0.0** | 2.0 | 4.2 | 0.2 | 1.0 | 5.1 | 5.6 | 6.1 | 7.7 | 8.4 | 6.9 | 15.2 | 27.0 | 29.7 | 7.0 | 13.5 | 31.8 | 30.1 | 6.0 | 11.0 | 29.6 |
| EWF D30 | C | 8.1 | **0.0** | 0.9 | 4.2 | 0.5 | 0.7 | 5.1 | 6.1 | 11.5 | 8.4 | 7.1 | 13.2 | 26.2 | 32.6 | 14.4 | 18.4 | 30.4 | 32.0 | 3.1 | 11.7 | 30.3 |
| W2V | S | 9.5 | 3.3 | 3.0 | 7.4 | 0.7 | 7.6 | 1.3 | 5.5 | 11.0 | 5.3 | 0.0 | 28.0 | 27.7 | 29.3 | 0.8 | 3.5 | 31.4 | 35.9 | 3.8 | 11.3 | 31.1 |
| W2V D100 | S | 6.3 | **0.0** | 2.1 | 5.7 | 0.7 | 0.9 | 1.3 | 2.8 | 2.2 | 2.8 | 0.0 | 28.0 | 27.7 | 29.3 | 0.2 | 3.5 | 31.4 | 35.9 | 3.8 | 9.7 | 31.1 |
| MAE | S | 5.0 | **0.0** | 0.4 | 2.9 | 0.3 | 2.3 | **0.0** | 4.5 | 7.1 | 4.6 | 0.2 | **0.0** | 30.2 | 30.4 | 14.4 | 14.4 | 34.5 | - | - | 8.9 | 31.7 |
| MAE D100 | S | 2.4 | **0.0** | 0.4 | **0.7** | 0.3 | **0.1** | **0.0** | 1.6 | 3.8 | 1.5 | 0.2 | - | 30.2 | 30.4 | 8.4 | 14.4 | 34.5 | 36.0 | 0.2 | 9.2 | 32.8 |
| EF D30 | E | 10.6 | **0.0** | 1.5 | 6.8 | 0.2 | 0.2 | 7.5 | 11.4 | 4.9 | 13.3 | 5.1 | 14.0 | 30.0 | 34.2 | 9.2 | 14.6 | 33.8 | 34.6 | 7.9 | 12.6 | 33.1 |
| EF D100 | E | 9.9 | **0.0** | 1.6 | 7.2 | 0.2 | 0.2 | 8.6 | 11.7 | 5.5 | 14.5 | 4.0 | 14.0 | 30.0 | 34.2 | 7.2 | 14.0 | 33.6 | 35.0 | 9.5 | 12.7 | 33.2 |
| EF | E | 10.6 | **0.0** | 1.5 | 6.8 | 0.2 | 0.2 | 8.6 | 11.4 | 4.9 | 13.3 | 4.0 | 14.0 | 30.0 | 34.2 | 9.2 | 14.0 | 33.6 | 35.0 | 9.5 | 12.7 | 33.2 |
| VC | S | 7.3 | **0.0** | 0.7 | 4.3 | 0.0 | 0.7 | 5.7 | 5.1 | 7.1 | 9.4 | 10.2 | 7.6 | 33.4 | 33.5 | 3.1 | 6.9 | 33.3 | 33.6 | 4.0 | 10.8 | 33.5 |
| VC | C | 7.9 | **0.0** | 0.6 | 3.3 | 0.0 | 0.5 | 6.5 | 3.2 | 6.0 | 9.3 | 8.4 | 7.1 | 32.0 | 33.7 | 2.0 | 5.1 | 33.9 | 34.5 | 3.8 | 10.4 | 33.5 |
| EWF | E | 7.8 | **0.0** | 1.0 | 5.8 | 0.1 | 0.8 | 14.1 | 8.4 | 5.5 | 13.1 | 3.1 | 14.5 | 28.3 | 37.1 | 14.0 | 16.2 | 32.7 | 36.2 | 7.1 | 12.9 | 33.6 |
| EWF D100 | E | 11.0 | **0.0** | 1.2 | 6.1 | 0.1 | 0.8 | 14.1 | 8.4 | 4.9 | 13.7 | 3.1 | 14.5 | 28.3 | 37.1 | 11.2 | 16.2 | 32.7 | 36.2 | 7.1 | 13.0 | 33.6 |
| EWF D30 | E | 7.8 | **0.0** | 1.0 | 5.8 | 0.2 | 0.7 | 14.2 | 8.4 | 5.5 | 13.1 | 5.0 | 13.3 | 28.6 | 36.8 | 14.0 | 16.8 | 32.8 | 36.2 | 6.7 | 13.0 | 33.6 |
| W2V D30 | S | 9.5 | 3.3 | 3.0 | 7.4 | 3.0 | 1.3 | 3.2 | 5.5 | 11.0 | 5.3 | 1.2 | 28.0 | 31.4 | 30.6 | 0.8 | 5.8 | 34.8 | 38.0 | 2.6 | 11.9 | 33.7 |
| AC | S | 5.3 | **0.0** | 0.2 | 3.8 | 0.1 | **0.1** | 12.7 | 10.7 | 7.1 | 9.8 | 1.3 | 10.6 | 33.2 | 36.8 | 6.3 | 10.2 | 38.2 | 38.4 | 6.9 | 12.2 | 36.7 |
| M | C | 1.9 | **0.0** | 0.2 | 2.0 | 0.0 | 2.4 | 20.7 | 13.0 | 18.1 | 24.8 | 6.0 | 9.0 | 36.5 | 34.4 | 3.7 | 8.8 | 40.9 | 37.1 | 4.5 | 13.9 | 37.2 |
| MAE D30 | S | 5.0 | **0.0** | 1.3 | 2.9 | 1.4 | 0.6 | 0.2 | 4.5 | 7.1 | 4.6 | 3.9 | - | 33.7 | 36.9 | 14.4 | 19.1 | 40.5 | 39.3 | 0.2 | 12.0 | 37.6 |
| VC | E | 9.4 | **0.0** | 0.3 | 2.8 | 0.0 | 0.3 | 11.5 | 6.1 | 8.8 | 19.9 | 11.7 | 6.6 | 35.6 | 37.2 | 3.3 | 5.4 | 39.5 | 39.6 | 7.9 | 12.9 | 38.0 |
| W2V | C | 13.0 | **0.0** | 4.0 | 7.6 | 3.5 | 2.9 | 8.4 | 13.8 | 11.0 | 10.1 | 0.9 | 6.8 | 38.4 | 33.6 | 1.8 | 9.8 | 40.7 | 40.3 | 7.9 | 13.4 | 38.3 |
| W2V D100 | C | 12.6 | **0.0** | 4.0 | 7.3 | 3.5 | 3.6 | 8.4 | 13.2 | 10.4 | 9.5 | 0.9 | 6.8 | 38.4 | 33.6 | 0.8 | 9.8 | 40.7 | 40.3 | 7.9 | 13.2 | 38.3 |
| WLM D30 | E | 20.6 | **0.0** | 2.5 | 5.4 | 1.7 | 1.5 | 6.5 | 8.7 | 19.2 | 24.6 | 1.9 | 20.3 | 37.8 | 37.9 | 0.1 | 3.9 | 39.6 | 38.2 | 23.1 | 15.5 | 38.4 |
| M | S | 37.8 | **0.0** | 0.5 | 3.4 | 0.1 | 4.5 | 16.1 | 29.5 | 21.4 | 22.8 | 9.7 | 21.2 | 38.8 | 35.5 | 6.5 | 14.3 | 42.1 | 39.7 | 31.9 | 19.8 | 39.0 |
| W2V D30 | C | 13.0 | **0.0** | 4.0 | 7.6 | 4.3 | 3.9 | 8.4 | 13.8 | 11.0 | 10.1 | 4.4 | 6.8 | 39.0 | 34.2 | 1.8 | 11.8 | 41.9 | 41.6 | 5.5 | 13.8 | 39.1 |
| WLM | E | 20.6 | **0.0** | 2.5 | 5.4 | 1.6 | 3.6 | 5.8 | 8.7 | 19.2 | 24.6 | 0.4 | 20.3 | 39.3 | 42.0 | 0.1 | 4.4 | 40.7 | 40.8 | 32.1 | 16.4 | 40.7 |
| WLM D100 | E | 32.7 | **0.0** | 2.5 | 6.2 | 1.6 | 1.1 | 5.8 | 12.7 | 15.9 | 29.2 | 0.4 | 20.3 | 39.3 | 42.0 | 0.0 | 4.4 | 40.7 | 40.8 | 32.1 | 17.3 | 40.7 |
| AC | C | 11.5 | **0.0** | 0.6 | 6.4 | 0.1 | 6.3 | 23.1 | 14.2 | 22.0 | 25.8 | 10.2 | 20.0 | 40.5 | 43.0 | 13.6 | 16.8 | 43.4 | 42.3 | 11.4 | 18.5 | 42.3 |
| CC | C | 10.8 | **0.0** | 1.7 | 11.4 | 3.2 | 9.0 | - | 22.2 | 22.5 | 26.6 | 11.3 | 20.6 | 42.4 | 38.2 | 16.1 | 27.7 | 46.9 | 42.0 | 13.3 | 20.3 | 42.4 |
| AC | E | 10.2 | **0.0** | 0.3 | 6.0 | 0.1 | 3.0 | 11.3 | 13.5 | 6.6 | 19.0 | 7.0 | 15.1 | 40.6 | 44.4 | 12.8 | 16.3 | 45.3 | 43.5 | 14.3 | 16.3 | 43.4 |
| M | E | 33.7 | **0.0** | 0.2 | 5.2 | 0.0 | 3.8 | 11.9 | 18.2 | 12.1 | 28.8 | 4.6 | 27.1 | 42.3 | 44.4 | 6.2 | 23.2 | 44.9 | 43.5 | 20.7 | 19.5 | 43.8 |
| W2V D30 | E | 25.0 | **0.0** | 4.4 | 10.7 | 4.5 | 6.7 | 19.7 | 32.8 | 16.5 | 27.1 | 3.8 | 29.1 | 44.5 | 40.3 | 3.7 | 25.0 | 47.6 | 46.4 | 23.3 | 21.6 | 44.7 |
| W2V D100 | E | 24.7 | **0.0** | 4.0 | 11.1 | 4.1 | 8.6 | 21.1 | 38.6 | 19.2 | 28.9 | 1.3 | 29.1 | 45.8 | 43.6 | 4.3 | 29.5 | 48.5 | 47.5 | 38.8 | 23.6 | 46.3 |
| W2V | E | 25.0 | **0.0** | 4.4 | 10.7 | 4.1 | 14.6 | 21.1 | 32.8 | 16.5 | 27.1 | 1.3 | 29.1 | 45.8 | 43.6 | 3.7 | 29.5 | 48.5 | 47.5 | 38.8 | 23.4 | 46.3 |
| CC | S | 22.2 | 3.3 | 3.7 | 15.5 | 5.1 | 11.8 | - | 28.3 | 33.5 | 30.2 | 17.8 | 25.5 | 46.8 | 42.8 | 22.0 | 30.5 | 50.4 | 48.0 | 20.5 | 25.4 | 47.0 |
| CC | E | 34.1 | **0.0** | 2.3 | 13.3 | 3.7 | 13.3 | - | 38.6 | 33.5 | 35.9 | 16.6 | 31.1 | 47.3 | 43.3 | 22.1 | 35.2 | 50.5 | 50.6 | 36.4 | 28.2 | 47.9 |
| MAE | C | 15.6 | **0.0** | 2.6 | 5.2 | 1.5 | 3.2 | 0.9 | 10.6 | 14.3 | 9.7 | 5.6 | 31.3 | 42.9 | 53.0 | 21.9 | 23.7 | 48.1 | 50.4 | 2.4 | 18.0 | 48.6 |
| MAE D100 | C | 14.0 | **0.0** | 1.8 | 4.7 | 1.5 | 0.7 | 0.9 | 10.3 | 11.0 | 7.4 | 5.6 | 31.3 | 42.9 | 53.0 | 18.9 | 23.7 | 48.1 | 50.4 | 2.4 | 17.3 | 48.6 |
| MAE D30 | C | 14.0 | **0.0** | 2.6 | 5.2 | 2.3 | 1.5 | 1.4 | 10.6 | 14.3 | 9.7 | 10.3 | 31.3 | 44.4 | 53.6 | 21.9 | 25.4 | 49.2 | 50.6 | 2.9 | 18.6 | 49.4 |
| MAE | E | 13.6 | **0.0** | 1.5 | 6.1 | 1.2 | 3.2 | 5.0 | 22.1 | 11.5 | 19.1 | 11.3 | 26.9 | 48.6 | 51.8 | 27.6 | 30.9 | 51.1 | 51.6 | 20.2 | 21.2 | 50.8 |
| MAE D100 | E | 15.1 | **0.0** | 1.5 | 6.6 | 1.2 | 1.5 | 5.0 | 27.0 | 12.1 | 19.3 | 11.3 | 26.9 | 48.6 | 51.8 | 27.0 | 30.9 | 51.1 | 51.6 | 20.2 | 21.5 | 50.8 |
| MAE D30 | E | 13.6 | **0.0** | 2.1 | 6.1 | 1.5 | 1.7 | 3.3 | 22.1 | 11.5 | 19.1 | 14.6 | 26.9 | 48.5 | 52.2 | 27.6 | 30.4 | 51.0 | 51.7 | 10.0 | 20.7 | 50.9 |

Table 24: Weighted Purity Results Across Subsets and Distances (↑better)

| Method | Dist | BC | BS1 | BS2 | BS3 | BS4 | BS5 | ES1 | HP | HS1 | HS2 | HU1 | HU2 | HU3 | HU4 | HW1 | HW2 | HW3 | HW4 | OC1 | Avg | Avg (Blind) |
|---|---|---|---|---|---|---|---|---|---|---|---|---|---|---|---|---|---|---|---|---|---|---|
| EWMTF D100 | S | 59.5 | 97.7 | 87.5 | 82.1 | 81.4 | 81.4 | 54.9 | 16.2 | 66.8 | 2.2 | 33.6 | 8.7 | 11.9 | **15.5** | 16.9 | 6.3 | 8.2 | **10.1** | 35.1 | 40.8 | **11.4** |
| EWMTF | S | 51.2 | 97.7 | 72.9 | 73.2 | 81.4 | 81.4 | 54.9 | 11.7 | 66.3 | 11.7 | 33.6 | 8.7 | 11.9 | **15.5** | 1.0 | 6.3 | 8.2 | **10.1** | 35.1 | 38.6 | **11.4** |
| VC | S | 59.2 | 96.6 | 83.8 | 82.0 | 82.6 | 81.4 | 24.8 | 17.8 | 59.4 | 12.7 | 9.9 | 11.3 | 12.3 | 14.5 | 14.0 | 8.1 | **8.9** | 9.1 | 40.4 | 38.4 | 11.2 |
| EWT | S | 56.2 | 98.9 | 90.3 | 83.1 | **89.2** | 86.7 | 52.5 | 11.5 | 64.7 | 13.1 | 2.7 | 9.4 | **13.0** | 14.7 | 6.9 | 6.0 | 7.9 | 9.0 | 32.4 | 39.4 | 11.1 |
| EWT D100 | S | 62.0 | 98.9 | 93.8 | 89.6 | **89.2** | 86.5 | 52.5 | 12.3 | 62.9 | 14.4 | 2.7 | 9.4 | **13.0** | 14.7 | 7.4 | 6.0 | 7.9 | 9.0 | 32.4 | 40.2 | 11.1 |
| EWTF | S | 57.8 | 98.9 | 90.3 | 80.9 | 87.6 | 86.7 | 50.8 | 1.9 | 45.1 | 12.9 | 2.9 | 9.3 | 12.5 | 14.6 | 6.9 | 6.7 | 7.7 | 9.2 | 36.7 | 37.9 | 11.0 |
| EWTF D100 | S | 63.7 | 98.9 | 90.3 | 86.1 | 87.6 | 86.7 | 50.8 | 2.0 | 64.9 | 14.6 | 2.9 | 9.3 | 12.5 | 14.6 | 7.6 | 6.7 | 7.7 | 9.2 | 36.7 | 39.6 | 11.0 |
| ETF | S | 56.2 | 98.7 | 86.1 | 74.2 | 81.2 | 89.0 | 52.9 | 12.6 | 60.8 | 11.6 | 4.0 | 0.8 | 12.5 | 15.2 | 7.9 | 0.1 | 7.5 | 8.1 | 9.1 | 36.2 | 10.8 |
| ETF D100 | S | 64.5 | 98.7 | 86.1 | 88.3 | 81.2 | 89.0 | 52.9 | 1.8 | 63.6 | 2.2 | 4.0 | 0.8 | 12.5 | 15.2 | 9.0 | 0.1 | 7.5 | 8.1 | 9.1 | 36.6 | 10.8 |
| ET | E | 54.4 | 99.2 | 83.6 | 84.3 | 80.3 | 86.9 | 44.5 | 12.4 | 64.6 | 2.5 | 4.1 | 0.8 | 12.2 | 14.3 | 1.0 | 0.1 | 8.1 | 8.6 | 39.2 | 36.9 | 10.8 |
| ET D100 | E | 57.9 | 99.2 | 83.6 | 80.2 | 80.3 | 86.1 | 44.5 | 1.8 | 63.6 | 13.7 | 4.1 | 0.8 | 12.2 | 14.3 | 8.8 | 0.1 | 8.1 | 8.6 | 39.2 | 37.2 | 10.8 |
| ETF | E | 53.4 | 99.2 | 84.0 | 82.9 | 79.7 | 86.9 | 47.6 | 13.2 | 66.2 | 2.6 | 4.2 | 9.6 | 12.3 | 14.5 | 1.0 | 6.1 | 7.5 | 8.6 | 40.6 | 37.9 | 10.7 |
| ETF D100 | E | 58.5 | 99.2 | 70.9 | 88.0 | 79.7 | 86.9 | 47.6 | 1.9 | 65.9 | 13.4 | 4.2 | 9.6 | 12.3 | 14.5 | 8.9 | 6.1 | 7.5 | 8.6 | 40.6 | 38.1 | 10.7 |
| CLP | S | 58.3 | 82.9 | 88.1 | 82.5 | 88.8 | **90.1** | 90.8 | 13.4 | 68.9 | 14.6 | 5.1 | 9.3 | 12.3 | 13.8 | 0.9 | 6.8 | 7.8 | 8.8 | 42.9 | 41.4 | 10.7 |
| CLP D100 | S | 65.5 | 82.9 | 88.1 | 90.5 | 88.8 | 86.9 | 90.8 | 13.9 | 68.3 | 16.9 | 5.1 | 9.3 | 12.3 | 13.8 | 1.2 | 6.8 | 7.8 | 8.8 | 42.9 | 42.1 | 10.7 |
| ETF | C | 55.6 | 99.2 | 70.9 | 75.7 | 78.8 | 87.7 | 47.5 | 12.7 | 66.2 | 2.4 | 4.4 | 0.8 | 11.6 | 14.2 | 8.5 | 0.1 | 7.8 | 8.7 | 35.4 | 36.2 | 10.6 |
| ETF D100 | C | 60.9 | 99.2 | 70.9 | 88.1 | 78.8 | 87.7 | 47.5 | 1.9 | 64.5 | 2.4 | 4.4 | 0.8 | 11.6 | 14.2 | 8.9 | 0.1 | 7.8 | 8.7 | 35.4 | 36.5 | 10.6 |
| EWTF D30 | E | 58.4 | 99.2 | 89.6 | 74.5 | 83.0 | 86.7 | 45.4 | 11.1 | 45.7 | 13.3 | 4.0 | 8.5 | 11.6 | 14.2 | 7.4 | 0.2 | 7.4 | 8.9 | 41.3 | 37.4 | 10.5 |
| EWT | C | 58.5 | 98.9 | 90.6 | 82.8 | 87.2 | 89.9 | 47.0 | 10.1 | 65.3 | 13.2 | 3.2 | 0.8 | 12.4 | 13.5 | 7.2 | 5.0 | 7.4 | 8.7 | 9.1 | 37.4 | 10.5 |
| EWT D100 | C | 47.6 | 98.9 | 91.1 | 87.8 | 87.7 | 47.0 | 11.0 | 66.1 | 14.4 | 3.2 | 0.8 | 12.4 | 13.5 | 7.7 | 5.0 | 7.4 | 8.7 | 9.1 | 37.2 | 10.5 |
| CLP D30 | C | 60.5 | 99.6 | 86.3 | 87.0 | 81.3 | 88.5 | 92.3 | 2.0 | **70.8** | 16.0 | 3.3 | 0.8 | 11.2 | 13.2 | 10.5 | 6.6 | 8.2 | 9.1 | 14.5 | 40.1 | 10.4 |
| EWT D30 | S | 56.2 | 99.2 | 90.3 | 83.1 | 79.8 | 85.4 | 42.9 | 11.5 | 64.7 | 13.1 | 3.6 | 9.4 | 10.9 | 14.4 | 6.9 | 5.9 | 7.7 | 8.7 | 34.5 | 38.3 | 10.4 |
| CLP | C | 60.5 | 99.6 | 86.3 | 87.0 | 85.4 | 87.7 | **92.7** | 2.0 | **70.8** | 16.0 | 6.5 | 0.8 | 11.2 | 14.0 | 10.5 | 6.3 | 7.4 | 8.8 | 42.0 | 41.9 | 10.3 |
| CLP D100 | C | 64.7 | 99.6 | 87.4 | 87.5 | 85.4 | 88.3 | **92.7** | 14.9 | 70.1 | 16.7 | 6.5 | 0.8 | 11.2 | 14.0 | 1.2 | 6.3 | 7.4 | 8.8 | 42.0 | **42.4** | 10.3 |
| MAE | S | 41.4 | 60.9 | 66.0 | 60.8 | 56.5 | 48.3 | 41.1 | 11.4 | 55.6 | 8.5 | 6.3 | **33.8** | 8.6 | 12.0 | 1.0 | 1.0 | 1.0 | 1.0 | 36.5 | 10.3 |
| ETF D30 | S | 56.2 | 98.5 | 80.3 | 74.2 | 74.2 | 84.0 | 43.6 | 12.6 | 60.8 | 11.6 | 5.9 | 0.8 | 10.7 | 14.5 | 7.9 | 5.7 | 7.0 | 8.7 | 40.8 | 36.7 | 10.2 |
| EW | E | 50.1 | 96.4 | 78.6 | 72.5 | 76.6 | 78.6 | 44.6 | 2.5 | 62.0 | 10.5 | 13.7 | 9.3 | 10.9 | 13.6 | 0.9 | 0.4 | 7.4 | 9.0 | 32.9 | 35.3 | 10.2 |
| EWMTF | E | 50.1 | 97.0 | 78.6 | 72.5 | 76.6 | 76.5 | 44.6 | 2.5 | 62.0 | 10.5 | 13.7 | 9.3 | 10.9 | 13.6 | 0.9 | 0.4 | 7.4 | 9.0 | 32.9 | 35.2 | 10.2 |
| EWMTF D100 | E | 49.1 | 97.0 | 78.6 | 75.7 | 76.6 | 76.5 | 44.6 | 2.3 | 62.0 | 11.4 | 13.7 | 9.3 | 10.9 | 13.6 | 0.9 | 0.4 | 7.4 | 9.0 | 32.9 | 35.4 | 10.2 |
| ETF D30 | C | 55.6 | 99.2 | 69.3 | 75.7 | 75.5 | 85.4 | 44.0 | 12.7 | 66.2 | 2.4 | 5.5 | 0.8 | 11.3 | 13.5 | 8.5 | 0.1 | 7.3 | 8.7 | 41.7 | 36.0 | 10.2 |
| CLP D30 | S | 58.3 | 82.9 | 85.3 | 82.5 | 79.3 | 88.0 | 88.5 | 13.4 | 68.9 | 14.6 | 3.0 | 9.3 | 11.3 | 14.0 | 0.9 | 6.3 | 7.2 | 8.3 | 42.4 | 40.2 | 10.2 |
| EWTF | C | 56.6 | 97.5 | 90.6 | 84.3 | 86.3 | 89.9 | 48.7 | 10.9 | 64.9 | 13.8 | 2.9 | 9.1 | 11.6 | 12.3 | 7.3 | 5.3 | 7.7 | 8.9 | 9.1 | 37.8 | 10.1 |
| EWTF D100 | C | 60.1 | 98.9 | 93.4 | 87.0 | 86.3 | 89.9 | 48.7 | 1.9 | 64.4 | 14.1 | 2.9 | 9.1 | 11.6 | 12.3 | 7.6 | 5.3 | 7.7 | 8.9 | 9.1 | 37.8 | 10.1 |
| ETF D30 | E | 53.4 | 99.2 | 79.5 | 82.9 | 76.8 | 88.3 | 41.5 | 13.2 | 66.2 | 2.6 | 5.4 | 9.6 | 10.8 | 13.5 | 1.0 | 5.8 | 7.3 | 8.8 | 39.9 | 37.1 | 10.1 |
| ET D30 | S | 54.1 | 98.7 | 80.2 | 80.9 | 74.4 | 85.9 | 41.2 | 12.6 | 44.9 | 12.0 | 5.4 | 9.5 | 11.1 | 13.4 | 7.9 | 5.4 | 7.5 | 8.2 | 42.0 | 36.6 | 10.0 |
| ET D30 | E | 54.4 | 99.2 | 69.1 | 84.3 | 73.8 | 87.2 | 40.8 | 12.4 | 64.6 | 2.5 | 5.8 | 0.8 | 11.6 | 13.7 | 1.0 | 0.1 | 6.7 | 8.1 | 41.3 | 35.6 | 10.0 |
| EWTF D30 | C | 56.6 | 98.9 | 89.0 | 84.3 | 82.6 | 85.7 | 43.9 | 10.9 | 64.9 | 13.8 | 4.4 | 8.4 | 10.7 | 13.3 | 7.3 | 4.7 | 7.2 | 8.5 | 32.2 | 38.3 | 9.9 |
| EMTF D30 | S | 26.3 | 97.7 | 60.5 | 68.6 | 66.9 | 73.8 | 36.8 | 12.8 | 61.6 | 6.7 | 0.7 | 0.8 | 10.2 | 14.5 | 8.2 | 5.5 | 6.2 | 8.1 | 36.5 | 31.7 | 9.9 |
| EWMTF D30 | S | 51.2 | 93.9 | 73.9 | 73.2 | 74.0 | 76.0 | 46.6 | 11.7 | 66.3 | 11.7 | 5.0 | 8.7 | 10.9 | 13.5 | 1.0 | 3.2 | 6.5 | 8.5 | 36.5 | 35.4 | 9.8 |
| E | C | 44.1 | 82.2 | 45.0 | 66.3 | 74.5 | 83.4 | 36.1 | 12.0 | 59.6 | 9.9 | 11.9 | 0.8 | 10.9 | 13.3 | 8.4 | 6.0 | 6.9 | 8.0 | 32.0 | 32.2 | 9.8 |
| EMTF | C | 44.1 | 82.2 | 67.6 | 66.3 | 74.5 | 84.1 | 36.1 | 12.0 | 59.6 | 9.9 | 11.9 | 0.8 | 10.9 | 13.3 | 8.4 | 6.0 | 6.9 | 8.0 | 32.0 | 33.4 | 9.8 |
| EMTF D100 | C | 44.7 | 97.5 | 67.6 | 72.1 | 74.5 | 84.1 | 36.1 | 13.3 | 60.5 | 10.0 | 11.9 | 0.8 | 10.9 | 13.3 | 9.4 | 6.0 | 6.9 | 8.0 | 32.0 | 34.7 | 9.8 |
| E | E | 43.1 | 98.1 | 83.6 | 67.0 | 71.0 | 82.9 | 4.7 | 1.9 | 60.2 | 9.6 | 1.0 | 1.0 | 10.7 | 11.9 | 1.2 | 0.2 | 7.3 | 8.6 | 31.3 | 31.3 | 9.6 |
| EMTF | E | 43.1 | 98.1 | 80.9 | 67.0 | 71.0 | 83.6 | 4.7 | 1.9 | 60.2 | 9.6 | 1.0 | 1.0 | 10.7 | 11.9 | 1.2 | 0.2 | 7.3 | 8.6 | 31.3 | 31.2 | 9.6 |
| EMTF D100 | E | 44.8 | 98.1 | 79.8 | 72.3 | 71.0 | 83.6 | 4.7 | 1.9 | 58.3 | 1.8 | 1.0 | 1.0 | 10.7 | 11.9 | 1.2 | 0.2 | 7.3 | 8.6 | 31.3 | 31.0 | 9.6 |
| EMTF D30 | C | 44.1 | 82.2 | 74.4 | 66.3 | 71.4 | 75.6 | 33.4 | 12.0 | 59.6 | 9.9 | 8.7 | 0.8 | 10.7 | 12.5 | 8.4 | 5.3 | 6.7 | 8.2 | 31.5 | 32.7 | 9.5 |
| EWF D30 | C | 46.6 | 94.9 | 70.6 | 70.4 | 64.3 | 66.3 | 3.5 | 10.2 | 56.9 | 8.4 | 7.8 | 7.1 | 8.9 | 13.7 | 6.0 | 4.2 | 6.5 | 8.6 | 37.0 | 31.2 | 9.4 |
| EW | C | 49.2 | 92.6 | 90.6 | 69.8 | 75.1 | 74.9 | 45.8 | 2.1 | 61.8 | 11.0 | 12.4 | 0.9 | 9.3 | 12.1 | 1.0 | 3.2 | 7.3 | 8.9 | 31.1 | 34.7 | 9.4 |
| EWMTF | C | 49.2 | 96.0 | 78.3 | 69.8 | 75.1 | 76.8 | 45.8 | 2.1 | 61.8 | 11.0 | 12.4 | 0.9 | 9.3 | 12.1 | 1.0 | 3.2 | 7.3 | 8.9 | 31.1 | 34.4 | 9.4 |
| EWMTF D100 | C | 51.4 | 93.9 | 78.0 | 74.7 | 75.1 | 76.8 | 45.8 | 2.1 | 61.0 | 12.0 | 12.4 | 0.9 | 9.3 | 12.1 | 1.0 | 3.2 | 7.3 | 8.9 | 31.1 | 34.6 | 9.4 |
| EMTF D30 | E | 43.1 | 97.9 | 75.2 | 67.0 | 71.1 | 75.9 | 35.0 | 1.9 | 60.2 | 9.6 | 9.3 | 1.0 | 10.7 | 11.3 | 1.2 | 0.2 | 7.3 | 8.1 | 32.7 | 32.6 | 9.4 |
| W2V D30 | E | 39.1 | 75.1 | 48.4 | 27.0 | 22.9 | 50.9 | 17.8 | 5.2 | 55.0 | 1.3 | 10.6 | 7.5 | 8.8 | 13.7 | 1.1 | 6.3 | 6.6 | 8.3 | 34.5 | 23.2 | 9.3 |
| EWMTF D30 | C | 49.2 | 96.0 | 74.8 | 69.8 | 72.9 | 75.6 | 39.5 | 2.1 | 61.8 | 11.0 | 0.6 | 0.9 | 10.4 | 11.7 | 1.0 | 3.2 | 8.3 | 8.3 | 32.0 | 33.0 | 9.3 |
| WLM D30 | E | 44.2 | 96.6 | 70.2 | 65.6 | 72.4 | 77.4 | 22.5 | 18.7 | 59.0 | 10.3 | 10.9 | 14.5 | 8.3 | 13.9 | 44.6 | 10.1 | 6.7 | 8.2 | 35.4 | 36.3 | 9.3 |
| WLM D30 | S | 46.5 | 96.8 | 75.6 | 65.8 | 71.0 | 79.0 | 21.9 | 18.6 | 60.9 | 9.8 | 10.4 | 27.0 | 8.7 | 13.3 | 44.5 | 9.5 | 6.3 | 8.6 | 32.4 | 37.2 | 9.2 |
| EWF | C | 46.6 | 96.0 | 70.6 | 70.4 | 69.3 | 68.4 | 3.6 | 10.2 | 56.9 | 8.4 | 6.6 | 6.0 | 8.5 | 13.5 | 6.0 | 4.1 | 6.3 | 8.4 | 36.1 | 31.4 | 9.2 |
| EWF D100 | C | 45.9 | 94.7 | 74.1 | 70.2 | 69.3 | 68.4 | 3.6 | 10.0 | 55.2 | 1.1 | 6.6 | 6.0 | 8.5 | 13.5 | 6.2 | 4.1 | 6.3 | 8.4 | 36.1 | 31.0 | 9.2 |
| EWF D30 | E | 44.9 | 96.2 | 72.8 | 67.3 | 63.6 | 65.3 | 3.6 | 9.1 | 55.7 | 8.6 | 5.1 | 7.1 | 9.2 | 13.2 | 6.0 | 3.5 | 6.3 | 7.9 | 32.9 | 30.4 | 9.1 |
| MAE D30 | S | 41.4 | 89.2 | 50.2 | 60.8 | 45.1 | 44.3 | 36.5 | 11.4 | 55.6 | 8.5 | 1.7 | 1.0 | 9.1 | 13.2 | 6.2 | 5.1 | 6.1 | 8.0 | 28.3 | 28.9 | 9.1 |
| WLM | C | 45.6 | 99.2 | 69.8 | 65.4 | 74.8 | 82.2 | 3.8 | **18.9** | 59.0 | 10.2 | 12.8 | 15.2 | 8.6 | 13.1 | 45.8 | 0.2 | 6.5 | 8.2 | 38.5 | 35.7 | 9.1 |
| WLM D100 | C | 46.7 | 99.2 | 76.9 | 71.1 | 74.8 | 80.6 | 3.8 | 16.9 | 57.8 | 9.7 | 12.8 | 15.2 | 8.6 | 13.1 | 51.6 | 0.2 | 6.5 | 8.2 | 38.5 | 36.4 | 9.1 |
| W2V D30 | S | 39.0 | 75.3 | 49.5 | 27.3 | 23.9 | 53.1 | 17.4 | 5.4 | 49.5 | 1.4 | 6.8 | 8.9 | 8.8 | 15.4 | 6.3 | 6.5 | 7.9 | 32.0 | 23.1 | 9.0 |
| WLM D30 | C | 45.6 | 99.2 | 69.8 | 65.4 | 71.7 | 77.9 | 3.8 | **18.9** | 59.0 | 10.2 | 11.3 | 15.2 | 9.3 | 11.5 | 45.8 | 10.2 | 6.9 | 8.3 | 39.9 | 35.8 | 9.0 |
| EWF | E | 44.9 | 96.8 | 72.8 | 67.3 | 67.0 | 66.2 | 3.7 | 9.1 | 55.7 | 8.6 | 8.0 | 6.4 | 8.9 | 12.5 | 6.0 | 3.7 | 6.3 | 8.3 | 33.3 | 30.8 | 9.0 |
| EWF D100 | E | 45.9 | 96.0 | 74.1 | 71.0 | 67.0 | 66.2 | 3.7 | 9.5 | 55.6 | 8.3 | 8.0 | 6.4 | 8.9 | 12.5 | 5.5 | 3.7 | 6.3 | 8.3 | 33.3 | 31.1 | 9.0 |
| W2V D30 | C | 40.1 | 75.7 | 49.3 | 27.6 | 23.9 | 52.3 | 17.2 | 4.5 | 54.9 | 8.6 | 5.0 | 7.8 | 9.3 | 12.2 | 16.1 | 6.5 | 7.9 | 31.1 | 24.0 | 9.0 |
| M | | 67.9 | 99.8 | **96.3** | 93.2 | 86.6 | 84.7 | 19.8 | 6.7 | 59.6 | 8.4 | 8.5 | 5.5 | 8.6 | 12.8 | 2.4 | 0.2 | 6.5 | 7.9 | 38.5 | 37.6 | 8.9 |
| WLM | E | 44.2 | 96.6 | 70.2 | 65.6 | 76.2 | 81.3 | 3.8 | 18.7 | 59.0 | 10.3 | 0.5 | 14.5 | 8.5 | 12.7 | 44.6 | 0.2 | 6.5 | 7.9 | 36.3 | 34.6 | 8.9 |
| WLM D100 | E | 45.8 | 99.2 | 77.4 | 72.4 | 76.2 | 80.8 | 3.8 | 15.7 | 43.8 | 10.1 | 0.5 | 14.5 | 8.5 | 12.7 | 50.6 | 0.2 | 6.5 | 7.9 | 36.3 | 34.9 | 8.9 |
| ET | S | 54.1 | 98.7 | 80.2 | 88.7 | 81.6 | 89.0 | 50.0 | 12.6 | 44.9 | 12.0 | 3.7 | 9.5 | 12.4 | 13.7 | 7.9 | 0.1 | 8.8 | 9.1 | 35.7 | 8.9 |
| ET D100 | S | 62.1 | 96.8 | 86.1 | 88.7 | 81.6 | 88.2 | 50.0 | 1.9 | 46.5 | 2.0 | 3.7 | 9.5 | 12.4 | 13.7 | 9.2 | 0.1 | 0.6 | 8.8 | 9.1 | 35.3 | 8.9 |
| MAE D30 | C | 34.1 | 89.9 | 52.9 | 58.0 | 45.7 | 41.0 | 36.0 | 1.7 | 56.3 | 8.7 | 2.3 | 8.0 | 8.5 | 13.2 | 6.9 | 5.3 | 6.2 | 7.5 | 32.0 | 27.1 | 8.8 |
| AC | C | 57.2 | 98.1 | 91.4 | 87.8 | 80.8 | 83.3 | 18.8 | 1.7 | 57.4 | 10.1 | 8.5 | 8.5 | 8.2 | 12.7 | 8.5 | 6.0 | 6.6 | 8.0 | 29.5 | 36.0 | 8.8 |
| CLP D30 | E | 61.9 | 99.6 | 87.9 | 86.9 | 82.9 | 89.7 | 92.5 | 1.8 | 46.0 | 15.9 | 3.4 | 0.8 | 12.1 | 14.0 | 0.9 | 6.5 | 7.5 | 1.6 | 14.5 | 38.2 | 8.8 |
| W2V | C | 40.1 | 75.7 | 49.3 | 27.6 | 23.2 | 53.4 | 17.2 | 4.5 | 54.9 | 8.6 | 10.0 | 7.8 | 8.7 | 12.1 | 16.1 | 6.4 | 6.5 | 7.8 | 31.3 | 24.3 | 8.8 |

*Continued on next page*

Table 24: (Continued) Weighted Purity Results Across Subsets and Distances (↑ better)

| Method | Dist | BC | BS1 | BS2 | BS3 | BS4 | BS5 | ES1 | HP | HS1 | HS2 | HU1 | HU2 | HU3 | HU4 | HW1 | HW2 | HW3 | HW4 | OC1 | Avg | Avg (Blind) |
|---|---|---|---|---|---|---|---|---|---|---|---|---|---|---|---|---|---|---|---|---|---|---|
| W2V D100 | C | 39.2 | 77.0 | 49.4 | 27.6 | 23.2 | 55.8 | 17.2 | 5.0 | 54.9 | 8.4 | 10.0 | 7.8 | 8.7 | 12.1 | 21.4 | 6.4 | 6.5 | 7.8 | 31.3 | 24.7 | 8.8 |
| EF D30 | S | 40.4 | 94.5 | 77.4 | 69.0 | 68.0 | 74.9 | 5.5 | 5.3 | 55.4 | 1.4 | 3.0 | 8.7 | 9.3 | 11.8 | 4.2 | 4.1 | 6.2 | 7.6 | 37.6 | 30.8 | 8.7 |
| EWF D30 | S | 45.2 | 94.3 | 70.9 | 68.0 | 64.7 | 68.8 | 6.2 | 10.5 | 57.1 | 1.6 | 4.6 | 6.9 | 9.2 | 11.3 | 6.4 | 4.5 | 6.3 | 8.1 | 39.0 | 30.7 | 8.7 |
| MAE D30 | E | 40.9 | 88.4 | 56.0 | 57.5 | 43.0 | 40.8 | 31.9 | 11.1 | 55.9 | 8.6 | 1.9 | 7.7 | 8.7 | 12.1 | 7.0 | 5.3 | 6.4 | 7.7 | 27.9 | 27.3 | 8.7 |
| EF D30 | E | 44.1 | 91.3 | 75.6 | 69.0 | 67.6 | 74.8 | 3.7 | 3.7 | 55.0 | 8.7 | 2.3 | 7.4 | 9.7 | 11.2 | 3.4 | 2.2 | 5.2 | 8.7 | 33.6 | 30.4 | 8.7 |
| MAE | C | 34.1 | 88.6 | 52.9 | 58.0 | 49.7 | 52.8 | 36.0 | 1.7 | 56.3 | 8.7 | 1.7 | 8.0 | 8.9 | 11.5 | 6.9 | 5.2 | 6.2 | 8.1 | 26.8 | 27.5 | 8.7 |
| MAE D100 | C | 40.1 | 88.6 | 62.6 | 63.3 | 49.7 | 51.4 | 36.0 | 11.6 | 56.4 | 8.8 | 1.7 | 8.0 | 8.9 | 11.5 | 7.1 | 5.2 | 6.2 | 8.1 | 26.8 | 29.1 | 8.7 |
| W2V | E | 39.1 | 75.1 | 48.4 | 27.0 | 22.8 | 53.1 | 16.3 | 5.2 | 55.0 | 1.3 | 11.5 | 7.5 | 8.0 | 11.4 | 1.1 | 6.4 | 6.3 | 8.5 | 26.1 | 22.6 | 8.6 |
| W2V D100 | E | 39.4 | 77.2 | 50.2 | 27.4 | 22.8 | 54.1 | 16.3 | 5.9 | 53.4 | 8.3 | 11.5 | 7.5 | 8.0 | 11.4 | 19.8 | 6.4 | 6.3 | 8.5 | 26.1 | 24.2 | 8.6 |
| EWT | E | 57.1 | 99.2 | 89.8 | 81.6 | 86.1 | 88.9 | 47.2 | 2.0 | 65.1 | 13.8 | 2.8 | 8.8 | 12.2 | 12.6 | 7.2 | 0.2 | 7.6 | 1.5 | 37.9 | 38.0 | 8.4 |
| EWT D100 | E | 60.0 | 99.4 | 90.3 | 88.2 | 86.1 | 88.9 | 47.2 | 11.7 | 45.7 | 14.4 | 2.8 | 8.8 | 12.2 | 12.6 | 8.3 | 0.2 | 7.6 | 1.5 | 37.9 | 38.1 | 8.4 |
| EWTF | E | 58.4 | 99.2 | 89.6 | 74.5 | 88.1 | 88.9 | 47.6 | 1.8 | 45.7 | 13.3 | 3.5 | 8.7 | 1.9 | 14.5 | 7.4 | 0.1 | 7.9 | 9.5 | 37.6 | 37.2 | 8.4 |
| EWTF D100 | E | 60.8 | 99.4 | 92.2 | 87.2 | 88.1 | 88.9 | 47.6 | 1.8 | 63.2 | 15.0 | 3.5 | 8.7 | 1.9 | 14.5 | 7.6 | 0.1 | 7.9 | 9.5 | 37.6 | 38.7 | 8.4 |
| EF | E | 44.1 | 96.0 | 78.0 | 69.0 | 69.8 | 75.3 | 3.7 | 3.7 | 55.0 | 8.7 | 0.7 | 7.4 | 8.6 | 10.4 | 3.4 | 3.9 | 6.3 | 8.4 | 12.2 | 29.7 | 8.4 |
| EF D100 | E | 39.4 | 95.1 | 73.1 | 73.7 | 69.8 | 75.3 | 3.7 | 6.1 | 45.4 | 8.5 | 0.7 | 7.4 | 8.6 | 10.4 | 5.9 | 3.9 | 6.3 | 8.4 | 12.2 | 29.2 | 8.4 |
| EF D30 | C | 42.2 | 93.2 | 77.9 | 70.4 | 69.3 | 74.4 | 3.8 | 6.9 | 56.0 | 8.6 | 1.4 | 6.0 | 6.6 | 12.3 | 4.1 | 2.2 |  | 8.4 | 34.9 | 30.8 | 8.4 |
| EF | S | 40.4 | 94.5 | 77.4 | 69.0 | 72.7 | 79.1 | 18.4 | 5.3 | 55.4 | 1.4 | 6.4 | 8.7 | 8.7 | 12.7 | 4.2 | 2.9 | 4.8 | 6.8 | 32.0 | 31.6 | 8.3 |
| EF D100 | S | 44.1 | 96.6 | 77.6 | 79.5 | 72.7 | 79.1 | 18.4 | 13.3 | 53.5 | 1.1 | 6.4 | 8.7 | 8.7 | 12.7 | 5.5 | 2.9 | 4.8 | 6.8 | 32.0 | 32.9 | 8.3 |
| MAE | E | 40.9 | 90.9 | 61.7 | 57.5 | 46.7 | 51.7 | 33.9 | 11.1 | 55.9 | 8.6 | 0.8 | 7.7 | 7.4 | 11.6 | 7.0 | 5.2 | 6.2 | 7.8 | 28.6 | 28.5 | 8.2 |
| MAE D100 | E | 23.0 | 88.2 | 61.7 | 62.8 | 46.7 | 50.0 | 33.9 | 11.4 | 45.9 | 1.4 | 0.8 | 7.7 | 7.4 | 11.6 | 7.1 | 5.2 | 6.2 | 7.8 | 28.6 | 26.7 | 8.2 |
| ET D30 | C | 55.5 | 99.2 | 80.0 | 82.4 | 77.0 | 88.8 | 42.6 | 1.8 | 64.3 | 2.1 | 5.4 | 0.9 | 10.5 | 13.6 | 8.0 | 5.9 | 0.6 | 8.3 | 39.5 | 36.1 | 8.2 |
| VC | C | 62.2 | 96.4 | 93.3 | 91.1 | 82.9 | 82.5 | 24.8 | 15.7 | 60.2 | 11.4 | 10.4 | 9.9 | 9.9 | 13.4 | 12.1 | 0.1 | 0.6 | 8.8 | 41.5 | 38.3 | 8.2 |
| EWMTF D30 | E | 50.1 | 96.6 | 74.0 | 72.5 | 73.0 | 74.2 | 43.8 | 2.5 | 62.5 | 10.5 | 0.5 | 9.3 | 10.0 | 13.4 | 0.9 | 2.9 | 0.8 | 8.4 | 32.4 | 33.6 | 8.1 |
| M | E | 66.9 | 99.8 | 96.1 | **93.3** | 87.5 | 84.6 | 20.2 | 11.7 | 57.3 | 1.1 | 2.5 | 0.8 | 6.0 | 12.3 | 9.3 | 6.3 | 6.2 | 7.9 | 36.3 | 37.2 | 8.1 |
| EWF | S | 45.2 | 91.8 | 76.1 | 68.0 | 71.2 | 69.8 | 2.1 | 10.5 | 57.1 | 1.6 | 5.2 | 1.1 | 8.3 | 12.8 | 6.4 | 2.1 | 4.3 | 6.7 | 30.6 | 30.0 | 8.0 |
| EWF D100 | S | 45.9 | 91.8 | 76.1 | 76.2 | 71.2 | 69.8 | 2.1 | 10.6 | 56.0 | 6.2 | 5.2 | 1.1 | 8.3 | 12.8 | 4.7 | 2.1 | 4.3 | 6.7 | 30.6 | 30.6 | 8.0 |
| EWT D30 | E | 57.1 | 97.7 | 89.8 | 81.6 | 82.9 | 85.4 | 42.2 | 2.0 | 65.1 | 13.8 | 4.2 | 8.8 | 10.9 | 5.0 | 7.2 | 0.3 | 7.5 | 8.8 | 32.4 | 37.0 | 8.0 |
| EMTF | S | 26.3 | 97.7 | 84.6 | 68.6 | 75.1 | 85.9 | 46.9 | 12.8 | 61.6 | 6.7 | **37.5** | 0.8 | 2.0 | 13.4 | 8.2 | 7.0 | 7.4 | 9.2 | 37.6 | 36.3 | 8.0 |
| EMTF D100 | S | 57.6 | 98.9 | 84.6 | 87.4 | 75.1 | 85.9 | 46.9 | 17.8 | 65.9 | 2.1 | **37.5** | 0.8 | 2.0 | 13.4 | 20.0 | 7.0 | 7.4 | 9.2 | 37.6 | 39.8 | 8.0 |
| EWT D30 | C | 58.5 | 98.9 | 89.8 | 82.8 | 82.1 | 82.5 | 43.8 | 10.1 | 65.3 | 13.2 | 3.3 | 0.8 | 11.3 | 4.5 | 7.2 | 0.1 | 6.6 | 9.3 | 32.4 | 37.0 | 7.9 |
| EWTF D30 | S | 57.8 | 98.9 | 88.9 | 80.9 | 82.3 | 82.4 | 45.2 | 1.9 | 45.1 | 12.9 | 3.8 | 8.6 | 9.9 | 4.9 | 6.9 | 0.1 | 6.9 | 9.3 | 39.7 | 36.1 | 7.7 |
| AC | S | 62.0 | **100.0** | 92.7 | 87.8 | 82.0 | 83.3 | 2.7 | 11.7 | 57.8 | 1.2 | 9.1 | 8.6 | 9.4 | 13.2 | 8.5 | 6.3 | 6.9 | 1.5 | 9.1 | 34.4 | 7.7 |
| EF | C | 42.2 | 93.2 | 73.7 | 70.4 | 71.5 | 76.5 | 14.8 | 6.9 | 56.0 | 8.6 | 4.3 | 6.0 | 7.3 | 9.0 | 4.1 | 3.9 | 5.8 | 8.3 | 34.7 | 31.4 | 7.6 |
| EF D100 | C | 39.2 | 95.1 | 73.7 | 74.3 | 71.5 | 76.5 | 14.8 | 3.8 | 55.0 | 1.1 | 4.3 | 6.0 | 7.3 | 9.0 | 6.1 | 3.9 | 5.8 | 8.3 | 34.7 | 31.1 | 7.6 |
| VC | E | 62.5 | **100.0** | 94.5 | 92.4 | 85.3 | 80.9 | 25.5 | 2.3 | 59.7 | 11.5 | 9.8 | 9.5 | 1.6 | 13.1 | 11.3 | 7.2 | 6.3 | 8.1 | 37.6 | 37.8 | 7.3 |
| WLM | S | 46.5 | 96.8 | 75.6 | 65.8 | 77.5 | 80.4 | 22.1 | 18.6 | 60.9 | 9.8 | 13.6 | 27.0 | 1.5 | 13.0 | 44.5 | **11.5** | 6.4 | 8.2 | 30.6 | 37.4 | 7.3 |
| WLM D100 | S | 47.5 | 98.7 | 79.4 | 71.2 | 77.5 | 80.4 | 22.1 | 16.2 | 61.3 | 10.4 | 13.6 | 0.9 | 1.5 | 13.0 | 53.5 | **11.5** | 6.4 | 8.2 | 30.6 | 37.0 | 7.3 |
| CLP | E | 61.9 | 99.6 | 88.3 | 86.9 | 84.9 | 88.9 | 92.5 | 1.8 | 46.0 | 15.9 | 6.5 | 0.8 | 12.6 | 13.5 | 0.9 | 6.8 | 0.9 | 1.7 | **43.1** | 39.7 | 7.2 |
| CLP D100 | E | 62.1 | 99.8 | 87.7 | 84.6 | 84.9 | 90.0 | 92.5 | 14.0 | 46.0 | **17.1** | 6.5 | 0.8 | 12.6 | 13.5 | 1.2 | 6.8 | 0.9 | 1.7 | **43.1** | 40.3 | 7.2 |
| M | S | 63.3 | **100.0** | 91.1 | 90.3 | 87.2 | 82.1 | 16.2 | 5.1 | 57.2 | 7.7 | 3.6 | 6.4 | 1.8 | 12.5 | 4.0 | 5.4 | 6.4 | 7.7 | 35.1 | 36.0 | 7.1 |
| MAE D100 | S | 42.2 | 60.9 | 66.0 | 76.4 | 56.5 | 57.4 | 41.1 | 11.5 | 56.2 | 9.0 | 6.3 | 1.0 | 8.6 | 12.0 | 1.0 | 0.1 | 6.2 | 1.5 | 26.1 | 29.9 | 7.1 |
| AC | E | 58.3 | 98.1 | 87.1 | 85.4 | 79.5 | 80.9 | 18.9 | 11.9 | 56.5 | 9.5 | 7.9 | 8.4 | 6.1 | 5.5 | 0.9 | 5.9 | 6.6 | 7.8 | 10.0 | 34.0 | 6.5 |
| ET | C | 55.5 | 99.2 | 84.4 | 82.4 | 79.9 | 87.7 | 48.9 | 1.8 | 64.3 | 2.1 | 4.1 | 0.9 | 2.4 | 13.9 | 8.0 | 0.1 | 7.7 | 1.4 | 37.2 | 35.9 | 6.4 |
| ET D100 | C | 61.2 | 99.2 | 84.4 | 85.2 | 79.9 | 87.8 | 48.9 | 1.9 | 63.5 | 13.4 | 4.1 | 0.9 | 2.4 | 13.9 | 8.7 | 0.1 | 7.7 | 1.4 | 37.2 | 36.9 | 6.4 |
| CC | E | 37.8 | 63.8 | 46.7 | 37.4 | 43.9 | 31.6 | 1.0 | 3.9 | 46.8 | 1.1 | 0.7 | 1.0 | 1.0 | 1.0 | 1.5 | 4.2 | 5.2 | 7.0 | 26.5 | 23.9 | 6.1 |
| W2V | S | 39.0 | 75.3 | 49.5 | 27.3 | 24.9 | 53.5 | 16.2 | 5.4 | 49.5 | 1.4 | 0.6 | 6.8 | 1.5 | 12.3 | 15.4 | 6.8 | 0.6 | 8.1 | 27.2 | 22.2 | 5.6 |
| W2V D100 | S | 39.1 | 61.7 | 47.4 | 27.8 | 24.9 | 58.2 | 16.2 | 5.9 | 46.5 | 7.6 | 0.6 | 6.8 | 1.5 | 12.3 | 22.3 | 6.8 | 0.6 | 8.1 | 27.2 | 22.2 | 5.6 |
| CC | S | 39.0 | 87.3 | 54.5 | 34.6 | 36.2 | 39.1 | 1.0 | 6.0 | 56.0 | 7.0 | 3.7 | 1.0 | 1.0 | 1.0 | 3.4 | 1.8 | 4.2 | 6.8 | 33.6 | 27.5 | 5.5 |
| CC | C | 38.1 | 70.4 | 56.3 | 36.2 | 40.8 | 8.1 | 1.0 | 7.4 | 47.4 | 5.7 | 6.7 | 1.0 | 1.0 | 1.0 | 5.8 | 4.0 | 0.6 | 6.3 | 18.6 | 23.5 | 3.5 |

Table 25: Avian Perception Alignment: Triplet Accuracy (High Consistency)

| Method | Triplet Acc. (%) |
|---|---|
| EW (C) | **80.9** |
| EW (E) | **80.9** |
| EW (S) | 80.6 |
| EWTF (S) | 74.8 |
| EWF (S) | 73.0 |
| EWTF (C) | 73.0 |
| EWTF (E) | 72.9 |
| EWF (C) | 72.8 |
| EWF (E) | 72.7 |
| EMB-LUA ∗ | 72.7 |
| ETF (D=100, PCA) (E) | 72.1 |
| ETF (E) | 71.7 |
| ETF (S) | 71.7 |
| ETF (C) | 71.6 |
| EF (C) | 71.6 |
| EF (E) | 71.4 |
| E (S) | 71.2 |
| EF (S) | 71.0 |
| E (E) | 70.1 |
| E (C) | 69.9 |
| Luscinia-U ∗ | 69.8 |
| CLAP (C) | 67.6 |
| CLAP (E) | 67.6 |
| CLAP (S) | 67.4 |
| CLP D100 (E) | 67.3 |
| ETF (D=100, PCA) (C) | 66.5 |
| CLP D30 (E) | 66.2 |
| ET (S) | 66.0 |
| Luscinia ∗ | 66.0 |
| ET (C) | 64.8 |
| ET (E) | 64.4 |
| SAP ∗ | 64.0 |
| CC (E) | 62.8 |
| CC (C) | 62.6 |
| CLP D30 (C) | 62.1 |
| CLP D100 (C) | 61.9 |
| CLP D30 (S) | 61.9 |
| M (E) | 61.4 |
| ETF (D=100, PCA) (S) | 60.4 |
| CLP D100 (S) | 60.2 |
| AC (S) | 60.0 |
| CC (S) | 59.2 |
| MAE D100 (E) | 59.0 |
| AC (C) | 58.8 |
| MAE D30 (E) | 58.4 |
| M (C) | 57.9 |
| MAE (S) | 57.7 |
| MAE (C) | 57.3 |
| VC (S) | 57.2 |
| WLM (S) | 57.1 |
| MAE (E) | 57.1 |
| Raven ∗ | 57.0 |

*Continued on next page*

Table 25: (Continued) Avian Perception Alignment: Triplet Accuracy (High Consistency)

| Method | Triplet Acc. (%) |
|---|---|
| VC (E) | 56.1 |
| MAE D100 (C) | 56.1 |
| MAE D30 (C) | 56.0 |
| WLM (C) | 55.8 |
| MAE D30 (S) | 55.3 |
| VC (C) | 55.3 |
| WLM (E) | 54.9 |
| W2V D100 (S) | 54.3 |
| W2V D30 (C) | 54.3 |
| W2V D100 (C) | 54.2 |
| MAE D100 (S) | 53.9 |
| WLM D30 (C) | 53.7 |
| WLM D100 (C) | 53.7 |
| WLM D30 (S) | 53.6 |
| WLM D100 (E) | 53.3 |
| W2V D30 (S) | 52.9 |
| W2V D100 (E) | 52.0 |
| WLM D100 (S) | 52.0 |
| W2V D30 (E) | 51.9 |
| WLM D30 (E) | 51.8 |
| AC (E) | 51.8 |

[*]Reference values from Zandberg et al. (2024) Zandberg et al. (2024).

Table 26: Predicting mouse strain. Classification accuracy (Top-1, %) and std over 5 splits. Only MLP at $\alpha = 0.001$ is shown.

| Method | k-NN | | | RF | | | MLP ($\alpha = 0.001$) |
|---|---|---|---|---|---|---|---|
| | k=3 | k=10 | k=30 | depth=10 | depth=15 | depth=20 | Top-1 (%) ± std |
| EWTF | 98.00 | 96.96 | 95.14 | 93.25 | 96.12 | 96.57 | 99.49 (0.12) |
| EWTF D30 | 97.19 | 96.30 | 94.82 | 89.87 | 94.86 | 94.99 | 97.90 (0.27) |
| EWTF D100 | 99.04 | 98.61 | 97.77 | 92.77 | 96.18 | 96.40 | 98.77 (0.22) |
| EWT | 98.00 | 96.96 | 95.14 | 93.25 | 96.12 | 96.57 | **99.49 (0.12)** |
| EWT D30 | 97.95 | 97.19 | 95.64 | 89.61 | 94.96 | 95.09 | 98.14 (0.11) |
| EWT D100 | 99.34 | 98.99 | 97.99 | 93.34 | 96.69 | 96.76 | 99.22 (0.13) |
| EWF | 97.18 | 95.82 | 93.73 | 91.49 | 95.25 | 95.80 | 99.18 (0.08) |
| EWF D30 | 96.06 | 94.14 | 91.61 | 89.53 | 94.41 | 94.76 | 95.95 (0.10) |
| EWF D100 | 97.66 | 96.55 | 94.28 | 90.60 | 95.02 | 95.27 | 97.46 (0.18) |
| EWMTF | 97.01 | 95.77 | 93.45 | 92.44 | 95.46 | 95.97 | 99.25 (0.19) |
| EWMTF D30 | 96.89 | 95.75 | 93.67 | 89.38 | 94.43 | 94.57 | 96.79 (0.31) |
| EWMTF D100 | 98.80 | 98.29 | 96.90 | 92.44 | 95.79 | 96.10 | 98.39 (0.09) |
| Spectrogram D=10* | 68.1 | 71.0 | 72.8 | 72.8 | 73.1 | 73.2 | 72.4 (0.4) |
| Spectrogram D=30* | 76.4 | 78.2 | 78.5 | 76.6 | 78.0 | 78.3 | 78.5 (0.8) |
| Spectrogram D=100* | 82.3 | 82.7 | 81.3 | 79.1 | 80.5 | 80.7 | 82.8 (0.1) |
| MUPET D=9* | 86.1 | 87.0 | 86.8 | 87.4 | 87.9 | 87.9 | 87.9 (0.2) |
| DeepSqueak D=10* | 79.0 | 80.7 | 81.0 | 81.2 | 82.1 | 81.9 | 82.4 (0.3) |
| Latent D=7* | 89.8 | 90.7 | 90.3 | 88.1 | 89.6 | 89.6 | 90.4 (0.3) |

Results with (*) are from Goffinet et al. (2021). Values are Top-1 accuracy (%) ± std.

Table 27: Predicting mouse identity. Classification accuracy (Top-1, %) and std over 5 splits.

| Feature Set | $\alpha = 0.01$ | | $\alpha = 0.001$ | | $\alpha = 0.0001$ | |
|---|---|---|---|---|---|---|
| | Top-1 | Top-5 | Top-1 | Top-5 | Top-1 | Top-5 |
| EF (D=100) | 35.0 (0.5) | 65.9 (0.3) | 35.3 (0.5) | 66.0 (0.2) | 35.0 (0.4) | 65.9 (0.4) |
| EF (D=30) | 30.1 (1.1) | 66.0 (0.5) | 30.0 (0.4) | 66.0 (0.4) | 30.4 (0.5) | 66.2 (0.4) |
| ETF (D=100) | 51.9 (0.9) | 79.3 (0.5) | 52.1 (0.4) | 79.5 (0.2) | 52.0 (0.4) | 79.6 (0.2) |
| ETF (D=30) | 41.6 (0.6) | 74.0 (0.4) | 42.5 (0.4) | 74.4 (0.5) | 42.5 (0.2) | 74.3 (0.2) |
| ET (D=100) | **53.1 (0.4)** | **80.0 (0.5)** | 52.8 (0.4) | 79.9 (0.3) | 52.6 (0.5) | 79.8 (0.3) |
| ET (D=30) | 42.8 (1.1) | 74.5 (0.5) | 42.9 (0.4) | 74.5 (0.2) | 42.9 (0.5) | 74.5 (0.4) |
| EMTF (D=100) | 48.2 (0.4) | 76.6 (0.3) | 48.2 (0.5) | 76.6 (0.2) | 47.9 (0.5) | 76.4 (0.2) |
| EMTF (D=30) | 38.3 (0.6) | 71.7 (0.3) | 37.9 (0.7) | 71.5 (0.4) | 38.1 (0.1) | 71.7 (0.2) |
| Spectrogram D=10* | 9.9 (0.2) | 36.6 (0.4) | 10.8 (0.1) | 38.6 (0.2) | 10.7 (0.2) | 38.7 (0.5) |
| Spectrogram D=30* | 14.9 (0.2) | 45.1 (0.5) | 17.3 (0.4) | 50.7 (0.6) | 17.3 (0.3) | 50.8 (0.3) |
| Spectrogram D=100* | 20.4 (0.4) | 55.0 (0.3) | 25.3 (0.3) | 62.9 (0.4) | 25.1 (0.3) | 63.2 (0.4) |
| MUPET D=9* | 14.7 (0.2) | 46.5 (0.3) | 19.0 (0.3) | 54.0 (0.2) | 20.6 (0.4) | 57.3 (0.4) |
| Latent D=8* | 17.0 (0.3) | 49.9 (0.4) | 22.7 (0.5) | 59.2 (0.6) | 24.0 (0.2) | 61.6 (0.4) |

Results with (*) are from Goffinet et al. (2021). Values are accuracy (%) $\pm$ std over 5 folds.

