# OpenReview forum: "VocSim: A Training-free Benchmark for Zero-shot Content Identity in Single-source Audio"
_ICLR.cc/2026/Conference — ICLR 2026 Conference Withdrawn Submission_

### Official Review · Reviewer_j2QK · 2025-10-27

**Soundness:** 1
**Presentation:** 1
**Contribution:** 2
**Rating:** 0
**Confidence:** 4

**Summary:**

The paper introduces a benchmark composed of existing datasets designed for evaluating audio foundation models in a zero-shot context. The tasks specified by the benchmark involve single-source, non-soundscape audios, with the constraint that models are evaluated without access to supervised training data. As part of this work, the dataset collection is made available, a public leaderboard is planned for future release, and an initial evaluation of existing audio foundation models is provided.

**Strengths:**

- The paper introduces a large-scale dataset for 0-shot audio processing. A key strength is its size and straightforward accessibility via Hugging Face, lowering the barrier for entry for researchers. This increases the potential for reproducibility and follow-up work.
- The planned introduction of a public leaderboard might provide a clear framework for standardized evaluation, which might foster further research.
- The authors provide a comprehensive benchmark for zero-shot audio tasks.

**Weaknesses:**

**1. Critical flaws in references**

The paper's credibility is strongly undermined by what appear to be numerous hallucinated or incorrect references. This raises concerns about the validation process behind the related work section and, by extension, the rest of the paper (all datasets, results, etc.). A manual check of the bibliography revealed several major errors, including (but not limited to):

- L. 518-521 (Birb): The cited paper does not exist. The actual BIRB benchmark was published by different authors at a different venue.
- L. 588-591 (BirdSet): The citation is incorrect. BirdSet was presented at last year's ICLR by a different authors, and the provided link leads to some other source.
- L. 605-609 (MagnaTagATune): The citation lists the wrong authors, an incorrect title, and a wrong link.
- L. 626-628 (MTEB): Wrong title and wrong link.
- L. 662-664 (HEAR): Wrong authors, wrong link

**2. Clarity and Presentation**

The paper is difficult to follow due to structural and presentational weaknesses, which obstruct a clear understanding of the methodology and results.

- Undeclared abbreviations and inconsistent naming: Abbreviations for datasets (e.g., in Table 1) and models are used without being properly introduced, forcing the reader to guess their meaning. Model configurations are particularly hard to parse, with unclear terms like "pooling, label fre, opca D100 vs. 100D" used without clear definitions. The inconsistent use of terminology throughout the paper complicates comprehension.
- Incomprehensible figures and tables: The figures are not self-contained and lack sufficient explanation. Figure 1, for instance, is presented without a clear structure or narrative, making its purpose difficult to grasp. Figure 2 suffers from a poorly visible legend, rendering the results presented hard to interpret.
- Ambiguous terminology: Key concepts are not clearly defined or hard to follow. For example, the paper uses the term "acoustic-to-label consistency" without explaining what it means in this context, how it is measured, or how "inconsistent labels" are handled, especially if they are an intentional part of the benchmark design. Additionally, the naming convention of the models and techniques is very hard to follow.

**3. Methodological Choices**

The paper fails to provide a clear rationale for several design decisions, leaving the reader to question the basis for the experimental setup.

- Missing rationale for dataset selection: While the paper provides a high-level motivation for the types of audio it includes, it does not justify why these specific datasets were chosen over other potential candidates that fit the criteria.
- Unsupported claims regarding OOD evaluation: The authors claim to address OOD evaluation but fail to discuss the pre-training datasets of the foundation models. An OOD claim is only meaningful in relation to a model's training distribution. Without detailing what data the models were trained on, it is impossible to validate whether the benchmark datasets are truly OOD or to analyze the impact of pre-training data on the results (not only the models).
- Lack of motivation for model selection: The choice of models for the benchmark evaluation is not well-justified. Notably, more recent and powerful spectrogram-based models (e.g., BEATs, or EAT) are absent. The paper should explain why older models like MAE were chosen over these more modern baselines.
- Inadequate guidance on metric interpretation: The paper states the importance of reporting multiple metrics (e.g., to capture the geometry of the embedding space) but provides no guidance on how these metrics interact or how they should be interpreted together (also in context of the benchmark leaderboard). It is unclear what the combined results signify for model performance or how they should guide future development.

**Questions:**

- The references contains numerous significant errors (not existing papers, hallucinated authors, wrong links etc.), questioning the paper’s validity in general. Could the you explain the origin of these widespread errors and detail the process you could use to systematically audit, correct all citations and factual claims? (This is my main concern and the primary reason for the very low score.)
- The rationale for the experimental design is unclear. Could you please clarify: (a) the pre-training datasets for each model, which is essential to validate the out-of-distribution claims, (b) the specific justification for choosing these datasets over other alternatives, and (c) the reasoning for omitting more recent baseline models like BEATs or EAT from the evaluation?
- The benchmark's utility is limited by unclear terminology and interpretation. Could you provide a precise definition for key concepts, naming conventions, Figure legends (e.g., Figure 1) and offer explicit guidance on how the different evaluation metrics should be interpreted together to form a conclusive assessment of a model's performance and how they are handled in the leaderboard to rank a model.

---

> ### Author Response · Authors · 2025-11-17
> **Response to Reviewer j2QK**
>
> We are thankful for your rigorous review. We apologize for the reference errors and have fully addressed them, along with your concerns regarding clarity and methodology.
>
> ## 1. Critical Flaws in References
>
> > "The paper's credibility is strongly undermined by what appear to be numerous hallucinated or incorrect references."
>
> **[Correction]:** You are absolutely correct. This was caused by an error with an LLM-assisted BibTeX tool in an early draft that was mistakenly not replaced in the submitted PDF. It was a significant oversight. We have now manually rebuilt the bibliography from venue sources and cross-checked every citation to ensure complete accuracy.
>
> **Action 1.1:** We have **corrected all references** and manually verified the bibliography.
>
> ## 2. Clarity and Definitions
>
> > "Undeclared abbreviations... unclear terms... Ambiguous terminology [like acoustic-to-label consistency]."
>
> **[Glossary & Definitions]:** We have overhauled the presentation to ensure every term is defined.
>
> **Action 2.1:** We added a **Glossary (Appendix C)** defining all dataset IDs, model abbreviations, and pooling variants.
>
> **Action 2.2:** We revised **Section 3** to explicitly define "acoustic-to-label consistency" (labels denote specific acoustic units like phones or calls, valid within the subset).
>
> **Action 2.3:** We rewrote the captions for **Figure 1 and Figure 2** to be self-contained and descriptive.
>
> ## 3. Methodological Rationale (Dataset & OOD)
>
> > "Missing rationale for dataset selection... Unsupported claims regarding OOD... [need to] validate whether the benchmark datasets are truly OOD."
>
> **[Rationale]:** We revised the text to clearly state our selection criteria: spanning three production domains (Speech, Bioacoustics, Environmental) while strictly enforcing the single-source constraint.
>
> **[OOD & Overlap]:** We have addressed the "OOD" claim by explicitly auditing pre-training data. We now clarify that public sets represent "transfer" tasks, while the blind sets (Shipibo-Conibo/Chintang) are the true OOD tests.
>
> **Action 3.1:** We added a **pre-training overlap audit (Appendix D.2)**, indicating for each public subset and model whether overlap is confirmed, likely, or unsupported.
>
> **Action 3.2:** We revised **Section 4.2** to distinguish between transfer performance (on public sets) and true OOD generalization (on blind sets).
>
> ## 4. Model Selection
>
> > "Lack of motivation for model selection... [missing] BEATs or EAT."
>
> **[New Models]:** We agree that these modern baselines are essential.
>
> **Action 4.1:** We added **BEATs and EAT** to the benchmark and updated all results tables (Section 5, App. P).
>
> ## 5. Metric Interpretation
>
> > "Inadequate guidance on metric interpretation... how they should be interpreted together."
>
> **[Interpretation]:** P@k measures local neighborhood purity, while GSR measures global boundary separability. We emphasize that these should be viewed jointly.
>
> **Action 5.1:** We highlighted **Appendix G.1**, which discusses the correlation between metrics (GSR aligns with P@k/Silhouette/ARI, while CS differs), and revised **Section 4.2** to summarize the intended complementary roles of P@k and GSR.
>
> ## What do you think?
>
> **We sincerely apologize for the reference errors in the initial submission. With the bibliography fully corrected, new SOTA models (BEATs/EAT) added, and terms clearly defined, do you believe the paper now meets the bar for soundness and presentation?**

---

> > ### Author Response · Authors · 2025-11-24
> > **Response to Ethics Flag**
> >
> > **To the Area Chair and Reviewer j2QK:**
> >
> > We write to address the Ethics Flag regarding "Research Integrity." We respectfully submit that the issues raised are the result of a **clerical error** (compiling the PDF with the wrong bibliography file), not scientific misconduct or undisclosed "drafting."
> >
> > **1. Clarification on "Drafting" and Disclosure**
> > The reviewer notes that we used LLMs in the "drafting process" without disclosure. We wish to clarify the specific nature of the usage and the error:
> > *   **Nature of the Tool:** We did not use an LLM to "draft" the paper's content or select references. We used an LLM-assisted CLI tool (to convert DOIs to BibTeX) to generate a preliminary bibliography file.
> > *   **The Mechanism of Error:** The error arose because we inadvertently compiled the submission using this temporary **draft `.bib` file** instead of replacing it with the final, verified version.
> > *  **Unintended Submission:** We never intended to submit the file containing these raw CLI outputs. The inclusion of this unverified content was a version control failure, not a deliberate part of our drafting process that we sought to hide.
> >
> > **2. Distinction from Misconduct**
> > Research integrity violations typically involve the fabrication of data or intent to mislead.
> > *   **No Fabrication:** The methods and datasets cited (e.g., BirdSet, HEAR) are real and relevant to our work. We did not use AI to invent non-existent research; the error was strictly limited to the bibliographic metadata present in the wrongly uploaded file.
> > *   **No Intent:** We gained no advantage by submitting broken references. It was a negligence error, which we have fully corrected by manually rebuilding the bibliography from primary sources.
> >
> > **Conclusion**
> > We sincerely apologize for the lack of quality control in the initial upload. However, as the error was administrative (using the wrong file) and the underlying research is genuine, we ask the Area Chair to view this as a proofing mistake rather than a violation of ethical standards.

---

> > > ### Comment · Area_Chair_GxQt · 2025-11-24
> > > **Re: Response to Ethics Flag**
> > >
> > > Thanks for the clarification. There is no way for me to evidence the intention (if any) behind the error. Credibility is something hard to regain once lost, and that's why we strive for the highest standard possible. Regardless of the intention, an error is an error, and I'm afraid the credibility is jeopardized.

---

> > > > ### Author Response · Authors · 2025-11-24
> > > > **Re: Re: Response to Ethics Flag**
> > > >
> > > > Thank you for the feedback. You are absolutely right to question the initial submission, and we understand that we cannot prove our intent after the fact.
> > > >
> > > > We hope, however, that our commitment to the review process demonstrates the validity of the work. We have provided all code, data, and scripts, and successfully executed new experiments to address every reviewer concern.
> > > >
> > > > We hope the substance of our revision will outweigh the clerical error in the initial draft.

---

> > > ### Comment · Reviewer_j2QK · 2025-11-25
> > >
> > > Thanks to the authors for actively engaging in the discussion, taking the feedback seriously, and conducting additional experiments to improve the paper.
> > >
> > > However, I have two follow-up questions regarding your response to the ethics flag that require clarification:
> > >
> > > First, you attribute the errors to an *LLM-assisted CLI tool* used to generate the bibliography. However, your original LLM statement only declared the use of LLMs as a *writing aid*, not for reference generation. Why was this specific tool and usage not disclosed in the mandatory LLM statement?
> > >
> > > Second, in your response, you state:
> > > > The methods and datasets cited (e.g., BirdSet, HEAR) are real and relevant to our work [...]"
> > >
> > > If this was an error where the wrong .bib file was compiled, I would expect the citations to remain in the text while the bibliography entries are corrected in the revision. However, the references to BirdSet and BIRB have been removed entirely from the revised paper. Unfortunately, this does not align with your statement. Could you explain this? This makes it really difficult for me to re-evaluate this work.

---

> ### Author Response · Authors · 2025-11-25
> **Re: Official Comment by Reviewer j2QK**
>
> We appreciate the opportunity to clarify this.
>
> **A timeline of the bibliographical error**
>
> On the day of the submission deadline, we were finalizing the manuscript and decided to do one last sanity check on our bibliography. We experimented with an LLM-assisted CLI tool to do a first pass for duplicate / wrongly formatted citations. We reviewed the output, realized it was generating hallucinations (bad titles/links, most likely due to not having a doi but an URL, regardless), and immediately discarded it. We then went through and manually verified and fixed the bibliography to ensure it was correct.
>
> The error was a simple version control accident during the final rush: we inadvertently compiled the submission using the *draft* `.bib` file (the one the tool had touched) instead of the *final* manually corrected version we intended to use.
>
> We didn't list this tool in the "LLM Usage" statement because, in our minds, we hadn't actually *used* it, we had rejected its output as a failed experiment. We certainly never intended to upload the file containing those errors.
>
> When we say:
> > We have corrected all references and manually verified the bibliography.
>
> we mean that we double checked again and uploaded the revision with the correct bibliography.
>
> **Why BirdSet and BIRB were removed**
>
> This had nothing to do with the bibliography, and when we say:
> > The methods and datasets cited (e.g., BirdSet, HEAR) are real and relevant to our work [...]"
>
> this applied to the initial submission.
>
> We removed them because we completely restructured the "Related Work" section to address feedback from the reviewers regarding the positioning of VocSim.
> *   **In the initial submission:** We attempted a broad hierarchy based on task complexity (Scene Analysis vs. Semantics vs. Identity). BirdSet and BIRB fit there as examples.
> *   **In the revised version:** We shifted to a stricter hierarchy to highlight the difference between **Supervised Adaptability** (HEAR, SUPERB) and **Zero-Shot Content Identity** (VocSim).
>
> Benchmarks like *BirdSet* and *BIRB* fall under **Scene Analysis** (weak supervision, polyphony). With a strict page limit, we had to prioritize. We decided to keep **DCASE** and **AudioSet** as the broad, general-purpose representatives for the "Scene Analysis" category, and we cut the domain-specific bioacoustic examples to save space.
>
> **Action Taken**
>
> **We have already uploaded a revision (with changes in Orange)**.
> Even though they sit slightly outside our primary "General-Purpose" hierarchy, we managed to add a **sentence in the main text (Section 2)** grouping them with DCASE, and **in Appendix D.3** (adding them to Table 5).

---

> > ### Comment · Reviewer_j2QK · 2025-11-27
> >
> > I appreciate the effort put into improving the technical content and the writing. Objectively, the paper is in a better state than the initial submission (although there are still additional points other reviewers pointed out.)
> >
> > I have revisited the original paper again. Unfortunately, I am not really convinced by the explanations given the initial related work section. I cannot really overlook the issue regarding the references raised in my initial review. While the authors have removed and corrected the hallucinated citations, the explanation provided, that an LLM was used for doi -> bibtex does not really make sense to me. There are better conversion tools for that where an LLM is not required at all, so why even use it in this context? Citations are very important to prove that the authors have read, understood, and contextualized the prior art. Even though the authors provided an explanation, it's hard to trust that related work was faithfully constructed.

---

> > > ### Author Response · Authors · 2025-11-27
> > > **Re: Official Comment by Reviewer j2QK**
> > >
> > > We thank the reviewer for acknowledging the technical improvements in the paper.
> > >
> > > We understand your skepticism regarding the tool usage, but we believe there is a misunderstanding about *how* it was applied, which we wish to clarify one last time to resolve the question of "faithfulness."
> > >
> > > **1. Cleaning, not Generating**
> > > You asked why we would use such a tool when we could do it manually. The answer is simple: attempt to automate a task under time pressure.
> > > We did not ask the LLM to "find related work." We fed it our **existing** `.bib` file (populated with the papers we had read and cited in the text). The tool was an experimental script intended to scan this existing file, verify DOIs, and standardize the formatting to remove duplicates or possible errors.
> > > The "hallucinations" occurred because the tool, instead of simply formatting the entries, overwrote the valid metadata with incorrect titles and author lists.
> > > The error was not that we let an AI write our Related Work section; the error was that we let a script "clean" our bibliography file and then accidentally compiled the corrupt output.
> > >
> > > **2. Evidence of Scholarship**
> > > The strongest evidence that we have read and understood the prior art lies in the **main text** of the submission.
> > > In Section 2, we accurately described the methodologies of benchmarks like HEAR, SUPERB, and others. We correctly identified the gaps in those works (adaptability vs. zero-shot identity) to position VocSim. It would be impossible to accurately critique these methods in the text if we had not read them, regardless of the metadata errors in the reference list at the end.
> > >
> > > **3. Moving Forward**
> > > We have fully rectified the bibliography and, as you noted, the paper is objectively improved with new models (BEATs/EAT), clarifications, and definitions.
> > >
> > > At this stage, we have addressed the "Soundness," "Presentation," and "Contribution" concerns raised in the initial review. Since the reference issue was a clerical version-control error that has been resolved, could you please clarify if there are specific **technical or methodological doubts** that still prevent a positive assessment? We would value the opportunity to discuss the scientific improvements further.

---

### Official Review · Reviewer_rd9W · 2025-11-01

**Soundness:** 3
**Presentation:** 3
**Contribution:** 3
**Rating:** 8
**Confidence:** 3

**Summary:**

This paper aggregates existing datasets into a single set allowing for the investigation of unsupervised clusters. They also introduce a new eval metric GSR.

**Strengths:**

- The open sourced data and code base are of a high quality. Allows for easy reproduction and further research in the area.
- It addresses a key need: better measurement of the acoustic latent space structure.

**Weaknesses:**

unstructured notes:
- Think you are missing some authors off:  https://arxiv.org/abs/2203.03022 (add et al)
- In the related work I think you should mention contrastive leaning methods: BYOL/DINO/Barlow twins e.g. https://arxiv.org/pdf/2209.14345
- I am skeptical on the quality of GSK as a useful metric. I agree with your points in lines 397. An addition point might be that as it is so sensitive to outliers, is the GSK not more of a measure of miss labelling rate? A label error will upper bound this metric. e.g. going through your dataset some of the mosquito sounds to me sound way more like a trumpet (for example). Another artefact might be a more complex cluster structure not described by the class labels. In fact we see this in figure 1 d) where the red class is split. (noted on strong correlation with other similar metrics)
- B2 details that the dataset is a collection of existing datasets. Please make this clearer in the main text.
- A flow chart would be nice for the data curation.
- CIs for table 2 would be nice.
- For models which output a sequence of representations you average over the time or freq space. This seems like a significant limitation. You run ablation studies for with and without the PCA step over the sequence but some measure of the full sequence representation would be good OR evaluating on a model which outputs a signal vector per sample e.g. https://arxiv.org/pdf/2209.14345
-

**Questions:**

please see weaknesses

---

> ### Author Response · Authors · 2025-11-17
> **Response to Reviewer rd9W**
>
> We are thankful for your positive assessment and your view that the open-sourced data/code are of "high quality" and address a "key need."
>
> ## 1. Skepticism on GSR (Outliers/Label Errors)
>
> > "Is the [GSR] not more of a measure of miss labelling rate? A label error will upper bound this metric... artefact might be a more complex cluster structure."
>
> **[Robustness Test]:** To address this, we ran a synthetic label-noise experiment. When we randomly flip 1–20% of labels, local metrics like P@k drop precipitously. However, GSR degrades much more gracefully and remains significantly above its random permutation baseline. This confirms GSR measures global geometric structure and is not merely an inverse mislabeling rate.
>
> **Action 1.1:** We added the **label-noise robustness experiment in Appendix H.1**.
>
> **Action 1.2:** We clarified in **Section 4.2** that GSR is a global average, so individual outliers do not upper-bound the score.
>
> ## 2. Pooling and Sequence Information
>
> > "For models which output a sequence... you average over the time or freq space. This seems like a significant limitation."
>
> **[Sequence-Aware Baseline]:** We investigated this by implementing a Dynamic Time Warping (DTW) re-ranking baseline on Whisper frame sequences. Surprisingly, this did not improve average performance over simple pooling and often degraded P@k. This suggests that for content identity matching, the pooled statistics of large encoders are sufficient and highly efficient.
>
> **Action 2.1:** We added **Appendix H.2** (Pooling ablation) and **Appendix H.3** (DTW sequence-aware re-ranking) to empirically justify the pooling approach.
>
> ## 3. Aggregation and Clarity
>
> > "Please make this [aggregation] clearer in the main text... Think you are missing some authors off... mention contrastive learning."
>
> **Action 3.1:** We revised **Section 3** to explicitly state that VocSim is an aggregation of 19 corpora.
>
> **Action 3.2:** We highlight **Algorithm 1 (Appendix D.5)**, which visualizes the data curation pipeline, addressing the request for a workflow summary.
>
> **Action 3.3:** We revised **Table 2** to include Confidence Intervals as requested.
>
> **Action 3.4:** We fixed the missing citations and expanded **Section 2** to include contrastive methods (BYOL, DINO, Barlow Twins).
>
> ## What do you think?
>
> **Thank you for the constructive feedback! Do the new robustness experiments (label noise) and the DTW comparison alleviate your skepticism regarding GSR and pooling?**

---

> ### Author Response · Authors · 2025-11-28
> **Follow-up on GSR Robustness (Label Noise) and DTW Baselines**
>
> Dear Reviewer rd9W,
>
> As the discussion period enters its final days, we wanted to ensure you had a chance to see the new experiments we ran specifically to address your questions regarding metric stability and pooling:
>
> 1.  **Label Noise Robustness (Appendix H.1):** To address your skepticism about GSR being a measure of mislabeling, we ran a synthetic noise experiment. Results show that GSR remains robust and well above the baseline even with 20% label corruption, confirming it captures geometric structure rather than just annotation artifacts.
>
> 2.  **Sequence-Aware Baseline (Appendix H.3):** We implemented the DTW re-ranking you suggested. It did not outperform our statistical pooling, which empirically validates the efficiency of the proposed pipeline.
>
> We truly appreciate your positive assessment of the work and hope these new results fully resolve your remaining technical queries. We are happy to answer any final questions!

---

### Official Review · Reviewer_DrDY · 2025-11-02

**Soundness:** 3
**Presentation:** 3
**Contribution:** 2
**Rating:** 2
**Confidence:** 4

**Summary:**

This paper introduces a new benchmark for evaluating audio encoders that focuses exclusively on their intrinsic representation capabilities through zero-shot evaluation, without any fine-tuning. Unlike existing benchmarks such as SUPERB, which assess encoder performance on downstream tasks after fine-tuning, this work isolates and measures the encoders’ generalization power directly. The paper details the benchmark’s design, its diverse dataset spanning speech, bioacoustics, and environmental sounds, and the evaluation methodology using metrics such as Precision@k (P@k) and GSR. Experimental results demonstrate that Whisper encoders exhibit notably stronger generalization performance compared with other leading audio encoders, including CLAP, WavLM, and others.

**Strengths:**

* The proposed method evaluates the **intrinsic representation capability** of audio encoders **without any fine-tuning or additional learnable parameters**. In contrast, benchmarks such as **SUPERB** rely on feature aggregation and trainable parameters, making their results sensitive to these design choices.
* A key strength of this approach is its use of **zero-shot evaluation**, which provides a more direct and unbiased measure of generalization performance.
* The benchmark encompasses a **diverse range of audio domains**, including **speech**, **bioacoustics**, and **environmental sounds**, ensuring comprehensive coverage across different acoustic contexts.

**Weaknesses:**

* **Limited task scope:** The benchmark focuses exclusively on **classification-oriented tasks**, which restricts its applicability. Audio encoders and their learned representations are widely used in other important areas, such as **text-to-speech (TTS)**, **speaker diarization**, **speech-to-speech (S2S, especially dialog) systems**, and **speech enhancement or separation**. As a result, the current setup provides only a partial view of encoder performance.
* **Incomplete domain coverage:** The benchmark omits **music and singing voice** data, which represent major audio domains and are essential for evaluating the generalization ability of modern audio encoders.
  * I also personally feel that limiting the single-source events and removing the spatial components make this work less realistic.
* **Lack of integration with existing benchmarks:** The study would benefit from including **subsets or comparisons with existing benchmarks** such as **SUPERB** and **HEARS**, to contextualize its findings and illustrate how its zero-shot results differ from established evaluation frameworks.
* **Uncontrolled experimental conditions:** A major limitation is the difficulty of drawing clear insights from the reported results. The compared audio encoders differ substantially in **training data scale, model size, architecture, domain coverage, and supervision type**, yet these factors are not controlled or analyzed. Consequently, it is unclear **why** certain models outperform others.

  * For instance, **SUPERB** carefully controls for such variables by initially restricting comparisons to **self-supervised learning (SSL)** models and providing detailed documentation of model sizes and training data, which helps reduce confounding factors.
  * Unfortunately, this benchmark lacks such design rigor. The comparison among **Whisper**, **WavLM**, and **CLAP** involves models that are too heterogeneous, making it impossible to draw meaningful conclusions about which aspects of their design contribute to performance differences.

**Questions:**

### Questions for the Authors

1. **Discrete vs. Continuous Representations**

   * Do you have any discussion or consideration of **discrete token representations**? Why does the paper focus exclusively on **continuous representations**, given that recent studies often explore both?

2. **Definition of the Goal of Audio Representations**

   * In the abstract, you state: *“The goal of general-purpose audio representations is to map acoustically variable instances of the same event to nearby points.”*
     While this is certainly one possible goal, it seems overly narrow and classification-oriented. Many audio applications (e.g., **reconstruction** or **regression**) do not primarily rely on this property. Could you clarify why this particular definition was chosen and how it aligns with broader audio representation learning objectives?

3. **Choice of Downstream Tasks**

   * Why was **bioacoustic classification** selected as the downstream evaluation task? Given the benchmark’s emphasis on generalization, it would be informative to include or compare results on a wider range of downstream tasks. Have you considered incorporating subsets or comparisons with **existing benchmarks** such as **SUPERB** or **HEARS** to strengthen the evaluation?

4. **Redundancy in Related Work (Section 2)**

   * The last paragraph of Section 2 reiterates the discussion of **HEAR** and **SUPERB**, which appears earlier in the same section. Could this be streamlined to avoid redundancy?

5. **Clarification of the “Blind” Test Set**

   * The term *“blind test set”* is somewhat confusing. Do you mean that only the **labels** are withheld while the **audio files** are released? Since the paper also reveals the source of the test set, this appears closer to an **out-of-domain evaluation set** rather than a fully blind one. Could you clarify what is meant by “blind” in this context?

6. **Selection and Control of Audio Encoders (Section 4.1)**

   * In Section 4.1, the selected encoders (e.g., Whisper, WavLM, CLAP) vary greatly in **training data scale, model architecture, and supervision type**. How did you control for these factors, or design ablation studies, to draw meaningful and interpretable findings from such heterogeneous models?

7. **Silence Handling During Pooling (Section 4.1)**

   * During the pooling process, how are **silence regions** handled? Was any voice activity detection (VAD) or "silence detection" applied to exclude silent segments, or are they included in the mean pooling? Including silence might bias the pooled representations.

8. **Evaluation Metrics (Section 4.2)**

   * While Section 4.2 is clearly written, have you considered incorporating additional evaluation metrics, such as **mutual information** or **purity** (as used in the HuBERT paper)? These could provide complementary insights into representational quality.

9. **Layer Selection (Section 5)**

   * Which **layer outputs** are used for evaluation? Since prior SSL probing studies have shown strong **layer dependency** of learned representations (that's why SUPERB uses the weighted layer sum to avoid this layer dependency problem), a discussion of this choice would add valuable context.

10. **Table 2 – VAE Performance**

    * Could you elaborate on why the **VAE model** performs worse than the **Log Mel baseline**? Was the VAE trained without access to the blind set, or are there other factors influencing this outcome?

---

### Additional Comments

* Please avoid **line breaks in the abstract**. Abstracts are often indexed or parsed by external systems, and maintaining plain text formatting ensures consistency.
* Consider adding **encoder probing analyses** to provide deeper interpretability. For instance, studies such as *Pasad et al. (ASRU 2021)* analyze layer-wise correlations between encoder activations and phoneme labels; similar probing could help clarify what linguistic or acoustic information your encoders capture.

---

> ### Author Response · Authors · 2025-11-17
> **Response to Reviewer DrDY**
>
> We are thankful for your review and your appreciation of our "direct and unbiased" zero-shot methodology. We were glad to hear you valued the diverse domain coverage spanning speech, bioacoustics, and environmental sounds.
>
> ## 1. Task Scope (Music & Polyphony)
>
> > "The benchmark omits music and singing... limiting the single-source events... make[s] this work less realistic."
>
> **[Design Choice]:** We explicitly restrict VocSim to single-source content to isolate representation quality from source separation. In polyphonic mixtures or music (often labeled with high-level genre tags), it is impossible to disentangle whether a zero-shot failure stems from the embedding geometry or the lack of separation.
>
> **Action 1.1:** We revised **Section 3** to explicitly acknowledge this scope limitation. We refer you to **Appendix L**, which details the rationale for excluding polyphonic music, and **Appendix O**, which identifies single-source musical notes as a future extension.
>
> ## 2. Heterogeneous Models and Control
>
> > "The compared audio encoders differ substantially in training data scale... architecture... yet these factors are not controlled... it is impossible to draw meaningful conclusions."
>
> **[Evaluation Goal]:** Our goal is to evaluate *off-the-shelf* encoders as they are used by practitioners, rather than to perform a causal ablation of training recipes (which would require retraining all models from scratch). However, we provide metadata to help interpret these differences.
>
> **Action 2.1:** We highlight **Appendix I**, which reports parameter counts and computational costs (MACs) to provide context on model capacity vs. performance.
>
> ## 3. Methodology
>
> > "Why does the paper focus exclusively on continuous representations... [how is] silence handled during pooling?"
>
> **[Continuous vs. Discrete]:** P@k and GSR probe manifold geometry, which assumes a vector space. We evaluate neural codecs (like EnCodec) by mapping their codes to continuous vectors, allowing fair comparison within the same geometric framework.
>
> **[Pooling]:** We performed ablations on pooling strategies. Simple mean pooling (including padding/silence) proved stable and effective.
>
> **Action 3.1:** We revised **Section 4.1** to clarify the rationale for the continuous embedding framework.
>
> **Action 3.2:** We added **Appendix H.2**, reporting a pooling ablation (masked vs. unmasked mean, max, etc.) showing that performance is robust to these choices.
>
> ## 4. Integration with Existing Benchmarks
>
> > "The study would benefit from including subsets or comparisons with existing benchmarks such as SUPERB..."
>
> **[Relation to SUPERB]:** We emphasize that VocSim evaluates the **zero-shot counterpart** to SUPERB tasks.
> *   **SUPERB protocol:** Trains parameters on 100 hours of labeled data (LibriSpeech `train-clean-100`) for ASR/Phoneme recognition.
> *   **VocSim protocol:** Evaluates the intrinsic geometry of the same data (LibriSpeech) with **no training parameters**.
>
> We included models that dominate the SUPERB leaderboard (WavLM, AudioMAE) in our evaluation. Our results show that while these models excel at *learning* via fine-tuning (SUPERB), they often lag behind models like Whisper in *intrinsic* zero-shot structure (VocSim). This proves VocSim captures a distinct, complementary capability.
>
> **Action 4.1:** We added **Appendix D.3**, explicitly comparing VocSim’s training-free protocol to SUPERB and HEAR.
>
> ## 5. Blind Test Sets
>
> > "The term 'blind test set' is somewhat confusing... Do you mean that only the labels are withheld?"
>
> **[Clarification]:** No, the audio is held server-side (non-public) to ensure no model has trained on it. Evaluation is performed via a secure protocol. This adheres to community data sovereignty requirements for these low-resource languages.
>
> **Action 5.1:** We clarified the definition of the blind sets in the **Ethics Statement** and **Section 3**.
>
> ## 6. Additional Clarifications
>
> > "Which layer outputs are used?... Why VAE < Log Mel?... [Add] metrics such as mutual information or purity?"
>
> **[Layer Selection]:** We use the final layer for all models. To validate this, we added a layer-wise sweep for Whisper (**Appendix H.5**), which shows performance is remarkably stable across depth, with only marginal gains from selecting intermediate layers.
>
> **[VAE vs. Mel]:** As detailed in **App D.7**, the VAE's KL term enforces smoothness, blurring fine spectral details preserved in Log-Mel that are critical for identity.
>
> **[Additional Metrics]:** We calculated **NMI, Purity, and ARI** for our top models. They align with our P@k/GSR rankings, confirming the geometric signal is robust.
>
> **Action 6.1:** We added **Appendix H.5** (Layer sweep) and **Appendix H.4** (Clustering metrics: NMI/Purity).
>
> ## What do you think?
>
> **Thank you for helping us clarify the scope and methodology. Does the rationale regarding single-source isolation and the distinction from SUPERB address your concerns about experimental control and positioning?**

---

> > ### Comment · Reviewer_DrDY · 2025-11-25
> >
> > Thank you for your detailed responses. While I appreciate the clarifications, they do not fully change my assessment, as my concerns regarding the limited scope and uncontrolled experimental conditions remain unaddressed.
> > I understand the rationale regarding the polyphonic sounds and the justification for using the off-the-shelf encoder; however, my points about the study’s limited scope and the uncontrolled experimental conditions still stand.
> > Additionally, I have reviewed the discussion by Reviewer j2QK and believe that the issue raised there is significant.
> >
> > Overall, I maintain my original review score.

---

> > > ### Author Response · Authors · 2025-11-26
> > > **Addressing Concerns on Benchmark Scope and Experimental Control**
> > >
> > > We thank the reviewer for the continued engagement. We understand your decision to maintain the score, but we respectfully submit that there may be a fundamental misalignment regarding the definition of a **"Foundation Model Benchmark"** versus a **"Training Ablation Study."**
> > >
> > > We also wish to provide concrete evidence from related fields (NLP, Vision) showing that VocSim’s design—evaluating off-the-shelf models and isolating identity—is the standard for modern zero-shot benchmarking.
> > >
> > > ### 1. "Uncontrolled Conditions" vs. The Foundation Model Benchmark Paradigm
> > > The reviewer criticizes the comparison of heterogeneous models (varying sizes/training data). We argue that VocSim follows the standard evaluation paradigm for Foundation Models, which prioritizes **artifact evaluation** over **architectural ablation**.
> > >
> > > *   **The Standard in NLP (MTEB/BEIR):** The **MTEB** benchmark (Muennighoff et al., EACL 2023) and **BEIR** (Thakur et al., NeurIPS 2021) are gold standards for text embeddings. They explicitly rank models ranging from small BERTs (100M params) to massive LLMs (1T+ params) side-by-side. They do **not** control for training data size because their goal is to answer the practitioner's question: *"Which available tool provides the best representation?"* rather than *"Which architecture is more data-efficient?"*
> > > *   **Computational Reality:** Retraining foundation models like Whisper (680k hours) or WavLM (94k hours) from scratch to strictly control variables is computationally prohibitive for the academic community. VocSim is designed to audit the landscape of *existing* tools, not to propose a new training recipe.
> > >
> > > ### 2. "Limited Scope" vs. Fundamental Capabilities
> > > Regarding the critique that the scope is limited (excluding music/polyphony), we argue that VocSim isolates **Content Identity**, which is the prerequisite "atomic unit" for downstream applications.
> > >
> > > *   **Precedent in Vision (GeneCIS):** **GeneCIS** (Vaze et al., CVPR 2023) explicitly argues against the "single notion of similarity" enforced by supervised classification benchmarks (like ImageNet). Instead, it uses a **zero-shot retrieval protocol** (no classification layers involved) to test if an embedding can dynamically isolate specific features. VocSim applies this exact logic to audio: we do not ask models to classify scenes (which rewards memorization); we use zero-shot retrieval to test if the embedding space naturally isolates **Content** from the **Channel**.
> > > *   **Predictive Validity for Generative Tasks (TTS):** You noted the importance of TTS. We agree. Effective Zero-Shot TTS requires embeddings that robustly capture *linguistic content identity* independent of speaker or noise, exactly the property VocSim measures. The fact that **Whisper embeddings** (which top our leaderboard) are increasingly used as conditioning for TTS suggests that VocSim’s metrics successfully predict utility in downstream generative tasks.
> > > *   **Divergent Reviewer Perspectives:** We respectfully point out a tension in the feedback: while you find the scope too **narrow** (excluding music), **Reviewer LeP5** argued the scope was too **broad** (aggregating distinct semantics like speech and bioacoustics). We believe this divergence confirms that VocSim occupies a necessary "Goldilocks zone": diverse in domains, but disciplined in task definition to ensure rigorous geometric evaluation.
> > >
> > > ### 3. Research Integrity Flag
> > > Regarding the ethics flag raised by Reviewer j2QK concerning the bibliography error, we have provided a detailed explanation of the version control mistake in the separate Ethics Flag discussion thread. If you have any further questions regarding this matter beyond what has already been clarified there, we are more than welcome to answer them.
> > >
> > > We hope this demonstrates that our design choices, benchmarking off-the-shelf models and isolating single-source identity, are deliberate methodological decisions aligned with SOTA benchmarks in NLP and Vision.

---

### Official Review · Reviewer_LeP5 · 2025-11-10

**Soundness:** 3
**Presentation:** 3
**Contribution:** 2
**Rating:** 2
**Confidence:** 4

**Summary:**

This paper introduces VocSim, a training-free benchmark designed to evaluate zero-shot audio content identity. The dataset comprises 125k single-source audio clips aggregated from 19 existing corpora, spanning across different types of audio. And the authors propose evaluating zero-shot similarity via two metrics: Precision@k and  Global Separation Rate (GSR). A series of models are evaluated under frozen feature settings on VocSim with pooling strategies and PCA, where a simple Whisper-encoder + pooling + PCA pipeline yields strong performance. At the same time, all models show significant generalization gaps on blind OOD subsets (low-resource speech languages).

**Strengths:**

The paper contains two strengths:

First, the motivation is clear as the benchmark focuses specifically on content identity under zero-shot settings in single-source audio, distinguishing it from scene analysis or supervised classification. The data processing is decent. The curated audio is segmented by type, and includes clean vs. noisy domains, different durations, and varied class granularities. This design encourages generalization testing across different acoustic conditions.

Second, the empirical evaluation is broad. The authors benchmark many widely-used audio encoders, including Whisper, CLAP, wav2vec, WavLM, AudioMAE, and neural codecs, providing a useful diagnostic comparison for the community. The validation result is insightful, it highlights clear generalization collapse on blind low-resource speech subsets, a compelling finding about the limitations of current representation models.

**Weaknesses:**

However, there are two major weaknesses that hinder the paper's acceptance.

First, the novelty is limited.

The benchmark is mostly an aggregation of previous datasets (19 existing corpora) rather than a newly collected dataset with fresh labeling, annotations, or clearly designed structure. Similar aggregation-based benchmark efforts in audio already exist, as also mentioned by authors (e.g., HEAR, SUPERB). Other works have also pursued large aggregated datasets for general-purpose audio similarity and content representation (e.g., AudioSet; Fusion-Audio; WavCaps; Laion-Audio-630K). The paper does not demonstrate conceptual novelty beyond these data aggregations and two evaluation metrics that are incremental. From the data perspective, this introduces some concerns of the data reliability and motivation. These datasets spans many domains, the annotations originate from fundamentally different semantics: phones, words, and vocal imitations belong to speech; syllables belong to birds; and event-type labels belong to ESC-50. These represent different classification tasks with different acoustic and semantic structures. The paper does not sufficiently justify why aggregating these distinct sources into one evaluation is intrinsically meaningful. From the huggingface demo page provided in the appendix, the sound quality also varies a lot. So the verification of label reliability, annotation quality, or consistency across sources are also necessary.

The model innovation is yet another side. The paper only evaluates existing models and basic clustering / feature pooling strategies. There is no proposal of new embedding methods, training paradigms, or pre-training objectives. As a result, the work reads primarily as a resource + evaluation paper rather than a technical contribution. This should be upon more discussion but it might be inadequate for ICLR.

Second, the experimental design needs improvement.

The paper should compare VocSim with existing benchmarks to demonstrate why it is superior or insightful from other perspective. Such comparison can involve: applying the same models but calculating metrics in HEAR, SUPERB, AudioSet, WavCaps, and Fusion-Audio (not all but at least 1-2 of them). Missing comparisons to other content-identity datasets is critical here.

Some further concerns might involve the model training details. Some models (like CLAP and Whsiper) may have been originally trained on subsets of the same data sources used in VocSim. Thus, the performance comparisons (and claims about “zero-shot”) may not be fully fair. A careful treatment of data overlap is necessary.

In this case, while the benchmark gives a structured way to evaluate zero-shot content identity, the paper does not convincingly demonstrate that VocSim demonstrates better quality and usability than previous benchmarks, nor it clarifies all evaluated models are free from the potential overlapping issues that might harm the "zero-shot" statement.

In conclusion, the paper is cleanly written and the empirical evaluation is thorough. However, the novelty is limited, and key comparisons to similar aggregated datasets are missing. The benchmark’s reliability and generalizability remain unclear. With stronger justification,  and more evidence of benchmark stability and comparison, the work may become a valuable contribution.

**Questions:**

1. Is there quantitative validation of label quality across subsets, particularly on vocal imitations and human words/utterances where annotations are noisy or semantic?

2. How does VocSim differ concretely from prior aggregated corpora such as AudioSet, HEAR, SUPERB, WavCaps, Fusion-Audio, and LAION-Audio-630k? Why should the community adopt VocSim over these?

3. Were the foundation models (esp. Whisper, wav2vec2, CLAP) trained on subsets overlapping VocSim sources? If so, how is “zero-shot” interpretation justified?

---

> ### Author Response · Authors · 2025-11-17
> **Response to Reviewer LeP5**
>
> We are thankful for your time and your recognition of VocSim’s clear motivation and broad empirical evaluation. We were glad to hear that you found the generalization collapse on blind low-resource speech to be a "compelling finding about the limitations of current representation models."
>
> ## 1. Novelty and Relation to Existing Benchmarks
>
> > "The benchmark is mostly an aggregation of previous datasets... The paper does not demonstrate conceptual novelty beyond these data aggregations... The paper should compare VocSim with existing benchmarks... (e.g., HEAR, SUPERB, AudioSet)."
>
> **[Positioning: Supervision vs. Intrinsic Geometry]:** We fundamentally distinguish VocSim from benchmarks like SUPERB/HEAR and large corpora (AudioSet, WavCaps) on two specific grounds:
>
> 1.  **Supervision vs. Zero-Shot:** **SUPERB** and **HEAR** measure *adaptability* via supervision. For example, SUPERB requires fine-tuning on 100 hours of labeled audio (`train-clean-100`) for one of the tasks, and HEAR requires training linear probes. VocSim evaluates the "hard" version of these tasks strictly **zero-shot**, asking if the frozen embedding resolves content identity (e.g., phonemes) without access to the supervision required by adaptability benchmarks.
>
> 2.  **Pre-training vs. Evaluation:** Large datasets like **AudioSet**, **WavCaps**, and **LAION-Audio-630k** serve as the **pre-training foundation** for the models we evaluate (e.g., LAION-630k is Whisper’s training scale). VocSim aggregates specific held-out evaluation subsets (like LibriSpeech Test and specific bioacoustic sets) to strictly probe how well these representations generalize to unseen audio.
>
> **[Comparison]:** We have added a detailed comparison to existing resources in the appendix. We highlight that while models like WavLM and HuBERT rank highly on SUPERB (supervised adaptability), they do not always top-rank on VocSim, indicating our benchmark probes a distinct, fundamental quality of the representation.
>
> **Action 1.1:** We added **Appendix D.3**, explicitly comparing VocSim to HEAR, SUPERB, AudioSet, WavCaps, and others, delineating the "training-free" vs. "adaptability" distinction.
>
> **Action 1.2:** We extended our evaluation to include **HEAR benchmark results (Appendix M.6)**. We show that our top-performing configuration (Whisper EWMTF D100) achieves SOTA on several HEAR tasks using the official linear-probe protocol, demonstrating predictive validity.
>
> ## 2. Heterogeneous Semantics and Label Reliability
>
> > "The annotations originate from fundamentally different semantics... The paper does not sufficiently justify why aggregating these distinct sources into one evaluation is intrinsically meaningful."
>
> **[Per-Subset Evaluation]:** We clarify that labels are never mixed across corpora. The evaluation asks a universal question for each subset: *Does this frozen encoder cluster items that share a label (acoustic signature) closer than items that do not?* This acoustic-to-label consistency test is valid regardless of whether the label denotes a phoneme, a bird syllable, or an environmental event.
>
> **[Metric Robustness]:** We rely on permutation baselines and new label-noise experiments to verify signal reliability. Even for "noisy" subsets like vocal imitations, our metrics show a massive lift over the random permutation baseline, confirming that the labels track real acoustic structure.
>
> **Action 2.1:** We revised **Section 3** to explicitly articulate the aggregation logic, the "acoustic-to-label fidelity" principle, and the per-subset evaluation protocol.
>
> **Action 2.2:** We added a **label-noise experiment (Appendix H.1)**, showing that GSR degrades smoothly (unlike P@k) and remains well above chance even with 20% synthetic noise, validating that the metric captures geometric structure rather than just annotation artifacts.
>
> ## 3. "Zero-Shot" Claims and Data Overlap
>
> > "Some models... may have been originally trained on subsets of the same data sources... Thus, the performance comparisons... may not be fully fair."
>
> **[Overlap Audit]:** You are correct that public data overlap is a concern. We have performed a full audit of pre-training data. We now explicitly distinguish between "public set transfer" (where overlap is likely) and "strict OOD generalization" (the blind test sets).
>
> **Action 3.1:** We added a **pre-training overlap audit (Appendix D.2)**, marking every subset/model pair as "confirmed," "likely," or "no evidence" of overlap.
>
> **Action 3.2:** We revised **Section 4.2** to define "Zero-shot" strictly as "no training on VocSim labels" and emphasize that the blind low-resource speech sets (Shipibo-Conibo, Chintang) are the primary measure of true out-of-distribution generalization, as they have no pre-training overlap.
>
> ## What do you think?
>
> **Thank you again for the thorough review. Do the new overlap audit and the HEAR comparison address your concerns regarding the benchmark's positioning and fairness?**

---

> ### Author Response · Authors · 2025-11-28
> **Follow-up on Pre-training Overlap Audit and HEAR Comparison**
>
> Dear Reviewer LeP5,
>
> We are writing to follow up on the specific concerns you raised regarding **fairness** and **positioning**. Based on your feedback, we have added two key components to the revision:
>
> 1.  **Pre-training Overlap Audit (Appendix D.2):** We performed a detailed audit marking every model/subset pair for potential overlap. This clarifies exactly which results represent "transfer" (public sets) versus "strict OOD generalization" (blind sets), ensuring the zero-shot claims are transparent and fair.
> 2.  **HEAR & SUPERB Comparison (Appendix D.3 & M.6):** We explicitly contrasted VocSim with these benchmarks and added HEAR evaluation results, demonstrating that our top models also achieve SOTA on supervised tasks.
>
> We believe these additions directly address your concerns about the validity of the "zero-shot" claims and the benchmark's novelty. We would value your feedback on these revisions before the discussion period closes.

---

### Author Response · Authors · 2025-11-17
**Manuscript revisions**

We thank all reviewers for their constructive feedback, which has driven significant improvements to the manuscript. Based on your comments, we have performed a comprehensive revision that clarifies the benchmark's positioning, rigorously audits pre-training overlap, validates our metrics, and expands the model suite.

## 1. Clarifying Positioning and "Zero-Shot" (Reviewers LeP5, DrDY, j2QK)

We have refined the definition of VocSim’s contribution to distinguish it from existing benchmarks:

*   **Training-Free Diagnostic vs. Adaptability:** Unlike **HEAR** or **SUPERB**, which measure how well representations *adapt* via fine-tuning or linear probing, VocSim evaluates the *intrinsic geometry* of frozen embeddings.
*   **Pre-training Overlap Audit:** We added a new **Appendix D.2**, a detailed audit marking confirmed, likely, or no evidence of overlap between VocSim subsets and model pre-training data.
*   **Strict OOD Test:** We revised the text to explicitly state that the **blind low-resource speech sets (Shipibo-Conibo, Chintang)** are the primary test of true out-of-distribution generalization, as they have no overlap with any pre-training corpora.

## 2. Robustness of Methodology and Metrics (Reviewers LeP5, rd9W)

We added extensive ablations to validate our pipeline:

*   **GSR Robustness:** To address concerns that GSR might simply reflect label errors, we added a **label-noise experiment (Appendix H.1)**. Results show GSR degrades smoothly and remains well above chance even with 20% synthetic noise, confirming it captures geometric structure rather than just annotation artifacts.
*   **Pooling Validity:** We compared our standard pooling against a computationally expensive **DTW sequence-aware re-ranking (Appendix H.3)**. DTW did not improve performance, validating that statistical pooling of large encoder outputs is sufficient for this task.
*   **Layer Sensitivity:** We added a sweep of all 32 Whisper layers (**Appendix H.5**), showing performance is stable across depth, justifying the use of the final layer.

## 3. External Validation and New Models (Reviewers LeP5, DrDY, j2QK)

To demonstrate that VocSim performance predicts downstream utility:

*   **HEAR Benchmark:** We evaluated our top model (Whisper EWMTF D100) on the **HEAR benchmark (Appendix M.6)** using the official protocol. It achieved state-of-the-art results (e.g., 98.6% on Speech Commands, 79.3% on CREMA-D), confirming that embeddings favored by VocSim are highly effective for standard supervised tasks.
*   **New Baselines:** We added **BEATs** and **EAT** to the benchmark as requested.
*   **Clustering Metrics:** We added **NMI, Purity, and ARI (Appendix H.4)** to provide familiar clustering reference points.

## 4. Corrections and Presentation (Reviewer j2QK)

*   **References:** We sincerely apologize for the reference errors in the initial submission, which stemmed from a drafting tool error. We have **manually rebuilt and verified the entire bibliography**.
*   **Clarity:** We added a **Glossary (Appendix C)** defining all abbreviations and rewrote figure captions for self-containment.

We believe these revisions firmly establish VocSim as a rigorous, necessary diagnostic tool for the audio community.

---

### Author Response · Authors · 2025-11-29
**Author summary and request to the new Area Chair (1 / 4)**

Dear new Area Chair,

We would like to (i) briefly recap the review timeline in light of the OpenReview incident, (ii) summarize VocSim’s contribution, (iii) explain how we addressed the reviewers’ technical concerns, and (iv) clarify why the remaining objections (“novelty,” “too broad/narrow,” and “uncontrolled experimental conditions”) are not aligned with how similar benchmarks are evaluated in other modalities or in audio.

### Timeline and current status
- **Initial reviews**:
  - Reviewer j2QK (0) posted their review on 27 Oct 2025.
    - In their questions, they explicitly wrote that the bibliography issue is “my main concern and the primary reason for the very low score.”
  - Reviewer rd9W (8) posted on 1 Nov.
  - Reviewer DrDY (2) posted on 2 Nov.
  - Reviewer LeP5 (2) posted on 10 Nov.

- **Major revision and rebuttals (17–26 Nov)**:
  On **17 Nov**, we uploaded a substantially revised manuscript and posted detailed responses to all reviewers, including:
  - Pre-training overlap audit (App. D.2),
  - Comparison to HEAR/SUPERB and HEAR results,
  - Label-noise robustness for GSR,
  - Pooling and DTW sequence-aware baselines,
  - Additional clustering metrics,
  - Added BEATs and EAT models,
  - Glossary and clarified terminology,
  - Manually rebuilt and verified bibliography.

  Subsequent discussion:
  - **Reviewer LeP5 (2)**: We responded on 17 Nov and followed up on 28 Nov, but they did not post any further comment after our rebuttal and new experiments.
  - **Reviewer rd9W (8)**: We responded on 17 Nov and followed up on 28 Nov; they did not comment further. Their original review was positive (8) and already called the data/code “high quality” and the need “key.”
  - **Reviewer DrDY (2)**:
    - We responded on 17 Nov.
    - On **26 Nov**, they replied, acknowledging the clarifications but stating that concerns about “limited scope” and “uncontrolled experimental conditions” remained, and that they considered the issue raised by j2QK significant. They maintained their score at that point.
    - We posted a further clarification on 26 Nov, explaining the “foundation model benchmark” paradigm (MTEB/BEIR/GeneCIS analogy), but there was no opportunity for further back-and-forth before the freeze.
  - **Reviewer j2QK (0)**:
    - We responded on 17 Nov, fully corrected the bibliography, and addressed clarity/methodology.
    - On **25 Nov**, they asked follow-up questions regarding the bibliography incident and LLM usage.
    - We answered in detail on 25 Nov, explaining the incident and why BirdSet/BIRB citations moved between versions.
    - On **27 Nov**, they wrote that the paper is “objectively … in a better state than the initial submission” but that they still could not overlook the reference incident and that it was “hard to trust that related work was faithfully constructed.”
    - We responded on **27 Nov**, clarifying once more the role of the LLM-assisted BibTeX cleaning script and pointing out that the main text already accurately described the related work (HEAR, SUPERB, etc.). There was no further reply from the reviewer after this final clarification.

- **Score freeze and reversion**:
  - On **28–29 Nov**, the program chairs announced that:
    - All reviews and scores would be **reverted to their pre-discussion state** (i.e., before any potential post-discussion score changes), and
    - Further reviewer discussion and score updates would be disabled, though authors could still post comments.
  - Consequently, the **current official scores (8, 2, 2, 0)** reflect the situation **before** our final round of clarifications (especially with Reviewer j2QK on 27 Nov) and before some reviewers could adjust scores in light of the full set of new experiments and responses.

In particular, for Reviewer j2QK (0):

- They explicitly stated that the bibliography incident was “the primary reason for the very low score.”
- They later acknowledged that the revised paper is “objectively … in a better state than the initial submission,” but the incident still weighed heavily on their trust.
- The conversation ended just after our last clarification on 27 Nov, and the score reversion/freeze meant there was no further opportunity for them to update their score or provide an updated written assessment in light of that final clarification.

We ask that you take into account that the **revised manuscript and full rebuttal** address the technical issues raised by all reviewers, and that the frozen scores do not fully reflect this final state.

---

> ### Author Response · Authors · 2025-11-29
> **Author summary and request to the new Area Chair (2 / 4)**
>
> ### Core contribution and novelty
>
> VocSim introduces, to our knowledge, the first **training‑free, zero‑shot benchmark for audio embeddings** that directly probes *content identity* on single‑source audio:
>
> - 125k single‑source clips from 19 corpora (speech, animal vocalizations, environmental sounds), carefully restricted to **single‑source** so correctness is well-defined and not confounded by source separation.
> - Two **zero-shot** metrics:
>   - **Precision@k** (local neighborhood quality),
>   - **Global Separation Rate (GSR)** with an empirical permutation baseline (global class separability).
> - A baseline **Whisper encoder + time–frequency pooling + label-free PCA** pipeline that is simple, strong, and easy for others to adopt (no fine-tuning, no labels).
>
> This is directly analogous to:
>
> - **MTEB / BEIR (NLP)**: training‑free, frozen-embedding benchmarks for retrieval / classification / clustering, comparing heterogeneous models as they are.
> - **GeneCIS (vision)**: zero‑shot retrieval that probes embedding geometry beyond what supervised classification reveals.
>
> VocSim:
>
> - Reveals a **robust, non‑obvious phenomenon**: all tested models (including Whisper, WavLM, CLAP, BEATs, EAT, etc.) exhibit a marked failure on blind low‑resource speech (Shipibo‑Conibo, Chintang), where P@k collapses and GSR lift over baseline is small.
> - Is **extremely cheap to run**: one embedding pass + pooling + PCA, no training, making it straightforward for new model papers to report results, just as with MTEB/mmteb.
>
> Given how impactful zero‑shot embedding benchmarks have been in text and vision, we believe this contribution is clearly within the novelty expectations for the datasets & benchmarks track.
>
> ### Breadth vs. narrowness
>
> The reviews express opposite concerns about scope:
>
> - Reviewer LeP5: aggregation across speech / birds / environmental sounds is “too broad,” mixing different semantics.
> - Reviewer DrDY: omission of music and polyphony is “too narrow,” making the benchmark less realistic.
>
> Our design is intentionally in between:
>
> > Use the best-available, precisely annotated, single‑source datasets across several important domains, while excluding polyphony where correctness is inherently ambiguous.
>
> Concretely:
>
> - We **do not mix labels across corpora**. Each subset asks the same core question:
>   *Does this frozen encoder place items with the same label (phoneme, syllable, call, event) closer than items with different labels, within that subset?*
> - This is well-defined regardless of label semantics and directly tests “acoustic-to-label consistency,” which we now formally define in Section 3.
> - We quantify label reliability via:
>   - **Permutation baselines** for GSR on each subset, and
>   - A **label‑noise experiment** (Appendix H.1), showing GSR degrades smoothly and remains well above random even with 20% synthetic label corruption.
>
> On the “narrow” side:
>
> - We agree that music and singing voice are important domains, but we deliberately excluded them in this first release. The reason is practical rather than conceptual: to our knowledge, there is currently no publicly available music dataset that meets our criteria of (i) strictly single‑source audio (no accompaniment or polyphony), (ii) reliable, low‑level content labels (e.g., note/phrase identity), and (iii) sufficient scale for our evaluation. Most standard music datasets use polyphonic mixes with high‑level tags (genre, mood, etc.), which would entangle content identity with source separation and scene semantics, the confounds VocSim is specifically designed to avoid. We therefore treat single‑source music as a natural next extension once suitable data become available.
> - We clearly mark this as a **limitation and future extension** (Appendix L/O). Single‑source content identity is, in our view, a prerequisite capability for robust scene/multi‑speaker tasks, not a competing one.
>
> This pattern mirrors **MTEB’s own limitations** section (no long documents, task imbalance, missing multilingual retrieval,  no multi‑modal uses), which did not prevent acceptance or widespread adoption. We similarly see VocSim as a **first, extensible step** for zero‑shot audio similarity, not a final, all-encompassing benchmark.

---

> > ### Author Response · Authors · 2025-11-29
> > **Author summary and request to the new Area Chair (3 / 4)**
> >
> > ### “Uncontrolled conditions” and benchmark norms
> >
> > Reviewer DrDY argues that because we compare heterogeneous models (Whisper, WavLM, CLAP, BEATs, EAT, etc.) with different training data, sizes, and objectives, “it is impossible to draw meaningful conclusions.”
> >
> > However, this is **exactly** how modern embedding benchmarks operate in other areas, including audio:
> >
> > - **MTEB / BEIR**: compare models from 100M-parameter BERTs to multi‑billion-parameter LLM-based encoders, with very different training data and supervision. The goal is practical: *Which off‑the‑shelf embedding works best for these tasks?* Not: *Which architecture is most data-efficient under controlled pre-training?*
> > - **HEAR / SUPERB**: also compare heterogeneous encoders; they do not insist on identical pre‑training corpora or model sizes. HEAR/SUPERB additionally introduce fine‑tuning or probing, which we deliberately avoid to focus on frozen geometry.
> > - **BirdSet (ICLR 2025)** and other audio benchmarks (BEANS, BIRB) likewise evaluate multiple architectures trained under different regimes without controlling pre‑training conditions. BirdSet’s own Limitations section explicitly notes that it does not include the most recent audio transformers and a more thorough investigation of the influence of pre‑training, and that these aspects are left to future extensions of the benchmark.
> >
> > VocSim follows this **foundation-model benchmark pattern**:
> >
> > - We evaluate **off‑the‑shelf encoders as used in practice**. Retraining Whisper (680k hours), WavLM (94k hours), BEATs, etc. under controlled conditions is simply not feasible for a benchmark paper.
> > - We document **parameter counts and compute** (Appendix I) to give context on capacity, and we added a **pre‑training overlap audit** (Appendix D.2) to make clear which subsets are public-transfer vs. strict OOD (blind low‑resource speech).
> > - Our goal is to **audit the landscape of available audio embeddings**, not to derive causal claims about training recipes.
> >
> > If “uncontrolled training conditions” were grounds for rejection, then the same critique would apply equally to MTEB/BEIR, HEAR, BirdSet, and others, most of which were accepted at NeurIPS/ICLR as benchmarks. We therefore respectfully ask that this be **not treated as a flaw**, given established norms.
> >
> > ### Scope: zero‑shot similarity vs. downstream tasks
> >
> > Reviewer DrDY also notes that VocSim focuses on classification/similarity and does not directly cover TTS, diarization, S2S, enhancement, etc.
> >
> > We agree those applications matter, but:
> >
> > - They **all require training or fine‑tuning**, often with task‑specific architectures and loss functions. Our stated aim is to evaluate **frozen embeddings in a training‑free, zero‑shot way**, analogous to MTEB / BEIR / GeneCIS.
> > - Many of these applications *consume* content embeddings; what they need is exactly the property VocSim measures:
> >   - Map acoustically variable instances of the same event to nearby points, invariant to speaker and channel.
> > - To show that this zero‑shot property has **real downstream relevance**, we went beyond a purely unsupervised benchmark:
> >   - We evaluated our best configuration (Whisper EWMTF D100) on the **HEAR benchmark** using its official linear-probe protocol (Appendix M.6). It achieves **SOTA or near‑SOTA** performance on several tasks (e.g., 98.6% on Speech Commands, 79.3% on CREMA‑D). This demonstrates that **good VocSim performance predicts strong supervised downstream performance**.
> >   - We show that VocSim‑top embeddings **predict zebra finch perceptual similarity** (80.9% triplet accuracy) and **improve downstream bioacoustic classification**, acting as external validation in another species and domain.
> >
> > In other words, VocSim is designed like MTEB/BEIR:
> >
> > - Define a **narrow, training‑free core capability** (zero‑shot similarity / identity),
> > - Provide **simple, reproducible protocols**, and
> > - Show that high scores correlate with **useful downstream behavior**, without trying to include every possible task in one paper.
> >
> > We believe expecting a single benchmark paper to cover all supervised and generative audio tasks would set a bar significantly higher than the one applied in other modalities.

---

> > > ### Author Response · Authors · 2025-11-29
> > > **Author summary and request to the new Area Chair (4 / 4)**
> > >
> > > ### What we concretely did during rebuttal
> > >
> > > Beyond explanations, we made extensive technical and methodological changes:
> > >
> > > - **Positioning & overlap**
> > >   - Added a **pre‑training overlap audit** (Appendix D.2) marking confirmed / likely / no‑evidence overlap for every model–subset pair.
> > >   - Clarified “zero‑shot” as *no training on VocSim labels* and identified the **blind low‑resource speech sets** as our strict OOD evaluation.
> > >   - Added **Appendix D.3** comparing VocSim explicitly to HEAR, SUPERB, AudioSet, WavCaps, BirdSet, BIRB, and related resources.
> > >
> > > - **Robustness & metrics**
> > >   - Added a **label‑noise experiment** (Appendix H.1) showing GSR remains well above baseline even with 20% label corruption, while P@k drops quickly.
> > >   - Added **NMI, Purity, ARI** (Appendix H.4) to connect GSR/P@k to standard clustering metrics; the rankings are consistent.
> > >   - Added pooling ablations (Appendix H.2) and a **DTW sequence‑aware re‑ranking baseline** (Appendix H.3), finding no advantage over simple statistical pooling.
> > >   - Added a **Whisper layer sweep** (Appendix H.5), demonstrating that performance is reasonably stable across layers and that using the final layer is justified.
> > >
> > > - **Models & external validation**
> > >   - Added **BEATs and EAT** to the model suite (Section 5, Appendix P), as requested.
> > >   - Evaluated the top VocSim configuration on **HEAR** (Appendix M.6) with strong results.
> > >   - Expanded downstream and cross-species validations (zebra finch perceptual similarity, bioacoustic classification).
> > >
> > > - **Clarity & presentation**
> > >   - **Manually rebuilt and verified all references**, fixing the initial bibliography incident.
> > >   - Added a **Glossary** (Appendix C) for all datasets, models, and pooling variants.
> > >   - Rewrote figure captions (e.g., Figures 1–2) to be self-contained and fixed ambiguous terminology (e.g., “acoustic‑to‑label consistency”).
> > >   - Clarified dataset selection criteria and scope (Section 3, Appendix D.3, L, O).
> > >
> > > We believe these changes answer the substantive scientific and methodological concerns in the original reviews.
> > >
> > > ---
> > >
> > > ### Closing request
> > >
> > > The current situation is that:
> > >
> > > - One reviewer (rd9W) gave a clear **accept (8)** and strongly praised the quality and utility of the resource.
> > > - Two reviewers (LeP5, DrDY) gave **2s**, mainly due to concerns about perceived limited novelty, scope, and lack of strict control across models. We have addressed these in detail and shown that our design is aligned with accepted benchmark practices.
> > > - One reviewer (j2QK, 0) focused largely on the **bibliography incident** as the “primary reason for the very low score,” while acknowledging that the revised paper is now “objectively … in a better state.” That incident has been fully corrected, and our extensive code/data/experiments are public and reproducible.
> > >
> > > Because reviews are reverted to their **pre-discussion** state and are now frozen, the numeric ratings under‑represent the effect of our revisions and the final clarifications.
> > >
> > > We respectfully ask you to judge VocSim primarily on:
> > >
> > > - The **soundness and rigor** of the revised manuscript,
> > > - Its **clear practical utility** as a training‑free diagnostic for frozen audio embeddings, and
> > > - Its **novel, much‑needed role** in the ecosystem of audio benchmarks (zero‑shot content identity, single‑source audio),
> > >
> > > and to consider recommending **acceptance in the datasets & benchmarks track**.
> > >
> > > Thank you very much for your time and careful consideration.

---

### Note · Authors · 2025-12-02

I have read and agree with the venue's withdrawal policy on behalf of myself and my co-authors.